# Benign Overfitting in Adversarial Training for Vision Transformers

## Abstract

Despite the remarkable success of Vision Transformers (ViTs) across a wide range of vision tasks, recent studies have revealed that they remain vulnerable to adversarial examples, much like Convolutional Neural Networks (CNNs). A common empirical defense strategy is adversarial training, yet the theoretical underpinnings of its robustness in ViTs remain largely unexplored. In this work, we present the first theoretical analysis of adversarial training under simplified ViT architectures. We show that, when trained under a signal-to-noise ratio that satisfies a certain condition and within a moderate $\ell_2$ perturbation budget, adversarial training enables ViTs to achieve nearly zero robust training loss and robust generalization error under certain regimes. Remarkably, this leads to strong generalization even in the presence of overfitting, a phenomenon known as *benign overfitting*, previously only observed in CNNs (with adversarial training). Experiments on both synthetic and real-world datasets further validate our theoretical findings.

## 1 Introduction

Vision Transformers (ViTs) have emerged as a powerful alternative to convolutional neural networks (CNNs) (Krizhevsky et al., 2012) for a wide range of computer vision tasks, including image classification, object detection, semantic segmentation, and vision-language modeling (Dosovitskiy et al., 2020; Liu et al., 2021; Hu et al., 2025a). Despite their strong performance, recent studies (Hu et al., 2023; Shao et al., 2021; Hu et al., 2025b; 2024) have revealed that ViTs, similar to CNNs, can still be vulnerable to small, carefully crafted perturbations. These perturbations, referring to adversarial examples (Szegedy et al., 2013), can often cause significant performance degradation.

A widely studied defense mechanism against such vulnerabilities is *adversarial training* (Goodfellow et al., 2014; 2016), which augments the training process with adversarially perturbed samples to improve model robustness. While adversarial training has proven effective in enhancing model robustness, it is frequently accompanied by a noticeable degradation in generalization performance on clean data (Raghunathan et al., 2019; Xu et al., 2023). Such a robustness-generalization trade-off (Fu & Wang, 2023; Zhang et al., 2021; Xiao et al., 2022) has been extensively studied, raising the question of whether it is possible to preserve robustness without sacrificing clean-data accuracy.

*Benign Overfitting* (Bartlett et al., 2020) refers to the phenomenon that overparameterized models can interpolate the training data (achieving near-zero empirical risk) yet still generalize well. In the standard (non-adversarial) setting, this behavior has been studied across various model architectures, including linear regression (Bartlett et al., 2020; Zhou & Ge, 2023), logistic regression Wang et al. (2021); Cao et al. (2021), ridge regression (Tsigler & Bartlett, 2023), kernel methods (Mei & Montanari, 2022), and neural networks (Li et al., 2021; Cao et al., 2022; Kou et al., 2023; Frei et al., 2023; Jiang et al., 2024; Frei & Vardi, 2024). In the adversarial setting, Chen et al. (2023) provides the first theoretical analysis that benign overfitting can arise in adversarial training for linear regression under sub-Gaussian mixture models. Subsequently, Wang et al. (2024) shows that adversarial training can still generalize well in the presence of inference-time attacks for two-layer neural networks under appropriate distributional assumptions. However, to the best of our knowledge, it remains unclear whether analogous behavior occurs in more advanced architectures such as ViTs.

Compared with linear and two-layer models, analyzing ViTs poses distinctive challenges for analyzing robust benign overfitting. Unlike CNNs or two-layer neural networks with activation functions followed by a linear model, ViTs incorporate attention heads with query, key, and value projection

matrices, leading to substantially more complex training dynamics. The effect of perturbations on attention heads differs significantly from their effect on linear layers or activation-plus-linear structures, and varying perturbation magnitudes can markedly influence the training dynamics of ViTs (A more detailed discussion is provided in Section 4). Thus, the benign overfitting behavior or its conditions for ViTs may be significantly different from previous models.

To fill the gap, in this work, we provide the first comprehensive theoretical analysis of robust benign overfitting for a simplified ViT model. Specifically,

1. We validate that *benign overfitting* can also arise in adversarially trained Vision Transformers when the signal-to-noise ratio and the perturbation magnitude satisfy certain conditions, similar to linear and two-layer neural network models. That is, the adversarially trained interpolator attains near-zero robust training loss while maintaining small robust test error.

2. By analyzing the adversarial training dynamics of ViTs, we identify three key regimes: 1) Small perturbations yield trajectories close to clean training; 2) Moderate perturbations cause the attention mechanism to fail, such that the ViT collapses to a linear model; 3) Large perturbations lead to significant generalization error beyond benign overfitting. In all cases, we provide explicit upper bounds on the clean and robust test error.

3. We empirically verify our theoretical findings on both synthetic data and MNIST, demonstrating agreement between the derived bound and observed conditions for the occurrence of benign overfitting.

## 2 RELATED WORK

**Benign Overfitting.** Benign overfitting refers to the phenomenon where a predictor perfectly fits (interpolates) noisy training data yet still achieves strong generalization performance on unseen data. Bartlett et al. (2020) analyzed benign overfitting in linear regression with Gaussian noise, showing that in high-dimensional (overparameterized) regimes, the excess risk of interpolation can be asymptotically optimal. Foundational studies have further established benign overfitting in various linear settings, including regression, sparse regression, logistic regression, ridge regression, and kernel methods (Belkin et al., 2018; Bartlett et al., 2020; Hastie et al., 2022; Ding et al., 2024b;a). In neural networks, Frei et al. (2023) studied benign overfitting in two-layer networks without relying on the lazy training assumption in finite-width regimes. Cao et al. (2022); Kou et al. (2023) examined benign overfitting in convolutional networks from a feature learning perspective. Recently, Jiang et al. (2024) investigated learning dynamics in Vision Transformers (ViTs), delineating the boundary between benign and harmful overfitting. Frei & Vardi (2024) also analyzed benign overfitting in trained Transformer classifiers within in-context learning setups.

**Benign Overfitting for Adversarial Training.** Chen et al. (2023) initiated the study of benign overfitting under adversarial training in the context of linear classifiers with sub-Gaussian mixture data, proving that under moderate perturbations, linear classifiers can achieve near-optimal standard and adversarial risks. Building on this line of work, Wang et al. (2024) extended the analysis to two-layer networks and demonstrated that, regardless of whether the activation function is smooth or non-smooth, adversarial training can achieve near-optimal robust generalization error. In contrast, Hao & Zhang (2024) showed that under non-negligible noise, linear regression and NTK regression models tend to overfit the training data, yielding estimators with inflated Lipschitz norms and consequently elevated adversarial risk. However, none of them considered and their conclusions do not hold for transformer architectures.

**Adversarial Robustness in Transformer.** Adversarial training has been widely used to improve Transformer robustness. For Vision Transformers (ViTs), Herrmann et al. (2022) proposed PyramidAT, combining consistent dropout and stochastic depth to alleviate performance degradation, while Wu et al. (2022) reduced computational cost via attention-guided dropping of patch embeddings. Gopal et al. (2025) introduced SAFER, a layer-selective fine-tuning method with sharpness-aware minimization to mitigate adversarial overfitting, and Islam et al. (2025) analyzed layer-wise perturbation propagation, proposing a neuron-level suppression mechanism. However, these works are purely empirical and lack theoretical guarantees. Recent studies have begun to analyze robustness from a theoretical perspective in linear Transformer in-context learning, showing that robustness can be enhanced through adversarial training against hijacking attacks (Anwar et al.), short-suffix

training to defend long-suffix jailbreaks (Fu et al., 2025), and multi-task adversarial pretraining without downstream AT (Kumano et al., 2025). Yet, these analyses are restricted to the in-context learning setting with linear Transformers, and theoretical guarantees for adversarial robustness in general Transformer architectures remain an open problem.

**Differences in Adversarial Robustness Between Transformer and CNN.** Several studies have compared Transformers and CNNs under adversarial attacks. Bai et al. (2021) observe that under unified training settings, Transformers are not inherently more robust, with their OOD generalization mainly attributed to self-attention. Mo et al. (2022) show that under standard training, Transformers do not necessarily outperform CNNs under adversarial attack, and propose training strategies to improve ViT robustness. Dingeto & Kim (2024) propose a regularization method that enables ViTs to exhibit stronger adversarial robustness than CNNs. These studies are largely empirical, focusing on performance differences, and do not analyze the learning dynamics or the role of attention in adversarial training.

## 3 PROBLEM SETUP

In this section, we introduce the necessary definitions and formally describe the Gradient Descent-based Adversarial Training under the multi-patch data distribution and the two-layer Vision Transformer model.

**Notations.** For two sequences $\{x_n\}$ and $\{y_n\}$, we write $x_n = O(y_n)$ if there exist absolute constants $C > 0$ and $N > 0$ such that $|x_n| \leq C|y_n|$ for all $n \geq N$. Similarly, we write $x_n = \Omega(y_n)$ if $y_n = O(x_n)$. We say $x_n = \Theta(y_n)$ if both $x_n = O(y_n)$ and $x_n = \Omega(y_n)$. Moreover, $x_n = o(y_n)$ if $\lim_{n \to \infty} |x_n/y_n| = 0$. Finally, we use $\widetilde{O}(\cdot)$, $\widetilde{\Omega}(\cdot)$, and $\widetilde{\Theta}(\cdot)$ to denote the corresponding notations with logarithmic factors suppressed.

**Definition 1** (Data Generation Model). *Let $\boldsymbol{\mu}_+, \boldsymbol{\mu}_- \in \mathbb{R}^d$ be fixed vectors representing the signals contained in data points, where $\|\boldsymbol{\mu}_+\|_2 = \|\boldsymbol{\mu}_-\|_2 = \|\boldsymbol{\mu}\|_2$ and $\langle \boldsymbol{\mu}_+, \boldsymbol{\mu}_- \rangle = 0$. Then each data point $(\mathbf{X}, y)$ with $\mathbf{X} = (\mathbf{x}_1, \mathbf{x}_2, \ldots, \mathbf{x}_M)^\top \in \mathbb{R}^{M \times d}$ and $y \in \{-1, 1\}$ is generated from the following distribution $D$:*

*(1) The label $y$ is generated as a Rademacher random variable, i.e., $\mathbb{P}(y = 1) = \mathbb{P}(y = -1) = \frac{1}{2}$.*

*(2) If $y = 1$ then $\mathbf{x}_1$ is given as $\boldsymbol{\mu}_+$, if $y = -1$ then $\mathbf{x}_1$ is given as $\boldsymbol{\mu}_-$, which represents signals.*

*(3) $\mathbf{x}_2, \ldots, \mathbf{x}_M$ are given by noise vectors $\boldsymbol{\xi}_2, \ldots, \boldsymbol{\xi}_M$, generated i.i.d from the Gaussian distribution $\mathcal{N}(0, \sigma_p^2 \cdot (\mathbf{I} - \boldsymbol{\mu}_+ \boldsymbol{\mu}_+^\top \cdot \|\boldsymbol{\mu}\|_2^{-2} - \boldsymbol{\mu}_- \boldsymbol{\mu}_-^\top \cdot \|\boldsymbol{\mu}\|_2^{-2}))$, which represent noises.*

Our data generation model is motivated by the patch-level structure of real image data, where some patches encode class-relevant signals (e.g. semantic information) while others capture irrelevant noise (e.g. background artifacts). Similar constructions have been widely employed in the feature learning literature to analyze the generalization behavior of overparameterized classifiers (Allen-Zhu & Li, 2020; Cao et al., 2022; Jelassi & Li, 2022; Kou et al., 2023; Zou et al., 2023; Jiang et al., 2024; Han et al., 2024; Ding et al., 2025). In our setup, the noise component $\boldsymbol{\xi}$ is modeled as a Gaussian variable, with its covariance structure designed to remain orthogonal to the signal component $\boldsymbol{\mu}$, ensuring that the data noise is independent of and unrelated to the feature.

**Two-layer Transformer.** Following the architecture introduced by Jiang et al. (2024), we consider a simplified two-layer Transformer consisting of a self-attention layer followed by a fixed linear layer, defined as the following, where $\theta = (\mathbf{W}_Q, \mathbf{W}_K, \mathbf{W}_V)$.

$$f(\mathbf{X}, \theta) = \frac{1}{M} \sum_{l=1}^{M} \varphi(\mathbf{x}_l^\top \mathbf{W}_Q \mathbf{W}_K^\top \mathbf{X}^\top) \mathbf{X} \mathbf{W}_V \boldsymbol{w}_O. \quad (1)$$

Here, $\varphi(\cdot) : \mathbb{R}^M \to \mathbb{R}^M$ denotes the softmax function; $\mathbf{W}_Q, \mathbf{W}_K \in \mathbb{R}^{d \times d_h}$ and $\mathbf{W}_V \in \mathbb{R}^{d \times d_v}$ represent the query, key, and value matrices, respectively; and $\boldsymbol{w}_O \in \mathbb{R}^{d_v}$ represents the weight vector of the linear layer. We use $\theta$ to denote the collection of all the model weights. This model is not reduced to a linear or single-layer attention architecture; instead, it more closely resembles the structure of a real Transformer, with a correspondingly more complex parameter update process.

**Loss Function.** Let $S = \{(\mathbf{X}_n, y_n)\}_{n=1}^N$ denote the training dataset drawn from the distribution $D$ defined in Definition 1, where $n$ indexes the samples (so $(\mathbf{X}_n, y_n)$ represents the $n$-th sample). In

---

**Algorithm 1** Gradient Descent-based Adversarial Training

---

**Require:** Step size $\eta$, perturbation budget per token $\tau$, number of iterations $T$, standard deviation $\sigma_V, \sigma_h$
1: Initialize network weights $(\mathbf{W}_Q^0)_{ij}, (\mathbf{W}_K^0)_{ij} \sim \mathcal{N}(0, \sigma_h^2), (\mathbf{W}_V^0)_{ij} \sim \mathcal{N}(0, \sigma_V^2)$   i.i.d.
2: **for** $t = 0, \ldots, T-1$ **do**
3:     **for** $n = 1, \ldots, N$ **do**
4:        Generate adversarial example for input $\mathbf{X}_n$:
       $\widetilde{\mathbf{X}}_n^t \leftarrow \arg\max_{\widetilde{\mathbf{X}}_n \in B(\mathbf{X}_n, \tau)} \ell\big(y_n f(\widetilde{\mathbf{X}}_n; \mathbf{W}_Q^t, \mathbf{W}_K^t, \mathbf{W}_V^t)\big)$
5:     **end for**
6:     Update all weight matrices simultaneously: $(\mathbf{W}_Q^{t+1}, \mathbf{W}_K^{t+1}, \mathbf{W}_V^{t+1}) \leftarrow (\mathbf{W}_Q^t, \mathbf{W}_K^t, \mathbf{W}_V^t) - \frac{\eta}{N} \sum_{n=1}^N (\nabla_{\mathbf{w}_Q}, \nabla_{\mathbf{w}_K}, \nabla_{\mathbf{w}_V}) \ell\big(y_n f(\widetilde{\mathbf{X}}_n^t; \mathbf{W}_Q^t, \mathbf{W}_K^t, \mathbf{W}_V^t)\big)$
7: **end for**
8: **return** $\theta(T) = (\mathbf{W}_Q^T, \mathbf{W}_K^T, \mathbf{W}_V^T)$

---

this work, we adopt the empirical cross-entropy loss as a surrogate for the non-differentiable $0/1$ loss, and train the two-layer Transformer by minimizing this loss:

$$L_S(\theta) = \frac{1}{N} \sum_{n=1}^N \ell(y_n f(\mathbf{X}_n, \theta)),$$

where $\ell(z) = \log(1 + \exp(-z))$ and $f(\mathbf{X}, \theta)$ is the two-layer Transformer. We measure the generalization ability of the two-layer Transformer using the *test error*, defined as the expected $0/1$ loss over the data distribution $D$:

$$L_D(\theta) = \mathbb{E}_{(\mathbf{x}, \mathbf{y}) \sim D} \mathbb{1}\left(y f(\mathbf{X}, \theta) \leq 0\right).$$

**Robust Loss.** We consider $\ell_2$ norm-bounded adversarial perturbations applied to each component of the input sequence $\mathbf{X} = [\mathbf{x}_1, \ldots, \mathbf{x}_M] \in \mathcal{X}^M$, where each $\mathbf{x}_m \in \mathbb{R}^d$ denotes a token (or patch) embedding. For a perturbation budget $\tau > 0$, the admissible perturbation set is $B(\mathbf{X}, \tau) := \big\{ \widetilde{\mathbf{X}} = [\widetilde{\mathbf{x}}_1, \ldots, \widetilde{\mathbf{x}}_M] \mid \|\widetilde{\mathbf{x}}_m - \mathbf{x}_m\|_2 \leq \tau, \forall m \in [M] \big\}$. Under this threat model, the robust $0/1$ loss is defined as $\ell_{\mathrm{rob}}^{0/1}(y f(\mathbf{X}, \theta)) := \max_{\widetilde{\mathbf{X}} \in B(\mathbf{X}, \tau)} \mathbb{1}\big(y f(\widetilde{\mathbf{X}}, \theta) \leq 0\big)$, and the robust loss is defined as $\ell_{\mathrm{rob}}(y_n f(\mathbf{X}_n, \theta)) := \max_{\widetilde{\mathbf{X}}_n \in B(\mathbf{X}_n, \tau)} \ell\big(y_n f(\mathbf{X}_n, \theta)\big)$. The robust test error and robust test loss are:

$$L_D^{\mathrm{rob}}(\theta) := \mathbb{E}_{(\mathbf{X}, y) \sim \mathcal{D}} \ell_{\mathrm{rob}}^{0/1}(y f(\mathbf{X}, \theta)), \quad L_S^{\mathrm{rob}}(\theta) := \frac{1}{N} \sum_{n=1}^N \ell_{\mathrm{rob}}(y_n f(\mathbf{X}_n, \theta))$$

**Adversarial Training.** We adopt a Gradient Descent-based Adversarial Training algorithm to update the network parameters, as summarized in Algorithm 1. Note that we initialize the network weights $\mathbf{W}_Q$, $\mathbf{W}_K$, and $\mathbf{W}_V$ with Gaussian distributions, where each entry of $\mathbf{W}_Q$ and $\mathbf{W}_K$ is drawn from $\mathcal{N}(0, \sigma_h^2)$, and each entry of $\mathbf{W}_V$ is drawn from $\mathcal{N}(0, \sigma_V^2)$. The algorithm iteratively constructs adversarial training examples by maximizing the training loss with respect to the input and updates the model parameters based on them. In our setting, only the attention projection matrices $\mathbf{W}_Q, \mathbf{W}_K, \mathbf{W}_V$ are updated during training, while the output vector $\boldsymbol{w}_O$ remains fixed.

## 4 MAIN RESULTS

In this section, we present our main theoretical results on the convergence and generalization of the ViT model, demonstrating how the signal-to-noise ratio $\mathrm{SNR} = \|\boldsymbol{\mu}\|_2 / (\sigma_p \sqrt{d})$ and the sample size $N$ influence its adversarial training dynamics. We first introduce the following conditions.

**Condition 1.** *Given a sufficiently small failure probability $\delta > 0$ and a target training loss $\epsilon > 0$, suppose that:*

*(1) The dimension $d$ and $d_h$ are sufficiently large satisfying $d = \widetilde{\Omega}\left(\epsilon^{-2} N^2 d_h\right)$ and $d_h = \widetilde{\Omega}\left(\max\{\{SNR^4, SNR^{-4}\}\right) N^2 \epsilon^{-2}$.*

*(2) The training sample size $N$ is large enough such that $N = \Omega(poly \log(d))$.*

*(3) The number of input tokens is bounded as $M = \Theta(1)$, and the $\ell_2$-norm of linear layer weights satisfies $\|\boldsymbol{w}_O\|_2 = \Theta(1)$.*

*(4) The learning rate $\eta$ is chosen sufficiently small so that $\eta \lesssim \widetilde{\mathcal{O}}(\min\{\|\boldsymbol{\mu}\|_2^{-2}, (\sigma_p^2 d)^{-1}\} \cdot d_h^{-\frac{1}{2}})$.*

*(5) The Gaussian initialization is appropriately chosen such that the standard deviation $\sigma_V$ satisfies $\sigma_V \leq \widetilde{\mathcal{O}}(\|\boldsymbol{w}_O\|_2^{-1} \cdot \min\{\|\boldsymbol{\mu}\|_2^{-1}, (\sigma_p \sqrt{d})^{-1}\} \cdot d_h^{-\frac{1}{4}} d^{-\frac{1}{2}})$ ,and the variance $\sigma_h^2$ satisfies $\min\{\|\boldsymbol{\mu}\|_2^{-2}, (\sigma_p^2 d)^{-1}\} \cdot d_h^{-\frac{1}{2}} \cdot (\log(6N^2 M^2/\delta))^{-2} \leq \sigma_h^2 \leq \min\{\|\boldsymbol{\mu}\|_2^{-2}, (\sigma_p^2 d)^{-1}\} \cdot d_h^{-\frac{1}{2}} \cdot (\log(6N^2 M^2/\delta))^{-\frac{3}{2}}$.*

*(6) The target training loss is satisfying $\epsilon \leq O(1/poly \log(d))$.*

Conditions (1) and (2) ensure that the learning problem is set in a sufficiently over-parameterized regime, allowing the model to fully capture the feature signal described in Definition 1. Condition (3) guarantees that each class contains enough samples with high probability. Conditions (4) and (5) simplify the analysis, though they can be generalized to the settings $M = \Omega(1)$, $\|\boldsymbol{w}_O\|_2 = o(1)$, or $\|\boldsymbol{w}_O\|_2 = \omega(1)$. Together, Conditions (4) and (5) further ensure that the Transformer can be effectively trained. Finally, Condition (6) ensures that the Transformer sufficiently overfits the training data. Similar conditions are widely made in the theoretical analysis of benign overfitting in neural networks (Allen-Zhu & Li, 2020; Cao et al., 2022; Frei et al., 2022; Jelassi & Li, 2022; Chatterji & Long, 2023; Zou et al., 2023; Kou et al., 2023; Frei & Vardi, 2024; Jiang et al., 2024).

**Theorem 2** (Benign Overfitting under Adversarial Training)**.** *Under Condition 1, we distinguish two cases:*

***Case 1.*** *If we have $N \cdot \mathrm{SNR}^2 = \Omega(1)$ and $\tau \leq O(\frac{\|\boldsymbol{\mu}\|_2}{\log d_h})$, ViT's attention head is effectively trainable . In this case, let $T = \Theta(\eta^{-1} \epsilon^{-1} \|\boldsymbol{\mu}\|_2^{-2} \|\boldsymbol{w}_O\|_2^{-2})$.*

***Case 2.*** *If we have $N \cdot \mathrm{SNR}^2 = \Omega(\frac{1}{\epsilon})$ and $\omega(\frac{\|\boldsymbol{\mu}\|_2}{\log d_h}) \leq \tau \leq O(\boldsymbol{\mu})$ , the attention head parameters barely update, causing the attention weights to remain nearly uniform and the ViT degenerates into a linear model . In this case, let $T = M \cdot \Theta(\eta^{-1} \epsilon^{-1} \|\boldsymbol{\mu}\|_2^{-2} \|\boldsymbol{w}_O\|_2^{-2})$.*

*In both cases, with probability at least $1 - d^{-1}$, the following holds:*

*1. The robust training loss converges to $\epsilon$:*

$$L_S^{\mathrm{rob}}(\theta(T)) \leq \epsilon.$$

*2. The clean test error satisfies:*

$$L_D(\theta(T)) \leq \exp\left(-C \cdot d\, \mathrm{SNR}^2\right).$$

*3. The robust test error satisfies:*

$$L_D^{\mathrm{rob}}(\theta(T)) \leq \exp\left(-C \cdot d\, \mathrm{SNR}^2 (1 - \frac{\tau}{\|\boldsymbol{\mu}\|_2})^2\right).$$

Theorem 2 demonstrates that , under the assumption on $\tau$, the model attains adversarial robustness, while the SNR condition guarantees that it prioritizes the signal over the noise, thereby leading to benign overfitting. The adversarially-trained two-layer Transformer model exhibits three key observations. 1) The model fits the training data well, with training loss converging to $\epsilon$. 2) For generalization guarantee, the clean test error decays rapidly with increasing $d$ and SNR, and obviously less than $\epsilon$ by Condition (2). This is consistent with classical benign overfitting phenomena established in prior work (Jiang et al., 2024). 3) For robustness guarantees, Theorem 2 provides an explicit upper bound on the robust test error as a function of the perturbation radius $\tau$ and the signal strength $\|\boldsymbol{\mu}\|_2$, showing that the bound increases with increasing $\tau$, which is consist with prior empirical observations(Madry et al., 2017; Schmidt et al., 2018).

**The impact of perturbation $\tau$ radius.** Here, we give more discussion on the perturbation radius $\tau$ to illustrate how it affects the different dynamics of ViTs, leading to the various results stated in Theorem 2. First, it can be observed that the softmax structure in attention is highly sensitive to perturbations according to Lemma 4. When $\tau \leq O(\frac{\|\boldsymbol{\mu}\|_2}{\log d_h})$, adversarial perturbations do not

dominate the learning dynamics, and the adversarial training trajectory of ViT remains close to standard clean training, corresponding to Case 1 in Theorem 2. In contrast, when $\omega(\frac{\|\boldsymbol{\mu}\|_2}{\log d_h}) \leq \tau \leq O(\boldsymbol{\mu})$, the updates between signal and noise in attention are effectively canceled out by the perturbation, forcing attention weights to remain close to their initialized uniform distribution, under which ViT degenerates into a linear model, corresponding to Case 2 in Theorem 2. The relevant lemmas will be provided in Section 5.2 later.

**The difference between ViT and degenerated linear model.** Although both cases can achieve benign overfitting, the attention mechanism in ViTs in Case 1 enables the model to learn the signal more rapidly, leading to faster convergence, and allows it to extract useful information even from sparser signals. According to Theorem 2, for the degenerated linear model, the convergence time $T$ is $M$ times slower than that of the ViT, and achieving benign overfitting requires a higher signal-to-noise ratio, $N \cdot \mathrm{SNR}^2 = \Omega(1/\epsilon)$, highlighting the advantages of the ViT architecture.

**Comparison with prior work.** The most closely related works to ours are Chen et al. (2023); Wang et al. (2024), which also investigated the phenomenon of benign overfitting under adversarial training. However, Chen et al. (2023) focused on linear regression models with moderate perturbations, and Wang et al. (2024) studied simplified neural networks, leaving more complex architectures unexplored. Our results on ViTs therefore complement this line of research. In addition, even in the more complex setting, our analysis does not rely on the implicit assumption of a large $\|\boldsymbol{\mu}\|_2$, i.e., $\|\boldsymbol{\mu}\|_2 = \Theta(d^r)$ for some $r \in (1/4, 1/2]$ in (Chen et al., 2023). Moreover, our results require a minimum convergence time of $T \sim O(N \max\{\mathrm{SNR}^2, \mathrm{SNR}^{-4}\})$, which is significantly smaller than the convergence time $T \sim O(\frac{d^2}{\|\boldsymbol{\mu}\|^2 \epsilon^2})$ reported in prior studies on CNN models by (Wang et al., 2024) in the over-parameterized regime.

Next, we will show that once the perturbation radius $\tau$ exceeds the signal strength $\|\boldsymbol{\mu}\|_2$, no classifier can achieve nontrivial robust accuracy. This implies that excessive adversarial training leads to poor model performance, consistent with the findings of Wang et al. (2024). Combining with Theorem 2, we can see that the assumption of the relation between $\tau$ and $\|\boldsymbol{\mu}\|_2$ is essential to understand benign overfitting. Also, our radius for benign overfitting is almost tight.

**Theorem 3.** *For any given classifier $f(\cdot; \boldsymbol{\theta})$, when $\tau \geq \|\boldsymbol{\mu}\|_2$, the robust test error satisfies $L_D^{rob}(\boldsymbol{\theta}) \geq 0.25$.*

## 5 PROOF SKETCH

In this section, we provide a proof sketch of the different adversarial training dynamics due to different perturbation radii. Based on the ViT formulation in Eq.(1), when the perturbations are relatively small, we show its impact on the learning of the $\mathbf{W}_Q$ and $\mathbf{W}_K$ matrices is limited. If more attention is allocated to signal tokens, the value matrix $\mathbf{W}_V$ tends to align more closely with a perturbed signal vector $\widetilde{\boldsymbol{\mu}}$, while its direction remains dominated by the true signal $\boldsymbol{\mu}$. Consequently, the gradients propagated to the attention heads associated with the signal are larger than those to the noise, forming a positive feedback loop that facilitates better generalization, similar to the learning dynamics observed in the clean training (Jiang et al., 2024). When the perturbations are moderate, adversarial training suppresses the learning of token-to-signal attention, keeping the attention weights near their initialization. When $\mathbf{W}_V$ and $\mathbf{W}_K$ are initialized with small Gaussian variance, the attention distribution remains nearly uniform, and the ViT degenerates into a linear model. In this regime, the model requires a larger SNR to align $\mathbf{W}_V \boldsymbol{w}_O$ with the signal $\boldsymbol{\mu}$, thereby achieving benign overfitting.

### 5.1 VECTORIZED Q & K AND SCALARIZED V WITH TIME-INDEPENDENT PERTURBED INPUTS

First, unlike CNNs that treat convolutional kernels as vectors for signal–noise decomposition, our setting requires handling the more complex interactions among the matrices $\mathbf{W}_V$, $\mathbf{W}_K$, and $\mathbf{W}_Q$. Therefore, we consider vectorizing $\mathbf{W}_K$, and $\mathbf{W}_Q$ and scalarizing $\mathbf{W}_V$. We fix a universal perturbed input $\widetilde{\boldsymbol{X}} = [\widetilde{\boldsymbol{\mu}}, \widetilde{\boldsymbol{\xi}}_{n,2}, \ldots, \widetilde{\boldsymbol{\xi}}_{n,M}]$, where $\widetilde{\boldsymbol{\mu}}_+ \in B(\boldsymbol{\mu}_+, \tau)$, $\widetilde{\boldsymbol{\mu}}_- \in B(\boldsymbol{\mu}_-, \tau)$, and $\widetilde{\boldsymbol{\xi}}_{n,i} \in B(\boldsymbol{\xi}_{n,i}, \tau)$ are chosen once and remain fixed for all iterations $t$. These perturbations are universal and do not correspond to iteration-specific adversarial examples. For example, for $\mathbf{W}_V$ we have the following.

**Definition 2** (Scalarized V). *Let $\mathbf{W}_V^{(t)}$ denote the V matrix of the ViT at the $t$-th iteration of adversarial training. For the fixed perturbed vectors above, there exist scalars $\gamma_{V,+}^{(t)}$, $\gamma_{V,-}^{(t)}$, and $\rho_{V,n,i}^{(t)}$ such that*

$$\widetilde{\boldsymbol{\mu}}_+^\top \mathbf{W}_V^{(t)} \boldsymbol{w}_O = \widetilde{\boldsymbol{\mu}}_+^\top \mathbf{W}_V^{(0)} \boldsymbol{w}_O + \gamma_{V,+}^{(t)} \|\boldsymbol{w}_O\|_2^2,$$

$$\widetilde{\boldsymbol{\mu}}_-^\top \mathbf{W}_V^{(t)} \boldsymbol{w}_O = \widetilde{\boldsymbol{\mu}}_-^\top \mathbf{W}_V^{(0)} \boldsymbol{w}_O + \gamma_{V,-}^{(t)} \|\boldsymbol{w}_O\|_2^2,$$

$$\widetilde{\boldsymbol{\xi}}_{n,i}^\top \mathbf{W}_V^{(t)} \boldsymbol{w}_O = \widetilde{\boldsymbol{\xi}}_{n,i}^\top \mathbf{W}_V^{(0)} \boldsymbol{w}_O + \rho_{V,n,i}^{(t)} \|\boldsymbol{w}_O\|_2^2,$$

*for $i \in [M] \setminus \{1\}$ and $n \in [N]$.*

*We further denote the $V_+^{(t)} := \widetilde{\boldsymbol{\mu}}_+^\top \mathbf{W}_V^{(t)} \boldsymbol{w}_O$, $V_-^{(t)} := \widetilde{\boldsymbol{\mu}}_-^\top \mathbf{W}_V^{(t)} \boldsymbol{w}_O$ and $V_{n,i}^{(t)} := \widetilde{\boldsymbol{\xi}}_{n,i}^\top \mathbf{W}_V^{(t)}$.*

With scalarized $V$ in Defintion 2, we can provide the dynamics of matrix $\mathbf{W}_V^{(t)}$ by analyzing the update of coefficients $\gamma^{(t)}$ as follows:

$$\gamma_{V,+}^{(t+1)} = \gamma_{V,+}^{(t)} - \frac{\eta}{NM} \sum_{n \in S_+} \widetilde{\ell}_n'^{(t)} \Bigg[ \sum_{l=1}^M \langle \widetilde{\boldsymbol{\mu}}_+, \widetilde{\boldsymbol{\mu}}_+^{(t)} \rangle \, \varphi(\widetilde{\mathbf{x}}_{n,l} \mathbf{W}_Q \mathbf{W}_K^\top (\widetilde{\mathbf{X}}_n)^\top)_1$$

$$+ \sum_{i=2}^M \sum_{l=1}^M \langle \widetilde{\boldsymbol{\mu}}_+, \widetilde{\boldsymbol{\xi}}_{n,i}^{(t)} \rangle \, \varphi(\widetilde{\mathbf{x}}_{n,l} \mathbf{W}_Q \mathbf{W}_K^\top (\widetilde{\mathbf{X}}_n)^\top)_i \Bigg].$$

First, the perturbed signal and noise components break the orthogonality between signal and noise, resulting in the appearance of terms of the form $\langle \widetilde{\boldsymbol{\mu}}_+, \widetilde{\boldsymbol{\xi}}_{n,i}^{(t)} \rangle$. The upper bound of $\langle \widetilde{\boldsymbol{\mu}}_+, \widetilde{\boldsymbol{\mu}}_+^{(t)} \rangle$ is given by $(\|\boldsymbol{\mu}\|_2 + \tau)^2$, while the upper bound of $\langle \widetilde{\boldsymbol{\mu}}_+, \widetilde{\boldsymbol{\xi}}_{n,i}^{(t)} \rangle$ is $(\|\boldsymbol{\mu}\|_2 \tau + \sigma_p \tau \sqrt{2\log(4NM/\delta)} + \tau^2)$. Therefore, when the perturbation magnitude is small, $\langle \widetilde{\boldsymbol{\mu}}_+, \widetilde{\boldsymbol{\mu}}_+^{(t)} \rangle$ dominates, and the training dynamics under adversarial training closely resemble those under clean training.

When bounding $V_+^{(t)}$ (similar for $V_-^{(t)}$), we can consider a special case $\widetilde{\boldsymbol{\mu}} = \widetilde{\boldsymbol{\mu}}^{(t)}$ and derive the bound via cumulative summation, yielding $|V_+^{(t)}| \leq |V_+^{(0)}| + \sum_{s=0}^{t-1} |\gamma_{V,+}^{(s+1)} - \gamma_{V,+}^{(s)}| \cdot \|\boldsymbol{w}_O\|_2^2$ by Definition 2. In fact, we establish that for any $\widetilde{\boldsymbol{\mu}} \in B(\boldsymbol{\mu}, \tau)$, the quantity $\widetilde{\boldsymbol{\mu}}^\top \mathbf{W}_V^{(t)} \boldsymbol{w}_O$ admits a uniform upper bound.

Similarly, we can define vectorized $\mathbf{W}_Q$ and $\mathbf{W}_K$ as follows.

**Definition 3** (Vectorized Q & K.). *Let $\mathbf{W}_Q^{(t)}$ and $\mathbf{W}_K^{(t)}$ be the QK matrices of the ViT at the $t$-th iteration of gradient descent. Then we define the vectorized Q and vectorized K as follows*

$$\boldsymbol{q}_+^{(t)} = \widetilde{\boldsymbol{\mu}}_+^\top \mathbf{W}_Q^{(t)}, \quad \boldsymbol{q}_-^{(t)} = \widetilde{\boldsymbol{\mu}}_-^\top \mathbf{W}_Q^{(t)}, \quad \boldsymbol{q}_{n,i}^{(t)} = \widetilde{\boldsymbol{\xi}}_{n,i}^\top \mathbf{W}_Q^{(t)},$$

$$\boldsymbol{k}_+^{(t)} = \widetilde{\boldsymbol{\mu}}_+^\top \mathbf{W}_K^{(t)}, \quad \boldsymbol{k}_-^{(t)} = \widetilde{\boldsymbol{\mu}}_-^\top \mathbf{W}_K^{(t)}, \quad \boldsymbol{k}_{n,i}^{(t)} = \widetilde{\boldsymbol{\xi}}_{n,i}^\top \mathbf{W}_K^{(t)}$$

*for $i \in [M] \setminus \{1\}, n \in [N]$.*

With Definition 2 and 3, we can analyze the learning dynamics of the transformed coefficients rather than the original matrices.

## 5.2 EFFECTS OF PERTURBATION MAGNITUDE ON ATTENTION

Our second key technique is to analyze how the attention mechanism behaves under perturbations of varying magnitudes. First, we present the following lemma, which shows that the attention mechanism is robust to small perturbations.

**Lemma 4** (Informal). *Under Condition 1, suppose the perturbation satisfies $\tau \leq O(\frac{\|\boldsymbol{\mu}\|_2}{\log(d_h)})$ and $t \geq \Omega\left(\frac{1}{\eta \|\boldsymbol{\mu}\|_2^2 \|\boldsymbol{w}_O\|_2^2 \log(6N^2M^2/\delta)}\right)$. Then, there exists a universal constant $C \leq e/2$ such that*

$$\max_{\widetilde{\mathbf{X}} \in B(\mathbf{X}, \tau)} \text{softmax}\left(\langle \mathbf{q}^{(t)}, \mathbf{k}^{(t)} \rangle\right) \Big/ \min_{\widetilde{\mathbf{X}} \in B(\mathbf{X}, \tau)} \text{softmax}\left(\langle \mathbf{q}^{(t)}, \mathbf{k}^{(t)} \rangle\right) \leq C.$$

This lemma implies that the relative change in attention weights under perturbations is uniformly bounded; hence, attention computed on perturbed inputs closely matches that on the clean inputs. Thus, when considering the attention from the perturbed patch to another, it can be well-approximated by the attention from the clean signal to the clean noise.

Next, we present the following lemma, which characterizes the behavior of the attention mechanism under moderate perturbations.

**Lemma 5** (Informal). *Under Condition 1, supposing the perturbation satisfies* $\omega(\frac{\|\boldsymbol{\mu}\|_2}{\log d_h}) \leq \tau \leq O(\boldsymbol{\mu})$, *for any* $t \geq 0$, *we have:*

$$\frac{1}{M} - o(1) \leq softmax(\langle \mathbf{q}^{(t)}, \mathbf{k}^{(t)} \rangle) \leq \frac{1}{M} + o(1)$$

This lemma demonstrates that, under moderate perturbations, the attention distribution remains invariant and stays close to its initial uniform form.

### 5.3 GENERALIZATION GUARANTEE

For a new data point $(\mathbf{X}, y)$ generated from the distribution defined in Definition 1, we interpret the test error as the probability that the noise component dominates the model output. For adversarial test data with added perturbations, we need to bound the maximal distance between them and the corresponding clean test data. First, from the analysis in Section 5.1, we know that bounding $V^{(t)} = \widetilde{\mathbf{x}}^\top \mathbf{W}_V^{(t)} \boldsymbol{w}_O$ implies that its deviation from the unperturbed counterpart $\mathbf{x}^\top \mathbf{W}_V \boldsymbol{w}_O$ is at most $\left| \langle \widetilde{\mathbf{x}} - \mathbf{x}, \mathbf{W}_V \boldsymbol{w}_O \rangle \right| \leq \tau \|\mathbf{W}_V \boldsymbol{w}_O\|_2$. Second, from the analysis in Section 5.2, we know that the attention component $softmax(\langle \boldsymbol{q}, \boldsymbol{k} \rangle)$ exhibits robustness under perturbations of bounded magnitude. Combining these insights, we state the following lemma.

**Lemma 6** (Informal). *Under Condition 1, if* $t \geq \Omega\left(\eta^{-1}\epsilon^{-1}\|\boldsymbol{\mu}\|_2^{-2}\|\boldsymbol{w}_O\|_2^{-2}\log^{-1}\left(\frac{6N^2M^2}{\delta}\right)\right)$ *and* $\tau \leq \frac{\|\boldsymbol{\mu}\|_2}{\log(d_h)}$, *we have:*

$$yf(\mathbf{X}, \theta(t)) - \min_{\widetilde{\mathbf{X}} \in B(\mathbf{X}, \tau)} yf(\widetilde{\mathbf{X}}, \theta(t)) \lesssim M\|\mathbf{W}_V^{(t)} \boldsymbol{w}_O\|_2 \tau.$$

As a result, the robust test error can be interpreted as the probability of misclassification when the output of the ViT under clean test data is perturbed by its maximal adversarial deviation.

**Lemma 7** (Informal). *Under the same conditions of Lemma 6, with high probability, we have for some constant* $c > 0$:

$$P(\exists \widetilde{\mathbf{X}} \in B(\mathbf{X}, \tau) : yf(\widetilde{\mathbf{X}}, \theta(t)) \leq 0) = P\left(yf(\mathbf{X}, \theta(t)) + (yf(\mathbf{X}, \theta(t)) - min_{\widetilde{\mathbf{X}} \in B(\mathbf{X}, \tau)} yf(\widetilde{\mathbf{X}}, \theta(t))) \leq 0\right)$$

$$\leq \exp\left(-c\left(\frac{(V_+^{(t)} - V_-^{(t)}) - \|\mathbf{W}_V^{(t)} \boldsymbol{w}_O\|_2 \tau}{\sigma_p \|\mathbf{W}_V^{(t)} \boldsymbol{w}_O\|_2}\right)^2\right).$$

Consequently, the robust test error differs from the clean test error only by an additive factor that scales with both the perturbation radius and the model complexity. More details are in Appendix.

## 6 EXPERIMENTS

### 6.1 EXPERIMENTAL SETUP

**Datasets.** For the experiments on synthetic data, we synthesize each sample following the distribution in Definition 1. Specifically, we first formalize the two signal vectors in Definition 1 as $\boldsymbol{\mu}_+ = \|\boldsymbol{\mu}\|_2 \cdot [1, 0, \ldots, 0]^\top$ and $\boldsymbol{\mu}_- = \|\boldsymbol{\mu}\|_2 \cdot [0, 1, 0, \ldots, 0]^\top$, where the signal dimension $d$ is set as 1024. Then, for each generated sample, the number $M$ of tokens in this sample is set as 16 and every noise vector in this sample is drawn independently from the Gaussian distribution $\boldsymbol{\xi}_i \sim \mathcal{N}(0, 0.4 \cdot \mathbb{I}_d)$. We generate up to 22 samples for training and 100 samples for evaluation.

For the experiments on real-world data, we use a subset of MNIST where only samples with label '0' or '1' are used. To better simulate the feature learning setting in our theory, we transform each image to a vector with the following procedures: (1) Vectorize the image to a "signal" vector and

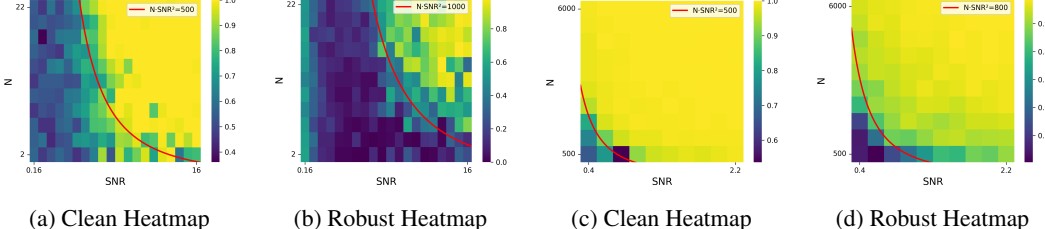

(a) Clean Heatmap     (b) Robust Heatmap     (c) Clean Heatmap     (d) Robust Heatmap

Figure 1: Clean and robust test accuracy under adversarial training across various signal-to-noise ratios (SNR) and sample sizes ($N$). (a)&(b): results on synthetic data; (c)&(d): results on real-world data. High test accuracy is colored in yellow, whereas low test accuracy is colored in purple.

normalize its $\ell_2$-norm to 5. (2) Draw a noise vector following the Gaussian distribution $\mathcal{N}(0, \mathbb{I}_d)$, where $d = 784$. (3) Concatenate the signal and noise vectors to form a new vector as the sample adoped in our experiments.

**Model architectures.** For synthetic data experiments, we adopt the two-layer Transformer defined in (1), where both of the hidden dimension $d_h$ and the value dimension $d_v$ are set as 128. For real-world data experiments, We implement a ViT model consisting of two attention layers, each with four self-attention heads, followed by an MLP layer with ReLU activation. The hidden dimension of this ViT is set as 128. Model parameters in all experiments are initialized using PyTorch's default method, followed by an additional scaling factor of $1/16$, matching the requirement of Condition (5).

**Model training & evaluation.** We use full-batch gradient descent to train all models in our experiments, with a learning rate of $0.1$. Each model will be trained until its training loss falls below a target threshold $0.01$. Besides, in each adversarial training step, we leverage projected gradient descent (PGD; Madry et al. 2017) to search adversarial examples with attack strength $\tau/\|\boldsymbol{\mu}\| = 0.05$ for 20 steps with a step size of $0.2\tau$, using per-token $L_2$-normalized gradient updates with projection onto the $L_2$ ball of radius $\tau$. To assess the performance of trained models, we report both their clean and robust classification errors calculated on test datasets.

### 6.2 RESULTS ANALYSIS

**Phase Transition in benign overfitting.** We perform adversarial training with different number $N$ of training data ranging from 2 to 22 and different SNR ranging from 0.16 to 16 on synthetic data. The clean and robust test accuracies of these models are collected and presented as heatmaps in Figures 1a and 1b. From the figure, we observe a clear decision boundary in the form $N \cdot \text{SNR}^2 = \Omega(1)$ separates the high-accuracy and low-accuracy regions. This observation aligns well with our Theorem 2 that $N \cdot \text{SNR}^2 = \Omega(1)$ is necessary for models in adversarial training to produce benign overfitting. We also conduct experiments on the real-world dataset MNIST, where similar conclusions can still be observed as shown in Figures 1c and 1d.

**Effects of signal-to-noise ratio and dataset size.** We then fix the number $N$ of training data and plot curves of the training loss/robust test accuracy versus the training iteration under different SNR in Figures 2a and 2b. From them, we observe that as the training iteration increases, the model overfits to training data, but its robustness does not consistently increase unless the SNR is large. We also fix the SNR and plot similar curves with different number of training data in Figures 2c and 2d, where we find that even the model is overfitting, its robustness improves only when the training data number $N$ is large. All these results indicate that benign overfitting can emerge in adversarial training only when the training data number $N$ and the data SNR are both not too small, which coincides with the requirements that $N \cdot \text{SNR}^2 = \Omega(1)$ or $N \cdot \text{SNR}^2 = \Omega(\frac{1}{\epsilon})$ in our Theorem 2.

We provide the following additional results in the appendix: Empirical validation of different learning dynamics under varying perturbation radii in Appendix B.1; Additional experiments on MNIST, CIFAR-10, and Tiny-ImageNet with the state-of-the-art APGD attack (Croce & Hein, 2020) in Appendix B.2; Additional experiments on multi-norm attacks in Appendix B.3; and Additional experiments on realistic ViT-base models(Dosovitskiy et al., 2020) in Appendix B.4.

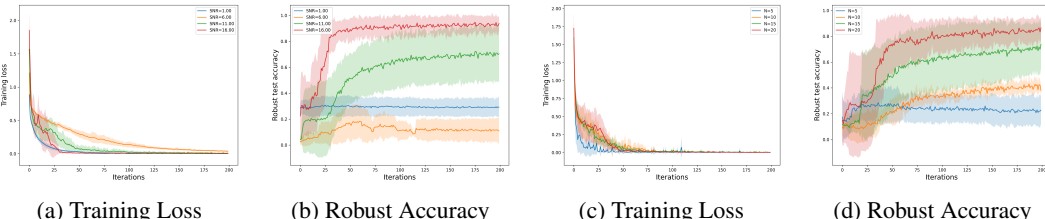

| (a) Training Loss | (b) Robust Accuracy | (c) Training Loss | (d) Robust Accuracy |

Figure 2: (a)&(b): Curves of robust training loss and robust test accuracy versus training iteration under fixed training data number $N = 22$. (c)&(d): Curves of robust training loss and robust test accuracy versus training iteration under fixed data SNR $= 12$.

## 7 CONCLUSION

Our paper presents the first comprehensive theoretical analysis of the generalization behavior after adversarial training on a two-layer Vision Transformer. We demonstrate that, under appropriate relationships between signal-to-noise ratio and perturbation magnitude, adversarially trained ViTs can interpolate the training data with vanishing robust loss while still achieving small robust test error. Our analysis reveals three perturbation regimes that clarify how adversarial training shapes the learning dynamics of attention heads, from clean-like behavior to linear collapse and eventual failure beyond the benign regime. Empirical results on synthetic data and MNIST corroborate the theory, aligning closely with the predicted conditions under which robust benign overfitting emerges. Empirically, experiments on synthetic data and MNIST corroborate the theory, matching the predicted conditions under which robust benign overfitting arises.

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

## A   LLMs Usage in the Paper

LLMs were used only occasionally to help polish the writing (propose new words, grammar and spelling correction). All technical ideas, experimental designs, analyses, conclusions, writing were developed and carried out entirely by the authors. The authors have full responsibility for the final text.

## B   Additional Experiments

### B.1   Empirical validation of theoretical regimes

In this section, we follow a similar setting as the experiments on synthetic data in Section 6, and add more experiments about training dynamics with different perturbation $\tau$ radius.

**Experiments setting.**

We focus on tracking the dynamics of attention entropy, training loss, and $W_V$ norm under benign overfitting regime with different perturbation $\tau$ radius, and the key parameters are as follows:

- $N = 25$
- $SNR = 16$
- $M = 2$
- $d = 1024$
- $d_h = 512$
- $d_v = 512$
- $\sigma_p = 0.05$
- $\frac{\tau}{\|\boldsymbol{\mu}\|_2} = 0.02, 0.1, 0.5$

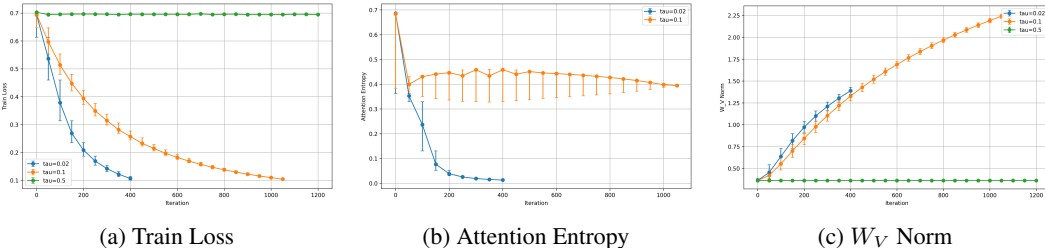

(a) Train Loss                (b) Attention Entropy                (c) $W_V$ Norm

Figure 3: Training dynamics with different perturbation $\tau$ radius: attention entropy, training loss, and $W_V$ norm.

**Experiments results.**

1. In Figure 3a, when $\frac{\tau}{\|\boldsymbol{\mu}\|_2} = 0.02$ or $0.1$, the adversarial training loss converges to zero, indicating that the model successfully interpolates all noise-corrupted training samples, consistent with the benign overfitting behavior in Theorem 2. In contrast, when $\frac{\tau}{\|\boldsymbol{\mu}\|_2} = 0.5$, the adversarial training loss fails to decrease, aligning with the non-convergence regime characterized in Theorem 3. Moreover, the case $\frac{\tau}{\|\boldsymbol{\mu}\|_2} = 0.02$ exhibits a faster convergence rate than the $\frac{\tau}{\|\boldsymbol{\mu}\|_2} = 0.1$ setting, highlighting the role of the attention mechanism in accelerating convergence, which is in agreement with our theoretical predictions in Theorem 2.

2. In Figure 3b, when $\frac{\tau}{\|\boldsymbol{\mu}\|_2} = 0.02$, the attention entropy decreases to nearly zero, indicating that the attention mechanism correctly concentrates on the signal patch. This behavior is consistent with Case 1 of Theorem 2, where the perturbation level is sufficiently small for the model to recover the underlying signal structure.

In contrast, when $\frac{\tau}{\|\boldsymbol{\mu}\|_2} = 0.1$, the attention entropy fails to decrease and instead remains high, demonstrating that moderate perturbations hinder the learning of attention weights. As a result, the

attention distribution remains nearly uniform rather than focusing on the signal patch. This phenomenon aligns with Case 2 of Theorem 2, where the perturbation magnitude prevents the attention mechanism from identifying the true signal.

3. In Figure 3c, when $\frac{\tau}{\|\boldsymbol{\mu}\|_2} = 0.1$, the $\|W_V\|_2$ norm exhibits the largest growth. This behavior is consistent with Case 2 of Theorem 2, where the ViT effectively collapses into a linear model and the value projection $W_V$ becomes the dominant component driving the learning dynamics.

In contrast, for $\frac{\tau}{\|\boldsymbol{\mu}\|_2} = 0.02$, the attention mechanism remains effective, so only mild updates to $W_V$ are required for the model to fit the noisy training data and achieve benign overfitting. When $\frac{\tau}{\|\boldsymbol{\mu}\|_2} = 0.5$, the perturbation is too large for the model to learn meaningful structure, resulting in $W_V$ failing to make progress during training.

## B.2 Additional Experiments on MNIST, CIFAR-10 and Tiny-ImageNet with APGD

In this section, we follow the same experimental setup as in Section 6 for the MNIST dataset and further extend our evaluation by conducting additional experiments on both MNIST, CIFAR-10 and Tiny-ImageNet under APGD attacks. These results demonstrate that our theoretical insights continue to hold for larger and more complex datasets, as well as under stronger adversarial attacks.

**Experiments setting.** We conduct adversarial training using the state-of-the-art APGD attack model. For the MNIST dataset, we consider an attack strength of $\frac{\tau}{\|\boldsymbol{\mu}\|_2} = 0.05$, and vary the number of training samples $N$ from 1000 to 6000 and SNR from 0.4 to 2. For the CIFAR-10 dataset, we set a weaker attack strength of $\frac{\tau}{\|\boldsymbol{\mu}\|_2} = 0.01$, and vary the number of training samples $N$ from 1000 to 10000 and the SNR from 0.4 to 10. For the Tiny-ImageNet dataset, we set a weaker attack strength of $\frac{\tau}{\|\boldsymbol{\mu}\|_2} = 0.10$, and vary the number of training samples $N$ from 100 to 1000 and the SNR from 0.4 to 10.

**Experiments results.** The clean and robust test accuracies on MNIST, CIFAR-10 and Tiny-ImageNet are collected and presented as heatmaps in Figures 4, 5 and 6. In both figures, we observe a clear phase transition phenomenon. Moreover, as both the sample size $N$ and the SNR increase, the clean and robust test error consistently decreases. This behavior is fully aligned with our theoretical analysis as well as the empirical findings reported in previous experiments.

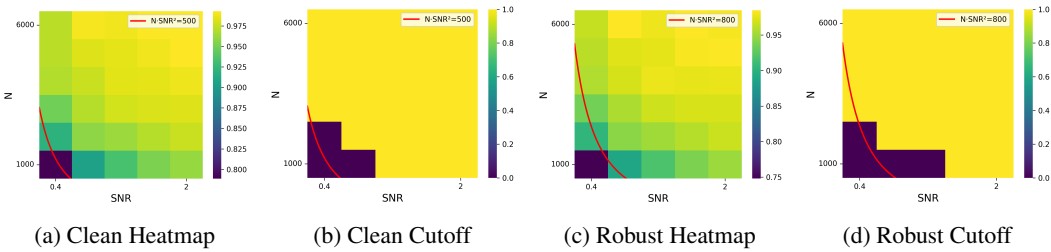

(a) Clean Heatmap     (b) Clean Cutoff     (c) Robust Heatmap     (d) Robust Cutoff

Figure 4: Clean and robust test accuracy heatmaps on MNIST with APGD attack across various signal-to-noise ratios (SNR) and sample sizes ($N$). (b)&(d) are a heatmap that applies a cutoff value 0.93.

## B.3 Additional Experiments on Multi-Norm Attacks

In this section, we follow the same experimental setup as in Section 6 for the MNIST dataset and further extend our evaluation by conducting additional experiments on both MNIST and CIFAR-10 under multi-norm attacks. These results demonstrate that although the model's robust test accuracy decreases under multi-norm attacks, our theoretical insights continue to hold.

**Experiments setting.** We perform adversarial training under a multi-norm PGD attack model spanning $l_1, l_2$ and $l_\infty$ perturbations.

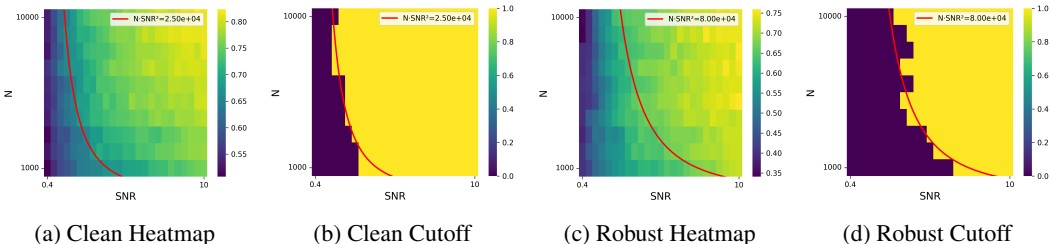

(a) Clean Heatmap      (b) Clean Cutoff      (c) Robust Heatmap      (d) Robust Cutoff

Figure 5: Clean and robust test accuracy heatmaps on CIFAR-10 with APGD attack across various signal-to-noise ratios (SNR) and sample sizes ($N$). (b)&(d) are a heatmap that applies a cutoff value 0.65.

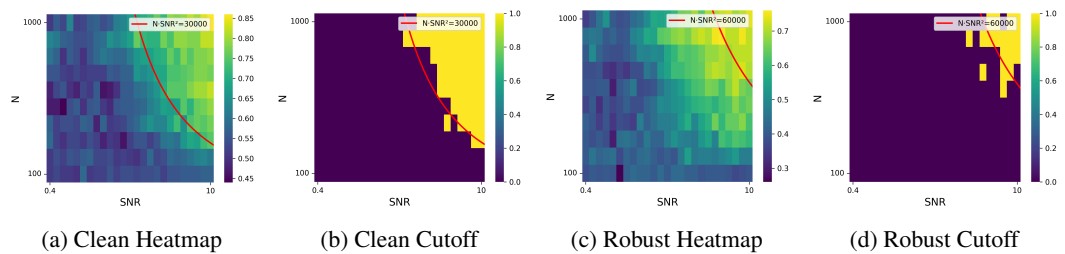

(a) Clean Heatmap      (b) Clean Cutoff      (c) Robust Heatmap      (d) Robust Cutoff

Figure 6: Clean and robust test accuracy heatmaps on Tiny Imagenet with APGD attack across various signal-to-noise ratios (SNR) and sample sizes ($N$). (b)&(d) are a heatmap that applies a cutoff value 0.70.

For the **MNIST** dataset, we consider a base attack strength of eps $= \frac{\tau_2}{\|\boldsymbol{\mu}\|_2} = 0.05$, and set $(\frac{\tau_1}{\|\boldsymbol{\mu}\|_2}, \frac{\tau_2}{\|\boldsymbol{\mu}\|_2}, \frac{\tau_\infty}{\|\boldsymbol{\mu}\|_2}) = (\text{eps} * 20, \text{eps}, \text{eps}/30)$. We vary vary the number of training samples $N$ from 1000 to 6000 and SNR from 0.4 to 2.

For the **CIFAR-10** dataset, we set a weaker base attack strength of eps $= \frac{\tau_2}{\|\boldsymbol{\mu}\|_2} = 0.01$, and set$(\frac{\tau_1}{\|\boldsymbol{\mu}\|_2}, \frac{\tau_2}{\|\boldsymbol{\mu}\|_2}, \frac{\tau_\infty}{\|\boldsymbol{\mu}\|_2}) = (\text{eps} * 20, \text{eps}, \text{eps}/30)$. We vary vary the number of training samples $N$ from 1000 to 10000 and the SNR from 0.4 to 10.

**Experiments results.** The clean and robust test accuracies on MNIST and CIFAR-10 are collected and presented as heatmaps in Figures 7. We observe that although the robust test accuracy decreases under multi-norm attacks, a phase transition phenomenon still persists, and increasing either the sample size $N$ or the SNR further reduces both clean and robust test error in a manner fully aligned with our theoretical results.

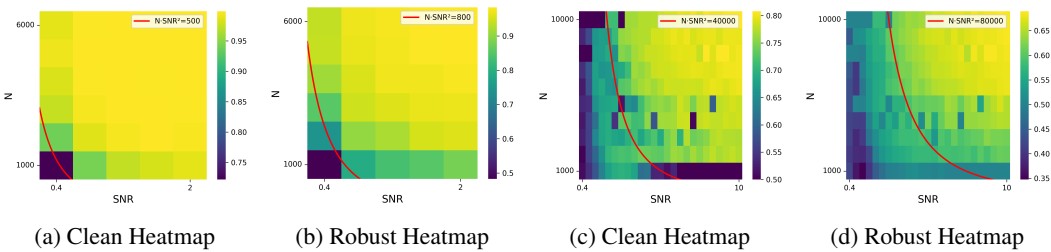

(a) Clean Heatmap      (b) Robust Heatmap      (c) Clean Heatmap      (d) Robust Heatmap

Figure 7: Clean and robust test accuracy under multi-norm attack adversarial training across various signal-to-noise ratios (SNR) and sample sizes ($N$). (a)&(b): results on MNIST data; (c)&(d): results on CIFAR-10 data.

### B.4 ADDITIONAL EXPERIMENTS ON REALISTIC VIT

In this section, we conduct real-world experiments on image classification benchmarks, including MNIST, CIFAR-10, and Tiny-ImageNet, using a realistic ViT architecture (Dosovitskiy et al., 2020). The results show that when our model is scaled from the simplified two-layer ViT to a full-fledged ViT model, our theoretical insights continue to hold.

**Experiments setting.** We adopt `google/vit-base-patch16-224-in21k` (Dosovitskiy et al., 2020) as the backbone model to extend our analysis to real-world scenarios. To align with our theoretical setting, we freeze all parameters except for the QKV matrices in the final attention layer. This setup effectively treats all preceding layers as a fixed feature-extraction encoder, whose output serves as the input to the last Transformer layer.

We conduct adversarial training on MNIST, CIFAR-10, and Tiny-ImageNet, using PGD as the threat model with a perturbation radius of $\frac{\|\boldsymbol{\mu}\|_2}{20}$ and 5 attack steps.

**Experiments results.** The clean and robust test accuracies of ViT-base on MNIST, CIFAR-10 and Tiny-ImageNet are collected and presented as heatmaps in Figures 8. We observe that when the model is scaled from the simplified two-layer ViT to a realistic ViT architecture, a phase transition phenomenon still persists, and increasing either the sample size $N$ or the SNR further reduces both clean and robust test error in a manner fully aligned with our theoretical results.

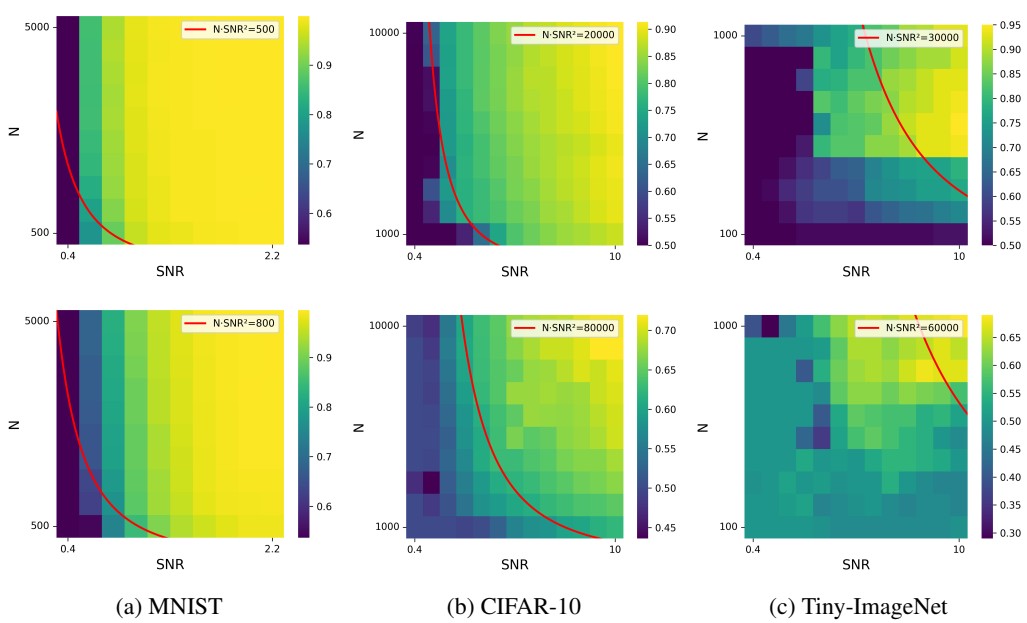

(a) MNIST         (b) CIFAR-10         (c) Tiny-ImageNet

Figure 8: Clean and robust test accuracy of ViT-base under adversarial training across various signal-to-noise ratios (SNR) and sample sizes (N). Top row: clean test accuracy. Bottom row: robust test accuracy.

## C EXTENSION TO MHA

We let the parameters be $\theta := \{(W_{Q,h}, W_{K,h}, W_{V,h})\}_{h=1}^{H}$, where $W_{Q,h}, W_{K,h} \in \mathbb{R}^{d \times d_h}$ and $W_{V,h} \in \mathbb{R}^{d \times d_v}$ for each $h \in [H]$. Here $H$ denotes the number of attention heads, which we treat as a fixed constant. Under this parameterization, the network can be written as:

$$f(\mathbf{X}, \theta) = \sum_{h=1}^{H} f_h(\mathbf{X}, \theta)$$

where,

$$f_h(\mathbf{X}, \theta) = \frac{1}{M} \sum_{l=1}^{M} \varphi(\mathbf{x}_l^\top \mathbf{W}_{Q,h} \mathbf{W}_{K,h}^\top \mathbf{X}^\top) \mathbf{X} \mathbf{W}_{V,h} \boldsymbol{w}_O.$$

The gradients in the multi-head attention module, $\frac{\partial f_h}{\partial W_{K,h}}, \frac{\partial f_h}{\partial W_{Q,h}}, \frac{\partial f_h}{\partial W_{V,h}}$, remain unchanged. However, the gradient of the loss with respect to the output of each single head, i.e., $\frac{\partial \ell}{\partial f_h}$, does change.

Intuitively, the model output increases by approximately an $H$-fold factor, which causes the scale of the loss $\ell'$ to decrease accordingly.

More concretely, following our analysis of the signal attention head, $\ell'^{(t)} = \frac{1}{M} \pm o(1)$ stay when $t \leq T_2 = \Theta\left(\frac{1}{\eta(\|\mu\|_2 + \tau)^2 \|w_O\|_2^2}\right)$. Thus, this implies $f_h^{(T_2)}(\mathbf{X}, \theta) = o(1)$. The $H$-fold increase in the multi-head model outputs does not alter this result, so the effect of the changes in $\frac{\partial \ell}{\partial f_h}$ can be ignored.

Therefore, under the MHA setting, the training dynamics of the model still follow those of the single-head attention case, and our conclusions remain unchanged.

# D   BASIC CALCULATION

## D.1   NOTION

## D.2   UPDATE RULES

We fix a universal perturbed input $\widetilde{\boldsymbol{X}} = \left[\widetilde{\boldsymbol{\mu}}, \widetilde{\boldsymbol{\xi}}_{n,2}, \ldots, \widetilde{\boldsymbol{\xi}}_{n,M}\right]$, where $\widetilde{\boldsymbol{\mu}}_+ \in B(\boldsymbol{\mu}_+, \tau)$, $\widetilde{\boldsymbol{\mu}}_- \in B(\boldsymbol{\mu}_-, \tau)$, and $\widetilde{\boldsymbol{\xi}}_{n,i} \in B(\boldsymbol{\xi}_{n,i}, \tau)$ are chosen once and remain fixed for all iterations $t$. These perturbations are universal and do not correspond to iteration-specific adversarial examples.

**Definition 4** (Scalarized V). *Let $\mathbf{W}_V^{(t)}$ be the V matrix of the ViT at the $t$-th iteration of adversarial training. Then there exist coefficients $\gamma_{V,+}^{(t)}, \gamma_{V,-}^{(t)}, \rho_{V,n,i}^{(t)}$ such that*

$$\widetilde{\boldsymbol{\mu}}_+^\top \mathbf{W}_V^{(t)} \boldsymbol{w}_O = \widetilde{\boldsymbol{\mu}}_+^\top \mathbf{W}_V^{(0)} \boldsymbol{w}_O + \gamma_{V,+}^{(t)} \|\boldsymbol{w}_O\|_2^2,$$

$$\widetilde{\boldsymbol{\mu}}_-^\top \mathbf{W}_V^{(t)} \boldsymbol{w}_O = \widetilde{\boldsymbol{\mu}}_-^\top \mathbf{W}_V^{(0)} \boldsymbol{w}_O + \gamma_{V,-}^{(t)} \|\boldsymbol{w}_O\|_2^2,$$

$$\widetilde{\boldsymbol{\xi}}_{n,i}^\top \mathbf{W}_V^{(t)} \boldsymbol{w}_O = \widetilde{\boldsymbol{\xi}}_{n,i}^\top \mathbf{W}_V^{(0)} \boldsymbol{w}_O + \rho_{V,n,i}^{(t)} \|\boldsymbol{w}_O\|_2^2$$

*for $i \in [M] \backslash \{1\}, n \in [N]$.*

*We further denote the $V_+^{(t)} := \widetilde{\boldsymbol{\mu}}_+^\top \mathbf{W}_V^{(t)} \boldsymbol{w}_O$, $V_-^{(t)} := \widetilde{\boldsymbol{\mu}}_-^\top \mathbf{W}_V^{(t)} \boldsymbol{w}_O$ and $V_{n,i}^{(t)} := \widetilde{\boldsymbol{\xi}}_{n,i}^\top \mathbf{W}_V^{(t)} \boldsymbol{w}_O$ with time-independent perturbations, and $\widetilde{V}_+^{(t)} := \widetilde{\boldsymbol{\mu}}_+^{(t)\top} \mathbf{W}_V^{(t)} \boldsymbol{w}_O$, $\widetilde{V}_-^{(t)} := \widetilde{\boldsymbol{\mu}}_-^{(t)\top} \mathbf{W}_V^{(t)} \boldsymbol{w}_O$ and $\widetilde{V}_{n,i}^{(t)} := \widetilde{\boldsymbol{\xi}}_{n,i}^{(t)\top} \mathbf{W}_V^{(t)} \boldsymbol{w}_O$ with time-dependent perturbations, where $\widetilde{\boldsymbol{\mu}}_+^{(t)}, \widetilde{\boldsymbol{\mu}}_-^{(t)}, \widetilde{\boldsymbol{\xi}}_{n,i}^{(t)}$ is the adversarial sample at $t$-th iteration. We refer to it as scalarized V.*

Similarly, we define the vectorized queries and keys as $\boldsymbol{q}^{(t)}, \boldsymbol{k}^{(t)}$ with time-independent perturbations, and as $\widetilde{\boldsymbol{q}}^{(t)}, \widetilde{\boldsymbol{k}}^{(t)}$ with time-dependent perturbations.

**Definition 5** (Vectorized Q & K). *Let $\mathbf{W}_Q^{(t)}$ and $\mathbf{W}_K^{(t)}$ be the QK matrices of the ViT at the $t$-th iteration of adversarial training. Then we define the vectorized Q and vectorized K as follows*

$$\boldsymbol{q}_+^{(t)} = \widetilde{\boldsymbol{\mu}}_+^\top \mathbf{W}_Q^{(t)}, \quad \boldsymbol{q}_-^{(t)} = \widetilde{\boldsymbol{\mu}}_-^\top \mathbf{W}_Q^{(t)}, \quad \boldsymbol{q}_{n,i}^{(t)} = \widetilde{\boldsymbol{\xi}}_{n,i}^\top \mathbf{W}_Q^{(t)},$$

$$\boldsymbol{k}_+^{(t)} = \widetilde{\boldsymbol{\mu}}_+^\top \mathbf{W}_K^{(t)}, \quad \boldsymbol{k}_-^{(t)} = \widetilde{\boldsymbol{\mu}}_-^\top \mathbf{W}_K^{(t)}, \quad \boldsymbol{k}_{n,i}^{(t)} = \widetilde{\boldsymbol{\xi}}_{n,i}^\top \mathbf{W}_K^{(t)},$$

$$\widetilde{\boldsymbol{q}}_+^{(t)} = \widetilde{\boldsymbol{\mu}}_+^{(t)\top} \mathbf{W}_Q^{(t)}, \quad \widetilde{\boldsymbol{q}}_-^{(t)} = \widetilde{\boldsymbol{\mu}}_-^{(t)\top} \mathbf{W}_Q^{(t)}, \quad \widetilde{\boldsymbol{q}}_{n,i}^{(t)} = \widetilde{\boldsymbol{\xi}}_{n,i}^{(t)\top} \mathbf{W}_Q^{(t)},$$

$$\widetilde{\boldsymbol{k}}_+^{(t)} = \widetilde{\boldsymbol{\mu}}_+^{(t)\top} \mathbf{W}_K^{(t)}, \quad \widetilde{\boldsymbol{k}}_-^{(t)} = \widetilde{\boldsymbol{\mu}}_-^{(t)\top} \mathbf{W}_K^{(t)}, \quad \widetilde{\boldsymbol{k}}_{n,i}^{(t)} = \widetilde{\boldsymbol{\xi}}_{n,i}^{(t)\top} \mathbf{W}_K^{(t)}$$

*for $i \in [M] \backslash \{1\}, n \in [N]$.*

Table 1: Notations

| Symbols | Definitions |
|---|---|
| $x_{n,i}$ | the i-th token in the n-th training sample 
 if $i \in [M]\setminus\{1\}$, $x_{n,i} = \boldsymbol{\xi}_{n,i}$. |
| $\varphi_{n,i}^{(t)}$ | the i-th row of attention for the n-th sample, i.e., $\varphi_{n,i}^{(t)} := \varphi(x_{n,i}^\top \mathbf{W}_Q^{(t)} \mathbf{W}_K^{(t)\top} X_n^\top)$ |
| $S_+, S_-$ | the training samples with +1 labels and -1 labels, 
 i.e., $S_+ := \{n \in [N] : y_n = 1\}$, $S_- := \{n \in [N] : y_n = -1\}$ |
| $\boldsymbol{q}_+^{(t)}, \boldsymbol{q}_-^{(t)}, \boldsymbol{q}_{n,i}^{(t)}$ | vectorized Q, defined as $\boldsymbol{q}_+^{(t)} = \widetilde{\boldsymbol{\mu}}_+^\top \mathbf{W}_Q^{(t)}, \boldsymbol{q}_-^{(t)} = \widetilde{\boldsymbol{\mu}}_-^\top \mathbf{W}_Q^{(t)}, \boldsymbol{q}_{n,i}^{(t)} = \widetilde{\boldsymbol{\xi}}_{n,i} \mathbf{W}_Q^{(t)}$ |
| $\boldsymbol{k}_+^{(t)}, \boldsymbol{k}_-^{(t)}, \boldsymbol{k}_{n,i}^{(t)}$ | vectorized K, defined as $\boldsymbol{k}_+^{(t)} = \widetilde{\boldsymbol{\mu}}_+^\top \mathbf{W}_K^{(t)}, \boldsymbol{k}_-^{(t)} = \widetilde{\boldsymbol{\mu}}_-^\top \mathbf{W}_K^{(t)}, \boldsymbol{k}_{n,i}^{(t)} = \widetilde{\boldsymbol{\xi}}_{n,i} \mathbf{W}_K^{(t)}$ |
| $V_+^{(t)}, V_-^{(t)}, V_{n,i}^{(t)}$ | scalarized V, defined as $V_+^{(t)} := \widetilde{\boldsymbol{\mu}}_+^\top \mathbf{W}_V^{(t)} \boldsymbol{w}_O, V_-^{(t)} := \widetilde{\boldsymbol{\mu}}_-^\top \mathbf{W}_V^{(t)} \boldsymbol{w}_O, V_{n,i}^{(t)} := \widetilde{\boldsymbol{\xi}}_{n,i}^\top \mathbf{W}_V^{(t)} \boldsymbol{w}_O$ |
| $\alpha_{\pm,\pm}^{(t)}, \alpha_{n,\pm,i}^{(t)}$ | linear combinations coefficients for the dynamics of $\boldsymbol{q}_+^{(t)}$ and $\boldsymbol{q}_-^{(t)}$, 
 i.e., $\boldsymbol{q}_\pm^{(t+1)} - \boldsymbol{q}_\pm^{(t)} = \alpha_{\pm,\pm}^{(t)} \boldsymbol{k}_\pm^{(t)} + \sum_{n \in S_\pm} \sum_{i=2}^M \alpha_{n,\pm,i}^{(t)} \boldsymbol{k}_{n,i}^{(t)}$ |
| $\alpha_{n,i,\pm}^{(t)}, \alpha_{n,i,n',i'}^{(t)}$ | linear combinations coefficients for the dynamics of $\boldsymbol{q}_{n,i}^{(t)}$, 
 i.e., $\boldsymbol{q}_{n,i}^{(t+1)} - \boldsymbol{q}_{n,i}^{(t)} = \alpha_{n,i,+}^{(t)} \boldsymbol{k}_+^{(t)} + \alpha_{n,i,-}^{(t)} \boldsymbol{k}_-^{(t)} + \sum_{n'=1}^N \sum_{i'=2}^M \alpha_{n,i,n',i'}^{(t)} \boldsymbol{k}_{n',i'}^{(t)}$ |
| $\beta_{\pm,\pm}^{(t)}, \beta_{n,\pm,i}^{(t)}$ | linear combinations coefficients for the dynamics of $\boldsymbol{k}_+^{(t)}$ and $\boldsymbol{k}_-^{(t)}$, 
 i.e., $\boldsymbol{k}_\pm^{(t+1)} - \boldsymbol{k}_\pm^{(t)} = \beta_{\pm,\pm}^{(t)} \boldsymbol{q}_\pm^{(t)} + \sum_{n \in S_\pm} \sum_{i=2}^M \beta_{n,\pm,i}^{(t)} \boldsymbol{q}_{n,i}^{(t)}$ |
| $\beta_{n,i,\pm}^{(t)}, \beta_{n,i,n',i'}^{(t)}$ | linear combinations coefficients for the dynamics of $\boldsymbol{k}_{n,i}^{(t)}$, 
 i.e., $\boldsymbol{k}_{n,i}^{(t+1)} - \boldsymbol{k}_{n,i}^{(t)} = \beta_{n,i,+}^{(t)} \boldsymbol{q}_+^{(t)} + \beta_{n,i,-}^{(t)} \boldsymbol{q}_-^{(t)} + \sum_{n'=1}^N \sum_{i'=2}^M \beta_{n,i,n',i'}^{(t)} \boldsymbol{q}_{n',i'}^{(t)}$ |
| $\mathrm{softmax}(\langle \boldsymbol{q}_\pm^{(t)}, \boldsymbol{k}_\pm^{(t)} \rangle)$ | a general references to $\frac{\exp(\langle \boldsymbol{q}_\pm^{(t)}, \boldsymbol{k}_\pm^{(t)} \rangle)}{\exp(\langle \boldsymbol{q}_\pm^{(t)}, \boldsymbol{k}_\pm^{(t)} \rangle) + \sum_{k=2}^M \exp(\langle \boldsymbol{q}_\pm^{(t)}, \boldsymbol{k}_{n,k}^{(t)} \rangle)}$ for $n \in S_+, i \in [M]\setminus\{1\}$, 
 and $\frac{\exp(\langle \boldsymbol{q}_\pm^{(t)}, \boldsymbol{k}_\pm^{(t)} \rangle)}{\exp(\langle \boldsymbol{q}_\pm^{(t)}, \boldsymbol{k}_\pm^{(t)} \rangle) + \sum_{k=2}^M \exp(\langle \boldsymbol{q}_\pm^{(t)}, \boldsymbol{k}_{n,k}^{(t)} \rangle)}$ for $n \in S_-, i \in [M]\setminus\{1\}$ |
| $\mathrm{softmax}(\langle \boldsymbol{q}_\pm^{(t)}, \boldsymbol{k}_{n,j}^{(t)} \rangle)$ | a general references to $\frac{\exp(\langle \boldsymbol{q}_\pm^{(t)}, \boldsymbol{k}_{n,j}^{(t)} \rangle)}{\exp(\langle \boldsymbol{q}_\pm^{(t)}, \boldsymbol{k}_\pm^{(t)} \rangle) + \sum_{k=2}^M \exp(\langle \boldsymbol{q}_\pm^{(t)}, \boldsymbol{k}_{n,k}^{(t)} \rangle)}$ for $n \in S_+, i, j \in [M]\setminus\{1\}$, 
 and $\frac{\exp(\langle \boldsymbol{q}_\pm^{(t)}, \boldsymbol{k}_{n,j}^{(t)} \rangle)}{\exp(\langle \boldsymbol{q}_\pm^{(t)}, \boldsymbol{k}_\pm^{(t)} \rangle) + \sum_{k=2}^M \exp(\langle \boldsymbol{q}_\pm^{(t)}, \boldsymbol{k}_{n,k}^{(t)} \rangle)}$ for $n \in S_-, i, j \in [M]\setminus\{1\}$ |
| $\mathrm{softmax}(\langle \boldsymbol{q}_{n,i}^{(t)}, \boldsymbol{k}_{n,j}^{(t)} \rangle)$ | a general references to $\frac{\exp(\langle \boldsymbol{q}_{n,i}^{(t)}, \boldsymbol{k}_{n,j}^{(t)} \rangle)}{\exp(\langle \boldsymbol{q}_{n,i}^{(t)}, \boldsymbol{k}_{n,+}^{(t)} \rangle) + \sum_{k=2}^M \exp(\langle \boldsymbol{q}_{n,i}^{(t)}, \boldsymbol{k}_{n,k}^{(t)} \rangle)}$ for $n \in S_+, i, j \in [M]\setminus\{1\}$, 
 and $\frac{\exp(\langle \boldsymbol{q}_{n,i}^{(t)}, \boldsymbol{k}_{n,j}^{(t)} \rangle)}{\exp(\langle \boldsymbol{q}_{n,i}^{(t)}, \boldsymbol{k}_{n,-}^{(t)} \rangle) + \sum_{k=2}^M \exp(\langle \boldsymbol{q}_{n,i}^{(t)}, \boldsymbol{k}_{n,k}^{(t)} \rangle)}$ for $n \in S_-, i, j \in [M]\setminus\{1\}$ |
| $\Lambda_{n,\pm,j}^{(t)}, \Lambda_{n,i,\pm,j}^{(t)}$ | $\Lambda_{n,\pm,j}^{(t)} := \langle \boldsymbol{q}_\pm^{(t)}, \boldsymbol{k}_\pm^{(t)} \rangle - \langle \boldsymbol{q}_\pm^{(t)}, \boldsymbol{k}_{n,j}^{(t)} \rangle, \Lambda_{n,i,\pm,j}^{(t)} := \langle \boldsymbol{q}_{n,i}^{(t)}, \boldsymbol{k}_\pm^{(t)} \rangle - \langle \boldsymbol{q}_{n,i}^{(t)}, \boldsymbol{k}_{n,j}^{(t)} \rangle$ |

**Definition 6** (Gradient Decomposition). *There exist coefficients $\alpha_{+,+}^{(t)}$, $\alpha_{n,+,i}^{(t)}$, $\alpha_{-,-}^{(t)}$, $\alpha_{n,-,i}^{(t)}$, $\alpha_{n,i,+}^{(t)}$, $\alpha_{n,i,-}^{(t)}$, $\alpha_{n,i,n',i'}^{(t)}$, $\beta_{+,+}^{(t)}$, $\beta_{n,+,i}^{(t)}$, $\beta_{-,-}^{(t)}$, $\beta_{n,-,i}^{(t)}$, $\beta_{n,i,+}^{(t)}$, $\beta_{n,i,-}^{(t)}$, $\beta_{n,i,n',i'}^{(t)}$ such that*

$$\Delta \boldsymbol{q}_+^{(t)} := \boldsymbol{q}_+^{(t+1)} - \boldsymbol{q}_+^{(t)} = \alpha_{+,+}^{(t)} \boldsymbol{k}_+^{(t)} + \sum_{n \in S_+} \sum_{i=2}^M \alpha_{n,+,i}^{(t)} \boldsymbol{k}_{n,i}^{(t)},$$

$$\Delta \boldsymbol{q}_-^{(t)} := \boldsymbol{q}_-^{(t+1)} - \boldsymbol{q}_-^{(t)} = \alpha_{-,-}^{(t)} \boldsymbol{k}_-^{(t)} + \sum_{n \in S_-} \sum_{i=2}^M \alpha_{n,-,i}^{(t)} \boldsymbol{k}_{n,i}^{(t)},$$

$$\Delta \boldsymbol{q}_{n,i}^{(t)} := \boldsymbol{q}_{n,i}^{(t+1)} - \boldsymbol{q}_{n,i}^{(t)} = \alpha_{n,i,+}^{(t)} \boldsymbol{k}_+^{(t)} + \alpha_{n,i,-}^{(t)} \boldsymbol{k}_-^{(t)} + \sum_{n'=1}^N \sum_{i'=2}^M \alpha_{n,i,n',i'}^{(t)} \boldsymbol{k}_{n',i'}^{(t)},$$

$$\Delta \boldsymbol{k}_+^{(t)} := \boldsymbol{k}_+^{(t+1)} - \boldsymbol{k}_+^{(t)} = \beta_{+,+}^{(t)} \boldsymbol{q}_+^{(t)} + \sum_{n \in S_+} \sum_{i=2}^M \beta_{n,+,i}^{(t)} \boldsymbol{q}_{n,i}^{(t)},$$

$$\Delta \boldsymbol{k}_-^{(t)} := \boldsymbol{k}_-^{(t+1)} - \boldsymbol{k}_-^{(t)} = \beta_{-,-}^{(t)} \boldsymbol{q}_-^{(t)} + \sum_{n \in S_-} \sum_{i=2}^M \beta_{n,-,i}^{(t)} \boldsymbol{q}_{n,i}^{(t)},$$

$$\Delta \boldsymbol{k}_{n,i}^{(t)} := \boldsymbol{k}_{n,i}^{(t+1)} - \boldsymbol{k}_{n,i}^{(t)} = \beta_{n,i,+}^{(t)} \boldsymbol{q}_{+}^{(t)} + \beta_{n,i,-}^{(t)} \boldsymbol{q}_{-}^{(t)} + \sum_{n'=1}^{N} \sum_{i'=2}^{M} \beta_{n,i,n',i'}^{(t)} \boldsymbol{q}_{n',i'}^{(t)}.$$

for $i, i' \in [M] \setminus \{1\}$ and $n, n' \in [N]$.

**Lemma 8** (Update Rule for V). *The coefficients $\gamma_{V,+}^{(t)}, \gamma_{V,-}^{(t)}, \rho_{V,n,i}^{(t)}$ defined in Definition 1 satisfy the following iterative equations:*

$$\gamma_{V,+}^{(t+1)} = \gamma_{V,+}^{(t)} - \frac{\eta \langle \widetilde{\boldsymbol{\mu}}_{+}, \widetilde{\boldsymbol{\mu}}_{+}^{(t)} \rangle}{NM} \sum_{n \in S_+} \widetilde{\ell}_n^{\prime(t)} \left( \frac{\exp(\langle \widetilde{\mathbf{q}}_{+}^{(t)}, \widetilde{\mathbf{k}}_{+}^{(t)} \rangle)}{\exp(\langle \widetilde{\mathbf{q}}_{+}^{(t)}, \widetilde{\mathbf{k}}_{+}^{(t)} \rangle) + \sum_{k=2}^{M} \exp(\langle \widetilde{\mathbf{q}}_{+}^{(t)}, \widetilde{\mathbf{k}}_{n,k}^{(t)} \rangle)} \right.$$

$$\left. + \sum_{j=2}^{M} \frac{\exp(\langle \widetilde{\mathbf{q}}_{n,j}^{(t)}, \widetilde{\mathbf{k}}_{+}^{(t)} \rangle)}{\exp(\langle \widetilde{\mathbf{q}}_{n,j}^{(t)}, \widetilde{\mathbf{k}}_{+}^{(t)} \rangle) + \sum_{k=2}^{M} \exp(\langle \widetilde{\mathbf{q}}_{n,j}^{(t)}, \widetilde{\mathbf{k}}_{n,k}^{(t)} \rangle)} \right)$$

$$+ \sum_{n \in S_+} \widetilde{\ell}_n^{\prime(t)} \sum_{i=2}^{M} \frac{-\eta \langle \widetilde{\boldsymbol{\mu}}_{+}, \widetilde{\boldsymbol{\xi}}_{n,i}^{(t)} \rangle}{NM} \left( \frac{\exp(\langle \widetilde{\mathbf{q}}_{+}^{(t)}, \widetilde{\mathbf{k}}_{n,i}^{(t)} \rangle)}{\exp(\langle \widetilde{\mathbf{q}}_{+}^{(t)}, \widetilde{\mathbf{k}}_{+}^{(t)} \rangle) + \sum_{k=2}^{M} \exp(\langle \widetilde{\mathbf{q}}_{+}^{(t)}, \widetilde{\mathbf{k}}_{n,k}^{(t)} \rangle)} \right.$$

$$\left. + \sum_{j=2}^{M} \frac{\exp(\langle \widetilde{\mathbf{q}}_{n,j}^{(t)}, \widetilde{\mathbf{k}}_{n,i}^{(t)} \rangle)}{\exp(\langle \widetilde{\mathbf{q}}_{n,j}^{(t)}, \widetilde{\mathbf{k}}_{+}^{(t)} \rangle) + \sum_{k=2}^{M} \exp(\langle \widetilde{\mathbf{q}}_{n,j}^{(t)}, \widetilde{\mathbf{k}}_{n,k}^{(t)} \rangle)} \right)$$

$$\gamma_{V,-}^{(t+1)} = \gamma_{V,-}^{(t)} - \frac{\eta \langle \widetilde{\boldsymbol{\mu}}_{-}, \widetilde{\boldsymbol{\mu}}_{-}^{(t)} \rangle}{NM} \sum_{n \in S_-} \widetilde{\ell}_n^{\prime(t)} \left( \frac{\exp(\langle \widetilde{\mathbf{q}}_{-}^{(t)}, \widetilde{\mathbf{k}}_{-}^{(t)} \rangle)}{\exp(\langle \widetilde{\mathbf{q}}_{-}^{(t)}, \widetilde{\mathbf{k}}_{-}^{(t)} \rangle) + \sum_{k=2}^{M} \exp(\langle \widetilde{\mathbf{q}}_{-}^{(t)}, \widetilde{\mathbf{k}}_{n,k}^{(t)} \rangle)} \right.$$

$$\left. + \sum_{j=2}^{M} \frac{\exp(\langle \widetilde{\mathbf{q}}_{n,j}^{(t)}, \widetilde{\mathbf{k}}_{-}^{(t)} \rangle)}{\exp(\langle \widetilde{\mathbf{q}}_{n,j}^{(t)}, \widetilde{\mathbf{k}}_{-}^{(t)} \rangle) + \sum_{k=2}^{M} \exp(\langle \widetilde{\mathbf{q}}_{n,j}^{(t)}, \widetilde{\mathbf{k}}_{n,k}^{(t)} \rangle)} \right)$$

$$+ \sum_{n \in S_+} \widetilde{\ell}_n^{\prime(t)} \sum_{i=2}^{M} \frac{-\eta \langle \widetilde{\boldsymbol{\mu}}_{-}, \widetilde{\boldsymbol{\xi}}_{n,i}^{(t)} \rangle}{NM} \left( \frac{\exp(\langle \widetilde{\mathbf{q}}_{-}^{(t)}, \widetilde{\mathbf{k}}_{n,i}^{(t)} \rangle)}{\exp(\langle \widetilde{\mathbf{q}}_{-}^{(t)}, \widetilde{\mathbf{k}}_{-}^{(t)} \rangle) + \sum_{k=2}^{M} \exp(\langle \widetilde{\mathbf{q}}_{-}^{(t)}, \widetilde{\mathbf{k}}_{n,k}^{(t)} \rangle)} \right.$$

$$\left. + \sum_{j=2}^{M} \frac{\exp(\langle \widetilde{\mathbf{q}}_{n,j}^{(t)}, \widetilde{\mathbf{k}}_{n,i}^{(t)} \rangle)}{\exp(\langle \widetilde{\mathbf{q}}_{n,j}^{(t)}, \widetilde{\mathbf{k}}_{-}^{(t)} \rangle) + \sum_{k=2}^{M} \exp(\langle \widetilde{\mathbf{q}}_{n,j}^{(t)}, \widetilde{\mathbf{k}}_{n,k}^{(t)} \rangle)} \right),$$

$$\rho_{V,n,i}^{(t+1)} = \rho_{V,n,i}^{(t)} - \frac{\eta}{NM} \sum_{n' \in S_+} \widetilde{\ell}_{n'}^{\prime(t)} \left( \left( \langle \widetilde{\boldsymbol{\xi}}_{n,i}, \widetilde{\boldsymbol{\mu}}_{+}^{(t)} \rangle \frac{\exp(\langle \widetilde{\mathbf{q}}_{+}^{(t)}, \widetilde{\mathbf{k}}_{+}^{(t)} \rangle)}{\exp(\langle \widetilde{\mathbf{q}}_{+}^{(t)}, \widetilde{\mathbf{k}}_{+}^{(t)} \rangle) + \sum_{k=2}^{M} \exp(\langle \widetilde{\mathbf{q}}_{+}^{(t)}, \widetilde{\mathbf{k}}_{n,k}^{(t)} \rangle)} \right. \right.$$

$$\left. + \sum_{j=2}^{M} \langle \widetilde{\boldsymbol{\xi}}_{n,i}, \widetilde{\boldsymbol{\mu}}_{+}^{(t)} \rangle \frac{\exp(\langle \widetilde{\mathbf{q}}_{n,j}^{(t)}, \widetilde{\mathbf{k}}_{+}^{(t)} \rangle)}{\exp(\langle \widetilde{\mathbf{q}}_{n,j}^{(t)}, \widetilde{\mathbf{k}}_{+}^{(t)} \rangle) + \sum_{k=2}^{M} \exp(\langle \widetilde{\mathbf{q}}_{n,j}^{(t)}, \widetilde{\mathbf{k}}_{n,k}^{(t)} \rangle)} \right)$$

$$+ \sum_{i=2}^{M} \left( \langle \widetilde{\boldsymbol{\xi}}_{n,i}, \widetilde{\boldsymbol{\xi}}_{n',i'}^{(t)} \rangle \frac{\exp(\langle \widetilde{\mathbf{q}}_{n',i'}^{(t)}, \widetilde{\mathbf{k}}_{n',i'}^{(t)} \rangle)}{\exp(\langle \widetilde{\mathbf{q}}_{n',i'}^{(t)}, \widetilde{\mathbf{k}}_{n',i'}^{(t)} \rangle) + \sum_{k=2}^{M} \exp(\langle \widetilde{\mathbf{q}}_{n',i'}^{(t)}, \widetilde{\mathbf{k}}_{n',k}^{(t)} \rangle)} \right.$$

$$\left. \left. \left. + \sum_{j=2}^{M} \langle \widetilde{\boldsymbol{\xi}}_{n,i}, \widetilde{\boldsymbol{\xi}}_{n',i'}^{(t)} \rangle \frac{\exp(\langle \widetilde{\mathbf{q}}_{n',j}^{(t)}, \widetilde{\mathbf{k}}_{n',i'}^{(t)} \rangle)}{\exp(\langle \widetilde{\mathbf{q}}_{n',j}^{(t)}, \widetilde{\mathbf{k}}_{n',i'}^{(t)} \rangle) + \sum_{k=2}^{M} \exp(\langle \widetilde{\mathbf{q}}_{n',j}^{(t)}, \widetilde{\mathbf{k}}_{n',k}^{(t)} \rangle)} \right) \right) \right)$$

$$- \frac{\eta}{NM} \sum_{n' \in S_-} (\cdot)$$

for $i \in [M] \setminus \{1\}, n \in [N]$.

*Proof.* The gradient of $\mathbf{W}_V$ can be obtained using the chain rule as follows

$$\nabla_{\mathbf{W}_V} L_S(\theta) = \frac{1}{N} \sum_{n=1}^{N} y_n \ell'(y_n f(\mathbf{X}_n, \theta)) \nabla_{\mathbf{W}_V} f(\mathbf{X}_n, \theta)$$

$$= \frac{1}{NM} \sum_{n=1}^{N} y_n \ell_n'(\theta) \left[ \boldsymbol{w}_O \sum_{l=1}^{M} \varphi(\mathbf{x}_{n,l} \mathbf{W}_Q \mathbf{W}_K^\top (\mathbf{X}_n)^\top) \mathbf{X}_n \right]^\top$$

Base on above, we have

$$\mathbf{x}^\top \nabla_{\mathbf{w}_V} \widetilde{L}_S(\theta) \boldsymbol{w}_O = \frac{1}{NM} \sum_{n \in S_+} \widetilde{\ell}_n'(\theta) \Big( \langle \mathbf{x}, \widetilde{\boldsymbol{\mu}}_+ \rangle \frac{\exp(\widetilde{\boldsymbol{\mu}}_+^\top \mathbf{W}_Q \mathbf{W}_K^\top \widetilde{\boldsymbol{\mu}}_+)}{\exp(\widetilde{\boldsymbol{\mu}}_+^\top \mathbf{W}_Q \mathbf{W}_K^\top \widetilde{\boldsymbol{\mu}}_+) + \sum_{k=2}^{M} \exp(\widetilde{\boldsymbol{\mu}}_+^\top \mathbf{W}_Q \mathbf{W}_K^\top \widetilde{\boldsymbol{\xi}}_{n,k})}$$

$$+ \sum_{j=2}^{M} \langle \mathbf{x}, \widetilde{\boldsymbol{\mu}}_+ \rangle \frac{\exp(\widetilde{\boldsymbol{\xi}}_{n,j}^\top \mathbf{W}_Q \mathbf{W}_K^\top \widetilde{\boldsymbol{\mu}}_+)}{\exp(\widetilde{\boldsymbol{\xi}}_{n,j}^\top \mathbf{W}_Q \mathbf{W}_K^\top \widetilde{\boldsymbol{\mu}}_+) + \sum_{k=2}^{M} \exp(\widetilde{\boldsymbol{\xi}}_{n,j}^\top \mathbf{W}_Q \mathbf{W}_K^\top \widetilde{\boldsymbol{\xi}}_{n,k})} \Big)$$

$$+ \sum_{i=2}^{M} \big( \langle \mathbf{x}, \widetilde{\boldsymbol{\xi}}_{n,i} \rangle \frac{\exp(\widetilde{\boldsymbol{\mu}}_+^\top \mathbf{W}_Q \mathbf{W}_K^\top \widetilde{\boldsymbol{\xi}}_{n,i})}{\exp(\widetilde{\boldsymbol{\mu}}_+^\top \mathbf{W}_Q \mathbf{W}_K^\top \widetilde{\boldsymbol{\mu}}_+) + \sum_{k=2}^{M} \exp(\widetilde{\boldsymbol{\mu}}_+^\top \mathbf{W}_Q \mathbf{W}_K^\top \widetilde{\boldsymbol{\xi}}_{n,k})}$$

$$+ \sum_{j=2}^{M} \langle \mathbf{x}, \widetilde{\boldsymbol{\xi}}_{n,i} \rangle \frac{\exp(\widetilde{\boldsymbol{\xi}}_{n,j}^\top \mathbf{W}_Q \mathbf{W}_K^\top \widetilde{\boldsymbol{\xi}}_{n,i})}{\exp(\widetilde{\boldsymbol{\xi}}_{n,j}^\top \mathbf{W}_Q \mathbf{W}_K^\top \widetilde{\boldsymbol{\mu}}_+) + \sum_{k=2}^{M} \exp(\widetilde{\boldsymbol{\xi}}_{n,j}^\top \mathbf{W}_Q \mathbf{W}_K^\top \widetilde{\boldsymbol{\xi}}_{n,k})} \big) \Big) \|\boldsymbol{w}_O\|_2^2$$

$$+ \frac{1}{NM} \sum_{n \in S_-} (\cdot)$$

where the second equality we expand $X_n$ into vectors and make inner products with $x$, the third equality we materializing all the $x_{n,i}$ (e.g., $x_{n,1} = \boldsymbol{\mu}_+$ for $n \in S_+$). Note the orthogonality between $\boldsymbol{\mu}$ and $\boldsymbol{\xi}_{n,i}$, we can remove many of the terms in this equation. For any $x = \widetilde{\boldsymbol{\mu}}_+' \in B(\boldsymbol{\mu}_+, \tau)$, we have

$$\widetilde{\boldsymbol{\mu}}_+^{'\top} \nabla_{\mathbf{w}_V} \widetilde{L}_S(\theta) \boldsymbol{w}_O = \frac{1}{NM} \sum_{n \in S_+} \widetilde{\ell}_n'(\theta) \Big( \big( \langle \widetilde{\boldsymbol{\mu}}_+', \widetilde{\boldsymbol{\mu}}_+ \rangle \frac{\exp(\widetilde{\boldsymbol{\mu}}_+^\top \mathbf{W}_Q \mathbf{W}_K^\top \widetilde{\boldsymbol{\mu}}_+)}{\exp(\widetilde{\boldsymbol{\mu}}_+^\top \mathbf{W}_Q \mathbf{W}_K^\top \widetilde{\boldsymbol{\mu}}_+) + \sum_{k=2}^{M} \exp(\widetilde{\boldsymbol{\mu}}_+^\top \mathbf{W}_Q \mathbf{W}_K^\top \widetilde{\boldsymbol{\xi}}_{n,k})}$$

$$+ \sum_{j=2}^{M} \langle \widetilde{\boldsymbol{\mu}}_+', \widetilde{\boldsymbol{\mu}}_+ \rangle \frac{\exp(\widetilde{\boldsymbol{\xi}}_{n,j}^\top \mathbf{W}_Q \mathbf{W}_K^\top \widetilde{\boldsymbol{\mu}}_+)}{\exp(\widetilde{\boldsymbol{\xi}}_{n,j}^\top \mathbf{W}_Q \mathbf{W}_K^\top \widetilde{\boldsymbol{\mu}}_+) + \sum_{k=2}^{M} \exp(\widetilde{\boldsymbol{\xi}}_{n,j}^\top \mathbf{W}_Q \mathbf{W}_K^\top \widetilde{\boldsymbol{\xi}}_{n,k})} \big)$$

$$+ \sum_{i=2}^{M} \big( \langle \widetilde{\boldsymbol{\mu}}_+', \widetilde{\boldsymbol{\xi}}_{n,i} \rangle \frac{\exp(\widetilde{\boldsymbol{\mu}}_+^\top \mathbf{W}_Q \mathbf{W}_K^\top \widetilde{\boldsymbol{\xi}}_{n,i})}{\exp(\widetilde{\boldsymbol{\mu}}_+^\top \mathbf{W}_Q \mathbf{W}_K^\top \widetilde{\boldsymbol{\mu}}_+) + \sum_{k=2}^{M} \exp(\widetilde{\boldsymbol{\mu}}_+^\top \mathbf{W}_Q \mathbf{W}_K^\top \widetilde{\boldsymbol{\xi}}_{n,k})}$$

$$+ \sum_{j=2}^{M} \langle \widetilde{\boldsymbol{\mu}}_+', \widetilde{\boldsymbol{\xi}}_{n,i} \rangle \frac{\exp(\widetilde{\boldsymbol{\xi}}_{n,j}^\top \mathbf{W}_Q \mathbf{W}_K^\top \widetilde{\boldsymbol{\xi}}_{n,i})}{\exp(\widetilde{\boldsymbol{\xi}}_{n,j}^\top \mathbf{W}_Q \mathbf{W}_K^\top \widetilde{\boldsymbol{\mu}}_+) + \sum_{k=2}^{M} \exp(\widetilde{\boldsymbol{\xi}}_{n,j}^\top \mathbf{W}_Q \mathbf{W}_K^\top \widetilde{\boldsymbol{\xi}}_{n,k})} \big) \Big) \|\boldsymbol{w}_O\|_2^2$$

Then we have

$$\widetilde{\boldsymbol{\mu}}_+^\top \mathbf{W}_V^{(t+1)} \boldsymbol{w}_O - \widetilde{\boldsymbol{\mu}}_+^\top \mathbf{W}_V^{(t)} \boldsymbol{w}_O = \widetilde{\boldsymbol{\mu}}_+^\top (-\eta \nabla_{\mathbf{w}_V} \widetilde{L}_S(\theta(t))) \boldsymbol{w}_O$$

$$= -\frac{\eta \langle \widetilde{\boldsymbol{\mu}}_+, \widetilde{\boldsymbol{\mu}}_+^{(t)} \rangle}{NM} \sum_{n \in S_+} \widetilde{\ell}_n^{'(t)} \left( \frac{\exp(\langle \widetilde{\mathbf{q}}_+^{(t)}, \widetilde{\mathbf{k}}_+^{(t)} \rangle)}{\exp(\langle \widetilde{\mathbf{q}}_+^{(t)}, \widetilde{\mathbf{k}}_+^{(t)} \rangle) + \sum_{k=2}^{M} \exp(\langle \widetilde{\mathbf{q}}_+^{(t)}, \widetilde{\mathbf{k}}_{n,k}^{(t)} \rangle)} \right.$$

$$+ \sum_{j=2}^{M} \frac{\exp(\langle \widetilde{\mathbf{q}}_{n,j}^{(t)}, \widetilde{\mathbf{k}}_+^{(t)} \rangle)}{\exp(\langle \widetilde{\mathbf{q}}_{n,j}^{(t)}, \widetilde{\mathbf{k}}_+^{(t)} \rangle) + \sum_{k=2}^{M} \exp(\langle \widetilde{\mathbf{q}}_{n,j}^{(t)}, \widetilde{\mathbf{k}}_{n,k}^{(t)} \rangle)} \right) \|\boldsymbol{w}_O\|_2^2$$

$$+ \sum_{n \in S_+} \widetilde{\ell}_n^{'(t)} \sum_{i=2}^{M} \frac{-\eta \langle \widetilde{\boldsymbol{\mu}}_+, \widetilde{\boldsymbol{\xi}}_{n,i}^{(t)} \rangle}{NM} \left( \frac{\exp(\langle \widetilde{\mathbf{q}}_+^{(t)}, \widetilde{\mathbf{k}}_{n,i}^{(t)} \rangle)}{\exp(\langle \widetilde{\mathbf{q}}_+^{(t)}, \widetilde{\mathbf{k}}_+^{(t)} \rangle) + \sum_{k=2}^{M} \exp(\langle \widetilde{\mathbf{q}}_+^{(t)}, \widetilde{\mathbf{k}}_{n,k}^{(t)} \rangle)} \right.$$

$$+ \sum_{j=2}^{M} \frac{\exp(\langle \widetilde{\mathbf{q}}_{n,j}^{(t)}, \widetilde{\mathbf{k}}_{n,i}^{(t)} \rangle)}{\exp(\langle \widetilde{\mathbf{q}}_{n,j}^{(t)}, \widetilde{\mathbf{k}}_+^{(t)} \rangle) + \sum_{k=2}^{M} \exp(\langle \widetilde{\mathbf{q}}_{n,j}^{(t)}, \widetilde{\mathbf{k}}_{n,k}^{(t)} \rangle)} \right) \|\boldsymbol{w}_O\|_2^2$$

Dividing by $\|\boldsymbol{w}_O\|_2^2$ we get

$$
\gamma_{V,+}^{(t+1)} = \gamma_{V,+}^{(t)} - \frac{\eta\langle\widetilde{\boldsymbol{\mu}}_+, \widetilde{\boldsymbol{\mu}}_+^{(t)}\rangle}{NM} \sum_{n\in S_+} \widetilde{\ell}_n'^{(t)} \left( \frac{\exp(\langle\widetilde{\mathbf{q}}_+^{(t)}, \widetilde{\mathbf{k}}_+^{(t)}\rangle)}{\exp(\langle\widetilde{\mathbf{q}}_+^{(t)}, \widetilde{\mathbf{k}}_+^{(t)}\rangle) + \sum_{k=2}^M \exp(\langle\widetilde{\mathbf{q}}_+^{(t)}, \widetilde{\mathbf{k}}_{n,k}^{(t)}\rangle)} \right.
$$

$$
\left. + \sum_{j=2}^M \frac{\exp(\langle\widetilde{\mathbf{q}}_{n,j}^{(t)}, \widetilde{\mathbf{k}}_+^{(t)}\rangle)}{\exp(\langle\widetilde{\mathbf{q}}_{n,j}^{(t)}, \widetilde{\mathbf{k}}_+^{(t)}\rangle) + \sum_{k=2}^M \exp(\langle\widetilde{\mathbf{q}}_{n,j}^{(t)}, \widetilde{\mathbf{k}}_{n,k}^{(t)}\rangle)} \right)
$$

$$
+ \sum_{n\in S_+} \widetilde{\ell}_n'^{(t)} \sum_{i=2}^M \frac{-\eta\langle\widetilde{\boldsymbol{\mu}}_+, \widetilde{\boldsymbol{\xi}}_{n,i}^{(t)}\rangle}{NM} \left( \frac{\exp(\langle\widetilde{\mathbf{q}}_+^{(t)}, \widetilde{\mathbf{k}}_{n,i}^{(t)}\rangle)}{\exp(\langle\widetilde{\mathbf{q}}_+^{(t)}, \widetilde{\mathbf{k}}_+^{(t)}\rangle) + \sum_{k=2}^M \exp(\langle\widetilde{\mathbf{q}}_+^{(t)}, \widetilde{\mathbf{k}}_{n,k}^{(t)}\rangle)} \right.
$$

$$
\left. + \sum_{j=2}^M \frac{\exp(\langle\widetilde{\mathbf{q}}_{n,j}^{(t)}, \widetilde{\mathbf{k}}_{n,i}^{(t)}\rangle)}{\exp(\langle\widetilde{\mathbf{q}}_{n,j}^{(t)}, \widetilde{\mathbf{k}}_+^{(t)}\rangle) + \sum_{k=2}^M \exp(\langle\widetilde{\mathbf{q}}_{n,j}^{(t)}, \widetilde{\mathbf{k}}_{n,k}^{(t)}\rangle)} \right)
$$

This proves the update rule for $\gamma_{V,+}^{(t)}$. The proof for $\gamma_{V,-}^{(t)}$ and $\rho_{V,n,i}^{(t)}$ is similar to it. $\qquad\square$

**Lemma 9** (Update Rule for QK, Lemma B.3 in Jiang et al. (2024)). *The dynamics of* $\boldsymbol{x}^\top \boldsymbol{W}_Q \boldsymbol{W}_K \boldsymbol{x}$ *can be characterized as follows:*

$$
\langle\boldsymbol{q}_+^{(t+1)}, \boldsymbol{k}_+^{(t+1)}\rangle - \langle\boldsymbol{q}_+^{(t)}, \boldsymbol{k}_+^{(t)}\rangle
$$

$$
= \alpha_+^{(t)}\|\boldsymbol{k}_+^{(t)}\|_2^2 + \sum_{n\in S_+}\sum_{i=2}^M \alpha_{n,+,i}^{(t)}\langle\boldsymbol{k}_+^{(t)}, \boldsymbol{k}_{n,i}^{(t)}\rangle
$$

$$
+ \beta_+^{(t)}\|\boldsymbol{q}_+^{(t)}\|_2^2 + \sum_{n\in S_+}\sum_{i=2}^M \beta_{n,+,i}^{(t)}\langle\boldsymbol{q}_+^{(t)}, \boldsymbol{q}_{n,i}^{(t)}\rangle \tag{2}
$$

$$
+ \left( \alpha_+^{(t)}\boldsymbol{k}_+^{(t)} + \sum_{n\in S_+}\sum_{i=2}^M \alpha_{n,+,i}^{(t)}\boldsymbol{k}_{n,i}^{(t)} \right)
$$

$$
\cdot \left( \beta_+^{(t)}\boldsymbol{q}_+^{(t)\top} + \sum_{n\in S_+}\sum_{i=2}^M \beta_{n,+,i}^{(t)}\boldsymbol{q}_{n,i}^{(t)\top} \right),
$$

$$
\langle\boldsymbol{q}_-^{(t+1)}, \boldsymbol{k}_-^{(t+1)}\rangle - \langle\boldsymbol{q}_-^{(t)}, \boldsymbol{k}_-^{(t)}\rangle
$$

$$
= \alpha_{-,-}^{(t)}\|\boldsymbol{k}_-^{(t)}\|_2^2 + \sum_{n\in S_-}\sum_{i=2}^M \alpha_{n,-,i}^{(t)}\langle\boldsymbol{k}_-^{(t)}, \boldsymbol{k}_{n,i}^{(t)}\rangle
$$

$$
+ \beta_{-,-}^{(t)}\|\boldsymbol{q}_-^{(t)}\|_2^2 + \sum_{n\in S_-}\sum_{i=2}^M \beta_{n,-,i}^{(t)}\langle\boldsymbol{q}_-^{(t)}, \boldsymbol{q}_{n,i}^{(t)}\rangle \tag{3}
$$

$$
+ \left( \alpha_{-,-}^{(t)}\boldsymbol{k}_-^{(t)} + \sum_{n\in S_-}\sum_{i=2}^M \alpha_{n,-,i}^{(t)}\boldsymbol{k}_{n,i}^{(t)} \right)
$$

$$
\cdot \left( \beta_{-,-}^{(t)}\boldsymbol{q}_-^{(t)\top} + \sum_{n\in S_-}\sum_{i=2}^M \beta_{n,-,i}^{(t)}\boldsymbol{q}_{n,i}^{(t)\top} \right),
$$

$$\langle \boldsymbol{q}_{n,i}^{(t+1)}, \boldsymbol{k}_+^{(t+1)} \rangle - \langle \boldsymbol{q}_{n,i}^{(t)}, \boldsymbol{k}_+^{(t)} \rangle$$

$$= \alpha_{n,i,+}^{(t)} \| \boldsymbol{k}_+^{(t)} \|_2^2 + \alpha_{n,i,-}^{(t)} \langle \boldsymbol{k}_+^{(t)}, \boldsymbol{k}_-^{(t)} \rangle + \sum_{n'=1}^{N} \sum_{l=2}^{M} \alpha_{n,i,n',l}^{(t)} \langle \boldsymbol{k}_+^{(t)}, \boldsymbol{k}_{n',l}^{(t)} \rangle$$

$$+ \beta_{+,+}^{(t)} \langle \boldsymbol{q}_+^{(t)}, \boldsymbol{q}_{n,i}^{(t)} \rangle + \sum_{n' \in S_+} \sum_{l=2}^{M} \beta_{n',+,l}^{(t)} \langle \boldsymbol{q}_{n,i}^{(t)}, \boldsymbol{q}_{n',l}^{(t)} \rangle$$

$$+ \left( \alpha_{n,i,+}^{(t)} \boldsymbol{k}_+^{(t)} + \alpha_{n,i,-}^{(t)} \boldsymbol{k}_-^{(t)} + \sum_{n'=1}^{N} \sum_{l=2}^{M} \alpha_{n,i,n',l}^{(t)} \boldsymbol{k}_{n',l}^{(t)} \right)$$

$$\cdot \left( \beta_{+,+}^{(t)} \boldsymbol{q}_+^{(t)\top} + \sum_{n' \in S_+} \sum_{l=2}^{M} \beta_{n',+,l}^{(t)} \boldsymbol{q}_{n',l}^{(t)\top} \right), \tag{4}$$

$$\langle \boldsymbol{q}_{n,i}^{(t+1)}, \boldsymbol{k}_-^{(t+1)} \rangle - \langle \boldsymbol{q}_{n,i}^{(t)}, \boldsymbol{k}_-^{(t)} \rangle$$

$$= \alpha_{n,i,-}^{(t)} \| \boldsymbol{k}_-^{(t)} \|_2^2 + \alpha_{n,i,+}^{(t)} \langle \boldsymbol{k}_+^{(t)}, \boldsymbol{k}_-^{(t)} \rangle + \sum_{n'=1}^{N} \sum_{l=2}^{M} \alpha_{n,i,n',l}^{(t)} \langle \boldsymbol{k}_-^{(t)}, \boldsymbol{k}_{n',l}^{(t)} \rangle$$

$$+ \beta_{-,-}^{(t)} \langle \boldsymbol{q}_-^{(t)}, \boldsymbol{q}_{n,i}^{(t)} \rangle + \sum_{n' \in S_-} \sum_{l=2}^{M} \beta_{n',-,l}^{(t)} \langle \boldsymbol{q}_{n,i}^{(t)}, \boldsymbol{q}_{n',l}^{(t)} \rangle$$

$$+ \left( \alpha_{n,i,+}^{(t)} \boldsymbol{k}_+^{(t)} + \alpha_{n,i,-}^{(t)} \boldsymbol{k}_-^{(t)} + \sum_{n'=1}^{N} \sum_{l=2}^{M} \alpha_{n,i,n',l}^{(t)} \boldsymbol{k}_{n',l}^{(t)} \right)$$

$$\cdot \left( \beta_{-,-}^{(t)} \boldsymbol{q}_-^{(t)\top} + \sum_{n' \in S_-} \sum_{l=2}^{M} \beta_{n',-,l}^{(t)} \boldsymbol{q}_{n',l}^{(t)\top} \right), \tag{5}$$

$$\langle \boldsymbol{q}_+^{(t+1)}, \boldsymbol{k}_{n,j}^{(t+1)} \rangle - \langle \boldsymbol{q}_+^{(t)}, \boldsymbol{k}_{n,j}^{(t)} \rangle$$

$$= \alpha_{+,+}^{(t)} \langle \boldsymbol{k}_+^{(t)}, \boldsymbol{k}_{n,j}^{(t)} \rangle + \sum_{n' \in S_+} \sum_{l=2}^{M} \alpha_{n',+,l}^{(t)} \langle \boldsymbol{k}_{n,j}^{(t)}, \boldsymbol{k}_{n',l}^{(t)} \rangle$$

$$+ \beta_{n,j,+}^{(t)} \| \boldsymbol{q}_+^{(t)} \|_2^2 + \beta_{n,j,-}^{(t)} \langle \boldsymbol{q}_+^{(t)}, \boldsymbol{q}_-^{(t)} \rangle + \sum_{n'=1}^{N} \sum_{l=2}^{M} \beta_{n,j,n',l}^{(t)} \langle \boldsymbol{q}_+^{(t)}, \boldsymbol{q}_{n',l}^{(t)} \rangle$$

$$+ \left( \alpha_{+,+}^{(t)} \boldsymbol{k}_+^{(t)} + \sum_{n' \in S_+} \sum_{l=2}^{M} \alpha_{n',+,l}^{(t)} \boldsymbol{k}_{n',l}^{(t)} \right)$$

$$\cdot \left( \beta_{n,j,+}^{(t)} \boldsymbol{q}_+^{(t)\top} + \beta_{n,j,-}^{(t)} \boldsymbol{q}_-^{(t)\top} + \sum_{n'=1}^{N} \sum_{l=2}^{M} \beta_{n,j,n',l}^{(t)} \boldsymbol{q}_{n',l}^{(t)\top} \right), \tag{6}$$

$$\langle \boldsymbol{q}_-^{(t+1)}, \boldsymbol{k}_{n,j}^{(t+1)} \rangle - \langle \boldsymbol{q}_-^{(t)}, \boldsymbol{k}_{n,j}^{(t)} \rangle$$

$$= \alpha_{-,-}^{(t)} \langle \boldsymbol{k}_-^{(t)}, \boldsymbol{k}_{n,j}^{(t)} \rangle + \sum_{n' \in S_-} \sum_{l=2}^{M} \alpha_{n',-,l}^{(t)} \langle \boldsymbol{k}_{n,j}^{(t)}, \boldsymbol{k}_{n',l}^{(t)} \rangle$$

$$+ \beta_{n,j,-}^{(t)} \|\boldsymbol{q}_-^{(t)}\|_2^2 + \beta_{n,j,+}^{(t)} \langle \boldsymbol{q}_+^{(t)}, \boldsymbol{q}_-^{(t)} \rangle + \sum_{n'=1}^{N} \sum_{l=2}^{M} \beta_{n,j,n',l}^{(t)} \langle \boldsymbol{q}_-^{(t)}, \boldsymbol{q}_{n',l}^{(t)} \rangle \qquad (7)$$

$$+ \left( \alpha_{-,-}^{(t)} \boldsymbol{k}_-^{(t)} + \sum_{n' \in S_-} \sum_{l=2}^{M} \alpha_{n',-,l}^{(t)} \boldsymbol{k}_{n',l}^{(t)} \right)$$

$$\cdot \left( \beta_{n,j,+}^{(t)} \boldsymbol{q}_+^{(t)\top} + \beta_{n,j,-}^{(t)} \boldsymbol{q}_-^{(t)\top} + \sum_{n'=1}^{N} \sum_{l=2}^{M} \beta_{n,j,n',l}^{(t)} \boldsymbol{q}_{n',l}^{(t)\top} \right),$$

$$\langle \boldsymbol{q}_{n,i}^{(t+1)}, \boldsymbol{k}_{n,j}^{(t+1)} \rangle - \langle \boldsymbol{q}_{n,i}^{(t)}, \boldsymbol{k}_{n,j}^{(t)} \rangle$$

$$= \alpha_{n,i,+}^{(t)} \langle \boldsymbol{k}_+^{(t)}, \boldsymbol{k}_{n,j}^{(t)} \rangle + \alpha_{n,i,-}^{(t)} \langle \boldsymbol{k}_-^{(t)}, \boldsymbol{k}_{n,j}^{(t)} \rangle + \sum_{n'=1}^{N} \sum_{l=2}^{M} \alpha_{n,i,n',l}^{(t)} \langle \boldsymbol{k}_{n',l}^{(t)}, \boldsymbol{k}_{n,j}^{(t)} \rangle$$

$$+ \beta_{n,j,+}^{(t)} \langle \boldsymbol{q}_+^{(t)}, \boldsymbol{q}_{n,i}^{(t)} \rangle + \beta_{n,j,-}^{(t)} \langle \boldsymbol{q}_-^{(t)}, \boldsymbol{q}_{n,i}^{(t)} \rangle + \sum_{n'=1}^{N} \sum_{l=2}^{M} \beta_{n,j,n',l}^{(t)} \langle \boldsymbol{q}_{n',l}^{(t)}, \boldsymbol{q}_{n,i}^{(t)} \rangle \qquad (8)$$

$$+ \left( \alpha_{n,i,+}^{(t)} \boldsymbol{k}_+^{(t)} + \alpha_{n,i,-}^{(t)} \boldsymbol{k}_-^{(t)} + \sum_{n'=1}^{N} \sum_{l=2}^{M} \alpha_{n,i,n',l}^{(t)} \boldsymbol{k}_{n',l}^{(t)} \right)$$

$$\cdot \left( \beta_{n,j,+}^{(t)} \boldsymbol{q}_+^{(t)\top} + \beta_{n,j,-}^{(t)} \boldsymbol{q}_-^{(t)\top} + \sum_{n'=1}^{N} \sum_{l=2}^{M} \beta_{n,j,n',l}^{(t)} \boldsymbol{q}_{n',l}^{(t)\top} \right),$$

for $i, j \in [M] \backslash \{1\}, n \in [N]$.

# E    CONCENTRATION INEQUALITIES

In this section, we will give some concentration inequalities that show some important properties of the data and the ViT parameters at random initialization.

**Lemma 10** (Lemma B.1 in Cao et al. (2022)). *Suppose that $\delta > 0$ and $n \geq 8 \log(4/\delta)$. Then with probability at least $1 - \delta$,*

$$\frac{N}{4} \leq |\{n \in [N] : y_n = 1\}|, |\{n \in [N] : y_n = -1\}| \leq \frac{3N}{4}.$$

**Lemma 11** (Initialization of V, Lemma C.2 in Jiang et al. (2024)). *Suppose that $\delta > 0$. Then with probability at least $1 - \delta$,*

$$|V_{\pm}^{(0)}| \leq d_h^{-\frac{1}{4}}, \quad |V_{n,i}^{(0)}| \leq d_h^{-\frac{1}{4}}$$

for $i \in [M] \backslash \{1\}, n \in [N]$.

**Lemma 12** (Initialization of QK, Lemma C.3 in Jiang et al. (2024)). *Suppose that $\delta > 0$. Then with probability at least $1 - \delta$,*

$$\frac{\|\boldsymbol{\mu}\|_2^2 \sigma_p^2 d_h}{2} \leq \|\boldsymbol{q}_{\pm}^{(0)}\|_2^2 \leq \frac{3\|\boldsymbol{\mu}\|_2^2 \sigma_p^2 d_h}{2},$$

$$\frac{\sigma_p^2 \sigma_h^2 d d_h}{2} \leq \|\boldsymbol{q}_{n,i}^{(0)}\|_2^2 \leq \frac{3\sigma_p^2 \sigma_h^2 d d_h}{2},$$

$$\frac{\|\boldsymbol{\mu}\|_2^2 \sigma_p^2 d_h}{2} \leq \|\boldsymbol{k}_{\pm}^{(0)}\|_2^2 \leq \frac{3\|\boldsymbol{\mu}\|_2^2 \sigma_p^2 d_h}{2},$$

$$\frac{\sigma_p^2 \sigma_h^2 d d_h}{2} \leq \|\boldsymbol{k}_{n,i}^{(0)}\|_2^2 \leq \frac{3\sigma_p^2 \sigma_h^2 d d_h}{2},$$

$$|\langle \boldsymbol{q}_+^{(0)}, \boldsymbol{q}_-^{(0)} \rangle| \leq 2\|\boldsymbol{\mu}\|_2^2 \sigma_h^2 \cdot \sqrt{d_h \log(6N^2M^2/\delta)},$$

$$|\langle \boldsymbol{q}_\pm^{(0)}, \boldsymbol{q}_{n,i}^{(0)} \rangle| \leq 2\|\boldsymbol{\mu}\|_2 \sigma_p \sigma_h^2 d^{\frac{3}{2}} \cdot \sqrt{d_h \log(6N^2M^2/\delta)},$$

$$|\langle \boldsymbol{k}_\pm^{(0)}, \boldsymbol{k}_\pm^{(0)} \rangle| \leq 2\|\boldsymbol{\mu}\|_2^2 \sigma_h^2 \cdot \sqrt{d_h \log(6N^2M^2/\delta)},$$

$$|\langle \boldsymbol{q}_\pm^{(0)}, \boldsymbol{k}_\pm^{(0)} \rangle| \leq 2\|\boldsymbol{\mu}\|_2^2 \sigma_h^2 \cdot \sqrt{d_h \log(6N^2M^2/\delta)},$$

$$|\langle \boldsymbol{q}_\pm^{(0)}, \boldsymbol{k}_\mp^{(0)} \rangle| \leq 2\|\boldsymbol{\mu}\|_2^2 \sigma_h^2 \cdot \sqrt{d_h \log(6N^2M^2/\delta)},$$

$$|\langle \boldsymbol{q}_{n,i}^{(0)}, \boldsymbol{k}_\pm^{(0)} \rangle| \leq 2\|\boldsymbol{\mu}\|_2 \sigma_p \sigma_h^2 d^{\frac{3}{2}} \cdot \sqrt{d_h \log(6N^2M^2/\delta)},$$

$$|\langle \boldsymbol{q}_{n,i}^{(0)}, \boldsymbol{q}_{n',j}^{(0)} \rangle| \leq 2\sigma_p^2 \sigma_h^2 d \cdot \sqrt{d_h \log(6N^2M^2/\delta)},$$

$$|\langle \boldsymbol{k}_{n,i}^{(0)}, \boldsymbol{k}_{n',j}^{(0)} \rangle| \leq 2\sigma_p^2 \sigma_h^2 d \cdot \sqrt{d_h \log(6N^2M^2/\delta)},$$

$$|\langle \boldsymbol{k}_\pm^{(0)}, \boldsymbol{k}_{n,i}^{(0)} \rangle| \leq 2\|\boldsymbol{\mu}\|_2 \sigma_p \sigma_h^2 d^{\frac{3}{2}} \cdot \sqrt{d_h \log(6N^2M^2/\delta)},$$

$$|\langle \boldsymbol{q}_\pm^{(0)}, \boldsymbol{k}_{n,i}^{(0)} \rangle| \leq 2\|\boldsymbol{\mu}\|_2 \sigma_p \sigma_h^2 d^{\frac{3}{2}} \cdot \sqrt{d_h \log(6N^2M^2/\delta)},$$

$$|\langle \boldsymbol{q}_{n,i}^{(0)}, \boldsymbol{k}_{n',j}^{(0)} \rangle| \leq 2\sigma_p^2 \sigma_h^2 d \cdot \sqrt{d_h \log(6N^2M^2/\delta)}$$

*for $i, j \in [M] \setminus \{1\}$ and $n, n' \in [N]$.*

**Lemma 13** (Lemma B.2 in Cao et al. (2022) and Lemma B.4 in Kou et al. (2023)). *Suppose that $\delta > 0$ and $d = \Omega(\log(4NM/\delta))$. Then with probability at least $1 - \delta$*

$$\frac{\sigma_p^2 d}{2} \leq \|\boldsymbol{\xi}_{n,i}\|_2^2 \leq \frac{3\sigma_p^2 d}{2},$$

$$|\langle \boldsymbol{\xi}_{n,i}, \boldsymbol{\xi}_{n',i'} \rangle| \leq 2\sigma_p^2 \cdot \sqrt{d \log(4N^2M^2/\delta)},$$

$$\frac{\sigma_p^2 d}{2} - 2\sigma_p \tau \sqrt{2\log(4NM/\delta)} - \tau^2 \leq \|\widetilde{\boldsymbol{\xi}}_{n,i}\|_2^2 \leq \frac{3\sigma_p^2 d}{2} + 2\sigma_p \tau \sqrt{2\log(4NM/\delta)} + \tau^2,$$

$$(\|\boldsymbol{\mu}\|_2 - \tau)^2 \leq \langle \widetilde{\boldsymbol{\mu}}_\pm, \widetilde{\boldsymbol{\mu}}'_\pm \rangle \leq (\|\boldsymbol{\mu}\|_2 + \tau)^2,$$

$$|\langle \widetilde{\boldsymbol{\mu}}_\pm, \widetilde{\boldsymbol{\xi}}_{n,i} \rangle| \leq \|\boldsymbol{\mu}\|_2 \tau + \sigma_p \tau \sqrt{2\log(4NM/\delta)} + \tau^2$$

$$|\langle \widetilde{\boldsymbol{\xi}}_{n,i}, \widetilde{\boldsymbol{\xi}}_{n',i'} \rangle| \leq 2\sigma_p^2 \cdot \sqrt{d \log(4N^2M^2/\delta)} + 2\sigma_p \tau \sqrt{2\log(4NM/\delta)} + \tau^2$$

*for $i, i' \in [M] \setminus \{1\}, n, n' \in [N], i \neq i'$ or $n \neq n'$.*

## F   BENIGN OVERFITTING IN CASE 1

In this section, we consider the benign overfitting regime under the condition that $N \cdot \mathrm{SNR}^2 = \Omega(1)$ and $\tau \leq O(\frac{\|\boldsymbol{\mu}\|_2}{\log d_h})$. We analyze the dynamics of $V_\pm, V_{n,i}$, the inner product $\boldsymbol{q}_\pm, \boldsymbol{q}_{n,i}$, and $\boldsymbol{k}_\pm, \boldsymbol{k}_{n,i}$ during adversarial training, and further give the upper bound for clean test error and robust test error. The proofs in this section are based on the results in Section E, which hold with high probability.

### F.1   STAGE I

In Stage I, $V_\pm^{(t)}, V_{n,i}^{(t)}$ begin to pull apart until $|V_\pm^{(t)}|$ is sufficiently larger than $|V_{n,i}^{(t)}|$. At the same time, the inner products of $q$ and $k$ maintain their magnitude.

**Lemma 14** (Gradient of Loss). *As long as $\max\{|V_+^{(t)}|, |V_-^{(t)}|, |V_{n,i}^{(t)}|\} = o(1)$, we have $-\ell'(y_n f(\widetilde{\mathbf{X}}_n, \theta(t)))$ remains $1/2 \pm o(1)$.*

*Proof.* Note that $\ell(z) = \log(1 + \exp(-z))$ and $-\ell' = \exp(-z)/(1 + \exp(-z))$, without loss of generality, we assume $y_n = 1$, we have

$$-\ell'(f(\widetilde{\mathbf{X}}_n, \theta(t))) = \frac{1}{1 + \exp\left(\frac{1}{M} \sum_{l=1}^{M} \varphi(\widetilde{\mathbf{x}}_{n,l}^\top \mathbf{W}_Q^{(t)} \mathbf{W}_K^{(t)\top} \widetilde{\mathbf{X}}_n) \widetilde{\mathbf{X}}_n^\top \mathbf{W}_V^{(t)} \boldsymbol{w}_O\right)}.$$

Note that

$$-\max\{|V_+^{(t)}|, |V_-^{(t)}|, |V_{n,i}^{(t)}|\} \leq \frac{1}{M}\sum_{l=1}^{M}\varphi(\widetilde{\mathbf{x}}_{n,l}^{\top}\mathbf{W}_Q^{(t)\top}\widetilde{\mathbf{X}}_n^{\top})\widetilde{\mathbf{X}}_n\mathbf{W}_V^{(t)}\boldsymbol{w}_O \leq \max\{|V_+^{(t)}|, |V_-^{(t)}|, |V_{n,i}^{(t)}|\}.$$

Then we have

$$-\ell'(f(\widetilde{\mathbf{X}}_n, \theta(t))) \geq \frac{1}{1+\exp(0+o(1))} \geq \frac{1}{2+o(1)} \geq \frac{1}{2} - o(1),$$

$$-\ell'(f(\widetilde{\mathbf{X}}_n, \theta(t))) \leq \frac{\exp(0+o(1))}{1+\exp(0+o(1))} \leq \frac{1+o(1)}{1+1+o(1)} \leq \frac{1}{2} + o(1).$$

$\square$

**Lemma 15** (Bound of Attention). *As long as* $|\langle\mathbf{q}_\pm^{(t)}, \mathbf{k}_\pm^{(t)}\rangle|, |\langle\mathbf{q}_{n,i}^{(t)}, \mathbf{k}_\pm^{(t)}\rangle|, |\langle\mathbf{q}_\pm^{(t)}, \mathbf{k}_{n,j}^{(t)}\rangle|, |\langle\mathbf{q}_{n,i}^{(t)}, \mathbf{k}_{n,j}^{(t)}\rangle| = o(1)$, *we have*

$$\frac{1}{M} - o(1) \leq \textit{softmax}(\langle\mathbf{q}_\pm^{(t)}, \mathbf{k}_\pm^{(t)}\rangle) \leq \frac{1}{M} + o(1),$$

$$\frac{1}{M} - o(1) \leq \textit{softmax}(\langle\mathbf{q}_{n,i}^{(t)}, \mathbf{k}_\pm^{(t)}\rangle) \leq \frac{1}{M} + o(1),$$

$$\frac{1}{M} - o(1) \leq \textit{softmax}(\langle\mathbf{q}_\pm^{(t)}, \mathbf{k}_{n,j}^{(t)}\rangle) \leq \frac{1}{M} + o(1),$$

$$\frac{1}{M} - o(1) \leq \textit{softmax}(\langle\mathbf{q}_{n,i}^{(t)}, \mathbf{k}_{n,j}^{(t)}\rangle) \leq \frac{1}{M} + o(1).$$

*Proof.* It is clear that $\exp(o(1)) = 1 + o(1)$. Therefore, as long as $|\langle\mathbf{q}_\pm^{(t)}, \mathbf{k}_\pm^{(t)}\rangle| = o(1)$, we have

$$\frac{1}{M} - o(1) = \frac{1}{1 + (M-1) + (M-1)o(1)} = \frac{1}{1 + (M-1)\exp(o(1))} =$$

$$= \frac{\exp(-o(1))}{\exp(-o(1)) + (M-1)\exp(o(1))} \leq \textit{softmax}(\langle\mathbf{q}_\pm^{(t)}, \mathbf{k}_\pm^{(t)}\rangle) \leq \frac{\exp(o(1))}{\exp(o(1)) + (M-1)\exp(-o(1))}$$

$$= \frac{\exp(o(1))}{\exp(o(1)) + (M-1)} = \frac{1+o(1)}{1+o(1)+(M-1)} = \frac{1}{M} + o(1)$$

Similarly, we have

$$\frac{1}{M} - o(1) \leq \text{softmax}(\langle\mathbf{q}_{n,i}^{(t)}, \mathbf{k}_\pm^{(t)}\rangle) \leq \frac{1}{M} + o(1),$$

$$\frac{1}{M} - o(1) \leq \text{softmax}(\langle\mathbf{q}_\pm^{(t)}, \mathbf{k}_{n,j}^{(t)}\rangle) \leq \frac{1}{M} + o(1),$$

$$\frac{1}{M} - o(1) \leq \text{softmax}(\langle\mathbf{q}_{n,i}^{(t)}, \mathbf{k}_{n,j}^{(t)}\rangle) \leq \frac{1}{M} + o(1).$$

$\square$

**Lemma 16** (Upper bound of V). *Let* $T_0 = \mathcal{O}\left(\frac{1}{\eta d_h^{\frac{1}{4}}(\|\boldsymbol{\mu}\|_2 + \tau)^2\|\boldsymbol{w}_O\|_2^2}\right)$. *Then under the same conditions as Theorem 2 we have*

$$|V_+^{(t)}|, |V_-^{(t)}|, |V_{n,i}^{(t)}| = \mathcal{O}(d_h^{-\frac{1}{4}})$$

*for* $t \in [0, T_0]$.

*Proof.* By Lemma 8, we have

$$|\gamma_{V,+}^{(t+1)} - \gamma_{V,+}^{(t)}| \leq -\frac{\eta\langle\widetilde{\boldsymbol{\mu}}_+, \widetilde{\boldsymbol{\mu}}_+^{(t)}\rangle}{NM}\sum_{n\in S_+}\widetilde{\ell}_n'^{(t)}\left(\frac{\exp(\langle\widetilde{\mathbf{q}}_+^{(t)}, \widetilde{\mathbf{k}}_+^{(t)}\rangle)}{\exp(\langle\widetilde{\mathbf{q}}_+^{(t)}, \widetilde{\mathbf{k}}_+^{(t)}\rangle) + \sum_{k=2}^{M}\exp(\langle\widetilde{\mathbf{q}}_+^{(t)}, \widetilde{\mathbf{k}}_{n,k}^{(t)}\rangle)}\right.$$

$$\left.+ \sum_{j=2}^{M}\frac{\exp(\langle\widetilde{\mathbf{q}}_{n,j}^{(t)}, \widetilde{\mathbf{k}}_+^{(t)}\rangle)}{\exp(\langle\widetilde{\mathbf{q}}_{n,j}^{(t)}, \widetilde{\mathbf{k}}_+^{(t)}\rangle) + \sum_{k=2}^{M}\exp(\langle\widetilde{\mathbf{q}}_{n,j}^{(t)}, \widetilde{\mathbf{k}}_{n,k}^{(t)}\rangle)}\right)\|\boldsymbol{w}_O\|_2^2$$

$$+ \frac{-\eta\langle\widetilde{\boldsymbol{\mu}}_+, \widetilde{\boldsymbol{\xi}}_{n,i}^{(t)}\rangle}{NM} \sum_{n\in S_+} \sum_{i=2}^{M} \left( \frac{\exp(\langle\widetilde{\mathbf{q}}_+^{(t)}, \widetilde{\mathbf{k}}_{n,i}^{(t)}\rangle)}{\exp(\langle\widetilde{\mathbf{q}}_+^{(t)}, \widetilde{\mathbf{k}}_+^{(t)}\rangle) + \sum_{k=2}^{M}\exp(\langle\widetilde{\mathbf{q}}_+^{(t)}, \widetilde{\mathbf{k}}_{n,k}^{(t)}\rangle)} \right.$$

$$\left. + \sum_{j=2}^{M} \frac{\exp(\langle\widetilde{\mathbf{q}}_{n,j}^{(t)}, \widetilde{\mathbf{k}}_{n,i}^{(t)}\rangle)}{\exp(\langle\widetilde{\mathbf{q}}_{n,j}^{(t)}, \widetilde{\mathbf{k}}_+^{(t)}\rangle) + \sum_{k=2}^{M}\exp(\langle\widetilde{\mathbf{q}}_{n,j}^{(t)}, \widetilde{\mathbf{k}}_{n,k}^{(t)}\rangle)} \right)$$

$$\leq \frac{\eta(\|\boldsymbol{\mu}\|_2 + \tau)^2}{NM} \cdot \frac{3N}{4}(1 + (M-1)) + \frac{\eta(\|\boldsymbol{\mu}\|_2\tau + \sigma_p\tau\sqrt{2\log(4NM/\delta)} + \tau^2)}{NM} \cdot \frac{3N}{4}M(M-1)$$

$$\leq O(\eta(\|\boldsymbol{\mu}\|_2 + \tau)^2),$$

where the second inequality is by Lemma 10 and Lemma 13 and $-\widetilde{\ell}_n^{\prime(t)} \leq 1$.

Similarly, we have

$$|\gamma_{V,-}^{(t+1)} - \gamma_{V,-}^{(t)}| \leq O(\eta(\|\boldsymbol{\mu}\|_2 + \tau)^2).$$

By Definition 4, we have

$$|V_+^{(t)}| = \left| V_+^{(0)} + \sum_{s=0}^{t-1}(\gamma_{V,+}^{(s+1)} - \gamma_{V,+}^{(s)})\|\boldsymbol{w}_O\|_2^2 \right|$$

$$\leq |V_+^{(0)}| + \sum_{s=0}^{t-1}|\gamma_{V,+}^{(s+1)} - \gamma_{V,+}^{(s)}| \cdot \|\boldsymbol{w}_O\|_2^2$$

$$\leq d_h^{-\frac{1}{4}} + O(\eta(\|\boldsymbol{\mu}\|_2 + \tau)^2) \cdot \|\boldsymbol{w}_O\|_2^2 \cdot O\left(\frac{1}{\eta d_h^{\frac{1}{4}}(\|\boldsymbol{\mu}\|_2 + \tau)^2\|\boldsymbol{w}_O\|_2^2}\right)$$

$$= O(d_h^{-\frac{1}{4}}),$$

where the first inequality is by triangle inequality, the second inequality is by Lemma 11. Similarly, we have $|V_-^{(t)}| = O(d_h^{-\frac{1}{4}})$. By Lemma 8, we have

$$|\rho_{V,n,i}^{(t+1)} - \rho_{V,n,i}^{(t)}| \leq \left| -\frac{\eta}{NM}\sum_{n'\in S_+}\widetilde{\ell}_{n'}^{\prime(t)}\left(\left(\langle\widetilde{\boldsymbol{\xi}}_{n,i}, \widetilde{\boldsymbol{\mu}}_+^{(t)}\rangle \frac{\exp(\langle\widetilde{\mathbf{q}}_+^{(t)}, \widetilde{\mathbf{k}}_+^{(t)}\rangle)}{\exp(\langle\widetilde{\mathbf{q}}_+^{(t)}, \widetilde{\mathbf{k}}_+^{(t)}\rangle) + \sum_{k=2}^{M}\exp(\langle\widetilde{\mathbf{q}}_+^{(t)}, \widetilde{\mathbf{k}}_{n,k}^{(t)}\rangle)}\right.\right.$$

$$\left.+ \sum_{j=2}^{M}\langle\widetilde{\boldsymbol{\xi}}_{n,i}, \widetilde{\boldsymbol{\mu}}_+^{(t)}\rangle \frac{\exp(\langle\widetilde{\mathbf{q}}_{n,j}^{(t)}, \widetilde{\mathbf{k}}_+^{(t)}\rangle)}{\exp(\langle\widetilde{\mathbf{q}}_{n,j}^{(t)}, \widetilde{\mathbf{k}}_+^{(t)}\rangle) + \sum_{k=2}^{M}\exp(\langle\widetilde{\mathbf{q}}_{n,j}^{(t)}, \widetilde{\mathbf{k}}_{n,k}^{(t)}\rangle)}\right)$$

$$+ \sum_{i=2}^{M}\left(\langle\widetilde{\boldsymbol{\xi}}_{n,i}, \widetilde{\boldsymbol{\xi}}_{n',i'}^{(t)}\rangle \frac{\exp(\langle\widetilde{\mathbf{q}}_{n',i'}^{(t)}, \widetilde{\mathbf{k}}_{n',i'}^{(t)}\rangle)}{\exp(\langle\widetilde{\mathbf{q}}_{n',i'}^{(t)}, \widetilde{\mathbf{k}}_{n',i'}^{(t)}\rangle) + \sum_{k=2}^{M}\exp(\langle\widetilde{\mathbf{q}}_{n',i'}^{(t)}, \widetilde{\mathbf{k}}_{n',k}^{(t)}\rangle)}\right.$$

$$\left.\left.\left.+ \sum_{j=2}^{M}\langle\widetilde{\boldsymbol{\xi}}_{n,i}, \widetilde{\boldsymbol{\xi}}_{n',i'}^{(t)}\rangle \frac{\exp(\langle\widetilde{\mathbf{q}}_{n',j}^{(t)}, \widetilde{\mathbf{k}}_{n',i'}^{(t)}\rangle)}{\exp(\langle\widetilde{\mathbf{q}}_{n',j}^{(t)}, \widetilde{\mathbf{k}}_{n',i'}^{(t)}\rangle) + \sum_{k=2}^{M}\exp(\langle\widetilde{\mathbf{q}}_{n',j}^{(t)}, \widetilde{\mathbf{k}}_{n',k}^{(t)}\rangle)}\right)\right)\right.$$

$$\left. - \frac{\eta}{NM}\sum_{n'\in S_-}(\cdot) \right|$$

$$\leq \frac{3\eta\sigma_p^2 d}{2NM}\cdot M + \frac{\eta}{NM}\cdot MN\cdot 2\sigma_p^2\cdot\sqrt{d\log(4N^2M^2/\delta)}$$

$$+ \frac{\eta(\|\boldsymbol{\mu}\|\tau + (2M-1)\sigma_p\tau\sqrt{3d/2} + M\tau^2)}{NM}\cdot NM$$

$$\leq \frac{2\eta\sigma_p^2 d}{N} + \frac{\eta(\|\boldsymbol{\mu}\|\tau + \sigma_p\tau\sqrt{2\log(4MN/\delta)} + \tau^2)}{NM}\cdot NM^2$$

$$= O(\eta(\max\{(\|\boldsymbol{\mu}\|_2 + \tau)^2, \frac{\sigma_p^2 d}{N}\}))$$

$$= O(\eta(\|\boldsymbol{\mu}\|_2 + \tau)^2)$$

where the second inequality is by Lemma 13 and $-\ell_n'^{(t)} \leq 1$, the third inequality is by $d = \widetilde{\Omega}(\epsilon^{-2}N^2 d_h) \geq 4N\sqrt{\log(4N^2M^2/\delta)}$, the last inequality is by $N \cdot \text{SNR}^2 = \Omega(1)$. Then by Definition 4, we have

$$
\begin{aligned}
|V_{n,i}^{(t)}| &= |V_{n,i}^{(0)} + \sum_{s=0}^{t-1}(\rho_{V,n,i}^{(s+1)} - \rho_{V,n,i}^{(s)})\|\boldsymbol{w}_O\|_2^2| \\
&\leq |V_{n,i}^{(0)}| + \sum_{s=0}^{t-1}|\rho_{V,n,i}^{(s+1)} - \rho_{V,n,i}^{(s)}| \cdot \|\boldsymbol{w}_O\|_2^2 \\
&\leq d_h^{-\frac{1}{4}} + O(\eta(\|\boldsymbol{\mu}\|_2 + \tau)^2) \cdot \|\boldsymbol{w}_O\|_2^2 \cdot O\left(\frac{1}{\eta d_h^{\frac{1}{4}}(\|\boldsymbol{\mu}\|_2 + \tau)^2\|\boldsymbol{w}_O\|_2^2}\right) \\
&= O(d_h^{-\frac{1}{4}}),
\end{aligned}
$$

where the first inequality is by triangle inequality, the second inequality is by Lemma C.2, which completes the proof. $\qquad\square$

**Lemma 17** (Inner Products Hold Magnitude). *Let $T_0 = O\left(\frac{1}{\eta d_h^{\frac{1}{4}}(\|\boldsymbol{\mu}\|_2+\tau)^2\|\boldsymbol{w}_O\|_2^2}\right)$. Then under the same conditions as Theorem 2, we have*

$$
\begin{aligned}
&|\langle \boldsymbol{q}_\pm^{(t)}, \boldsymbol{k}_\pm^{(t)}\rangle|, |\langle \boldsymbol{q}_{n,i}^{(t)}, \boldsymbol{k}_\pm^{(t)}\rangle|, |\langle \boldsymbol{q}_\pm^{(t)}, \boldsymbol{k}_{n,j}^{(t)}\rangle|, |\langle \boldsymbol{q}_{n,i}^{(t)}, \boldsymbol{k}_{n',j}^{(t)}\rangle| \\
&= O\left(\max\{\|\boldsymbol{\mu}\|_2^2, \sigma_p^2 d\} \cdot \sigma_h^2 \cdot \sqrt{d_h}\log(6N^2M^2/\delta)\right),
\end{aligned}
$$

$$
\begin{aligned}
&|\langle \boldsymbol{q}_\pm^{(t)}, \boldsymbol{q}_\mp^{(t)}\rangle|, |\langle \boldsymbol{q}_{n,i}^{(t)}, \boldsymbol{q}_\mp^{(t)}\rangle|, |\langle \boldsymbol{q}_{n,i}^{(t)}, \boldsymbol{q}_{n',j}^{(t)}\rangle| \\
&= O\left(\max\{\|\boldsymbol{\mu}\|_2^2, \sigma_p^2 d\} \cdot \sigma_h^2 \cdot \sqrt{d_h}\log(6N^2M^2/\delta)\right),
\end{aligned}
$$

$$
\begin{aligned}
&|\langle \boldsymbol{k}_\pm^{(t)}, \boldsymbol{k}_\mp^{(t)}\rangle|, |\langle \boldsymbol{k}_{n,i}^{(t)}, \boldsymbol{k}_\pm^{(t)}\rangle|, |\langle \boldsymbol{k}_{n,i}^{(t)}, \boldsymbol{k}_{n',j}^{(t)}\rangle| \\
&= O\left(\max\{\|\boldsymbol{\mu}\|_2^2, \sigma_p^2 d\} \cdot \sigma_h^2 \cdot \sqrt{d_h}\log(6N^2M^2/\delta)\right),
\end{aligned}
$$

$$
\|\boldsymbol{q}_\pm^{(t)}\|_2^2, \|\boldsymbol{k}_\pm^{(t)}\|_2^2 = \Theta(\|\boldsymbol{\mu}\|_2^2\sigma_h^2 d_h),
$$

$$
\|\boldsymbol{q}_{n,i}^{(t)}\|_2^2, \|\boldsymbol{k}_{n,i}^{(t)}\|_2^2 = \Theta(\sigma_p^2\sigma_h^2 d d_h)
$$

*for $i, j \in [M]\backslash\{1\}$, $n, n' \in [N]$ and $t \in [0, T_0]$.*

The proof for Lemma 17 is in Section I.4. Note that $\sigma_h^2 \leq \min\{\|\boldsymbol{\mu}\|_2^{-2}, (\sigma_p^2 d)^{-1}\} \cdot d_h^{-\frac{1}{2}} \cdot (\log(6N^2M^2/\delta))^{-\frac{3}{2}}$, thus $O\left(\max\{\|\boldsymbol{\mu}\|_2^2, \sigma_p^2 d\} \cdot \sigma_h^2 \cdot \sqrt{d_h}\log(6N^2M^2/\delta)\right) = o(1)$.

**Lemma 18** (V's Beginning of Learning Signals). *Under the same conditions as Theorem 2, there exists*

$$
T_1 = \frac{10M(3M+1)N}{\eta d_h^{\frac{1}{4}}\left(N(\|\boldsymbol{\mu}\|_2 - \tau)^2 - (N+30M^2)(\|\boldsymbol{\mu}\|\tau + 2\sigma_p\tau\sqrt{2\log(4NM/\delta)} + \tau^2) - 60M^2\sigma_p^2 d\right)\|\boldsymbol{w}_O\|_2}
$$

*such that the first element of the vector $\widetilde{X}_n\mathbf{W}_V^{(t)}\boldsymbol{w}_O$ dominates its other elements, that is,*

$$
V_+^{(t)} \geq 3M \cdot |V_{n,i}^{(t)}|, \quad \text{for all } n \in S_+,\ i \in [M]\backslash\{1\},
$$

$$
V_-^{(t)} \leq -3M \cdot |V_{n,i}^{(t)}|, \quad \text{for all } n \in S_-,\ i \in [M]\backslash\{1\}.
$$

*Proof.* Let $C$ be a constant larger than $10M(3M+1)$. As long as

$$
N(\|\boldsymbol{\mu}\|_2 - \tau)^2 - (N+30M^2)(\|\boldsymbol{\mu}\|\tau + 2\sigma_p\tau\sqrt{2\log(4NM/\delta)} + \tau^2) - 60M^2\sigma_p^2 d \geq \frac{10M(3M+1)}{C}N(\|\boldsymbol{\mu}\|_2 + \tau)^2.
$$

Thus, we further get

$$T_1 = \frac{10M(3M+1)N}{\eta d_h^{\frac{1}{4}} \left( N(\|\boldsymbol{\mu}\|_2 - \tau)^2 - (N + 30M^2)(\|\boldsymbol{\mu}\|\tau + 2\sigma_p\tau\sqrt{2\log(4NM/\delta)} + \tau^2) - 60M^2\sigma_p^2 d \right) \|\boldsymbol{w}_O\|_2}$$

$$\leq \frac{C}{\eta d_h^{\frac{1}{4}}(\|\boldsymbol{\mu}\|_2 + \tau)^2 \|\boldsymbol{w}_O\|_2} = \mathcal{O}\left( \frac{1}{\eta d_h^{\frac{1}{4}}(\|\boldsymbol{\mu}\|_2 + \tau)^2 \|\boldsymbol{w}_O\|_2} \right),$$

which satisfies the time condition in Lemma 16 and Lemma 17. Then by Lemma 14 and Lemma 15 we have:

$$-\ell_n'^{(t)} = \frac{1}{2} \pm o(1),$$

$$\frac{1}{M} - o(1) \leq \text{softmax}(\langle \boldsymbol{q}_+^{(t)}, \boldsymbol{k}_+^{(t)} \rangle) \leq \frac{1}{M} + o(1),$$

$$\frac{1}{M} - o(1) \leq \text{softmax}(\langle \boldsymbol{q}_{n,i}^{(t)}, \boldsymbol{k}_+^{(t)} \rangle) \leq \frac{1}{M} + o(1),$$

$$\frac{1}{M} - o(1) \leq \text{softmax}(\langle \boldsymbol{q}_{n,i}^{(t)}, \boldsymbol{k}_{n,j}^{(t)} \rangle) \leq \frac{1}{M} + o(1).$$

For $i, j \in [M] \setminus \{1\}$, $n \in [N]$, and $t \in [0, T_1]$, plugging these into the update rule for $\gamma_{V_+}^{(t)}$ shown in Lemma 8, we have:

$$\gamma_{V_+}^{(t+1)} - \gamma_{V_+}^{(t)} = -\frac{\eta\langle \widetilde{\boldsymbol{\mu}}_+, \widetilde{\boldsymbol{\mu}}_+^{(t)} \rangle}{NM} \sum_{n \in S_+} \widetilde{\ell}_n'^{(t)} \left( \frac{\exp(\langle \widetilde{\mathbf{q}}_+^{(t)}, \widetilde{\mathbf{k}}_+^{(t)} \rangle)}{\exp(\langle \widetilde{\mathbf{q}}_+^{(t)}, \widetilde{\mathbf{k}}_+^{(t)} \rangle) + \sum_{k=2}^{M} \exp(\langle \widetilde{\mathbf{q}}_+^{(t)}, \widetilde{\mathbf{k}}_{n,k}^{(t)} \rangle)} \right.$$

$$\left. + \sum_{j=2}^{M} \frac{\exp(\langle \widetilde{\mathbf{q}}_{n,j}^{(t)}, \widetilde{\mathbf{k}}_+^{(t)} \rangle)}{\exp(\langle \widetilde{\mathbf{q}}_{n,j}^{(t)}, \widetilde{\mathbf{k}}_+^{(t)} \rangle) + \sum_{k=2}^{M} \exp(\langle \widetilde{\mathbf{q}}_{n,j}^{(t)}, \widetilde{\mathbf{k}}_{n,k}^{(t)} \rangle)} \right)$$

$$+ \sum_{n \in S_+} \widetilde{\ell}_n'^{(t)} \sum_{i=2}^{M} \frac{-\eta\langle \widetilde{\boldsymbol{\mu}}_+, \widetilde{\boldsymbol{\xi}}_{n,i}^{(t)} \rangle}{NM} \left( \frac{\exp(\langle \widetilde{\mathbf{q}}_+^{(t)}, \widetilde{\mathbf{k}}_{n,i}^{(t)} \rangle)}{\exp(\langle \widetilde{\mathbf{q}}_+^{(t)}, \widetilde{\mathbf{k}}_+^{(t)} \rangle) + \sum_{k=2}^{M} \exp(\langle \widetilde{\mathbf{q}}_+^{(t)}, \widetilde{\mathbf{k}}_{n,k}^{(t)} \rangle)} \right.$$

$$\left. + \sum_{j=2}^{M} \frac{\exp(\langle \widetilde{\mathbf{q}}_{n,j}^{(t)}, \widetilde{\mathbf{k}}_{n,i}^{(t)} \rangle)}{\exp(\langle \widetilde{\mathbf{q}}_{n,j}^{(t)}, \widetilde{\mathbf{k}}_+^{(t)} \rangle) + \sum_{k=2}^{M} \exp(\langle \widetilde{\mathbf{q}}_{n,j}^{(t)}, \widetilde{\mathbf{k}}_{n,k}^{(t)} \rangle)} \right)$$

$$\geq \frac{\eta(\|\boldsymbol{\mu}\|_2 - \tau)_2^2}{NM} \cdot \frac{N}{4} \cdot \left( \frac{1}{2} \pm o(1) \right) \cdot M\left( \frac{1}{M} \pm o(1) \right)$$

$$- \frac{\eta(\|\boldsymbol{\mu}\|\tau + 2\sigma_p\tau\sqrt{2\log(4NM/\delta)} + \tau^2)}{NM} \cdot \frac{N}{4} \cdot \left( \frac{1}{2} \pm o(1) \right) \cdot M(M-1)\left( \frac{1}{M} \pm o(1) \right)$$

$$\geq \frac{\eta\left( (\|\boldsymbol{\mu}\|_2 - \tau)_2^2 - (\|\boldsymbol{\mu}\|\tau + 2\sigma_p\tau\sqrt{2\log(4NM/\delta)} + \tau^2) \right)}{10M}$$

Then by Definition 4 and summing over $T_1$ steps, we have:

$$V_+^{(T_1)} \geq -|V_+^{(0)}| + T_1 \cdot \frac{\eta\left( (\|\boldsymbol{\mu}\|_2 - \tau)_2^2 - (\|\boldsymbol{\mu}\|\tau + 2\sigma_p\tau\sqrt{2\log(4NM/\delta)} + \tau^2) \right)}{10M} \cdot \|\boldsymbol{w}_O\|_2^2$$

$$= -d_h^{-\frac{1}{4}} + T_1 \cdot \frac{\eta\left( (\|\boldsymbol{\mu}\|_2 - \tau)_2^2 - (\|\boldsymbol{\mu}\|\tau + 2\sigma_p\tau\sqrt{2\log(4NM/\delta)} + \tau^2) \right)}{10M} \cdot \|\boldsymbol{w}_O\|_2^2$$

$$(9)$$

Similarly, we have:

$$V_-^{(T_1)} \leq d_h^{-\frac{1}{4}} - T_1 \cdot \frac{\eta\left((\|\boldsymbol{\mu}\|_2 - \tau)_2^2 - (\|\boldsymbol{\mu}\|\tau + 2\sigma_p\tau\sqrt{2\log(4NM/\delta)} + \tau^2)\right)}{10M} \cdot \|\boldsymbol{w}_O\|_2^2 \quad (10)$$

Similarly, by the bound

$$|\rho_{V_{n,i}^{(t+1)}} - \rho_{V_{n,i}^{(t)}}| \leq \frac{2\eta\sigma_p^2 d}{N} + \eta(\|\boldsymbol{\mu}\|\tau + 2\sigma_p\tau\sqrt{2\log(4NM/\delta)} + \tau^2)$$

given in equation (9), we have

$$\begin{aligned}
|V_{n,i}^{(T_1)}| &\leq |V_{n,i}^{(0)}| + T_1 \cdot \left(\frac{2\eta\sigma_p^2 d}{N} + \eta(\|\boldsymbol{\mu}\|\tau + 2\sigma_p\tau\sqrt{2\log(4NM/\delta)} + \tau^2)\right) \cdot \|\boldsymbol{w}_O\|_2^2 \\
&= d_h^{-\frac{1}{4}} + T_1 \cdot \left(\frac{2\eta\sigma_p^2 d}{N} + \eta(\|\boldsymbol{\mu}\|\tau + 2\sigma_p\tau\sqrt{2\log(4NM/\delta)} + \tau^2)\right) \cdot \|\boldsymbol{w}_O\|_2^2
\end{aligned} \quad (11)$$

According to equations (9), (10), and (11), it is easy to verify that

$$V_+^{(T_1)} - 3M \cdot |V_{n,i}^{(T_1)}| \geq 0 \quad \text{and} \quad V_-^{(T_1)} + 3M \cdot |V_{n,i}^{(T_1)}| \leq 0,$$

which completes the proof.

$$\square$$

## F.2 STAGE II

In stage II, $\langle\boldsymbol{q}_+, \boldsymbol{k}_+\rangle, \langle\boldsymbol{q}_{n,i}, \boldsymbol{k}_+\rangle$ grows while $\langle\boldsymbol{q}_+, \boldsymbol{k}_{n,j}\rangle, \langle\boldsymbol{q}_{n,i}, \boldsymbol{k}_{n,j}\rangle$ decreases, resulting in attention focusing more and more on the signals and less on the noises. By the results of stage I, we have the following conditions at the beginning of stage II

$$V_+^{(T_1)} \geq 3M \cdot |V_{n,i}^{(T_1)}|,$$

$$V_-^{(T_1)} \leq -3M \cdot |V_{n,i}^{(T_1)}|,$$

$$|V_+^{(T_1)}|, |V_-^{(T_1)}|, |V_{n,i}^{(T_1)}| = O(d_h^{-\frac{1}{4}}),$$

$$\begin{aligned}
&|\langle\boldsymbol{q}_\pm^{(T_1)}, \boldsymbol{k}_\pm^{(T_1)}\rangle|, |\langle\boldsymbol{q}_{n,i}^{(T_1)}, \boldsymbol{k}_\pm^{(T_1)}\rangle|, |\langle\boldsymbol{q}_\pm^{(T_1)}, \boldsymbol{k}_{n,j}^{(T_1)}\rangle|, |\langle\boldsymbol{q}_{n,i}^{(T_1)}, \boldsymbol{k}_{n',j}^{(T_1)}\rangle| \\
&= O\left(\max\{\|\boldsymbol{\mu}\|_2^2, \sigma_p^2 d\} \cdot \sigma_h^2 \cdot \sqrt{d_h \log(6N^2M^2/\delta)}\right),
\end{aligned}$$

$$\begin{aligned}
&|\langle\boldsymbol{q}_\pm^{(T_1)}, \boldsymbol{q}_\mp^{(T_1)}\rangle|, |\langle\boldsymbol{q}_{n,i}^{(T_1)}, \boldsymbol{q}_\pm^{(T_1)}\rangle|, |\langle\boldsymbol{q}_{n,i}^{(T_1)}, \boldsymbol{q}_{n',j}^{(T_1)}\rangle| \\
&= O\left(\max\{\|\boldsymbol{\mu}\|_2^2, \sigma_p^2 d\} \cdot \sigma_h^2 \cdot \sqrt{d_h \log(6N^2M^2/\delta)}\right),
\end{aligned}$$

$$\begin{aligned}
&|\langle\boldsymbol{k}_\pm^{(T_1)}, \boldsymbol{k}_\pm^{(T_1)}\rangle|, |\langle\boldsymbol{k}_{n,i}^{(T_1)}, \boldsymbol{k}_\pm^{(T_1)}\rangle|, |\langle\boldsymbol{k}_{n,i}^{(T_1)}, \boldsymbol{k}_{n',j}^{(T_1)}\rangle| \\
&= O\left(\max\{\|\boldsymbol{\mu}\|_2^2, \sigma_p^2 d\} \cdot \sigma_h^2 \cdot \sqrt{d_h \log(6N^2M^2/\delta)}\right),
\end{aligned}$$

$$\|\boldsymbol{q}_\pm^{(T_1)}\|_2, \|\boldsymbol{k}_\pm^{(T_1)}\|_2 = \Theta(\|\boldsymbol{\mu}\|_2^2 \sigma_h^2 d_h),$$

$$\|\boldsymbol{q}_{n,i}^{(T_1)}\|_2, \|\boldsymbol{k}_{n,i}^{(T_1)}\|_2 = \Theta(\sigma_p^2 \sigma_h^2 d d_h)$$

for $i, j \in [M]\backslash\{1\}, n, n' \in [N]$.

*Notations. To better characterize the gap between different inner products, we define the following notations:*

- denote $\Lambda_{n,+,j}^{(t)} = \langle \boldsymbol{q}_+^{(t)}, \boldsymbol{k}_+^{(t)} \rangle - \langle \boldsymbol{q}_+^{(t)}, \boldsymbol{k}_{n,j}^{(t)} \rangle, \quad n \in S_+$.

- denote $\Lambda_{n,-,j}^{(t)} = \langle \boldsymbol{q}_-^{(t)}, \boldsymbol{k}_-^{(t)} \rangle - \langle \boldsymbol{q}_-^{(t)}, \boldsymbol{k}_{n,j}^{(t)} \rangle, \quad n \in S_-$.

- denote $\Lambda_{n,i,+,j}^{(t)} = \langle \boldsymbol{q}_{n,i}^{(t)}, \boldsymbol{k}_+^{(t)} \rangle - \langle \boldsymbol{q}_{n,i}^{(t)}, \boldsymbol{k}_{n,j}^{(t)} \rangle, \quad n \in S_+$.

- denote $\Lambda_{n,i,-,j}^{(t)} = \langle \boldsymbol{q}_{n,i}^{(t)}, \boldsymbol{k}_-^{(t)} \rangle - \langle \boldsymbol{q}_{n,i}^{(t)}, \boldsymbol{k}_{n,j}^{(t)} \rangle, \quad n \in S_-$.

**Lemma 19** (Upper bound of V). *Let* $T_0 = O\left( \frac{1}{\eta(\|\boldsymbol{\mu}\|_2+\tau)^2 \|\boldsymbol{w}_O\|_2^2 \log(6N^2M^2/\delta)} \right)$. *Then under the same conditions as Theorem 2, we have*

$$|V_+^{(t)}|, |V_-^{(t)}|, |V_{n,i}^{(t)}| = o(1)$$

*for* $t \in [0, T_0]$.

The proof of Lemma 19 is similar to that of Lemma 16, except that the time $T_0$ is changed. Let $T_2 = \Omega\left( \frac{1}{\eta(\|\boldsymbol{\mu}\|_2+\tau)^2 \|\boldsymbol{w}_O\|_2^2 \log(6N^2M^2/\delta)} \right)$, then by Lemma 19 and Lemma 14 we have $\frac{1}{2} - o(1) \leq -\widetilde{\ell}_n^{\prime(t)} \leq \frac{1}{2} + o(1)$ for $n \in [N], t \in [T_1, T_2]$, which can simplify the calculations of $\alpha$ and $\beta$ defined in Definition 6 by their bounds. Next we prove the following four propositions $\mathcal{B}(t), \mathcal{C}(t), \mathcal{D}(t), \mathcal{E}(t)$ by induction on $t$ for $t \in [T_1, T_2]$:

- $\mathcal{B}(t)$:
$$V_+^{(t)} \geq \eta C_3 (\|\boldsymbol{\mu}\|_2 - \tau)^2 \|\boldsymbol{w}_O\|_2^2 (t - T_1)$$
$$V_- \leq \eta C_3 (\|\boldsymbol{\mu}\|_2 - \tau)^2 \|\boldsymbol{w}_O\|_2^2 (t - T_1)$$
$$V_+^{(t)} \geq 3M \cdot |V_{n,i}^{(t)}|,$$
$$V_-^{(t)} \leq -3M \cdot |V_{n,i}^{(t)}|,$$
$$|V_+^{(t)}| \leq O(d_h^{-\frac{1}{4}}) + \eta C_4 (\|\boldsymbol{\mu}\|_2 + \tau)^2 \|\boldsymbol{w}_O\|_2^2 (t - T_1)$$
$$|V_-^{(t)}| \leq O(d_h^{-\frac{1}{4}}) + \eta C_4 (\|\boldsymbol{\mu}\|_2 + \tau)^2 \|\boldsymbol{w}_O\|_2^2 (t - T_1)$$
for $i \in [M] \backslash \{1\}, n \in [N]$.

- $\mathcal{C}(t)$:
$$\|\boldsymbol{q}_\pm^{(t)}\|_2, \|\boldsymbol{k}_\pm^{(t)}\|_2 = \Theta(\|\boldsymbol{\mu}\|_2^2 \sigma_h^2 d_h),$$
$$\|\boldsymbol{q}_{n,i}^{(t)}\|_2, \|\boldsymbol{k}_{n,i}^{(t)}\|_2 = \Theta(\sigma_p^2 \sigma_h^2 ddd_h),$$
$$|\langle \boldsymbol{q}_+^{(t)}, \boldsymbol{q}_-^{(t)} \rangle|, |\langle \boldsymbol{q}_+^{(t)}, \boldsymbol{q}_{n,i}^{(t)} \rangle|, |\langle \boldsymbol{q}_{n,i}^{(t)}, \boldsymbol{q}_{n',j}^{(t)} \rangle| = o(1),$$
$$|\langle \boldsymbol{k}_+^{(t)}, \boldsymbol{k}_-^{(t)} \rangle|, |\langle \boldsymbol{k}_+^{(t)}, \boldsymbol{k}_{n,i}^{(t)} \rangle|, |\langle \boldsymbol{k}_{n,i}^{(t)}, \boldsymbol{k}_{n',j}^{(t)} \rangle| = o(1),$$
for $i, j \in [M] \backslash \{1\}, n, n' \in [N], i \neq j$ or $n \neq n'$.

- $\mathcal{D}(t)$:
$$\langle \boldsymbol{q}_+^{(t+1)}, \boldsymbol{k}_+^{(t+1)} \rangle \geq \langle \boldsymbol{q}_+^{(t)}, \boldsymbol{k}_+^{(t)} \rangle$$
$$\langle \boldsymbol{q}_{n,i}^{(t+1)}, \boldsymbol{k}_+^{(t+1)} \rangle \geq \langle \boldsymbol{q}_{n,i}^{(t)}, \boldsymbol{k}_+^{(t)} \rangle$$
$$\langle \boldsymbol{q}_+^{(t+1)}, \boldsymbol{k}_{n,j}^{(t+1)} \rangle \leq \langle \boldsymbol{q}_+^{(t)}, \boldsymbol{k}_{n,j}^{(t)} \rangle$$
$$\langle \boldsymbol{q}_{n,i}^{(t+1)}, \boldsymbol{k}_{n,j}^{(t+1)} \rangle \leq \langle \boldsymbol{q}_{n,i}^{(t)}, \boldsymbol{k}_{n,j}^{(t)} \rangle$$

$$\Lambda_{n,\pm,j}^{(t+1)} \geq \log \left( \exp(\Lambda_{n,\pm,j}^{(T_1)}) + \frac{\eta^2 C_8 (\|\boldsymbol{\mu}\|_2 + \tau)^2 \|\boldsymbol{\mu}\|_2^2 \|\boldsymbol{w}_O\|_2^2 d_h^{\frac{1}{2}}}{N (\log(6N^2M^2/\delta))^2} \cdot (t - T_1)(t - T_1 + 1) \right)$$

$$\Lambda_{n,i,\pm,j}^{(t+1)} \geq \log \Bigg( \exp\left( \Lambda_{n,i,\pm,j}^{(T_1)} \right)$$

$$+ \frac{\eta^2 C_8 \left( \sigma_p^2 d + \sigma_p \tau \sqrt{2 \log(4NM/\delta)} + \tau^2 \right) \|\boldsymbol{\mu}\|_2^2 \|\boldsymbol{w}_O\|_2^2 d_h}{N \left( \log(6N^2M^2/\delta) \right)^2} \cdot (t - T_1)(t - T_1 + 1) \Bigg).$$

for $i, j \in [M] \backslash \{1\}, n \in [N]$.

- $\mathcal{E}(t)$:

$$|\langle \boldsymbol{q}_{\pm}^{(t)}, \boldsymbol{k}_{\pm}^{(t)}\rangle|, |\langle \boldsymbol{q}_{n,j}^{(t)}, \boldsymbol{k}_{\pm}^{(t)}\rangle|, |\langle \boldsymbol{q}_{n,i}^{(t)}, \boldsymbol{k}_{n,j}^{(t)}\rangle|, |\langle \boldsymbol{q}_{n,i}^{(t)}, \boldsymbol{k}_{n',j}^{(t)}\rangle| \le \log(d_h^{\frac{1}{4}})$$

$$|\langle \boldsymbol{q}_{\pm}^{(t)}, \boldsymbol{k}_{\pm}^{(t)}\rangle|, |\langle \boldsymbol{q}_{n,i}^{(t)}, \boldsymbol{k}_{n,j}^{(t)}\rangle| = o(1)$$

for $i, j \in [M]\backslash\{1\}, n, \bar{n} \in [N], n \ne \bar{n}$.

By the results of Stage I, we know that $\mathcal{B}(T_1), \mathcal{C}(T_1), \mathcal{E}(T_1)$ are true. To prove that $\mathcal{B}(t), \mathcal{C}(t), \mathcal{D}(t)$ and $\mathcal{E}(t)$ are true in Stage II, we will prove the following claims holds for $t \in [T_1, T_2]$:

**Claim 1.** $\mathcal{D}(T_1), \ldots, \mathcal{D}(t-1), \mathcal{E}(T_1), \ldots, \mathcal{E}(t) \implies \mathcal{B}(t+1)$

**Claim 2.** $\mathcal{B}(T_1), \ldots, \mathcal{B}(t), \mathcal{C}(T_1), \ldots, \mathcal{C}(t), \mathcal{D}(T_1), \ldots, \mathcal{D}(t-1), \mathcal{E}(T_1), \ldots, \mathcal{E}(t) \implies \mathcal{D}(t)$

**Claim 3.** $\mathcal{B}(T_1), \ldots, \mathcal{B}(t), \mathcal{D}(T_1), \ldots, \mathcal{D}(t-1), \mathcal{E}(T_1), \ldots, \mathcal{E}(t) \implies \mathcal{C}(t+1)$

**Claim 4.** $\mathcal{B}(T_1), \ldots, \mathcal{B}(t), \mathcal{C}(T_1), \ldots, \mathcal{C}(t), \mathcal{D}(T_1), \ldots, \mathcal{D}(t-1), \mathcal{E}(T_1), \ldots, \mathcal{E}(t) \implies \mathcal{E}(t+1)$

First, we emphasize that when the perturbation is sufficiently small, the softmax of the inner product $\langle \mathbf{q}, \mathbf{k}\rangle$ remains robust with respect to such perturbations. This is formally established in Lemma 4. Importantly, this lemma enables us to uniformly control the behavior of the softmax function, rather than being restricted to the specific case of $\widetilde{q}^{(t)}$ and $\widetilde{k}^{(t)}$ at iteration $t$.

**Lemma 20.** *Suppose the perturbation satisfies* $\tau \le O(\frac{\|\boldsymbol{\mu}\|_2}{\log d_h})$ *and* $t \ge \Omega\left(\frac{1}{\eta(\|\boldsymbol{\mu}\|_2 + \tau)^2 \|\boldsymbol{w}_O\|_2^2 \log(6N^2 M^2/\delta)}\right)(\mathcal{E}(t)$ *holds). Then there exists a universal constant* $C \le e/2$ *such that*

$$\max_{\widetilde{X} \in B(X, \tau)} softmax\left(\langle \mathbf{q}_+^{(t)}, \mathbf{k}_{n,j}^{(t)}\rangle\right) / \min_{\widetilde{X} \in B(X, \tau)} softmax\left(\langle \mathbf{q}_+^{(t)}, \mathbf{k}_{n,j}^{(t)}\rangle\right) \le C.$$

*Proof.* For any perturbed query-key pair, the softmax weight can be written as

$$softmax\left(\langle \mathbf{q}_+^{(t)}, \mathbf{k}_{n,j}^{(t)}\rangle\right) = \frac{\exp\left(\langle \boldsymbol{q}_+^{(t)}, \boldsymbol{k}_{n,j}^{(t)}\rangle\right)}{\exp\left(\langle \boldsymbol{q}_+^{(t)}, \boldsymbol{k}_+^{(t)}\rangle\right) + \sum_{j'=2}^{M} \exp\left(\langle \boldsymbol{q}_+^{(t)}, \boldsymbol{k}_{n,j'}^{(t)}\rangle\right)}$$

$$= \frac{1}{\exp(\langle \boldsymbol{q}_+^{(t)}, \boldsymbol{k}_+^{(t)}\rangle - \langle \boldsymbol{q}_+^{(t)}, \boldsymbol{k}_{n,j}^{(t)}\rangle) + \sum_{j'=2}^{M} \exp(\langle \boldsymbol{q}_+^{(t)}, \boldsymbol{k}_{n,j'}^{(t)}\rangle - \langle \boldsymbol{q}_+^{(t)}, \boldsymbol{k}_{n,j}^{(t)}\rangle)}. \tag{12}$$

Next, we analyze the difference in the logits. Expanding the perturbation terms yields

$$\begin{aligned}
&\langle \boldsymbol{q}_+^{(t)}, \boldsymbol{k}_+^{(t)}\rangle - \langle \boldsymbol{q}_+^{(t)}, \boldsymbol{k}_{n,j}^{(t)}\rangle \\
&= \langle \boldsymbol{q}_+, \boldsymbol{k}_+ - \boldsymbol{k}_{n,j}\rangle + \langle \tau_+ \mathbf{W}_Q^{(t)}, \boldsymbol{k}_+ - \boldsymbol{k}_{n,j}\rangle \\
&\quad + \langle \boldsymbol{q}_+, \tau_+ \mathbf{W}_K^{(t)} - \tau_{n,j} \mathbf{W}_K^{(t)}\rangle + \langle \tau_+ \mathbf{W}_Q^{(t)}, \tau_+ \mathbf{W}_K^{(t)} - \tau_{n,j} \mathbf{W}_K^{(t)}\rangle \\
&= \langle \boldsymbol{q}_+^{(t)}, \boldsymbol{k}_+^{(t)} - \boldsymbol{k}_{n,j}^{(t)}\rangle \pm o(1).
\end{aligned} \tag{13}$$

The first equality follows from decomposing the perturbed terms, while the second uses the perturbation bound $\tau \le O(\frac{\|\boldsymbol{\mu}\|_2}{\log d_h})$, together with the assumption that the logit magnitudes satisfy

$$|\langle \boldsymbol{q}_{\pm}^{(t)}, \boldsymbol{k}_{\pm}^{(t)}\rangle|, \ |\langle \boldsymbol{q}_{\pm}^{(t)}, \boldsymbol{k}_{n,j}^{(t)}\rangle|, \ |\langle \boldsymbol{q}_{n,i}^{(t)}, \boldsymbol{k}_{\pm}^{(t)}\rangle|, \ |\langle \boldsymbol{q}_{n,i}^{(t)}, \boldsymbol{k}_{n,j}^{(t)}\rangle| \le \log\left(d_h^{1/2}\right).$$

An analogous relation holds for differences involving $\widetilde{\boldsymbol{k}}_{n,j'}^{(t)}$ as well.

Substituting equation 13 into equation 12, we obtain

$$\frac{1}{C} softmax\left(\langle \mathbf{q}_+^{(t)}, \mathbf{k}_{n,j}^{(t)}\rangle\right) \le softmax\left(\langle \mathbf{q}_+^{(t)}, \mathbf{k}_{n,j}^{(t)}\rangle\right) \le C softmax\left(\langle \mathbf{q}_+^{(t)}, \mathbf{k}_{n,j}^{(t)}\rangle\right),$$

for some absolute constant $1 \le C \le e/2$. This establishes the claim.

Thus, in the following discussion, we show that the updates related to $\boldsymbol{q}^{(t)}$ and $\boldsymbol{k}^{(t)}$ during adversarial training can be bounded, while the effect of the perturbation does not accumulate over time. Since $C$ is a very small function, when computing the single-step update, we approximate the perturbed softmax at the previous time step by its clean state. $\qquad\square$

### F.2.1 PROOF OF CLAIM 1

By the results of Stage I, we have

$$|\langle \boldsymbol{q}_{\pm}^{(T_1)}, \boldsymbol{k}_{\pm}^{(T_1)} \rangle|, |\langle \boldsymbol{q}_{\pm}^{(T_1)}, \boldsymbol{k}_{n,j}^{(T_1)} \rangle|, |\langle \boldsymbol{q}_{n,i}^{(T_1)}, \boldsymbol{k}_{\pm}^{(T_1)} \rangle|, |\langle \boldsymbol{q}_{n,i}^{(T_1)}, \boldsymbol{k}_{n,j}^{(T_1)} \rangle| = o(1)$$

Assume that $\mathcal{D}(T_1), \dots, \mathcal{D}(t-1)$ $(t \in [T_1, T_2])$ are true, then $\langle \boldsymbol{q}_{\pm}^{(s)}, \boldsymbol{k}_{\pm}^{(s)} \rangle$, $\langle \boldsymbol{q}_{n,i}^{(s)}, \boldsymbol{k}_{\pm}^{(s)} \rangle$ are monotonically non-decreasing and $\langle \boldsymbol{q}_{\pm}^{(s)}, \boldsymbol{k}_{n,j}^{(s)} \rangle$, $\langle \boldsymbol{q}_{n,i}^{(s)}, \boldsymbol{k}_{n,j}^{(s)} \rangle$ are monotonically non-increasing for $s \in [T_1, t-1]$, so we have

$$\langle \boldsymbol{q}_{\pm}^{(s)}, \boldsymbol{k}_{\pm}^{(s)} \rangle, \langle \boldsymbol{q}_{n,i}^{(s)}, \boldsymbol{k}_{\pm}^{(s)} \rangle \geq -o(1),$$
$$\langle \boldsymbol{q}_{\pm}^{(s)}, \boldsymbol{k}_{n,j}^{(s)} \rangle, \langle \boldsymbol{q}_{n,i}^{(s)}, \boldsymbol{k}_{n,j}^{(s)} \rangle \leq o(1),$$

for $s \in [T_1, t]$. Further we have the lower bounds for the attention on signal $\boldsymbol{\mu}_+$ as follows for $s \in [T_1, t]$:

$$\begin{aligned}
\text{softmax}(\langle \boldsymbol{q}_{\pm}^{(s)}, \boldsymbol{k}_{\pm}^{(s)} \rangle) &\geq \frac{\exp(-o(1))}{\exp(-o(1)) + (M-1)\exp(o(1))} \\
&= \frac{1}{1 + (M-1)\exp(o(1))} \\
&= \frac{1}{1 + (M-1) + (M-1)o(1)} \\
&= \frac{1}{M} - o(1),
\end{aligned} \tag{14}$$

where the second equality is by $\exp(o(1)) = 1 + o(1)$. Similarly, we have

$$\text{softmax}(\langle \boldsymbol{q}_{n,i}^{(s)}, \boldsymbol{k}_{\pm}^{(s)} \rangle) \geq \frac{1}{M} - o(1). \tag{15}$$

Plugging them in the update rule for $\gamma_{V_+}^{(s)}$ shown in Lemma 8 and we have

$$\gamma_{V_+}^{(s+1)} - \gamma_{V_+}^{(s)}$$
$$= -\frac{\eta \langle \widetilde{\boldsymbol{\mu}}_+, \widetilde{\boldsymbol{\mu}}_+^{(t)} \rangle}{NM} \sum_{n \in S_+} \widetilde{\ell}_n^{\prime(t)} \left( \frac{\exp(\langle \boldsymbol{q}_+^{(t)}, \boldsymbol{k}_+^{(t)} \rangle)}{\exp(\langle \boldsymbol{q}_+^{(t)}, \boldsymbol{k}_+^{(t)} \rangle) + \sum_{k=2}^{M} \exp(\langle \boldsymbol{q}_+^{(t)}, \boldsymbol{k}_{n,k}^{(t)} \rangle)} \right.$$
$$\left. + \sum_{j=2}^{M} \frac{\exp(\langle \boldsymbol{q}_{n,j}^{(t)}, \boldsymbol{k}_+^{(t)} \rangle)}{\exp(\langle \boldsymbol{q}_{n,j}^{(t)}, \boldsymbol{k}_+^{(t)} \rangle) + \sum_{k=2}^{M} \exp(\langle \boldsymbol{q}_{n,j}^{(t)}, \boldsymbol{k}_{n,k}^{(t)} \rangle)} \right)$$
$$+ \sum_{n \in S_+} \widetilde{\ell}_n^{\prime(t)} \sum_{i=2}^{M} \frac{-\eta \langle \widetilde{\boldsymbol{\mu}}_+, \widetilde{\boldsymbol{\xi}}_{n,i}^{(t)} \rangle}{NM} \left( \frac{\exp(\langle \boldsymbol{q}_+^{(t)}, \boldsymbol{k}_{n,i}^{(t)} \rangle)}{\exp(\langle \boldsymbol{q}_+^{(t)}, \boldsymbol{k}_+^{(t)} \rangle) + \sum_{k=2}^{M} \exp(\langle \boldsymbol{q}_+^{(t)}, \boldsymbol{k}_{n,k}^{(t)} \rangle)} \right.$$
$$\left. + \sum_{j=2}^{M} \frac{\exp(\langle \boldsymbol{q}_{n,j}^{(t)}, \boldsymbol{k}_{n,i}^{(t)} \rangle)}{\exp(\langle \boldsymbol{q}_{n,j}^{(t)}, \boldsymbol{k}_+^{(t)} \rangle) + \sum_{k=2}^{M} \exp(\langle \boldsymbol{q}_{n,j}^{(t)}, \boldsymbol{k}_{n,k}^{(t)} \rangle)} \right)$$
$$\geq \frac{\eta (\|\boldsymbol{\mu}\|_2 - \tau)_2^2}{NMC} \cdot \frac{N}{4} \cdot \left( \frac{1}{2} \pm o(1) \right) \cdot M \left( \frac{1}{M} \pm o(1) \right)$$
$$- \frac{\eta (\|\boldsymbol{\mu}\|\tau + \sigma_p \tau \sqrt{2\log(4NM/\delta)} + \tau^2)}{NMC} \cdot \frac{N}{4} \cdot \left( \frac{1}{2} \pm o(1) \right) \cdot M(M-1) \left( \frac{1}{M} \pm o(1) \right)$$
$$\geq \frac{\eta \left( (\|\boldsymbol{\mu}\|_2 - \tau)^2 - (\|\boldsymbol{\mu}\|\tau + \sigma_p \tau \sqrt{2\log(4NM/\delta)} + \tau^2) \right)}{10MC} = O\left( \frac{\eta (\|\boldsymbol{\mu}\|_2 - \tau)^2}{10MC} \right),$$

for $s \in [T_1, t]$. The first inequality is by Lemma 4. Then by Definition 4 and taking a summation, we have

$$\begin{aligned}
V_+^{(t+1)} &\geq V_+^{(T_1)} + (t - T_1 + 1) \frac{\eta (\|\boldsymbol{\mu}\|_2 - \tau)^2 \|\boldsymbol{w}_O\|_2^2}{10MC} \\
&\geq \eta C_3 (\|\boldsymbol{\mu}\|_2 - \tau)^2 \|\boldsymbol{w}_O\|_2^2 (t - T_1 + 1),
\end{aligned} \tag{16}$$

where the last inequality is by $V_+^{(T_1)} \geq 0$ and $M = \Theta(1)$. Similarly, we have

$$V_-^{(t+1)} \leq -\eta C_3 (\|\boldsymbol{\mu}\|_2 - \tau)^2 \|\boldsymbol{w}_O\|_2^2 (t - T_1 + 1). \tag{17}$$

By $|\rho_{V_{n,i}^{(t+1)}} - \rho_{V_{n,i}^{(t)}}| \leq \eta(\|\boldsymbol{\mu}\|\tau + \sigma_p \tau \sqrt{2\log(4NM/\delta)} + \tau^2 + \frac{2\sigma_p^2 d}{N})$ in (9) and taking a summation, we have

$$|V_{n,i}^{(t+1)}| \leq |V_{n,i}^{(t)}| + (t - T_1 + 1)\eta(\|\boldsymbol{\mu}\|\tau + \sigma_p \tau \sqrt{2\log(4NM/\delta)} + \tau^2 + \frac{2\sigma_p^2 d}{N})\|\boldsymbol{w}_O\|_2^2. \tag{18}$$

Combining (16) and (18) we have

$$\begin{aligned}
V_+^{(t+1)} &- 3M \cdot |V_{n,i}^{(t+1)}| \\
&\geq V_+^{(T_1)} + (t - T_1 + 1)\frac{\eta(\|\boldsymbol{\mu}\|_2 - \tau)^2}{10M}\|\boldsymbol{w}_O\|_2^2 \\
&\quad - 3M \cdot (|V_{n,i}^{(T_1)}| + (t - T_1 + 1)\eta(\max\{\|\boldsymbol{\mu}\|\tau + \tau^2, \frac{2C_p^2\sigma_p^2 d}{N}\})\|\boldsymbol{w}_O\|_2^2) \\
&\geq V_+^{(T_1)} - 3M \cdot |V_{n,i}^{(T_1)}| + (t - T_1 + 1)\frac{\eta(\|\boldsymbol{\mu}\|_2 - \tau)^2}{10M}\|\boldsymbol{w}_O\|_2^2 \\
&\quad - 3M \cdot (|V_{n,i}^{(T_1)}| + (t - T_1 + 1)\eta(\max\{\|\boldsymbol{\mu}\|\tau + \tau^2, \frac{2C_p^2\sigma_p^2 d}{N}\})\|\boldsymbol{w}_O\|_2^2) \\
&\geq 0,
\end{aligned} \tag{19}$$

where the last inequality is by $V_+^{(T_1)} \geq 3M \cdot |V_{n,i}^{(T_1)}|$ and requires $N \cdot \text{SNR}^2 \geq 60M^2 C_p^2$. The proof for $V_-^{(t+1)} \leq -3M \cdot |V_{n,i}^{(t+1)}|$ is the same.

Next, we prove the upper bound for $V_+$ and $V_{n,i}$. Based on the upper bound of attention($< 1$) and $-\widetilde{\ell}_n' \leq 1$ we have

$$\begin{aligned}
\gamma_{V_+}^{(s+1)} &\leq \gamma_{V_+}^{(s)} - \frac{\eta}{NM} \sum_{n \in S_+} \widetilde{\ell}_n'^{(s)}((\|\boldsymbol{\mu}\|_2 + \tau)^2 + \sum_{j=2}^{M}(\|\boldsymbol{\mu}\|_2 + \tau)^2) \\
&\quad - \frac{\eta}{NM} \sum_{n \in S_+} \widetilde{\ell}_n'^{(s)}(\|\boldsymbol{\mu}\|\tau + \sigma_p \tau \sqrt{2\log(4NM/\delta)} + \tau^2) \cdot M \\
&\leq \gamma_{V_+}^{(s)} + \frac{3\eta(\|\boldsymbol{\mu}\|_2 + \tau)^2}{4} \\
&\leq \gamma_{V_+}^{(s)} + \eta C_4(\|\boldsymbol{\mu}\|_2 + \tau)^2
\end{aligned} \tag{20}$$

Then we can get that

$$\begin{aligned}
|V_+^{(t+1)}| &\leq V_+^{(T_1)} + (\gamma_{V_+}^{(t+1)} - \gamma_{V_+}^{(T_1)})\|\boldsymbol{w}_O\|_2 \\
&\leq V_+^{(T_1)} + \sum_{s=T_1}^{t} \eta C_4(\|\boldsymbol{\mu}\|_2 + \tau)^2 \|\boldsymbol{w}_O\|_2^2 \\
&\leq O(d_h^{-\frac{1}{4}}) + \eta C_4(\|\boldsymbol{\mu}\|_2 + \tau)^2 \|\boldsymbol{w}_O\|_2^2 (t - T_1 + 1)
\end{aligned} \tag{21}$$

where the first inequality is by the monotonicity of $\gamma_{V_+}$ and the definition of $V_+$, the last inequality is by the result of stage 1 where $V_+^{(T_1)} = O(d^{-1})$. Similarly, we have

$$|V_-^{(t+1)}| \leq O(d_h^{-\frac{1}{4}}) + \eta C_4(\|\boldsymbol{\mu}\|_2 + \tau)^2 \|\boldsymbol{w}_O\|_2^2 (t - T_1 + 1) \tag{22}$$

which completes the proof for the upper bound of $V_+$.

Expanding (18) yields

$$
\begin{aligned}
|V_{n,i}^{(t+1)}| &\leq |V_{n,i}^{(T_1)}| + \eta(\max\{\|\boldsymbol{\mu}\|\tau + \tau^2, \frac{2C_p^2\sigma_p^2 d}{N}\}) \cdot \|\boldsymbol{w}_O\|_2^2(t - T_1 + 1) \\
&\leq O(d_h^{-\frac{1}{4}}) + \eta C_4(\|\boldsymbol{\mu}\|_2 + \tau)_2^2\|\boldsymbol{w}_O\|_2(t - T_1 + 1)
\end{aligned}
\tag{23}
$$

where the last inequality is by the result of phase 1 where $|V_{n,i}^{(T_1)}| = O(d_h^{-\frac{1}{4}})$ and the condition that $N \cdot \text{SNR}^2 \geq \Omega(1)$.

### F.2.2 PROOF OF CLAIM 2

By the results of I.6, we have the dynamic of $\langle \boldsymbol{q}, \boldsymbol{k} \rangle$ as follows

$$
\begin{aligned}
&\langle \boldsymbol{q}_+^{(s+1)}, \boldsymbol{k}_+^{(s+1)} \rangle - \langle \boldsymbol{q}_+^{(s)}, \boldsymbol{k}_+^{(s)} \rangle \\
&\geq \frac{\eta^2 C_6(\|\boldsymbol{\mu}\|_2 - \tau)^4\|\boldsymbol{\mu}\|_2^2\|\boldsymbol{w}_O\|_2^2\sigma_h^2 d_h(s - T_1)}{N} \cdot \frac{1}{\exp(\Lambda_{n,+,j}^{(s)})},
\end{aligned}
\tag{24}
$$

$$
\begin{aligned}
&\langle \boldsymbol{q}_-^{(s+1)}, \boldsymbol{k}_-^{(s+1)} \rangle - \langle \boldsymbol{q}_-^{(s)}, \boldsymbol{k}_-^{(s)} \rangle \\
&\geq \frac{\eta^2 C_6(\|\boldsymbol{\mu}\|_2 - \tau)^4\|\boldsymbol{\mu}\|_2^2\|\boldsymbol{w}_O\|_2^2\sigma_h^2 d_h(s - T_1)}{N} \cdot \frac{1}{\exp(\Lambda_{n,-,j}^{(s)})},
\end{aligned}
\tag{25}
$$

$$
\begin{aligned}
&\langle \boldsymbol{q}_+^{(s+1)}, \boldsymbol{k}_{n,j}^{(s+1)} \rangle - \langle \boldsymbol{q}_+^{(s)}, \boldsymbol{k}_{n,j}^{(s)} \rangle \\
&\leq -\frac{\eta^2 C_6(\sigma_p^2 d + \sigma_p\tau\sqrt{2\log(4NM/\delta)} + \tau^2)(\|\boldsymbol{\mu}\|_2 - \tau)^2\|\boldsymbol{\mu}\|_2^2\|\boldsymbol{w}_O\|_2^2\sigma_h^2 d_h(s - T_1)}{N} \cdot \frac{1}{\exp(\Lambda_{n,+,j}^{(s)})},
\end{aligned}
\tag{26}
$$

$$
\begin{aligned}
&\langle \boldsymbol{q}_-^{(s+1)}, \boldsymbol{k}_{n,j}^{(s+1)} \rangle - \langle \boldsymbol{q}_-^{(s)}, \boldsymbol{k}_{n,j}^{(s)} \rangle \\
&\leq -\frac{\eta^2 C_6(\sigma_p^2 d + \sigma_p\tau\sqrt{2\log(4NM/\delta)} + \tau^2)(\|\boldsymbol{\mu}\|_2 - \tau)^2\|\boldsymbol{\mu}\|_2^2\|\boldsymbol{w}_O\|_2^2\sigma_h^2 d_h(s - T_1)}{N} \cdot \frac{1}{\exp(\Lambda_{n,-,j}^{(s)})}
\end{aligned}
\tag{27}
$$

$$
\begin{aligned}
&\langle \boldsymbol{q}_{n,i}^{(s+1)}, \boldsymbol{k}_+^{(s+1)} \rangle - \langle \boldsymbol{q}_{n,i}^{(s)}, \boldsymbol{k}_+^{(s)} \rangle \\
&\geq \frac{\eta^2 C_6(\sigma_p^2 d + \sigma_p\tau\sqrt{2\log(4NM/\delta)} + \tau^2)(\|\boldsymbol{\mu}\|_2 - \tau)^2\|\boldsymbol{\mu}\|_2^2\|\boldsymbol{w}_O\|_2^2\sigma_h^2 d_h(s - T_1)}{N} \cdot \frac{1}{\exp(\Lambda_{n,i,+,j})}
\end{aligned}
\tag{28}
$$

$$
\begin{aligned}
&\langle \boldsymbol{q}_{n,i}^{(s+1)}, \boldsymbol{k}_-^{(s+1)} \rangle - \langle \boldsymbol{q}_{n,i}^{(s)}, \boldsymbol{k}_-^{(s)} \rangle \\
&\geq \frac{\eta^2 C_6(\sigma_p^2 d + \sigma_p\tau\sqrt{2\log(4NM/\delta)} + \tau^2)(\|\boldsymbol{\mu}\|_2 - \tau)^2\|\boldsymbol{\mu}\|_2^2\|\boldsymbol{w}_O\|_2^2\sigma_h^2 d_h(s - T_1)}{N} \cdot \frac{1}{\exp(\Lambda_{n,i,-,j})}
\end{aligned}
\tag{29}
$$

$$
\begin{aligned}
&\langle \boldsymbol{q}_{n,i}^{(s+1)}, \boldsymbol{k}_{n,j}^{(s+1)} \rangle - \langle \boldsymbol{q}_{n,i}^{(s)}, \boldsymbol{k}_{n,j}^{(s)} \rangle \\
&\leq -\frac{\eta^2 C_6(\sigma_p^2 d + \sigma_p\tau\sqrt{2\log(4NM/\delta)} + \tau^2)^2\|\boldsymbol{\mu}\|_2^2\|\boldsymbol{w}_O\|_2^2\sigma_h^2 d_h(s - T_1)}{N} \cdot \frac{1}{\exp(\Lambda_{n,i,\pm,j})}
\end{aligned}
\tag{30}
$$

for $s \in [T_1, t]$. The seven equations above show that $\langle \boldsymbol{q}_\pm^{(s)}, \boldsymbol{k}_\pm^{(s)} \rangle$, $\langle \boldsymbol{q}_{n,i}^{(s)}, \boldsymbol{k}_\pm^{(s)} \rangle$ are monotonically increasing and $\langle \boldsymbol{q}_\pm^{(s)}, \boldsymbol{k}_{n,j}^{(s)} \rangle$, $\langle \boldsymbol{q}_{n,i}^{(s)}, \boldsymbol{k}_{n,j}^{(s)} \rangle$ are monotonically decreasing. Next, we provide the logarithmic increasing lower bounds of $\Lambda_{n,\pm,j}^{(s+1)}$ and $\Lambda_{n,\pm,j}^{(s+1)}$.

We have

$$
\begin{aligned}
\Lambda_{n,+,j}^{(s+1)} - \Lambda_{n,+,j}^{(s)} &= (\langle \boldsymbol{q}_+^{(s+1)}, \boldsymbol{k}_+^{(s+1)} \rangle - \langle \boldsymbol{q}_+^{(s)}, \boldsymbol{k}_+^{(s)} \rangle) - (\langle \boldsymbol{q}_{n,j}^{(s+1)}, \boldsymbol{k}_{n,j}^{(s+1)} \rangle - \langle \boldsymbol{q}_{n,j}^{(s)}, \boldsymbol{k}_{n,j}^{(s)} \rangle) \\
&\geq \frac{\eta^2 C_6 (\|\boldsymbol{\mu}\|_2 - \tau)^4 \|\boldsymbol{\mu}\|_2^2 \|\boldsymbol{w}_O\|_2^2 \sigma_h^2 d_h (s - T_1)}{N} \cdot \frac{1}{\exp(\Lambda_{n,+,j}^{(s)})} \\
&\quad + \frac{\eta^2 C_6 (\sigma_p^2 d + \sigma_p \tau \sqrt{2 \log(4NM/\delta)} + \tau^2)(\|\boldsymbol{\mu}\|_2 - \tau)^2 \|\boldsymbol{\mu}\|_2^2 \|\boldsymbol{w}_O\|_2^2 \sigma_h^2 d_h}{N} (s - T_1)(s - T_1 + 1) \cdot \frac{1}{\exp(\Lambda_{n,+,j}^{(s)})} \\
&\geq \frac{\eta^2 C_7 \max\{\|\boldsymbol{\mu}\|_2^2, \sigma_p^2 d\}(\|\boldsymbol{\mu}\|_2 - \tau)^2 \|\boldsymbol{\mu}\|_2^2 \|\boldsymbol{w}_O\|_2^2 \sigma_h^2 d_h (s - T_1)}{N} \cdot \frac{1}{\exp(\Lambda_{n,+,j}^{(s)})} \\
&\geq \frac{\eta^2 C_7 (\|\boldsymbol{\mu}\|_2 - \tau)^2 \|\boldsymbol{\mu}\|_2^2 \|\boldsymbol{w}_O\|_2^2 d_h^{\frac{1}{2}} (s - T_1)^2}{N(\log(6N^2 M^2/\delta))^2} \cdot \frac{1}{\exp(\Lambda_{n,+,j}^{(s)})}
\end{aligned}
\tag{31}
$$

where the last inequality is by $\sigma_h^2 \geq \min\{\|\boldsymbol{\mu}\|_2^{-2}, (\sigma_p^2 d)^{-1}\} d_h^{-\frac{1}{2}} (\log(6N^2 M^2/\delta))^{-2}$. Multiply both sides simultaneously by $\exp(\Lambda_{n,+,j}^{(s)})$ and get

$$
\exp(\Lambda_{n,+,j}^{(s)})(\Lambda_{n,+,j}^{(s+1)} - \Lambda_{n,+,j}^{(s)}) \geq \frac{\eta^2 C_7 (\|\boldsymbol{\mu}\|_2 - \tau)^2 \|\boldsymbol{\mu}\|_2^2 \|\boldsymbol{w}_O\|_2^2 d_h^{\frac{1}{2}} (s - T_1)}{N(\log(6N^2 M^2/\delta))^2} \cdot \frac{1}{\exp(\Lambda_{n,+,j}^{(s)})}. \tag{32}
$$

Taking a summation from $T_1$ to $t$ and get

$$
\sum_{s=T_1}^{t} \exp(\Lambda_{n,+,j}^{(s)})(\Lambda_{n,+,j}^{(s+1)} - \Lambda_{n,+,j}^{(s)}) \geq \sum_{s=T_1}^{t} \frac{\eta^2 C_7 (\|\boldsymbol{\mu}\|_2 - \tau)^2 \|\boldsymbol{\mu}\|_2^2 \|\boldsymbol{w}_O\|_2^2 d_h^{\frac{1}{2}} (s - T_1)}{N(\log(6N^2 M^2/\delta))^2} \tag{33}
$$

$$
\geq \frac{\eta^2 C_8 (\|\boldsymbol{\mu}\|_2 - \tau)^2 \|\boldsymbol{\mu}\|_2^2 \|\boldsymbol{w}_O\|_2^2 d_h^{\frac{1}{2}}}{N(\log(6N^2 M^2/\delta))^2} \cdot (t - T_1)(t - T_1 + 1). \tag{34}
$$

By the property that $\Lambda_{n,+,j}^{(t)}$ is monotonically increasing, we have

$$
\int_{\Lambda_{n,+,j}^{(T_1)}}^{T_1} \exp(x) dx \geq \sum_{s=T_1}^{t} \exp(\Lambda_{n,+,j}^{(s)})(\Lambda_{n,+,j}^{(s+1)} - \Lambda_{n,+,j}^{(s)}) \geq \frac{\eta^2 C_8 (\|\boldsymbol{\mu}\|_2 - \tau)^2 \|\boldsymbol{\mu}\|_2^2 \|\boldsymbol{w}_O\|_2^2 d_h^{\frac{1}{2}}}{N(\log(6N^2 M^2/\delta))^2} \cdot (t - T_1)(t - T_1 + 1). \tag{35}
$$

By $\int_{\Lambda_{n,+,j}^{(T_1)}}^{\Lambda_{n,+,j}^{(t+1)}} \exp(x) dx = \exp(\Lambda_{n,+,j}^{(t+1)}) - \exp(\Lambda_{n,+,j}^{(T_1)})$ we get

$$
\Lambda_{n,+,j}^{(t+1)} \geq \log\left(\exp(\Lambda_{n,+,j}^{(T_1)}) + \frac{\eta^2 C_8 (\|\boldsymbol{\mu}\|_2 - \tau)^2 \|\boldsymbol{\mu}\|_2^2 \|\boldsymbol{w}_O\|_2^2 d_h^{\frac{1}{2}}}{N(\log(6N^2 M^2/\delta))^2} \cdot (t - T_1)(t - T_1 + 1)\right).
$$

Similarly, we have

$$
\Lambda_{n,-,j}^{(t+1)} \geq \log\left(\exp(\Lambda_{n,-,j}^{(T_1)}) + \frac{\eta^2 C_8 (\|\boldsymbol{\mu}\|_2 - \tau)^2 \|\boldsymbol{\mu}\|_2^2 \|\boldsymbol{w}_O\|_2^2 d_h^{\frac{1}{2}}}{N(\log(6N^2 M^2/\delta))^2} \cdot (t - T_1)(t - T_1 + 1)\right).
$$

Thus, we have

$$
\begin{aligned}
\Lambda_{n,i,+,j}^{(s+1)} - \Lambda_{n,i,+,j}^{(s)} &= \langle (\boldsymbol{q}_{n,i}^{(s+1)}, \boldsymbol{k}_+^{(s+1)}) - (\boldsymbol{q}_{n,i}^{(s)}, \boldsymbol{k}_+^{(s)}) \rangle - \langle (\boldsymbol{q}_{n,i}^{(s+1)}, \boldsymbol{k}_{n,j}^{(s+1)}) - (\boldsymbol{q}_{n,i}^{(s)}, \boldsymbol{k}_{n,j}^{(s)}) \rangle \\
&\geq \frac{\eta^2 C_6 (\sigma_p^2 d + \sigma_p \tau \sqrt{2 \log(4NM/\delta)} + \tau^2)(\|\boldsymbol{\mu}\|_2 - \tau)^2 \|\boldsymbol{\mu}\|_2^2 \|\boldsymbol{w}_O\|_2^2 \sigma_h^2 d_h (s - T_1)}{N} \cdot \frac{1}{\exp(\Lambda_{n,i,+,j}^{(s)})} \\
&\quad + \frac{\eta^2 C_6 (\sigma_p^2 d + \sigma_p \tau \sqrt{2 \log(4NM/\delta)} + \tau^2)^2 \|\boldsymbol{\mu}\|_2^2 \|\boldsymbol{w}_O\|_2^2 \sigma_h^2 d_h (s - T_1)}{N} \cdot \frac{1}{\exp(\Lambda_{n,i,+,j}^{(s)})} \\
&\geq \frac{\eta^2 C_7 \max\{\|\boldsymbol{\mu}\|_2^2, \sigma_p^2 d\}(\sigma_p^2 d + \sigma_p \tau \sqrt{2 \log(4NM/\delta)} + \tau^2) \|\boldsymbol{\mu}\|_2^2 \|\boldsymbol{w}_O\|_2^2 \sigma_h^2 d_h (s - T_1)}{N} \cdot \frac{1}{\exp(\Lambda_{n,i,+,j}^{(s)})} \\
&\geq \frac{\eta^2 C_7 (\sigma_p^2 d + \sigma_p \tau \sqrt{2 \log(4NM/\delta)} + \tau^2) \|\boldsymbol{\mu}\|_2^2 \|\boldsymbol{w}_O\|_2^2 d_h^{\frac{1}{2}} (s - T_1)}{N (\log(6N^2 M^2/\delta))^2} \cdot \frac{1}{\exp(\Lambda_{n,i,+,j}^{(s)})}.
\end{aligned}
\tag{36}
$$

Then using the similar method as for $\Lambda_{n,i,+,j}^{(t)}$, we get

$$
\Lambda_{n,i,+,j}^{(t+1)} \geq \log \left( \exp(\Lambda_{n,i,+,j}^{(T_1)}) + \frac{\eta^2 C_8 (\sigma_p^2 d + \sigma_p \tau \sqrt{2 \log(4NM/\delta)} + \tau^2) \|\boldsymbol{\mu}\|_2^2 \|\boldsymbol{w}_O\|_2^2 d_h^{\frac{1}{2}}}{N (\log(6N^2 M^2/\delta))^2} \cdot (t - T_1)(t - T_1 + 1) \right),
$$

which complete the proof. The proof for Claim 3 is in Section I.9.

### F.2.3 PROOF OF CLAIM 4

By the results of I.7, we have

$$
\langle \boldsymbol{q}_+^{(s+1)}, \boldsymbol{k}_+^{(s+1)} \rangle - \langle \boldsymbol{q}_+^{(s)}, \boldsymbol{k}_+^{(s)} \rangle \leq \frac{\eta C_{10} \|\boldsymbol{\mu}\|_2^2 (\|\boldsymbol{\mu}\|_2 + \tau)^2 \sigma_h^2 d_h}{\exp(\langle \boldsymbol{q}_+^{(s)}, \boldsymbol{k}_+^{(s)} \rangle)}
$$

for $s \in [T_1, t]$. Further we have

$$
\begin{aligned}
\exp(\langle \boldsymbol{q}_+^{(s+1)}, \boldsymbol{k}_+^{(s+1)} \rangle) &\leq \exp \left( \langle \boldsymbol{q}_+^{(s)}, \boldsymbol{k}_+^{(s)} \rangle + \frac{\eta C_{10} \|\boldsymbol{\mu}\|_2^2 (\|\boldsymbol{\mu}\|_2 + \tau)^2 \sigma_h^2 d_h}{\exp(\langle \boldsymbol{q}_+^{(s)}, \boldsymbol{k}_+^{(s)} \rangle)} \right) \\
&= \exp(\langle \boldsymbol{q}_+^{(s)}, \boldsymbol{k}_+^{(s)} \rangle) \cdot \exp \left( \frac{\eta C_{10} \|\boldsymbol{\mu}\|_2^2 (\|\boldsymbol{\mu}\|_2 + \tau)^2 \sigma_h^2 d_h}{\exp(\langle \boldsymbol{q}_+^{(s)}, \boldsymbol{k}_+^{(s)} \rangle)} \right) \tag{37} \\
&\leq C_{11} \exp(\langle \boldsymbol{q}_+^{(s)}, \boldsymbol{k}_+^{(s)} \rangle).
\end{aligned}
$$

For the last inequality, by $\eta \leq \widetilde{O}(\min\{\|\boldsymbol{\mu}\|_2^{-2}, (\sigma_p^2 d)^{-1}\} \cdot d_h^{-\frac{1}{2}})$, $\sigma_h^2 \leq \min\{\|\boldsymbol{\mu}\|_2^{-2}, (\sigma_p^2 d)^{-1}\} d_h^{-\frac{1}{2}} (\log(6N^2 M^2/\delta))^{-\frac{3}{2}}$, $\langle \boldsymbol{q}_+^{(T_1)}, \boldsymbol{k}_+^{(T_1)} \rangle = o(1)$ and the monotonicity of $\langle \boldsymbol{q}_+^{(s)}, \boldsymbol{k}_+^{(s)} \rangle$ for $s \in [T_1, t]$, we have $\exp \left( \frac{\eta C_{10} \|\boldsymbol{\mu}\|_2^2 (\|\boldsymbol{\mu}\|_2 + \tau)^2 \sigma_h^2 d_h}{\exp(\langle \boldsymbol{q}_+^{(s)}, \boldsymbol{k}_+^{(s)} \rangle)} \right) \leq \exp(o(1)) \leq C_{11}$. Multiplying both sides by $\left( \langle \boldsymbol{q}_+^{(s+1)}, \boldsymbol{k}_+^{(s+1)} \rangle - \langle \boldsymbol{q}_+^{(s)}, \boldsymbol{k}_+^{(s)} \rangle \right)$ simultaneously gives

$$
\begin{aligned}
\exp(\langle \boldsymbol{q}_+^{(s+1)}, \boldsymbol{k}_+^{(s+1)} \rangle) &\left( \langle \boldsymbol{q}_+^{(s+1)}, \boldsymbol{k}_+^{(s+1)} \rangle - \langle \boldsymbol{q}_+^{(s)}, \boldsymbol{k}_+^{(s)} \rangle \right) \\
&\leq C_{11} \exp \left( \langle \boldsymbol{q}_+^{(s)}, \boldsymbol{k}_+^{(s)} \rangle \right) \cdot \left( \langle \boldsymbol{q}_+^{(s+1)}, \boldsymbol{k}_+^{(s+1)} \rangle - \langle \boldsymbol{q}_+^{(s)}, \boldsymbol{k}_+^{(s)} \rangle \right) \tag{38} \\
&\leq \eta C_{12} \|\boldsymbol{\mu}\|_2^2 (\|\boldsymbol{\mu}\|_2 + \tau)^2 \sigma_h^2 d_h,
\end{aligned}
$$

where the last inequality is by plugging (37). Taking a summation, we obtain

$$
\int_{\langle \boldsymbol{q}_+^{(T_1)}, \boldsymbol{k}_+^{(T_1)} \rangle}^{\langle \boldsymbol{q}_+^{(t+1)}, \boldsymbol{k}_+^{(t+1)} \rangle} \exp(x) \, dx
$$

$$
\leq \sum_{s=T_1}^{t} \exp(\langle \boldsymbol{q}_+^{(s+1)}, \boldsymbol{k}_+^{(s+1)} \rangle) \left( \langle \boldsymbol{q}_+^{(s+1)}, \boldsymbol{k}_+^{(s+1)} \rangle - \langle \boldsymbol{q}_+^{(s)}, \boldsymbol{k}_+^{(s)} \rangle \right)
$$

$$
\leq \sum_{s=T_1}^{t} \eta C_{12} \|\boldsymbol{\mu}\|_2^2 (\|\boldsymbol{\mu}\|_2 + \tau)^2 \sigma_h^2 d_h \tag{39}
$$

$$
\leq T_2 \cdot \eta C_{12} \|\boldsymbol{\mu}\|_2^2 (\|\boldsymbol{\mu}\|_2 + \tau)^2 \sigma_h^2 d_h
$$

$$
\leq \frac{d_h^{1/2}}{\log^2(6N^2 M^2/\delta)}.
$$

where the first inequality is due to $\langle \boldsymbol{q}_+^{(s)}, \boldsymbol{k}_+^{(s)} \rangle$ is monotone increasing, the last inequality is by $T_2 = \Theta(\eta^{-1} \|\boldsymbol{\mu}\|_2^{-2} \|\boldsymbol{w}_O\|_2^{-2} \log(6N^2 M^2/\delta)^{-1})$ and $\sigma_h^2 \leq \min\{\|\boldsymbol{\mu}\|_2^{-2}, (\sigma_p^2 d)^{-1}\} d_h^{-\frac{1}{2}} (\log(6N^2 M^2/\delta))^{-\frac{3}{2}}$. By

$$
\int_{\langle \boldsymbol{q}_+^{(T_1)}, \boldsymbol{k}_+^{(T_1)} \rangle}^{\langle \boldsymbol{q}_+^{(t+1)}, \boldsymbol{k}_+^{(t+1)} \rangle} \exp(x) dx = \exp(\langle \boldsymbol{q}_+^{(t+1)}, \boldsymbol{k}_+^{(t+1)} \rangle) - \exp(\langle \boldsymbol{q}_+^{(T_1)}, \boldsymbol{k}_+^{(T_1)} \rangle),
$$

we have

$$
\langle \boldsymbol{q}_+^{(t+1)}, \boldsymbol{k}_+^{(t+1)} \rangle \leq \log \left( \langle \boldsymbol{q}_+^{(T_1)}, \boldsymbol{k}_+^{(T_1)} \rangle + \frac{d_h^{\frac{1}{2}}}{\log(6N^2 M^2/\delta)} \right) \leq \log \left( d_h^{\frac{1}{2}} \right), \tag{40}
$$

By the results of I.7, we also have

$$
\langle \boldsymbol{q}_-^{(s+1)}, \boldsymbol{k}_-^{(s+1)} \rangle - \langle \boldsymbol{q}_-^{(s)}, \boldsymbol{k}_-^{(s)} \rangle \leq \frac{\eta C_{10} \|\boldsymbol{\mu}\|_2^2 (\|\boldsymbol{\mu}\|_2 + \tau)^2 \sigma_h^2 d_h}{\exp(\langle \boldsymbol{q}_-^{(s)}, \boldsymbol{k}_-^{(s)} \rangle)} \tag{41}
$$

$$
\langle \boldsymbol{q}_\pm^{(s+1)}, \boldsymbol{k}_{n,j}^{(s+1)} \rangle - \langle \boldsymbol{q}_\pm^{(s)}, \boldsymbol{k}_{n,j}^{(s)} \rangle \geq -\frac{\eta C_{10} \sigma_p^2 d (\|\boldsymbol{\mu}\|_2 + \tau)^2 \sigma_h^2 d_h}{N} \cdot \exp(\langle \boldsymbol{q}_\pm^{(s)}, \boldsymbol{k}_{n,j}^{(s)} \rangle). \tag{42}
$$

$$
\langle \boldsymbol{q}_{n,i}^{(s+1)}, \boldsymbol{k}_\pm^{(s+1)} \rangle - \langle \boldsymbol{q}_{n,i}^{(s)}, \boldsymbol{k}_\pm^{(s)} \rangle \leq \frac{\eta C_{10} \sigma_p^2 d (\|\boldsymbol{\mu}\|_2 + \tau)^2 \sigma_h^2 d_h}{N \exp(\langle \boldsymbol{q}_{n,i}^{(s)}, \boldsymbol{k}_\pm^{(s)} \rangle)}. \tag{43}
$$

$$
\langle \boldsymbol{q}_{n,i}^{(s+1)}, \boldsymbol{k}_{n,j}^{(s+1)} \rangle - \langle \boldsymbol{q}_{n,i}^{(s)}, \boldsymbol{k}_{n,j}^{(s)} \rangle \geq -\frac{\eta C_{10} \sigma_p^2 d (\sigma_p^2 d + \sigma_p \tau \sqrt{2 \log(4NM/\delta)} + \tau^2) \sigma_h^2 d_h}{N} \cdot \exp(\langle \boldsymbol{q}_{n,i}^{(s)}, \boldsymbol{k}_{n,j}^{(s)} \rangle). \tag{44}
$$

Then using the similar method as for $\langle \boldsymbol{q}_+^{(t+1)}, \boldsymbol{k}_+^{(t+1)} \rangle$, we get

$$
\langle \boldsymbol{q}_-^{(t+1)}, \boldsymbol{k}_-^{(t+1)} \rangle \leq \log \left( d_h^{\frac{1}{2}} \right), \tag{45}
$$

$$
\langle \boldsymbol{q}_\pm^{(t+1)}, \boldsymbol{k}_{n,j}^{(t+1)} \rangle \geq -\log \left( d_h^{\frac{1}{2}} \right), \tag{46}
$$

$$
\langle \boldsymbol{q}_{n,i}^{(t+1)}, \boldsymbol{k}_\pm^{(t+1)} \rangle \leq \log \left( d_h^{\frac{1}{2}} \right), \tag{47}
$$

$$
\langle \boldsymbol{q}_{n,i}^{(t+1)}, \boldsymbol{k}_{n,j}^{(t+1)} \rangle \geq -\log \left( d_h^{\frac{1}{2}} \right). \tag{48}
$$

Next we provide the upper bound for $|\langle \boldsymbol{q}_\pm^{(t+1)}, \boldsymbol{k}_\pm^{(t+1)} \rangle|, |\langle \boldsymbol{q}_{n,i}^{(t+1)}, \boldsymbol{k}_{n',j}^{(t+1)} \rangle|$. By the results of I.8, we have

$$
\sum_{s=T_1}^{t} |\alpha_{+,+}^{(s)}|, \sum_{s=T_1}^{t} |\alpha_{-,-}^{(s)}|, \sum_{s=T_1}^{t} |\beta_{+,+}^{(s)}|, \sum_{s=T_1}^{t} |\beta_{-,-}^{(s)}|, \sum_{s=T_1}^{t} |\beta_{n,i,+}^{(s)}|, \sum_{s=T_1}^{t} |\beta_{n,i,-}^{(s)}| = O \left( N^{\frac{1}{2}} d_h^{-\frac{1}{4}} \right), \tag{49}
$$

for $i \in [M]\backslash\{1\}, n \in S_\pm$.

$$\sum_{s=T_1}^{t} |\alpha_{n,+,i}^{(s)}|, \sum_{s=T_1}^{t} |\alpha_{n,-,i}^{(s)}| = O\left(N^{-\frac{1}{2}} d_h^{-\frac{1}{4}}\right), \tag{50}$$

for $i \in [M]\backslash\{1\}, n \in S_\pm$.

$$\sum_{s=T_1}^{t} |\beta_{n,+,i}^{(s)}|, \sum_{s=T_1}^{t} |\beta_{n,-,i}^{(s)}| = O\left(\text{SNR} \cdot N^{-\frac{1}{2}} d_h^{-\frac{1}{4}}\right) \tag{51}$$

for $i \in [M]\backslash\{1\}, n \in S_\pm$.

$$\sum_{s=T_1}^{t} |\alpha_{n,i,n',j}^{(s)}|, \sum_{s=T_1}^{t} |\beta_{n,j,n',i}^{(s)}| = O\left(d^{-\frac{1}{2}} d_h^{-\frac{1}{4}} \log(6N^2 M^2/\delta)\right) \tag{52}$$

for $i, j \in [M]\backslash\{1\}, n, n' \in [N], n \neq n'$. Plugging these into the update rule of $\langle \boldsymbol{q}_\pm^{(t)}, \boldsymbol{k}_\pm^{(t)} \rangle, \langle \boldsymbol{q}_{n,i}^{(t)}, \boldsymbol{k}_{n,j}^{(t)} \rangle$ and assume that propositions $\mathcal{C}(T_1), \ldots, \mathcal{C}(t)$ hold, we have

$$
\begin{aligned}
|\langle \boldsymbol{q}_+^{(t+1)}, \boldsymbol{k}_-^{(t+1)} \rangle| &\leq |\langle \boldsymbol{q}_+^{(T_1)}, \boldsymbol{k}_-^{(T_1)} \rangle| + \sum_{s=T_1}^{t} |\langle \boldsymbol{q}_+^{(s+1)}, \boldsymbol{k}_-^{(s+1)} \rangle - \langle \boldsymbol{q}_+^{(s)}, \boldsymbol{k}_-^{(s)} \rangle| \\
&\leq |\langle \boldsymbol{q}_+^{(T_1)}, \boldsymbol{k}_-^{(T_1)} \rangle| \\
&\quad + \sum_{s=T_1}^{t} \left| \alpha_{+,+}^{(s)} \langle \boldsymbol{k}_+^{(s)}, \boldsymbol{k}_-^{(s)} \rangle + \sum_{n \in S_+} \sum_{i=2}^{M} \alpha_{n,+,i}^{(s)} \langle \boldsymbol{k}_{n,i}^{(s)}, \boldsymbol{k}_-^{(s)} \rangle \right. \\
&\quad + \beta_{-,-}^{(s)} \langle \boldsymbol{q}_+^{(s)}, \boldsymbol{q}_-^{(s)} \rangle + \sum_{n \in S_-} \sum_{i=2}^{M} \beta_{n,-,i}^{(s)} \langle \boldsymbol{q}_{n,i}^{(s)}, \boldsymbol{q}_+^{(s)} \rangle \\
&\quad + \left( \alpha_{+,+}^{(s)} \boldsymbol{k}_+^{(s)} + \sum_{n \in S_+} \sum_{i=2}^{M} \alpha_{n,+,i}^{(s)} \boldsymbol{k}_{n,i}^{(s)} \right) \\
&\quad \left. \cdot \left( \beta_{-,-}^{(s)\top} \boldsymbol{q}_-^{(s)\top} + \sum_{n \in S_-} \sum_{i=2}^{M} \beta_{n,-,i}^{(s)\top} \boldsymbol{q}_{n,i}^{(s)\top} \right) \right| \\
&\leq |\langle \boldsymbol{q}_+^{(T_1)}, \boldsymbol{k}_-^{(T_1)} \rangle| \\
&\quad + \sum_{s=T_1}^{t} |\alpha_{+,+}^{(s)}| |\langle \boldsymbol{k}_+^{(s)}, \boldsymbol{k}_-^{(s)} \rangle| + \sum_{n \in S_+} \sum_{i=2}^{M} \sum_{s=T_1}^{t} |\alpha_{n,+,i}^{(s)}| |\langle \boldsymbol{k}_{n,i}^{(s)}, \boldsymbol{k}_-^{(s)} \rangle| \\
&\quad + \sum_{s=T_1}^{t} |\beta_{-,-}^{(s)}| |\langle \boldsymbol{q}_+^{(s)}, \boldsymbol{q}_-^{(s)} \rangle| + \sum_{n \in S_-} \sum_{i=2}^{M} \sum_{s=T_1}^{t} |\beta_{n,-,i}^{(s)}| |\langle \boldsymbol{q}_{n,i}^{(s)}, \boldsymbol{q}_+^{(s)} \rangle| \\
&\quad + \{\text{lower order term}\} \\
&= |\langle \boldsymbol{q}_+^{(T_1)}, \boldsymbol{k}_-^{(T_1)} \rangle| \\
&\quad + O\left(N^{\frac{1}{2}} d_h^{-\frac{1}{4}}\right) \cdot o(1) + N \cdot M \cdot O\left(N^{-\frac{1}{2}} d_h^{-\frac{1}{4}}\right) \cdot o(1) \\
&\quad + O\left(N^{\frac{1}{2}} d_h^{-\frac{1}{4}}\right) \cdot o(1) + N \cdot M \cdot O\left(\text{SNR} \cdot N^{-\frac{1}{2}} d_h^{-\frac{1}{4}}\right) \cdot o(1) \\
&= |\langle \boldsymbol{q}_+^{(T_1)}, \boldsymbol{k}_-^{(T_1)} \rangle| + o\left(N^{\frac{1}{2}} d_h^{-\frac{1}{4}}\right) + o\left(\text{SNR} \cdot N^{\frac{1}{2}} d_h^{-\frac{1}{4}}\right) \\
&= o(1),
\end{aligned}
\tag{53}
$$

where the first inequality is by triangle inequality, the last equality is by $|\langle \boldsymbol{q}_+^{(T_1)}, \boldsymbol{k}_-^{(T_1)}\rangle| = o(1)$ and $d_h = \widetilde{\Omega}\left(\max\{\mathrm{SNR}^4, \mathrm{SNR}^{-4}\}N^2\epsilon^{-2}\right)$. Similarly we have $|\langle \boldsymbol{q}_-^{(t+1)}, \boldsymbol{k}_+^{(t+1)}\rangle| = o(1)$.

$$
\begin{aligned}
|\langle \boldsymbol{q}_{n,i}^{(t+1)}, \boldsymbol{k}_{\overline{n},j}^{(t+1)}\rangle| &\leq |\langle \boldsymbol{q}_{n,i}^{(T_1)}, \boldsymbol{k}_{\overline{n},j}^{(T_1)}\rangle| + \sum_{s=T_1}^{t}|\langle \boldsymbol{q}_{n,i}^{(s+1)}, \boldsymbol{k}_{\overline{n},j}^{(s+1)}\rangle - \langle \boldsymbol{q}_{n,i}^{(s)}, \boldsymbol{k}_{\overline{n},j}^{(s)}\rangle| \\
&\leq |\langle \boldsymbol{q}_{n,i}^{(T_1)}, \boldsymbol{k}_{\overline{n},j}^{(T_1)}\rangle| \\
&\quad + \sum_{s=T_1}^{t}\Bigg| \alpha_{n,i,+}^{(s)}\langle \boldsymbol{k}_+^{(s)}, \boldsymbol{k}_{\overline{n},j}^{(s)}\rangle + \alpha_{n,i,-}^{(s)}\langle \boldsymbol{k}_-^{(s)}, \boldsymbol{k}_{\overline{n},j}^{(s)}\rangle + \sum_{n'=1}^{N}\sum_{l=2}^{M}\alpha_{n,i,n',l}^{(s)}\langle \boldsymbol{k}_{n',l}^{(s)}, \boldsymbol{k}_{\overline{n},j}^{(s)}\rangle \\
&\qquad + \beta_{\overline{n},j,+}^{(s)}\langle \boldsymbol{q}_+^{(s)}, \boldsymbol{q}_{n,i}^{(s)}\rangle + \beta_{\overline{n},j,-}^{(s)}\langle \boldsymbol{q}_-^{(s)}, \boldsymbol{q}_{n,i}^{(s)}\rangle + \sum_{n'=1}^{N}\sum_{l=2}^{M}\beta_{\overline{n},j,n',l}^{(s)}\langle \boldsymbol{q}_{n',l}^{(s)}, \boldsymbol{q}_{n,i}^{(s)}\rangle \\
&\qquad + \left( \alpha_{n,i,+}^{(s)}\boldsymbol{k}_+^{(s)} + \alpha_{n,i,-}^{(s)}\boldsymbol{k}_-^{(s)} + \sum_{n'=1}^{N}\sum_{l=2}^{M}\alpha_{n,i,n',l}^{(s)}\boldsymbol{k}_{n',l}^{(s)} \right) \\
&\qquad \cdot \left( \beta_{\overline{n},j,+}^{(s)\top}\boldsymbol{q}_+^{(s)\top} + \beta_{\overline{n},j,-}^{(s)\top}\boldsymbol{q}_-^{(s)\top} + \sum_{n'=1}^{N}\sum_{l=2}^{M}\beta_{\overline{n},j,n',l}^{(s)\top}\boldsymbol{q}_{n',l}^{(s)\top} \right)\Bigg| \\
&\leq |\langle \boldsymbol{q}_{n,i}^{(T_1)}, \boldsymbol{k}_{\overline{n},j}^{(T_1)}\rangle| \\
&\quad + \sum_{s=T_1}^{t}|\alpha_{n,i,+}^{(s)}||\langle \boldsymbol{k}_+^{(s)}, \boldsymbol{k}_{\overline{n},j}^{(s)}\rangle| + \sum_{s=T_1}^{t}|\alpha_{n,i,-}^{(s)}||\langle \boldsymbol{k}_-^{(s)}, \boldsymbol{k}_{\overline{n},j}^{(s)}\rangle| \\
&\quad + \sum_{s=T_1}^{t}|\alpha_{n,i,\overline{n},j}^{(s)}||\langle \boldsymbol{k}_{\overline{n},j}^{(s)}, \boldsymbol{k}_{\overline{n},j}^{(s)}\rangle| + \sum_{l=2}^{M}\sum_{s=T_1}^{t}|\alpha_{n,i,n,l}^{(s)}||\langle \boldsymbol{k}_{n,l}^{(s)}, \boldsymbol{k}_{\overline{n},j}^{(s)}\rangle| \\
&\quad + \sum_{n'\neq\overline{n}\wedge(l\neq j\vee n'\neq\overline{n})}\sum_{s=T_1}^{t}\sum_{n'=1}^{N}\sum_{l=2}^{M}|\alpha_{n,i,n',l}^{(s)}||\langle \boldsymbol{k}_{n',l}^{(s)}, \boldsymbol{k}_{\overline{n},j}^{(s)}\rangle| \\
&\quad + \sum_{s=T_1}^{t}|\beta_{\overline{n},j,+}^{(s)}||\langle \boldsymbol{q}_+^{(s)}, \boldsymbol{q}_{n,i}^{(s)}\rangle| + \sum_{s=T_1}^{t}|\beta_{\overline{n},j,-}^{(s)}||\langle \boldsymbol{q}_-^{(s)}, \boldsymbol{q}_{n,i}^{(s)}\rangle| \\
&\quad + \sum_{s=T_1}^{t}|\beta_{\overline{n},j,n,i}^{(s)}||\langle \boldsymbol{q}_{n,i}^{(s)}, \boldsymbol{q}_{n,i}^{(s)}\rangle| + \sum_{l=2}^{M}\sum_{s=T_1}^{t}|\beta_{\overline{n},j,n,l}^{(s)}||\langle \boldsymbol{q}_{n,l}^{(s)}, \boldsymbol{q}_{n,i}^{(s)}\rangle| \\
&\quad + \sum_{n'\neq\overline{n}\wedge(l\neq i\vee n'\neq n)}\sum_{s=T_1}^{t}\sum_{n'=1}^{N}\sum_{l=2}^{M}|\beta_{\overline{n},j,n',l}^{(s)}||\langle \boldsymbol{q}_{n',l}^{(s)}, \boldsymbol{q}_{n,i}^{(s)}\rangle| \\
&\quad + \{\text{lower order term}\} \\
&= |\langle \boldsymbol{q}_{n,i}^{(T_1)}, \boldsymbol{k}_{\overline{n},j}^{(T_1)}\rangle| \\
&\quad + O(d_h^{-\frac{1}{4}})\cdot o(1) + O(d^{-\frac{1}{2}}d_h^{-\frac{1}{4}}\log(6N^2M^2/\delta))\cdot\Theta(\sigma_p^2\sigma_h^2dd_h) \\
&\quad + M\cdot O(d_h^{-\frac{1}{4}})\cdot o(1) + N\cdot M\cdot O(d^{-\frac{1}{2}}d_h^{-\frac{1}{4}}\log(6N^2M^2/\delta))\cdot o(1) \\
&\quad + O(N^{\frac{1}{2}}d_h^{-\frac{1}{4}})\cdot o(1) \\
&= |\langle \boldsymbol{q}_{n,i}^{(T_1)}, \boldsymbol{k}_{\overline{n},j}^{(T_1)}\rangle| + o(d_h^{-\frac{1}{4}}) \\
&\quad + O(d^{-\frac{1}{2}}d_h^{\frac{1}{4}}) + o(Nd^{-\frac{1}{2}}d_h^{-\frac{1}{4}}\log(6N^2M^2/\delta)) \\
&= o(1),
\end{aligned}
$$

$$(54)$$

where the first inequality is by triangle inequality, the second equality is by $\sigma_h^2 \leq \min\{\|\boldsymbol{\mu}\|_2^{-2}, (\sigma_p^2 d)^{-1}\} \cdot d_h^{-\frac{1}{2}} \cdot (\log(6N^2M^2/\delta))^{-\frac{3}{2}}$, the last equality is by $|\langle \boldsymbol{q}_{n,i}^{(T_1)}, \boldsymbol{k}_{\overline{n},j}^{(T_1)} \rangle| = o(1)$, $d = \widetilde{\Omega}(\epsilon^{-2} N^2 d_h)$ and $d_h = \widetilde{\Omega}\left(\max\{\text{SNR}^4, \text{SNR}^{-4}\} N^2 \epsilon^{-2}\right)$.

### F.3 STAGE III

In Stage III, the outputs of ViT grow up and the loss derivatives are no longer at $o(1)$. We will carefully compute the growth rate of $V_\pm$ and $V_{n,i}$ while keeping monitoring the monotonicity of $\langle q, k \rangle$. By substituting $t = T_2 = \Theta\left(\frac{1}{\eta(\|\boldsymbol{\mu}\|_2 + \tau)^2 \|\boldsymbol{w}_O\|_2^2}\right)$ into propositions $\mathcal{B}(t), \mathcal{C}(t), \mathcal{D}(t), \mathcal{E}(t)$ in Stage II, we have the following conditions at the beginning of stage III

$$|V_+^{(T_2)}|, |V_-^{(T_2)}|, |V_{n,i}^{(T_2)}| = o(1),$$

$$V_+^{(T_2)} \geq 3M \cdot |V_{n,i}^{(T_2)}|,$$

$$V_-^{(T_2)} \leq -3M \cdot |V_{n,i}^{(T_2)}|,$$

$$\|\boldsymbol{q}_+^{(T_2)}\|_2^2, \|\boldsymbol{k}_+^{(T_2)}\|_2^2 = \Theta(\|\boldsymbol{\mu}\|_2^2 \sigma_h^2 d_h),$$

$$\|\boldsymbol{q}_{n,i}^{(T_2)}\|_2^2, \|\boldsymbol{k}_{n,i}^{(T_2)}\|_2^2 = \Theta(\sigma_p^2 \sigma_h^2 d d_h),$$

$$|\langle \boldsymbol{q}_+^{(T_2)}, \boldsymbol{q}_-^{(T_2)} \rangle|, |\langle \boldsymbol{q}_+^{(T_2)}, \boldsymbol{q}_{n,i}^{(T_2)} \rangle|, |\langle \boldsymbol{q}_{n,i}^{(T_2)}, \boldsymbol{q}_{n',j}^{(T_2)} \rangle| = o(1),$$

$$|\langle \boldsymbol{k}_+^{(T_2)}, \boldsymbol{k}_-^{(T_2)} \rangle|, |\langle \boldsymbol{k}_+^{(T_2)}, \boldsymbol{k}_{n,i}^{(T_2)} \rangle|, |\langle \boldsymbol{k}_{n,i}^{(T_2)}, \boldsymbol{k}_{n',j}^{(T_2)} \rangle| = o(1),$$

for $i, j \in [M] \setminus \{1\}, n, n' \in [N], i \neq j$ or $n \neq n'$.

$$\Lambda_{n,\pm,j}^{(T_2)} \geq \log\left(\exp(\Lambda_{n,\pm,j}^{(T_1)}) + \Theta\left(\frac{d_h^{\frac{1}{2}}}{N(\log(6N^2M^2/\delta))^3}\right)\right)$$

$$\Lambda_{n,i,\pm,j}^{(T_2)} \geq \log\left(\exp(\Lambda_{n,i,\pm,j}^{(T_1)}) + \Theta\left(\frac{\sigma_p^2 d d_h^{\frac{1}{2}}}{N\|\boldsymbol{\mu}\|_2^2 (\log(6N^2M^2/\delta))^3}\right)\right)$$

$$|\langle \boldsymbol{q}_+^{(T_2)}, \boldsymbol{k}_+^{(T_2)} \rangle|, |\langle \boldsymbol{q}_+^{(T_2)}, \boldsymbol{k}_{n,j}^{(T_2)} \rangle|, |\langle \boldsymbol{q}_{n,i}^{(T_2)}, \boldsymbol{k}_+^{(T_2)} \rangle|, |\langle \boldsymbol{q}_{n,i}^{(T_2)}, \boldsymbol{k}_{n',j}^{(T_2)} \rangle| \leq \log(d_h^{\frac{1}{2}})$$

$$|\langle \boldsymbol{q}_+^{(T_2)}, \boldsymbol{k}_-^{(T_2)} \rangle|, |\langle \boldsymbol{q}_{n,i}^{(T_2)}, \boldsymbol{k}_{\overline{n},j}^{(T_2)} \rangle| = o(1)$$

for $i, j \in [M] \setminus \{1\}, n, \overline{n} \in [N], n \neq \overline{n}$.

Let $T_3 = \Theta\left(\frac{1}{\eta\epsilon(\|\boldsymbol{\mu}\|_2 + \tau)^2 \|\boldsymbol{w}_O\|_2^2}\right)$. Next we prove the following four propositions $\mathcal{F}(t), \mathcal{G}(t), \mathcal{H}(t), \mathcal{I}(t)$ by induction on $t$ for $t \in [T_2, T_3]$:

- $\mathcal{F}(t)$:

$$V_+^{(t)} \geq 3M \cdot |V_{n,i}^{(t)}|,$$

$$V_-^{(t)} \leq -3M \cdot |V_{n,i}^{(t)}|,$$

$$|V_{n,i}^{(t)}| = o(1),$$

$$\log\left(\exp(V_+^{(T_2)}) + \eta C_{17}(\|\boldsymbol{\mu}\|_2 - \tau)^2 \|\boldsymbol{w}_O\|_2^2 (t - T_2)\right) \leq V_+^{(t)} \leq 2\log\left(O\left(\frac{1}{\epsilon}\right)\right),$$

$$-2\log\left(O\left(\frac{1}{\epsilon}\right)\right) \leq V_-^{(t)} \leq -\log\left(\exp(-V_-^{(T_2)}) + \eta C_{17}(\|\boldsymbol{\mu}\|_2 - \tau)^2 \|\boldsymbol{w}_O\|_2^2 (t - T_2)\right)$$

  for $i \in [M] \setminus \{1\}, n \in [N]$.
- $\mathcal{G}(t)$:

$$\|\boldsymbol{q}_\pm^{(t)}\|_2^2, \|\boldsymbol{k}_\pm^{(t)}\|_2^2 = \Theta(\|\boldsymbol{\mu}\|_2^2 \sigma_h^2 d_h),$$

$$\|\boldsymbol{q}_{n,i}^{(t)}\|_2^2, \|\boldsymbol{k}_{n,i}^{(t)}\|_2^2 = \Theta\left(\sigma_p^2 \sigma_h^2 dd_h\right),$$

$$|\langle \boldsymbol{q}_+^{(t)}, \boldsymbol{q}_-^{(t)}\rangle|, |\langle \boldsymbol{q}_\pm^{(t)}, \boldsymbol{q}_{n,i}^{(t)}\rangle|, |\langle \boldsymbol{q}_{n,i}^{(t)}, \boldsymbol{q}_{n',j}^{(t)}\rangle| = o(1),$$

$$|\langle \boldsymbol{k}_+^{(t)}, \boldsymbol{k}_-^{(t)}\rangle|, |\langle \boldsymbol{k}_\pm^{(t)}, \boldsymbol{k}_{n,i}^{(t)}\rangle|, |\langle \boldsymbol{k}_{n,i}^{(t)}, \boldsymbol{k}_{n',j}^{(t)}\rangle| = o(1)$$

for $i, j \in [M]\backslash\{1\}, n, n' \in [N], i \neq j$ or $n \neq n'$.

- $\mathcal{H}(t)$:

$$\langle \boldsymbol{q}_\pm^{(t+1)}, \boldsymbol{k}_\pm^{(t+1)}\rangle \geq \langle \boldsymbol{q}_\pm^{(t)}, \boldsymbol{k}_\pm^{(t)}\rangle,$$

$$\langle \boldsymbol{q}_{n,i}^{(t+1)}, \boldsymbol{k}_\pm^{(t+1)}\rangle \geq \langle \boldsymbol{q}_{n,i}^{(t)}, \boldsymbol{k}_\pm^{(t)}\rangle,$$

$$\langle \boldsymbol{q}_\pm^{(t+1)}, \boldsymbol{k}_{n,j}^{(t+1)}\rangle \leq \langle \boldsymbol{q}_\pm^{(t)}, \boldsymbol{k}_{n,j}^{(t)}\rangle,$$

$$\langle \boldsymbol{q}_{n,i}^{(t+1)}, \boldsymbol{k}_{n,j}^{(t+1)}\rangle \leq \langle \boldsymbol{q}_{n,i}^{(t)}, \boldsymbol{k}_{n,j}^{(t)}\rangle$$

for $i, j \in [M]\backslash\{1\}, n \in [N]$.

- $\mathcal{I}(t)$:

$$|\langle \boldsymbol{q}_\pm^{(t)}, \boldsymbol{k}_\pm^{(t)}\rangle|, |\langle \boldsymbol{q}_\pm^{(t)}, \boldsymbol{k}_{n,j}^{(t)}\rangle|, |\langle \boldsymbol{q}_{n,i}^{(t)}, \boldsymbol{k}_\pm^{(t)}\rangle|, |\langle \boldsymbol{q}_{n,i}^{(t)}, \boldsymbol{k}_{n,j}^{(t)}\rangle| \leq \log(\epsilon^{-1} d_h^{\frac{1}{2}}),$$

$$|\langle \boldsymbol{q}_\pm^{(t)}, \boldsymbol{k}_\mp^{(t)}\rangle|, |\langle \boldsymbol{q}_{n,i}^{(t)}, \boldsymbol{k}_{\overline{n},j}^{(t)}\rangle| = o(1)$$

for $i, j \in [M]\backslash\{1\}, n, n' \in [N], n \neq \overline{n}$.

By the results of Stage II, we know that $\mathcal{F}(T_2), \mathcal{G}(T_2), \mathcal{I}(T_2)$ are true. To prove that $\mathcal{F}(t), \mathcal{G}(t),$ $\mathcal{H}(t)$ and $\mathcal{I}(t)$ are true in stage 3, we will prove the following claims holds for $t \in [T_2, T_3]$:

- Claim 5. $\mathcal{H}(T_2), \ldots, \mathcal{H}(t-1), \mathcal{I}(T_2), \ldots, \mathcal{I}(t) \implies \mathcal{F}(t+1)$
- Claim 6. $\mathcal{F}(t), \mathcal{G}(t), \mathcal{H}(T_2), \ldots, \mathcal{H}(t-1), \mathcal{I}(T_2), \ldots, \mathcal{I}(t-1) \implies \mathcal{H}(t)$
- Claim 7. $\mathcal{F}(T_2), \ldots, \mathcal{F}(t), \mathcal{G}(t), \mathcal{H}(T_2), \ldots, \mathcal{H}(t-1), \mathcal{I}(T_2), \ldots, \mathcal{I}(t) \implies \mathcal{G}(t+1)$
- Claim 8. $\mathcal{F}(T_2), .., \mathcal{F}(t), \mathcal{G}(T_2), .., \mathcal{G}(t), \mathcal{H}(T_2), .., \mathcal{H}(t-1), \mathcal{I}(T_2), .., \mathcal{I}(t) \implies \mathcal{I}(t+1)$

### F.3.1 PROOF OF CLAIM 5

The proofs for $V_+^{(t)} \geq 3M \cdot |V_{n,i}^{(t)}|$ and $V_-^{(t)} \leq -3M \cdot |V_{n,i}^{(t)}|$ are the same as for F.2.1. Based on $\mathcal{H}(T_2), \ldots, \mathcal{H}(t)$ where $\langle \boldsymbol{q}_\pm^{(s)}, \boldsymbol{k}_\pm^{(s)}\rangle$ and $\langle \boldsymbol{q}_{n,i}^{(s)}, \boldsymbol{k}_\pm^{(s)}\rangle$ are monotonically non-decreasing and $\max_j \langle \boldsymbol{q}_\pm^{(s)}, \boldsymbol{k}_{n,j}^{(s)}\rangle, \max_j \langle \boldsymbol{q}_{n,i}^{(s)}, \boldsymbol{k}_{n,j}^{(s)}\rangle$ are monotonically non-increasing for $s \in [T_2, t-1]$, we have

$$\Lambda_{n,\pm,j}^{(s)} \geq \Lambda_{n,\pm,j}^{(T_2)} \geq \log\left(\exp(\Lambda_{n,\pm,j}^{(T_1)}) + \Theta\left(\frac{d_h^{\frac{1}{2}}}{N(\log(6N^2M^2/\delta))^3}\right)\right), \tag{55}$$

$$\Lambda_{n,i,\pm,j}^{(s)} \geq \Lambda_{n,i,\pm,j}^{(T_2)} \geq \log\left(\exp(\Lambda_{n,i,\pm,j}^{(T_1)}) + \Theta\left(\frac{\sigma_p^2 dd_h^{\frac{1}{2}}}{N\|\boldsymbol{\mu}\|_2^2(\log(6N^2M^2/\delta))^3}\right)\right) \tag{56}$$

for $i, j \in [M]\backslash\{1\}, n \in [N], s \in [T_2, t]$. We further get

$$\frac{\exp(\langle \boldsymbol{q}_\pm^{(s)}, \boldsymbol{k}_\pm^{(s)}\rangle)}{\exp(\langle \boldsymbol{q}_\pm^{(s)}, \boldsymbol{k}_\pm^{(s)}\rangle) + \sum_{j'=2}^M \exp(\langle \boldsymbol{q}_\pm^{(s)}, \boldsymbol{k}_{n,j'}^{(s)}\rangle)} \leq \frac{\exp(\langle \boldsymbol{q}_\pm^{(s)}, \boldsymbol{k}_{n,j}^{(s)}\rangle)}{C\exp(\langle \boldsymbol{q}_\pm^{(s)}, \boldsymbol{k}_\pm^{(s)}\rangle)} = \frac{1}{C\exp(\Lambda_{n,\pm,j}^{(s)})}$$

$$\leq \frac{1}{C\exp(\Lambda_{n,\pm,j}^{(T_1)}) + \Theta\left(\frac{d_h^{\frac{1}{2}}}{N(\log(6N^2M^2/\delta))^3}\right)} = O\left(\frac{N(\log(6N^2M^2/\delta))^3}{d_h^{\frac{1}{2}}}\right). \tag{57}$$

For the first inequality, by the monotonicity of $\langle \boldsymbol{q}_\pm^{(s)}, \boldsymbol{k}_\pm^{(s)}\rangle$ ($\langle \boldsymbol{q}_\pm^{(s)}, \boldsymbol{k}_\pm^{(s)}\rangle$ is increasing and $\langle \boldsymbol{q}_\pm^{(s)}, \boldsymbol{k}_{n,j}^{(s)}\rangle$ is decreasing), there exist a constant $C$ such that $C\exp(\langle \boldsymbol{q}_\pm^{(s)}, \boldsymbol{k}_\pm^{(s)}\rangle) \geq \exp(\langle \boldsymbol{q}_\pm^{(s)}, \boldsymbol{k}_\pm^{(s)}\rangle) +$

$\sum_{j'=2}^{M} \exp(\langle \boldsymbol{q}_{\pm}^{(s)}, \boldsymbol{k}_{n,j'}^{(s)} \rangle)$. The second inequality is by plugging 55. Similarly, we have

$$\frac{\exp(\langle \boldsymbol{q}_{n,i}^{(s)}, \boldsymbol{k}_{n,j}^{(s)} \rangle)}{\exp(\langle \boldsymbol{q}_{n,i}^{(s)}, \boldsymbol{k}_{\pm}^{(s)} \rangle) + \sum_{j'=2}^{M} \exp(\langle \boldsymbol{q}_{n,i}^{(s)}, \boldsymbol{k}_{n,j'}^{(s)} \rangle)} \leq \frac{1}{C \exp(\Lambda_{n,i,\pm,j}^{(s)})}$$

$$= O\left( \frac{N \|\boldsymbol{\mu}\|_2^2 (\log(6N^2 M^2/\delta))^3}{\sigma_p^2 d d_h^{\frac{1}{2}}} \right).$$
(58)

Plugging (57) and (58) into the update rule of $\rho_{V,n,i}$ in Lemma 8 and get

$$|\rho_{V,n,i}^{(s+1)} - \rho_{V,n,i}^{(s)}| \leq \frac{\eta}{NM} |\widetilde{\ell}_n^{'(s)}| \cdot \langle \widetilde{\boldsymbol{\xi}}_{n,i}, \widetilde{\boldsymbol{\xi}}_{n,i}^{(t)} \rangle \cdot \left( O\left( \frac{N(\log(6N^2 M^2/\delta))^3}{d_h^{\frac{3}{2}}} \right) + O\left( \frac{N\|\boldsymbol{\mu}\|_2^2(\log(6N^2 M^2/\delta))^3}{\sigma_p^2 d d_h^{\frac{1}{2}}} \right) \right)$$

$$+ \frac{\eta}{NM} \sum_{n' \neq n \vee i \neq i'} |\widetilde{\ell}_{n'}^{'(t)}| \cdot \langle \widetilde{\boldsymbol{\xi}}_{n,i}, \widetilde{\boldsymbol{\xi}}_{n',i'}^{(t)} \rangle \cdot \left( O\left( \frac{N(\log(6N^2 M^2/\delta))^3}{d_h^{\frac{3}{2}}} \right) + O\left( \frac{N\|\boldsymbol{\mu}\|_2^2(\log(6N^2 M^2/\delta))^3}{\sigma_p^2 d d_h^{\frac{1}{2}}} \right) \right) + o(1)$$

$$\leq \frac{3\eta(\sigma_p^2 d + o(1))}{2NM} \left( O\left( \frac{N(\log(6N^2 M^2/\delta))^3}{d_h^{\frac{1}{2}}} \right) + O\left( \frac{N\|\boldsymbol{\mu}\|_2^2(\log(6N^2 M^2/\delta))^3}{\sigma_p^2 d d_h^{\frac{1}{2}}} \right) \right)$$

$$+ \frac{\eta}{NM} NM (2\sigma_p^2 \sqrt{d \log(4N^2 M^2/\delta)} \sqrt{d} + o(1)) \cdot \left( O\left( \frac{N(\log(6N^2 M^2/\delta))^3}{d_h^{\frac{1}{2}}} \right) + O\left( \frac{N\|\boldsymbol{\mu}\|_2^2(\log(6N^2 M^2/\delta))^3}{\sigma_p^2 d d_h^{\frac{1}{2}}} \right) \right)$$

$$\leq \frac{2\eta}{NM} \left( O\left( \frac{N(\sigma_p^2 d + o(1))(\log(6N^2 M^2/\delta))^3}{d_h^{\frac{3}{2}}} \right) + O\left( \frac{N\|\boldsymbol{\mu}\|_2^2(\log(6N^2 M^2/\delta))^3}{d_h^{\frac{3}{2}}} \right) \right)$$

$$= O\left( \frac{\eta(\sigma_p^2 d + o(1))(\log(6N^2 M^2/\delta))^3}{d_h^{\frac{1}{2}}} + \frac{\eta\|\boldsymbol{\mu}\|_2^2(\log(6N^2 M^2/\delta))^3}{d_h^{\frac{1}{2}}} \right)$$
(59)

where the second inequality is by Lemma 13 and $|\widetilde{\ell}_n^{(t)}| \leq 1$. For the last inequality, since $d = \widetilde{\Omega}(\epsilon^{-2} N^2 d_h)$, we have $N \cdot M \cdot 2\sigma_p^2 \sqrt{d \log(4N^2 M^2/\delta)} \leq \frac{1}{2} \sigma_p^2 d$. By Definition 4 and taking a summation we have

$$|V_{n,i}^{(t+1)}| \leq |V_{n,i}^{(T_2)}| + \sum_{s=T_2}^{t} |\rho_{V,n,i}^{(s+1)} - \rho_{V,n,i}^{(s)}| \cdot \|\boldsymbol{w}_O\|_2^2$$

$$\leq |V_{n,i}^{(T_2)}| + T_3 \cdot |\rho_{V,n,i}^{(t+1)} - \rho_{V,n,i}^{(t)}| \cdot \|\boldsymbol{w}_O\|_2^2$$

$$\leq o(1) + \Theta\left( \frac{1}{\eta\epsilon(\|\boldsymbol{\mu}\|_2 + \tau)^2 \|\boldsymbol{w}_O\|_2^2} \right) \cdot O\left( \frac{\eta(\sigma_p^2 d + o(1))(\log(6N^2 M^2/\delta))^3}{d_h^{\frac{1}{2}}} + \frac{\eta\|\boldsymbol{\mu}\|_2^2(\log(6N^2 M^2/\delta))^3}{d_h^{\frac{1}{2}}} \right) \cdot \|\boldsymbol{w}_O\|_2^2$$

$$= o(1) + O\left( \frac{(\sigma_p^2 d + o(1))(\log(6N^2 M^2/\delta))^3}{\epsilon(\|\boldsymbol{\mu}\|_2 + \tau)^2 d_h^{\frac{1}{2}}} + \frac{(\log(6N^2 M^2/\delta))^3}{\epsilon d_h^{\frac{1}{2}}} \right)$$

$$= o(1) + o(1)$$

$$= o(1),$$
(60)

where the first equality is by $N \cdot \mathrm{SNR}^2 \geq \Omega(1)$, the second equality is by $d_h = \widetilde{\Omega}\big(\max\{\mathrm{SNR}^4, \mathrm{SNR}^{-4}\} N^2 \epsilon^{-2}\big)$. Then we have a constant upper bound for the sum of $V_{n,i}$ as follows:

$$\sum_{i \in [M] \setminus \{1\}} |V_{n,i}^{(s)}| = (M-1) \cdot o(1) \leq C_{15},$$

for $n \in [N], s \in [T_2, t]$.

Expanding (57) and (58), we have

$$\frac{\exp(\langle \boldsymbol{q}_{\pm}^{(s)}, \boldsymbol{k}_{n,j}^{(s)} \rangle)}{\exp(\langle \boldsymbol{q}_{\pm}^{(s)}, \boldsymbol{k}_{\pm}^{(s)} \rangle) + \sum_{j'=2}^{M} \exp(\langle \boldsymbol{q}_{\pm}^{(s)}, \boldsymbol{k}_{n,j'}^{(s)} \rangle)} = O\left( \frac{N(\log(6N^2 M^2/\delta))^3}{d_h^{3/2}} \right) = o(1), \quad (61)$$

where the equality is by $d_h = \widetilde{\Omega}\big(\max\{\mathrm{SNR}^4, \mathrm{SNR}^{-4}\} N^2 \epsilon^{-2}\big)$.

Similarly,

$$\frac{\exp(\langle \boldsymbol{q}_{n,i}^{(s)}, \boldsymbol{k}_{n,j}^{(s)}\rangle)}{\exp(\langle \boldsymbol{q}_{n,i}^{(s)}, \boldsymbol{k}_{\pm}^{(s)}\rangle) + \sum_{j'=2}^{M} \exp(\langle \boldsymbol{q}_{n,i}^{(s)}, \boldsymbol{k}_{n,j'}^{(s)}\rangle)} = O\left(\frac{N\|\boldsymbol{\mu}\|_2^2 (\log(6N^2 M^2/\delta))^3}{\sigma_p^2 d\, d_h^{1/2}}\right) = o(1), \quad (62)$$

where the equality is by the same choice of $d_h$.

Then we have

$$\mathrm{softmax}(\langle \boldsymbol{q}_{\pm}^{(s)}, \boldsymbol{k}_{\pm}^{(s)}\rangle) \geq 1 - (M-1)\cdot o(1) \geq 1 - o(1), \quad (63)$$

$$\mathrm{softmax}(\langle \boldsymbol{q}_{n,i}^{(s)}, \boldsymbol{k}_{\pm}^{(s)}\rangle) \geq 1 - (M-1)\cdot o(1) \geq 1 - o(1), \quad (64)$$

for $i \in [M]\backslash\{1\}, n \in [N], s \in [T_2, t]$.

Thus, in the following discussion, we omit the noise-related terms as $o(1)$.

Next we provide the bounds for $-\widetilde{\ell}_n^{'(s)}$. Note that $\ell(z) = \log(1 + \exp(-z))$ and $-\ell'(z) = \exp(-z)/(1 + \exp(-z))$. Without loss of generality, assume $y_n = 1$. We have

$$
\begin{aligned}
-\ell'(f(\widetilde{\boldsymbol{X}}_n, \theta(s))) &= \frac{1}{1 + \exp\Big(\frac{1}{M}\sum_{l=1}^{M} \varphi(\widetilde{\boldsymbol{x}}_{n,l}^{(s)\top}\mathbf{W}_Q^{(s)}\mathbf{W}_K^{(s)\top}(\widetilde{\boldsymbol{X}}_n^{(s)})^\top)\widetilde{\boldsymbol{X}}_n^{(s)}\mathbf{W}_V^{(s)}\boldsymbol{w}_O\Big)} \\
&= \frac{1}{M}\Big(\mathrm{softmax}(\langle \boldsymbol{q}_{\pm}^{(s)}, \boldsymbol{k}_{\pm}^{(s)}\rangle) + \sum_{l=2}^{M}\mathrm{softmax}(\langle \boldsymbol{q}_{n,l}^{(s)}, \boldsymbol{k}_{\pm}^{(s)}\rangle)\Big) \cdot \widetilde{\boldsymbol{\mu}}_{+}^{(s)\top}\mathbf{W}_V^{(s)}\boldsymbol{w}_O \\
&\quad + \sum_{j\in[M]\backslash\{1\}}\Big(\mathrm{softmax}(\langle \boldsymbol{q}_{n,j}^{(s)}, \boldsymbol{k}_{n,j}^{(s)}\rangle) + \sum_{l=2}^{M}\mathrm{softmax}(\langle \boldsymbol{q}_{n,l}^{(s)}, \boldsymbol{k}_{n,j}^{(s)}\rangle)\Big) \cdot \widetilde{\boldsymbol{\xi}}_{n,j}^{(s)\top}\mathbf{W}_V^{(s)}\boldsymbol{w}_O \\
&= \frac{1}{M}\Big(M \cdot (1 - o(1)) \cdot V_{+}^{(s)} + M \cdot o(1) \cdot \sum_{j\in[M]\backslash\{1\}} V_{n,i}^{(s)}\Big) \\
&\geq \tfrac{1}{2}V_{+}^{(s)},
\end{aligned}
\quad (65)
$$

for $s \in [T_2, t]$, where the second equality is by plugging equations above, and the inequality follows from $V_{+}^{(s)} \geq 3M \cdot |V_{n,i}^{(s)}|$.

Similarly, we have

$$\frac{1}{M}\sum_{l=1}^{M} \varphi(\widetilde{\boldsymbol{x}}_{n,l}^{(s)\top}\mathbf{W}_Q^{(s)}\mathbf{W}_K^{(s)\top}(\widetilde{\boldsymbol{X}}_n^{(s)})^\top)\widetilde{\boldsymbol{X}}_n^{(s)}\mathbf{W}_V^{(s)}\boldsymbol{w}_O \leq \max_{i\in[M]\backslash\{1\}}\{V_{+}^{(s)}, V_{n,i}^{(s)}\} = V_{+}^{(s)}. \quad (66)$$

Then we have

$$
\begin{aligned}
-\ell'(f(\widetilde{X}_n, \theta(s))) &= \frac{1}{1 + \exp\Big(\frac{1}{M}\sum_{l=1}^{M} \varphi(\widetilde{x}_{n,l}^{(s)\top}\mathbf{W}_Q^{(s)}\mathbf{W}_K^{(s)\top}(\widetilde{X}_n^{(s)})^\top)\widetilde{X}_n^{(s)}\mathbf{W}_V^{(s)}\boldsymbol{w}_O\Big)} \\
&\geq \frac{1}{1 + \exp(V_{+}^{(s)})} \\
&\geq \frac{C_{16}}{\exp(V_{+}^{(s)})}
\end{aligned}
\quad (67)
$$

where the first inequality is by plugging (66). For the last inequality, note that $V_{+}^{(T_2)} \geq 0$ and $V_{+}^{(s)}$ is monotonically increasing, so there exist a constant $C_{16}$ such that $\frac{1}{1+\exp(V_{+}^{(s)})} \geq \frac{C_{16}}{\exp(V_{+}^{(s)})}$. We also

have the upper bound

$$
\begin{aligned}
-\ell'(f(\boldsymbol{X}_n, \theta(s))) &= \frac{1}{1 + \exp\left(\frac{1}{M} \sum_{l=1}^{M} \varphi(\boldsymbol{x}_{n,l}^\top \mathbf{W}_Q^{(s)} \mathbf{W}_K^{(s)\top}(\boldsymbol{X}_n)^\top) \boldsymbol{X}_n \mathbf{W}_V^{(s)} \boldsymbol{w}_O\right)} \\
&\leq \frac{1}{1 + \exp(V_+^{(s)}/2)} \\
&\leq \frac{1}{\exp(V_+^{(s)}/2)}
\end{aligned}
\tag{68}
$$

Then by the update rule of $\gamma_{V,+}^{(t)}$ and in Lemma 8 and get

$$
\begin{aligned}
\gamma_{V,+}^{(s+1)} - \gamma_{V,+}^{(s)} &= -\eta\langle\widetilde{\boldsymbol{\mu}}_+, \widetilde{\boldsymbol{\mu}}_+^{(s)}\rangle \sum_{n \in S_+} \widetilde{\ell}_n'^{(s)} \left(\frac{\exp(\langle\boldsymbol{q}_+^{(s)}, \boldsymbol{k}_+^{(s)}\rangle)}{\exp(\langle\boldsymbol{q}_+^{(s)}, \boldsymbol{k}_+^{(s)}\rangle) + \sum_{k=2}^{M} \exp(\langle\boldsymbol{q}_+^{(s)}, \boldsymbol{k}_{n,k}^{(s)}\rangle)}\right. \\
&\qquad \left. + \sum_{j=2}^{M} \frac{\exp(\langle\boldsymbol{q}_{n,j}^{(s)}, \boldsymbol{k}_+^{(s)}\rangle)}{\exp(\langle\boldsymbol{q}_{n,j}^{(s)}, \boldsymbol{k}_+^{(s)}\rangle) + \sum_{k=2}^{M} \exp(\langle\boldsymbol{q}_{n,j}^{(s)}, \boldsymbol{k}_{n,k}^{(s)}\rangle)}\right) + o(1) \\
&\geq -\eta(\|\boldsymbol{\mu}\|_2 - \tau)^2 \sum_{n \in S_+} \widetilde{\ell}_n'^{(s)}(M \cdot (1 - o(1))) \\
&\geq \eta(\|\boldsymbol{\mu}\|_2 - \tau)^2 \cdot \frac{N}{4} \cdot (1 - o(1)) \cdot \frac{C_{16}}{\exp(V_+^{(s)})} \\
&\geq \eta C_{17}(\|\boldsymbol{\mu}\|_2 - \tau)_2^2 \frac{1}{\exp(V_+^{(s)})}
\end{aligned}
\tag{69}
$$

where the second inequality is by (67). Then by definition 4, we get

$$
V_+^{(s+1)} - V_+^{(s)} = (\gamma_{V,+}^{(s+1)} - \gamma_{V,+}^{(s)})\|\boldsymbol{w}_O\|_2^2 \geq \frac{\eta C_{17}(\|\boldsymbol{\mu}\|_2 - \tau)^2 \|\boldsymbol{w}_O\|_2^2}{\exp(V_+^{(s)})}
\tag{70}
$$

Multiply both sides simultaneously by $\exp(V_+^{(s)})$ and get

$$
\exp(V_+^{(s)})(V_+^{(s+1)} - V_+^{(s)}) \geq \eta C_{17}(\|\boldsymbol{\mu}\|_2 - \tau)^2 \|\boldsymbol{w}_O\|_2^2
\tag{71}
$$

Taking a summation from $T_2$ to $t$ and get

$$
\begin{aligned}
\sum_{s=T_2}^{t} \exp(V_+^{(s)})(V_+^{(s+1)} - V_+^{(s)}) &\geq \sum_{s=T_2}^{t} \eta C_{17}(\|\boldsymbol{\mu}\|_2 - \tau)^2 \|\boldsymbol{w}_O\|_2^2 \\
&\geq \eta C_{17}(\|\boldsymbol{\mu}\|_2 - \tau)^2 \|\boldsymbol{w}_O\|_2^2 (t - T_2 + 1)
\end{aligned}
\tag{72}
$$

By the property that $V_+^{(s)}$ is monotonically increasing, we have

$$
\begin{aligned}
\int_{V_+^{(T_2)}}^{V_+^{(t+1)}} \exp(x) dx &\geq \sum_{s=T_2}^{t} \exp(V_+^{(s)})(V_+^{(s+1)} - V_+^{(s)}) \\
&\geq \eta C_{17}(\|\boldsymbol{\mu}\|_2 - \tau)^2 \|\boldsymbol{w}_O\|_2^2 (t - T_2 + 1)
\end{aligned}
\tag{73}
$$

By $\int_{V_+^{(T_2)}/2}^{V_+^{(t+1)}/2} \exp(x) dx = \exp(V_+^{(t+1)}/2) - \exp(V_+^{(T_2)}/2)$ we get

$$
\begin{aligned}
V_+^{(t+1)} &\geq \log\left(\exp(V_+^{(T_2)}/2) + \eta C_{17}(\|\boldsymbol{\mu}\|_2 - \tau)^2 \|\boldsymbol{w}_O\|_2^2 (t - T_2 + 1)\right) \\
&\geq \eta C_{17}(\|\boldsymbol{\mu}\|_2 - \tau)^2 \|\boldsymbol{w}_O\|_2^2 (t - T_2 + 1)
\end{aligned}
\tag{74}
$$

Similarly, we have

$$
V_-^{(t+1)} \leq -\log\left(\exp\left(V_-^{(T_2)}\right) + \eta C_{17}(\|\boldsymbol{\mu}\|_2 - \tau)_2^2 \|\boldsymbol{w}_O\|_2^2 (t - T_2 + 1)\right)
\tag{75}
$$

Next we provide upper bounds for $V_+^{(t+1)}$ and $V_-^{(t+1)}$. By the update rule of $\gamma_{V,+}^{(t)}$ and in Lemma 8 we have

$$\gamma_{V,+}^{(s+1)} - \gamma_{V,+}^{(s)} = -\eta\langle\widetilde{\boldsymbol{\mu}}_+, \widetilde{\boldsymbol{\mu}}_+^{(s)}\rangle \sum_{n\in S_+} \widetilde{\ell}_n^{'(s)} \left( \frac{\exp\left(\langle \boldsymbol{q}_+^{(s)}, \boldsymbol{k}_+^{(s)}\rangle\right)}{\exp\left(\langle \boldsymbol{q}_+^{(s)}, \boldsymbol{k}_+^{(s)}\rangle\right) + \sum_{k=2}^M \exp\left(\langle \boldsymbol{q}_+^{(s)}, \boldsymbol{k}_{n,k}^{(s)}\rangle\right)} \right.$$

$$\left. + \sum_{j=2}^M \frac{\exp\left(\langle \boldsymbol{q}_{n,j}^{(s)}, \boldsymbol{k}_+^{(s)}\rangle\right)}{\exp\left(\langle \boldsymbol{q}_{n,j}^{(s)}, \boldsymbol{k}_+^{(s)}\rangle\right) + \sum_{k=2}^M \exp\left(\langle \boldsymbol{q}_{n,j}^{(s)}, \boldsymbol{k}_{n,k}^{(s)}\rangle\right)} \right) + o(1)$$

$$\leq \eta(\|\boldsymbol{\mu}\|_2 + \tau)^2 \sum_{n\in S_+} \left(-\widetilde{\ell}_n^{'(s)}\right) \cdot M$$

$$\leq \eta(\|\boldsymbol{\mu}\|_2 + \tau)^2 \cdot \frac{3N}{4} \cdot \exp\left(-V_+^{(s)}/2\right)$$

$$= \frac{3\eta(\|\boldsymbol{\mu}\|_2 + \tau)^2}{4} \exp\left(-V_+^{(s)}/2\right). \tag{76}$$

where the second inequality is by (68). Then by definition 4, we get

$$V_+^{(s+1)} - V_+^{(s)} = (\gamma_{V,+}^{(s+1)} - \gamma_{V,+}^{(s)})\|\boldsymbol{w}_O\|_2^2$$
$$\leq \frac{3\eta(\|\boldsymbol{\mu}\|_2 + \tau)^2\|\boldsymbol{w}_O\|_2^2}{4\exp(V_+^{(s)}/2)} \tag{77}$$

Further we have

$$\exp(V_+^{(s+1)}/2) \leq \exp(V_+^{(s)}/2 + \frac{3\eta(\|\boldsymbol{\mu}\|_2 + \tau)^2\|\boldsymbol{w}_O\|_2^2}{8\exp(V_+^{(s)}/2)}$$

$$= \exp(V_+^{(s)}/2) \cdot \exp(\frac{3\eta(\|\boldsymbol{\mu}\|_2 + \tau)^2\|\boldsymbol{w}_O\|_2^2}{8\exp(V_+^{(s)}/2)}) \tag{78}$$

$$\leq C_{18}\exp(V_+^{(s)}/2)$$

For the last inequality, by $\eta \leq \widetilde{O}(\min\{\|\boldsymbol{\mu}\|_2^{-2}, (\sigma_p^2 d)^{-1}\} \cdot d_h^{-\frac{1}{2}})$, $V_+^{(T_2)} = \Theta(1)$ and the monotonicity of $V_+^{(s)}$, we have $\exp(\frac{3\eta\|\boldsymbol{\mu}\|_2^2\|\boldsymbol{w}_O\|_2^2}{8\exp(V_+^{(s)}/2)}) \leq C_{18}$. Multiplying both sides by $(V_+^{(s+1)}/2 - V_+^{(s)}/2)$ simultaneously gives

$$\exp(V_+^{(s)})(V_+^{(s+1)}/2 - V_+^{(s)}/2) \leq C_{18}\exp(V_+^{(s)}/2)(V_+^{(s+1)}/2 - V_+^{(s)}/2)$$
$$\leq \frac{3\eta C_{18}(\|\boldsymbol{\mu}\|_2 + \tau)^2\|\boldsymbol{w}_O\|_2^2}{8} \tag{79}$$

where the last inequality is by plugging (78). Taking a summation we have

$$\int_{V_+^{(T_2)}/2}^{V_+^{(t+1)}/2} \exp(x)dx \leq \sum_{s=T_2}^{T_3} \exp(V_+^{(s+1)}/2)(V_+^{(s+1)}/2 - V_+^{(s)}/2)$$

$$\leq \sum_{s=T_2}^{T_3} \frac{3\eta C_{18}(\|\boldsymbol{\mu}\|_2 + \tau)^2\|\boldsymbol{w}_O\|_2^2}{8}$$

$$\leq \Theta\left(\frac{1}{\eta\epsilon(\|\boldsymbol{\mu}\|_2 + \tau)^2\|\boldsymbol{w}_O\|_2^2}\right) \cdot \frac{3\eta C_{18}(\|\boldsymbol{\mu}\|_2 + \tau)^2\|\boldsymbol{w}_O\|_2^2}{8} = O\left(\frac{1}{\epsilon}\right) \tag{80}$$

By $\int_{V_+^{(T_2)}/2}^{V_+^{(t+1)}/2} \exp(x)dx = \exp(V_+^{(t+1)}/2) - \exp(V_+^{(T_2)}/2)$ we have

$$V_+^{(t+1)} \leq 2\log\left(\exp(V_+^{(T_2)}/2) + O\left(\frac{1}{\epsilon}\right)\right) = 2\log\left(O\left(\frac{1}{\epsilon}\right)\right)$$

Similarly, we have

$$V_-^{(t+1)} \geq -2\log\left(O\left(\frac{1}{\epsilon}\right)\right)$$

### F.3.2 PROOF OF CLAIM 6

By $\mathcal{H}(T_2), \ldots, \mathcal{H}(t-1)$, we have $\text{softmax}(\langle \boldsymbol{q}_\pm^{(t)}, \boldsymbol{k}_\pm^{(t)}\rangle), \text{softmax}(\langle \boldsymbol{q}_{n,i}^{(t)}, \boldsymbol{k}_\pm^{(t)}\rangle) = 1 - o(1)$ and $\text{softmax}(\langle \boldsymbol{q}_\pm^{(t)}, \boldsymbol{k}_{n,j}^{(t)}\rangle), \text{softmax}(\langle \boldsymbol{q}_{n,i}^{(t)}, \boldsymbol{k}_{n,j}^{(t)}\rangle) = o(1)$, which have been proved in F.3.1. By the results of I.5, we have the signs of $\alpha$ and $\beta$ as follows:

$$\alpha_{+,+}^{(t)}, \alpha_{-,-}^{(t)}, \beta_{+,+}^{(t)}, \beta_{-,-}^{(t)}, \alpha_{n,i,+}^{(t)}, \alpha_{n,i,-}^{(t)}, \beta_{n,+,i}^{(t)}, \beta_{n,-,i}^{(t)} \geq 0,$$
$$\alpha_{n,+,i}^{(t)}, \alpha_{n,-,i}^{(t)}, \alpha_{n,i,n,j}^{(t)}, \beta_{n,i,+}^{(t)}, \beta_{n,i,-}^{(t)}, \beta_{n,j,n,i}^{(t)} \leq 0.$$

Then combined with $\mathcal{G}(T)$ and we have the dynamics of $\langle \boldsymbol{q}, \boldsymbol{k}\rangle$ as follows:

$$\langle \boldsymbol{q}_+^{(t+1)}, \boldsymbol{k}_+^{(t+1)}\rangle - \langle \boldsymbol{q}_+^{(t)}, \boldsymbol{k}_+^{(t)}\rangle = \alpha_{+,+}^{(t)}\|\boldsymbol{k}_+^{(t)}\|_2^2 + \sum_{n \in S_+}\sum_{i=2}^M \alpha_{n,+,i}^{(t)}\langle \boldsymbol{k}_+^{(t)}, \boldsymbol{k}_{n,i}^{(t)}\rangle$$

$$+ \beta_{+,+}^{(t)}\|\boldsymbol{q}_+^{(t)}\|_2^2 + \sum_{n \in S_+}\sum_{i=2}^M \beta_{n,+,i}^{(t)}\langle \boldsymbol{q}_+^{(t)}, \boldsymbol{q}_{n,i}^{(t)}\rangle$$

$$+ \left(\alpha_{+,+}^{(t)}\boldsymbol{k}_+^{(t)} + \sum_{n \in S_+}\sum_{i=2}^M \alpha_{n,+,i}^{(t)}\boldsymbol{k}_{n,i}^{(t)}\right) \tag{81}$$

$$\cdot \left(\beta_{+,+}^{(t)}\boldsymbol{q}_+^{(t)} + \sum_{n \in S_+}\sum_{i=2}^M \beta_{n,+,i}^{(t)}\boldsymbol{q}_{n,i}^{(t)}\right)^\top$$

$$= \alpha_{+,+}^{(t)}\|\boldsymbol{k}_+^{(t)}\|_2^2 + \beta_{+,+}^{(t)}\|\boldsymbol{q}_+^{(t)}\|_2^2 + \{\text{lower order term}\}$$

$$\geq 0$$

Similarly, we have

$$\langle \boldsymbol{q}_\pm^{(t+1)}, \boldsymbol{k}_\pm^{(t+1)}\rangle - \langle \boldsymbol{q}_\pm^{(t)}, \boldsymbol{k}_\pm^{(t)}\rangle \geq 0,$$
$$\langle \boldsymbol{q}_{n,i}^{(t+1)}, \boldsymbol{k}_\pm^{(t+1)}\rangle - \langle \boldsymbol{q}_{n,i}^{(t)}, \boldsymbol{k}_\pm^{(t)}\rangle \geq, 0$$
$$\langle \boldsymbol{q}_\pm^{(t+1)}, \boldsymbol{k}_{n,j}^{(t+1)}\rangle - \langle \boldsymbol{q}_\pm^{(t)}, \boldsymbol{k}_{n,j}^{(t)}\rangle \leq 0,$$
$$\langle \boldsymbol{q}_{n,i}^{(t+1)}, \boldsymbol{k}_{n,j}^{(t+1)}\rangle - \langle \boldsymbol{q}_{n,i}^{(t)}, \boldsymbol{k}_{n,j}^{(t)}\rangle \leq 0$$

which completes the proof. The proof for Claim 7 is in Section I.12

### F.3.3 PROOF OF CLAIM 8

By the results of I.10, we have

$$\langle \boldsymbol{q}_+^{(t+1)}, \boldsymbol{k}_+^{(t+1)}\rangle - \langle \boldsymbol{q}_+^{(t)}, \boldsymbol{k}_+^{(t)}\rangle \leq \frac{\eta C_{10}(\|\boldsymbol{\mu}\|_2 + \tau)^2\|\boldsymbol{\mu}\|_2^2\sigma_h^2 d_h \log\left(O\left(\frac{1}{\epsilon}\right)\right)}{\exp\left(\langle \boldsymbol{q}_+^{(t)}, \boldsymbol{k}_+^{(t)}\rangle\right)}, \tag{82}$$

Further we have

$$\exp(\langle \boldsymbol{q}_+^{(t+1)}, \boldsymbol{k}_+^{(t+1)}\rangle) \leq \exp\left(\langle \boldsymbol{q}_+^{(t)}, \boldsymbol{k}_+^{(t)}\rangle + \frac{\eta C_{10}(\|\boldsymbol{\mu}\|_2 + \tau)^2\|\boldsymbol{\mu}\|_2^2\sigma_h^2 d_h \log\left(O\left(\frac{1}{\epsilon}\right)\right)}{\exp(\langle \boldsymbol{q}_+^{(t)}, \boldsymbol{k}_+^{(t)}\rangle)}\right)$$

$$= \exp\left(\langle \boldsymbol{q}_+^{(t)}, \boldsymbol{k}_+^{(t)}\rangle\right) \cdot \exp\left(\frac{\eta C_{10}(\|\boldsymbol{\mu}\|_2 + \tau)^2\|\boldsymbol{\mu}\|_2^2\sigma_h^2 d_h \log\left(O\left(\frac{1}{\epsilon}\right)\right)}{\exp(\langle \boldsymbol{q}_+^{(t)}, \boldsymbol{k}_+^{(t)}\rangle)}\right)$$

$$\leq C_{11}\exp\left(\langle \boldsymbol{q}_+^{(t)}, \boldsymbol{k}_+^{(t)}\rangle\right).$$

For the last inequality, by $\eta \leq \widetilde{O}(\min\{\|\boldsymbol{\mu}\|_2^{-2}, (\sigma_p^2 d)^{-1}\} \cdot d_h^{-\frac{1}{2}})$, $\sigma_h^2 \leq \min\{\|\boldsymbol{\mu}\|_2^{-2}, (\sigma_p^2 d)^{-1}\} \cdot d_h^{-\frac{1}{2}} \cdot (\log(6N^2 M^2/\delta))^{-\frac{3}{2}}$, $\langle \boldsymbol{q}_+^{(T_1)}, \boldsymbol{k}_+^{(T_1)} \rangle = o(1)$ and the monotonicity of $\langle \boldsymbol{q}_+^{(s)}, \boldsymbol{k}_+^{(s)} \rangle$ for $s \in [T_1, t]$, we have $\exp\left( \frac{\eta C_{10} \|\boldsymbol{\mu}\|_2^4 \sigma_h^2 d_h \log(O(\frac{1}{\epsilon}))}{\exp(\langle \boldsymbol{q}_+^{(t)}, \boldsymbol{k}_+^{(t)} \rangle)} \right) \leq \exp(o(1)) \leq C_{11}$. Multiplying both sides by $\left( \langle \boldsymbol{q}_+^{(t+1)}, \boldsymbol{k}_+^{(t+1)} \rangle - \langle \boldsymbol{q}_+^{(t)}, \boldsymbol{k}_+^{(t)} \rangle \right)$ simultaneously gives

$$
\begin{aligned}
&\exp(\langle \boldsymbol{q}_+^{(t+1)}, \boldsymbol{k}_+^{(t+1)} \rangle) \left( \langle \boldsymbol{q}_+^{(t+1)}, \boldsymbol{k}_+^{(t+1)} \rangle - \langle \boldsymbol{q}_+^{(t)}, \boldsymbol{k}_+^{(t)} \rangle \right) \\
&\leq C_{11} \exp\left( \langle \boldsymbol{q}_+^{(t)}, \boldsymbol{k}_+^{(t)} \rangle \right) \cdot \left( \langle \boldsymbol{q}_+^{(t+1)}, \boldsymbol{k}_+^{(t+1)} \rangle - \langle \boldsymbol{q}_+^{(t)}, \boldsymbol{k}_+^{(t)} \rangle \right) \\
&\leq \eta C_{12}(\|\boldsymbol{\mu}\|_2 + \tau)^2 \|\boldsymbol{\mu}\|_2^2 \sigma_h^2 d_h \log\left( O\left( \frac{1}{\epsilon} \right) \right),
\end{aligned} \tag{83}
$$

where the last inequality is by plugging (82). Taking a summation we have

$$
\begin{aligned}
\int_{\langle \boldsymbol{q}_+^{(T_2)}, \boldsymbol{k}_+^{(T_2)} \rangle}^{\langle \boldsymbol{q}_+^{(t+1)}, \boldsymbol{k}_+^{(t+1)} \rangle} \exp(x)\,dx &\leq \sum_{s=T_2}^{t} \exp(\langle \boldsymbol{q}_+^{(s+1)}, \boldsymbol{k}_+^{(s+1)} \rangle) \left( \langle \boldsymbol{q}_+^{(s+1)}, \boldsymbol{k}_+^{(s+1)} \rangle - \langle \boldsymbol{q}_+^{(s)}, \boldsymbol{k}_+^{(s)} \rangle \right) \\
&\leq \sum_{s=T_2}^{t} \eta C_{12}(\|\boldsymbol{\mu}\|_2 + \tau)^2 \|\boldsymbol{\mu}\|_2^2 \sigma_h^2 d_h \log\left( O\left( \tfrac{1}{\epsilon} \right) \right) \\
&\leq T_3 \cdot \eta C_{12}(\|\boldsymbol{\mu}\|_2 + \tau)^2 \|\boldsymbol{\mu}\|_2^2 \sigma_h^2 d_h \log\left( O\left( \tfrac{1}{\epsilon} \right) \right) \\
&= O\left( \frac{d_h^{\frac{1}{2}} \log\left( O\left( \tfrac{1}{\epsilon} \right) \right)}{\epsilon(\log(6N^2 M^2/\delta))^{\frac{3}{2}}} \right).
\end{aligned} \tag{84}
$$

where the first inequality is due to $\langle \boldsymbol{q}_+^{(s)}, \boldsymbol{k}_+^{(s)} \rangle$ is monotone increasing, the last equality is by $T_3 = \Theta(\eta^{-1} \epsilon^{-1}(\|\boldsymbol{\mu}\|_2 + \tau)^{-2} \|\boldsymbol{w}_O\|_2^{-2})$, $\|\boldsymbol{w}_O\|_2^2 = \Theta(1)$ and $\sigma_h^2 \leq \min\{\|\boldsymbol{\mu}\|_2^{-2}, (\sigma_p^2 d)^{-1}\} \cdot d_h^{-\frac{1}{2}} \cdot (\log(6N^2 M^2/\delta))^{-\frac{3}{2}}$. By $\int_{\langle \boldsymbol{q}_+^{(T_2)}, \boldsymbol{k}_+^{(T_2)} \rangle}^{\langle \boldsymbol{q}_+^{(t+1)}, \boldsymbol{k}_+^{(t+1)} \rangle} \exp(x)\,dx = \exp(\langle \boldsymbol{q}_+^{(t+1)}, \boldsymbol{k}_+^{(t+1)} \rangle) - \exp(\langle \boldsymbol{q}_+^{(T_2)}, \boldsymbol{k}_+^{(T_2)} \rangle)$, we have

$$
\langle \boldsymbol{q}_+^{(t+1)}, \boldsymbol{k}_+^{(t+1)} \rangle \leq \log\left( \exp(\langle \boldsymbol{q}_+^{(T_2)}, \boldsymbol{k}_+^{(T_2)} \rangle) + O\left( \frac{d_h^{\frac{1}{2}} \log\left( O\left( \tfrac{1}{\epsilon} \right) \right)}{\epsilon(\log(6N^2 M^2/\delta))^{\frac{3}{2}}} \right) \right) \leq \log(\epsilon^{-1} d_h^{\frac{1}{2}})
$$

where the last inequality is by $\langle \boldsymbol{q}_+^{(T_2)}, \boldsymbol{k}_+^{(T_2)} \rangle \leq \log(d_h^{\frac{1}{2}})$. By the results of I.10, we also have

$$
\langle \boldsymbol{q}_-^{(t+1)}, \boldsymbol{k}_-^{(t+1)} \rangle - \langle \boldsymbol{q}_-^{(t)}, \boldsymbol{k}_-^{(t)} \rangle \leq \frac{\eta C_{10} \|\boldsymbol{\mu}\|_2^2 (\|\boldsymbol{\mu}\|_2 + \tau)^2 \sigma_h^2 d_h \log\left( O\left( \tfrac{1}{\epsilon} \right) \right)}{\exp(\langle q-^{(t)}, \boldsymbol{k}_-^{(t)} \rangle)}. \tag{85}
$$

$$
\langle \boldsymbol{q}_\pm^{(s+1)}, \boldsymbol{k}_{n,j}^{(s+1)} \rangle - \langle \boldsymbol{q}_\pm^{(s)}, \boldsymbol{k}_{n,j}^{(s)} \rangle \geq -\frac{\eta C_{10} \sigma_p^2 d(\|\boldsymbol{\mu}\|_2 + \tau)^2 \sigma_h^2 d_h \log\left( O\left( \tfrac{1}{\epsilon} \right) \right)}{N} \cdot \exp(\langle \boldsymbol{q}_\pm^{(s)}, \boldsymbol{k}_{n,j}^{(s)} \rangle). \tag{86}
$$

$$
\langle \boldsymbol{q}_{n,i}^{(s+1)}, \boldsymbol{k}_\pm^{(s+1)} \rangle - \langle \boldsymbol{q}_{n,i}^{(s)}, \boldsymbol{k}_\pm^{(s)} \rangle \leq \frac{\eta C_{10} \sigma_p^2 d(\|\boldsymbol{\mu}\|_2 + \tau)^2 \sigma_h^2 d_h \log\left( O\left( \tfrac{1}{\epsilon} \right) \right)}{N \exp(\langle \boldsymbol{q}_{n,i}^{(s)}, \boldsymbol{k}_\pm^{(s)} \rangle)}. \tag{87}
$$

$$
\begin{aligned}
\langle \boldsymbol{q}_{n,i}^{(s+1)}, \boldsymbol{k}_{n,j}^{(s+1)} \rangle - \langle \boldsymbol{q}_{n,i}^{(s)}, \boldsymbol{k}_{n,j}^{(s)} \rangle &\geq -\frac{\eta C_{10} \sigma_p^2 d(\sigma_p^2 d + \sigma_p \tau \sqrt{2\log(4NM/\delta)} + \tau^2)\sigma_h^2 d_h \log\left( O\left( \tfrac{1}{\epsilon} \right) \right)}{N} \\
&\quad \cdot \exp(\langle \boldsymbol{q}_{n,i}^{(s)}, \boldsymbol{k}_{n,j}^{(s)} \rangle).
\end{aligned} \tag{88}
$$

Then using the similar method as for $\langle \boldsymbol{q}_+^{(t+1)}, \boldsymbol{k}_+^{(t+1)} \rangle$, we get

$$\begin{aligned}
\langle \boldsymbol{q}_-^{(t+1)}, \boldsymbol{k}_-^{(t+1)} \rangle &\leq \log(\epsilon^{-1} d_h^{\frac{1}{2}}), \\
\langle \boldsymbol{q}_\pm^{(t+1)}, \boldsymbol{k}_{n,j}^{(t+1)} \rangle &\geq -\log(\epsilon^{-1} d_h^{\frac{1}{2}}), \\
\langle \boldsymbol{q}_{n,i}^{(t+1)}, \boldsymbol{k}_\pm^{(t+1)} \rangle &\leq \log(\epsilon^{-1} d_h^{\frac{1}{2}}), \\
\langle \boldsymbol{q}_{n,i}^{(t+1)}, \boldsymbol{k}_{n,j}^{(t+1)} \rangle &\geq -\log(\epsilon^{-1} d_h^{\frac{1}{2}}),
\end{aligned} \tag{89}$$

Next we provide the upper bound for $|\langle \boldsymbol{q}_\pm^{(t+1)}, \boldsymbol{k}_\pm^{(t+1)} \rangle|, |\langle \boldsymbol{q}_{n,i}^{(t+1)}, \boldsymbol{k}_{n',j}^{(t+1)} \rangle|$. By the results of I.11, we have

$$\sum_{s=T_2}^t |\beta_{n,+,i}^{(t)}|, \sum_{s=T_2}^t |\beta_{n,-,i}^{(t)}| = O\left( \frac{\mathrm{SNR}^2 (\log(6N^2 M^2/\delta))^3 \log\left(O\left(\frac{1}{\epsilon}\right)\right)}{\epsilon d_h^{\frac{1}{2}}} \right), \tag{90}$$

for $i \in [M]\backslash\{1\}, n \in S_\pm$.

$$\begin{aligned}
&\sum_{s=T_2}^t |\alpha_{+,+}^{(t)}|, \sum_{s=T_2}^t |\alpha_{-,-}^{(t)}|, \sum_{s=T_2}^t |\beta_{+,+}^{(t)}|, \sum_{s=T_2}^t |\beta_{-,-}^{(t)}|, \sum_{s=T_2}^t |\alpha_{n,i,+}^{(t)}|, \sum_{s=T_2}^t |\beta_{n,i,-}^{(t)}| \\
&= O\left( \frac{N(\log(6N^2 M^2/\delta))^3 \log\left(O\left(\frac{1}{\epsilon}\right)\right)}{\epsilon d_h^{\frac{1}{2}}} \right),
\end{aligned} \tag{91}$$

for $i \in [M]\backslash\{1\}, n \in S_\pm$.

$$\begin{aligned}
&\sum_{s=T_2}^t |\alpha_{n,+,i}^{(t)}|, \sum_{s=T_2}^t |\alpha_{n,-,i}^{(t)}|, \sum_{s=T_2}^t |\alpha_{n,i,+}^{(t)}|, \sum_{s=T_2}^t |\alpha_{n,i,-}^{(t)}|, \sum_{s=T_2}^t |\alpha_{n,i,n,j}^{(t)}|, \sum_{s=T_2}^t |\beta_{n,j,n,i}^{(t)}| \\
&= O\left( \frac{(\log(6N^2 M^2/\delta))^3 \log\left(O\left(\frac{1}{\epsilon}\right)\right)}{\epsilon d_h^{\frac{1}{2}}} \right),
\end{aligned} \tag{92}$$

for $i, j \in [M]\backslash\{1\}, n \in S_\pm$.

$$\sum_{s=T_2}^t |\alpha_{n,i,n',j}^{(t)}|, \sum_{s=T_2}^t |\beta_{n,j,n',i}^{(t)}| = O\left( \frac{(\log(6N^2 M^2/\delta))^4 \log\left(O\left(\frac{1}{\epsilon}\right)\right)}{\epsilon d_h^{\frac{1}{2}}} \right) \tag{93}$$

for $i, j \in [M] \backslash \{1\}, n, n' \in [N], n \neq n'$. Plugging these and proposition $\mathcal{G}(t)$ into the update rule of $|\langle \boldsymbol{q}_\pm^{(t)}, \boldsymbol{k}_\mp^{(t)} \rangle|, |\langle \boldsymbol{q}_{n,i}^{(t)}, \boldsymbol{k}_{\bar{n},j}^{(t)} \rangle|$ and get

$$|\langle \boldsymbol{q}_+^{(t+1)}, \boldsymbol{k}_-^{(t+1)} \rangle| \leq |\langle \boldsymbol{q}_+^{(T_2)}, \boldsymbol{k}_-^{(T_2)} \rangle| + \sum_{s=T_2}^{t} |\langle \boldsymbol{q}_+^{(s+1)}, \boldsymbol{k}_-^{(s+1)} \rangle - \langle \boldsymbol{q}_+^{(s)}, \boldsymbol{k}_-^{(s)} \rangle|$$

$$\leq |\langle \boldsymbol{q}_+^{(T_2)}, \boldsymbol{k}_-^{(T_2)} \rangle|$$

$$+ \sum_{s=T_2}^{t} \left| \alpha_{+,+}^{(s)} \langle \boldsymbol{k}_+^{(s)}, \boldsymbol{k}_-^{(s)} \rangle + \sum_{n \in S_+} \sum_{i=2}^{M} \alpha_{n,+,i}^{(s)} \langle \boldsymbol{k}_{n,i}^{(s)}, \boldsymbol{k}_-^{(s)} \rangle \right.$$

$$+ \beta_{-,-}^{(s)} \langle \boldsymbol{q}_+^{(s)}, \boldsymbol{q}_-^{(s)} \rangle + \sum_{n \in S_-} \sum_{i=2}^{M} \beta_{n,-,i}^{(s)} \langle \boldsymbol{q}_{n,i}^{(s)}, \boldsymbol{q}_-^{(s)} \rangle \Bigg|$$

$$+ \left( \alpha_{+,+}^{(s)} \boldsymbol{k}_+^{(s)} + \sum_{n \in S_+} \sum_{i=2}^{M} \alpha_{n,+,i}^{(s)} \boldsymbol{k}_{n,i}^{(s)} \right)$$

$$\cdot \left( \beta_{-,-}^{(s)} \boldsymbol{q}_-^{(s)\top} + \sum_{n \in S_-} \sum_{i=2}^{M} \beta_{n,-,i}^{(s)} \boldsymbol{q}_{n,i}^{(s)\top} \right) \Bigg|$$

$$\leq |\langle \boldsymbol{q}_+^{(T_2)}, \boldsymbol{k}_-^{(T_2)} \rangle|$$

$$+ \sum_{s=T_2}^{t} |\alpha_{+,+}^{(t)}| |\langle \boldsymbol{k}_+^{(t)}, \boldsymbol{k}_-^{(t)} \rangle| + \sum_{n \in S_+} \sum_{i=2}^{M} \sum_{s=T_2}^{t} |\alpha_{n,+,i}^{(t)}| |\langle \boldsymbol{k}_{n,i}^{(t)}, \boldsymbol{k}_-^{(t)} \rangle|$$

$$+ \sum_{s=T_2}^{t} |\beta_{-,-}^{(t)}| |\langle \boldsymbol{q}_+^{(t)}, \boldsymbol{q}_-^{(t)} \rangle| + \sum_{n \in S_-} \sum_{i=2}^{M} \sum_{s=T_2}^{t} |\beta_{n,-,i}^{(t)}| |\langle \boldsymbol{q}_{n,i}^{(t)}, \boldsymbol{q}_+^{(t)} \rangle|$$

$$+ \{\text{lower order term}\}$$

$$= |\langle \boldsymbol{q}_+^{(T_2)}, \boldsymbol{k}_-^{(T_2)} \rangle|$$

$$+ O \left( \frac{N (\log(6N^2 M^2/\delta))^3 \log \left( O \left( \frac{1}{\epsilon} \right) \right)}{\epsilon d_h^{\frac{1}{2}}} \right) \cdot o(1) + N \cdot M \cdot O \left( \frac{\text{SNR}^2 (\log(6N^2 M^2/\delta))^3 \log \left( O \left( \frac{1}{\epsilon} \right) \right)}{\epsilon d_h^{\frac{1}{2}}} \right) \cdot o(1)$$

$$+ O \left( \frac{N (\log(6N^2 M^2/\delta))^3 \log \left( O \left( \frac{1}{\epsilon} \right) \right)}{\epsilon d_h^{\frac{1}{2}}} \right) \cdot o(1) + N \cdot M \cdot O \left( \frac{\text{SNR}^2 (\log(6N^2 M^2/\delta))^3 \log \left( O \left( \frac{1}{\epsilon} \right) \right)}{\epsilon d_h^{\frac{1}{2}}} \right) \cdot o(1)$$

$$= |\langle \boldsymbol{q}_+^{(T_2)}, \boldsymbol{k}_-^{(T_2)} \rangle| + o \left( \frac{N (\log(6N^2 M^2/\delta))^3 \log \left( O \left( \frac{1}{\epsilon} \right) \right)}{\epsilon d_h^{\frac{1}{2}}} \right) + o \left( \frac{N \cdot \text{SNR}^2 (\log(6N^2 M^2/\delta))^3 \log \left( O \left( \frac{1}{\epsilon} \right) \right)}{\epsilon d_h^{\frac{1}{2}}} \right)$$

$$= o(1), \tag{94}$$

where the first inequality is by triangle inequality, the second inequality is by results in D.2, the last equality is by $|\langle \boldsymbol{q}_+^{(T_2)}, \boldsymbol{k}_-^{(T_2)} \rangle| = o(1)$ and $d_h = \widetilde{\Omega} \left( \max\{\text{SNR}^4, \text{SNR}^{-4}\} N^2 \epsilon^{-2} \right)$. Similarly we have $|\langle \boldsymbol{q}_-^{(t+1)}, \boldsymbol{k}_+^{(t+1)} \rangle| = o(1)$ and $|\langle \boldsymbol{q}_{n,i}^{(t+1)}, \boldsymbol{k}_{\bar{n},j}^{(t+1)} \rangle| = o(1)$.

**Lemma 21** (Convergence of Training Loss, Lemma D.7 in (Jiang et al., 2024))**.** *There exist* $T = \frac{C_{19}}{\eta \epsilon (\|\boldsymbol{\mu}\|_2 + \tau)^2 \|\boldsymbol{w}_O\|_2^2}$ *such that*

$$L_S(\theta(T)) \leq \epsilon \tag{95}$$

*Proof.* As we have the same conditions at the end of stage III as (Jiang et al., 2024), thus we have: Substituting $t = T = \frac{C_{19}}{\eta \epsilon (\|\boldsymbol{\mu}\|_2 + \tau)^2 \|\boldsymbol{w}_O\|_2^2}$ into propositions $\mathcal{F}(t)$ and get

$$V_+^{(t)} \geq \log \left( \exp(V_+^{(T_2)}) + \eta C_{17} (\|\boldsymbol{\mu}\|_2 - \tau)^2 \|\boldsymbol{w}_O\|_2^2 (t - T_2) \right)$$

$$\geq \log\left(\exp(V_+^{(T_2)}) + \frac{C_{20}}{\epsilon}\right)$$

$$\geq \log\left(\frac{C_{20}}{\epsilon}\right),$$

$$|V_{n,i}^{(t)}| = O(1).$$

Thus, for $n \in S_+$, we bound $f(\widetilde{\boldsymbol{X}}_n, \theta(t))$ as follows :

$$f(\widetilde{\boldsymbol{X}}_n, \theta) = \frac{1}{M}\sum_{l=1}^{M}\varphi(\widetilde{\boldsymbol{x}}_{n,l}^\top \mathbf{W}_Q \mathbf{W}_K^\top (\widetilde{\boldsymbol{X}}_n)^\top)\widetilde{\boldsymbol{X}}_n \mathbf{W}_V \boldsymbol{w}_O \geq \log\left(\frac{1}{\epsilon}\right) \tag{96}$$

And

$$\widetilde{\ell}_n^{(t)} = \log\left(1 + \exp(-f(\widetilde{\boldsymbol{X}}_n, \theta(t)))\right)$$

$$\leq \exp(-f(\widetilde{\boldsymbol{X}}_n, \theta(t)))$$

$$\leq \exp\left(-\log\left(\frac{1}{\epsilon}\right)\right)$$

$$\leq \epsilon.$$

Similarly, we have $\widetilde{\ell}_n^{(t)} \leq \epsilon$ for $n \in S_-$. Therefore, we have

$$L_S(\theta(T)) = \frac{1}{N}\sum_{n=1}^{N}\widetilde{\ell}_n^{(t)} \leq \epsilon.$$

$\square$

## F.4 TEST ERROR

In this section, we denote clean $V_+, V_-$ and $V_{\boldsymbol{\xi}}$ as $\boldsymbol{\mu}_+^\top \mathbf{W}_V \boldsymbol{w}_O, \boldsymbol{\mu}_-^\top \mathbf{W}_V \boldsymbol{w}_O$ and $\boldsymbol{\xi}^\top \mathbf{W}_V \boldsymbol{w}_O$, and perturbed $\widetilde{V}_+, \widetilde{V}_-$ and $\widetilde{V}_{\boldsymbol{\xi}}$ as $\widetilde{\boldsymbol{\mu}}_+^\top \mathbf{W}_V \boldsymbol{w}_O, \widetilde{\boldsymbol{\mu}}_-^\top \mathbf{W}_V \boldsymbol{w}_O$ and $\widetilde{\boldsymbol{\xi}}^\top \mathbf{W}_V \boldsymbol{w}_O$.

### F.4.1 CLEAN TEST ERROR

**Theorem 22.** *Under Assumption 1, in the theoretical analysis of test error in the second and third stages of benign overfitting, we define $g(\boldsymbol{\xi})$ as $V_{\boldsymbol{\xi}}^{(t)} = \left\langle \boldsymbol{\xi}, \mathbf{W}_V^{(t)} \boldsymbol{w}_O \right\rangle$. Then, we know that for any $x \geq 0$, if $g : \mathbb{R}^n \to \mathbb{R}$ is a Lipschitz function and $c$ is a constant, the following inequality holds for the test loss.*

$$\mathbb{P}(\sum_{j=2}^{M}\alpha_i(g(\boldsymbol{\xi}_j) - \mathbb{E}g(\boldsymbol{\xi}_j)) \geq x) \leq \exp\left(-\frac{cx^2}{\sigma_p^2(\sum_{j=2}^{M}\alpha_j^2)\left\|\mathbf{W}_V^{(t)}\boldsymbol{w}_O\right\|_2^2}\right).$$

*Proof.* According to Theorem 5.2.2 in Vershynin (2018), we know that for any $x \geq 0$, if $g : \mathbb{R}^n \to \mathbb{R}$ is a Lipschitz function, it holds that

$$\mathbb{P}(\sum_{i=1}^{N}\alpha_i(g(\boldsymbol{\xi}_i) - \mathbb{E}g(\boldsymbol{\xi}_i)) \geq x) \leq \exp\left(-\frac{cx^2}{\sigma_p^2(\sum_{i=1}^{N}\alpha_i^2)\|g\|_{\text{Lip}}^2}\right) \tag{97}$$

where $g(\boldsymbol{\xi})$ is defined as $|V_{\boldsymbol{\xi}}^{(t)}| = |\langle \boldsymbol{\xi}, \mathbf{W}_V^{(t)} \boldsymbol{w}_O \rangle|$, we have

$$|g(\boldsymbol{\xi}) - g(\boldsymbol{\xi}')| = \left|\left|\langle \boldsymbol{\xi}, \mathbf{W}_V^{(t)} \boldsymbol{w}_O \rangle\right| - \left|\langle \boldsymbol{\xi}', \mathbf{W}_V^{(t)} \boldsymbol{w}_O \rangle\right|\right|$$

$$\leq \left|\left\langle \boldsymbol{\xi} - \boldsymbol{\xi}', \mathbf{W}_V^{(t)} \boldsymbol{w}_O \right\rangle\right|$$

$$\leq \|\mathbf{W}_V^{(t)} \boldsymbol{w}_O\|_2 \|\boldsymbol{\xi} - \boldsymbol{\xi}'\|_2$$

So, we can get

$$\|g\|_{\text{Lip}} \leq \left\|\mathbf{W}_V^{(t)} \boldsymbol{w}_O\right\|_2. \tag{98}$$

By plugging (98) into (97), we can get the result. □

The following inequality holds according to the update rules of the $V$ vector in Lemma 8, the first equality is derived from the update of triangle inequality, and the second equality is due to the initialization of the $V$ vector.

$$
\begin{aligned}
\left\|\mathbf{W}_V^{(t)} \boldsymbol{w}_O\right\|_2 &\leq \left\|\mathbf{W}_V^{(0)} \boldsymbol{w}_O\right\|_2 + \sum_{t'=0}^{t-1} \left\|\mathbf{W}_V^{(t'+1)} \boldsymbol{w}_O - \mathbf{W}_V^{(t')} \boldsymbol{w}_O\right\|_2 \\
&= \left\|\mathbf{W}_V^{(0)} \boldsymbol{w}_O\right\|_2 + tO\left(\eta \cdot \max\left\{\|\boldsymbol{\mu}\|_2, \sigma_p \sqrt{d}\right\} \cdot \|\boldsymbol{w}_O\|^2\right) \\
&= O\left(\sigma_V \|\boldsymbol{w}_O\|_2 \sqrt{d} + t\eta \|\boldsymbol{w}_O\|^2 \max\left\{\|\boldsymbol{\mu}\|_2, \sigma_p \sqrt{d}\right\}\right) \\
&\leq O\left(t\eta \|\boldsymbol{w}_O\|^2 \max\left\{\|\boldsymbol{\mu}\|_2, \sigma_p \sqrt{d}\right\}\right)
\end{aligned} \tag{99}
$$

Since $g(\boldsymbol{\xi})$ as $|V_{\boldsymbol{\xi}}^{(t)}| = |\langle \boldsymbol{\xi}, \mathbf{W}_V^{(t)} \boldsymbol{w}_O \rangle|$, and since $\left\langle \boldsymbol{\xi}, \mathbf{W}_V^{(t)} \boldsymbol{w}_O \right\rangle \sim \mathcal{N}(0, \|\mathbf{W}_V^{(t)} \boldsymbol{w}_O\|_2^2 \sigma_p^2)$, so we can get:

$$\mathbb{E}g(\boldsymbol{\xi}) = \mathbb{E}|\langle \boldsymbol{\xi}, \mathbf{W}_V^{(t)} \boldsymbol{w}_O \rangle| = \sqrt{\frac{2}{\pi}} \|\mathbf{W}_V^{(t)} \boldsymbol{w}_O\|_2 \sigma_p$$

The test error can be interpreted as the probability that the noise term dominates the signal term. Formally, this corresponds to the event that the cumulative contribution of the random perturbation exceeds the deterministic signal margin. After centralization, we can apply Theorem 22 to obtain a high-probability upper bound on the test error.

$$
\begin{aligned}
P(y(f(\theta, \mathbf{X})) \leq 0) &= P\left(\left[(\sum_{i=1}^M S_{i,1})(V_+^{(t)} - V_-^{(t)}) + \sum_{j=2}^M ((\sum_{i=1}^M S_{i,j}) V_{\boldsymbol{\xi}_j}^{(t)})\right] \leq 0\right) \\
&\leq P\left(\sum_{j=2}^M ((\sum_{i=1}^M S_{i,j}) |V_{\boldsymbol{\xi}_j}^{(t)}|) \geq (\sum_{i=1}^M S_{i,1}) \left(V_+^{(t)} - V_-^{(t)}\right)\right) \\
&= P\left(\sum_{j=2}^M \alpha_j (g(\boldsymbol{\xi}_j) - E\{g(\boldsymbol{\xi}_j)\}) \geq \alpha_1 \left(V_+^{(t)} - V_-^{(t)}\right) - \sigma_p \sqrt{\frac{2}{\pi}} (\sum_{j=2}^M \alpha_j) \left\|\mathbf{W}_V^{(t)} \boldsymbol{w}_O\right\|_2\right) \\
&\leq \exp\left[-\frac{c_2 \left(\sum_r \alpha_1 \left(V_+^{(t)} - V_-^{(t)}\right) - \sigma_p \sqrt{\frac{2}{\pi}} (\sum_{j=2}^M \alpha_j) \left\|\mathbf{W}_V^{(t)} \boldsymbol{w}_O\right\|_2\right)^2}{\sigma_p^2 (\sum_{j=2}^M \alpha_j^2) \left(\left\|\mathbf{W}_V^{(t)} \boldsymbol{w}_O\right\|_2\right)^2}\right] \\
&\leq \exp\left(\frac{c_4}{\pi}\right) \cdot \exp\left[-\frac{c_5}{2} \left(\frac{\sum_r \alpha_1 \left(V_+^{(t)} - V_-^{(t)}\right)}{\sigma_p \sqrt{\sum_{j=2}^M \alpha_j^2} \left\|\mathbf{W}_{-V}^{(t)} \boldsymbol{w}_O\right\|_2}\right)^2\right]
\end{aligned}
$$

We denote $\alpha_j$ as $\sum_i S_{i,j}$, where $S_{i,j}$ is $softmax(\langle \mathbf{q}_i^{(t)}, \mathbf{k}_j^{(t)} \rangle)$. When the subscript is 1, it represents the signal, and when the subscript is from 2 to M, it represents noise.

Then, by the lower bound of $V_{\pm}^{(t)}$ and upper bound of $\left\|\mathbf{W}_V^{(t)} \boldsymbol{w}_O\right\|_2$, we can further bound the test error with following inequality:

$$P(y(f(\theta, \mathbf{X}) \leq 0) \leq \exp\left(\frac{c_4}{\pi}\right) \cdot \exp\left[-\frac{c_5}{2}\left(\frac{\sum_r \alpha_1\left(V_+^{(t)} - V_-^{(t)}\right)}{\sigma_p \sqrt{\sum_{j=2}^M \alpha_j^2} \left\|\mathbf{W}_V^{(t)} \boldsymbol{w}_O\right\|_2}\right)^2\right]$$

$$\leq \exp\left(\frac{c_{12}}{\pi}\right) \exp\left[-\frac{c_{13}}{2} O\left(\frac{V_+^{(t)} - V_-^{(t)}}{\frac{\sigma_p (V_+^{(t)} - V_-^{(t)})}{\|\boldsymbol{\mu}\|_2}}\right)^2\right]$$

$$= \exp\left(\frac{c_{12}}{\pi}\right) \exp\left[-\frac{c_{13}}{2} O\left(dSNR^2\right)\right]$$

where the second inequality is by $\mathbf{W}_V^{(t)} \boldsymbol{w}_O$ is almost aligned with $V_+^{(t)}$ and $V_-^{(t)}$, thus $\|\mathbf{W}_V^{(t)} \boldsymbol{w}_O\|_2 \sim \frac{V_+^{(t)} - V_-^{(t)}}{\|\boldsymbol{\mu}\|_2}$.

### F.4.2 ROBUST TEST ERROR

We start by writing the prediction score under perturbation $\widetilde{X}$ as

$$yf(\theta, \widetilde{X}) = \left(\sum_{i=1}^M \widetilde{S}_{i1}\right)\left(\widetilde{V}_+^{(t)} + \widetilde{V}_-^{(t)}\right) + \sum_{j=2}^M \left(\sum_{i=1}^M \widetilde{S}_{ij}\right)\widetilde{V}_{\boldsymbol{\xi}_j}^{(t)},$$

We first take the first term as an example and derive an upper bound on the maximum discrepancy between the perturbed input $\widetilde{X} \in B(X, \tau)$ and the clean input.

$$S_{11} V_+^{(t)} - \widetilde{S}_{11} \widetilde{V}_+^{(t)} = (S_{11} - \widetilde{S}_{11}) V_+^{(t)} + \widetilde{S}_{11}(V_+^{(t)} - \widetilde{V}_+^{(t)})$$

$$\leq (1 - 1/C) S_{11} V_+^{(t)} + \widetilde{S}_{11} |\langle \widetilde{\boldsymbol{\mu}}_+ - \boldsymbol{\mu}_+, \mathbf{W}_V^{(t)} \boldsymbol{w}_O \rangle|$$

$$\leq (1 - 1/C) S_{11} V_+^{(t)} + \widetilde{S}_{11} \|\mathbf{W}_V^{(t)} \boldsymbol{w}_O\| \tau,$$

where the inequality comes from Lemma 20, and the definition of $V_+^{(t)}$. This shows that the deviation can be bounded linearly in the perturbation magnitude, with only a small residual term since $(C - 1) = o(1)$. Similarly, an analogous upper bound holds for $\widetilde{S}_{i1} \widetilde{V}_{\pm}^{(t)} - S_{i1} V_{\pm}^{(t)}$ and $\widetilde{S}_{ij} \widetilde{V}_{\boldsymbol{\xi}_j}^{(t)} - S_{ij} V_{\boldsymbol{\xi}_j}^{(t)}$ for $i \in [M]$, $j \in [M] \backslash \{1\}$.

Aggregating the deviations across all components, the worst-case perturbation satisfies

$$yf(\theta, X) - \min_{\widetilde{X} \in B(X, \tau)} yf(\theta, \widetilde{X})$$

$$\leq \left(\sum_i S_{i1}\right) \|\mathbf{W}_V^{(t)} \boldsymbol{w}_O\| \tau + \sum_{j=2}^M \left(\sum_i S_{ij}\right) \|\mathbf{W}_V^{(t)} \boldsymbol{w}_O\| \tau + (1 - 1/C) S_{11}(\widetilde{V}_+^{(t)} + \widetilde{V}_+^{(t)}) + o(1)$$

$$\lesssim \left(\sum_{j=1}^M \sum_i S_{ij}\right) \|\mathbf{W}_V^{(t)} \boldsymbol{w}_O\| \tau + (1 - 1/C)\left(\sum_i S_{i1}\right)(\widetilde{V}_+^{(t)} + \widetilde{V}_-^{(t)})$$

$$= M \|\mathbf{W}_V^{(t)} \boldsymbol{w}_O\| \tau + (1 - 1/C)\left(\sum_i S_{i1}\right)(\widetilde{V}_+^{(t)} + \widetilde{V}_-^{(t)})$$

$$(100)$$

where the first inequality comes from $\widetilde{V}_{\boldsymbol{\xi}_j}^{(t)} = o(1)$ for $j \in [M] \backslash \{1\}$. Thus the adversarial effect scales with both the cumulative magnitude of the perturbed coefficients and the operator norm of the weight matrices. Then we can bound the robust test error:

$$P\left(\min_{\widetilde{X} \in B(X, \tau)} yf(\theta, \widetilde{X}) \leq 0\right) = P\left(yf(\theta, X) + \left(yf(\theta, X) - \min_{\widetilde{X} \in B(X, \tau)} yf(\theta, \widetilde{X})\right) \leq 0\right)$$

$$\leq P\bigg(\sum_{j=2}^{M}\alpha_j\big(g(\boldsymbol{\xi}_j)-E\{g(\boldsymbol{\xi}_j)\}\big)\geq \alpha_1\big(V_+^{(t)}-V_-^{(t)}\big)-\sigma_p\sqrt{\tfrac{2}{\pi}}\Big(\sum_{j=2}^{M}\alpha_j\Big)\big\|\mathbf{W}_V^{(t)}\boldsymbol{w}_O\big\|_2$$

$$-M\big\|\mathbf{W}_V^{(t)}\boldsymbol{w}_O\big\|_2\tau-(1-1/C)\alpha_1\big(V_+^{(t)}-V_-^{(t)}\big)\bigg)$$

$$\leq \exp\big(\tfrac{c_{12}}{\pi}\big)\exp\left[-\tfrac{c_{13}}{2}\left(\frac{\alpha_1(V_+^{(t)}-V_-^{(t)})}{\sigma_p\sqrt{\sum_{j=2}^{M}\alpha_j^2}\|\mathbf{W}_V^{(t)}\boldsymbol{w}_O\|_2}-\frac{M\|\mathbf{W}_V^{(t)}\boldsymbol{w}_O\|\tau}{\sigma_p\sqrt{\sum_{j=2}^{M}\alpha_j^2}\|\mathbf{W}_V^{(t)}\boldsymbol{w}_O\|_2}\right)^2\right]$$

$$\leq \exp\big(\tfrac{c_{12}}{\pi}\big)\exp\left[-\tfrac{c_{13}}{2}\,O\Big(\sqrt{d}SNR(1-\tfrac{\tau}{\|\boldsymbol{\mu}\|_2})\Big)^2\right]$$

The first inequality follows from (100). The third inequality uses the fact that $C \leq e/2$, which we absorb into the constant term. The last inequality follows from the bound on $V_\pm^{(t)}$ and $\|\mathbf{W}_V^{(t)}\boldsymbol{w}_O\|_2$.

This completes the proof.

## G  BENIGN OVERFITTING IN CASE 2

Stage I stay same with F.1. Next, we aim to prove that under condition $\tau = (1-o(1))\|\boldsymbol{\mu}\|$, the attention component in a ViT will remain in its initialization state and fail to learn meaningful signal-to-signal or noise-to-signal interactions. This is because, under such perturbations, any newly emerging margin can be immediately neutralized, preventing the signal-to-signal attention from accumulating advantages. In this regime, the ViT effectively degenerates into a linear model.

At the end of Stage I, since $V_+^{T_1} \geq 3M|V_{n,i}^{T_1}|$ and $q = o(1)$ at this point, the perturbation has no significant effect. Consequently, $\langle \boldsymbol{q}_+, \boldsymbol{k}_+ \rangle$ and $\langle \boldsymbol{q}_{n,i}, \boldsymbol{k}_+ \rangle$ experiences a temporary increase, but it does not exceed $\Theta(\log C)$ (as we will demonstrate later). Therefore, we assume that after Stage II, when $\langle q, k \rangle$ has stabilized and the loss derivatives are no longer at the $o(1)$ scale. However, $|V_{n,i}^{T_3}|$ is not $o(1)$; it is of the same order as $|V_+^{T_3}|$, which implies that a larger SNR is required. The following conditions hold at the beginning of Stage III.

$$|V_+^{(T_2)}|,|V_-^{(T_2)}|,|V_{n,i}^{(T_2)}| = o(1),$$

$$V_+^{(T_2)} \geq 3M\cdot|V_{n,i}^{(T_2)}|,$$

$$V_-^{(T_2)} \leq -3M\cdot|V_{n,i}^{(T_2)}|,$$

$$\|\boldsymbol{q}_+^{(T_2)}\|_2^2,\|\boldsymbol{k}_+^{(T_2)}\|_2^2 = \Theta(\log C),$$

$$\|\boldsymbol{q}_{n,i}^{(T_2)}\|_2^2,\|\boldsymbol{k}_{n,i}^{(T_2)}\|_2^2 = \Theta(\log C),$$

$$|\langle\boldsymbol{q}_+^{(T_2)},\boldsymbol{q}_-^{(T_2)}\rangle|,|\langle\boldsymbol{q}_+^{(T_2)},\boldsymbol{q}_{n,i}^{(T_2)}\rangle|,|\langle\boldsymbol{q}_{n,i}^{(T_2)},\boldsymbol{q}_{n',j}^{(T_2)}\rangle| = o(1),$$

$$|\langle\boldsymbol{k}_+^{(T_2)},\boldsymbol{k}_-^{(T_2)}\rangle|,|\langle\boldsymbol{k}_+^{(T_2)},\boldsymbol{k}_{n,i}^{(T_2)}\rangle|,|\langle\boldsymbol{k}_{n,i}^{(T_2)},\boldsymbol{k}_{n',j}^{(T_2)}\rangle| = o(1),$$

for $i,j \in [M]\backslash\{1\}, n,n' \in [N], i \neq j$ or $n \neq n'$.

$$|\langle\boldsymbol{q}_+^{(T_2)},\boldsymbol{k}_+^{(T_2)}\rangle|,|\langle\boldsymbol{q}_+^{(T_2)},\boldsymbol{k}_{n,j}^{(T_2)}\rangle|,|\langle\boldsymbol{q}_{n,i}^{(T_2)},\boldsymbol{k}_+^{(T_2)}\rangle|,|\langle\boldsymbol{q}_{n,i}^{(T_2)},\boldsymbol{k}_{n',j}^{(T_2)}\rangle| = \Theta(\log C)$$

$$|\langle\boldsymbol{q}_+^{(T_2)},\boldsymbol{k}_-^{(T_2)}\rangle|,|\langle\boldsymbol{q}_{n,i}^{(T_2)},\boldsymbol{k}_{\overline{n},j}^{(T_2)}\rangle| = o(1)$$

for $i,j \in [M]\backslash\{1\}, n,\overline{n} \in [N], n \neq \overline{n}$.

Let $T_3 = \Theta\left(\frac{M}{\eta\epsilon(\|\boldsymbol{\mu}\|_2+\tau)^2\|\boldsymbol{w}_O\|_2^2}\right)$. Next we prove the following four propositions $\mathcal{J}(t), \mathcal{K}(t), \mathcal{L}(t)$ by induction on $t$ for $t \in [T_2, T_3]$:

- $\mathcal{J}(t)$:

$$V_+^{(t)} \geq 3M \cdot |V_{n,i}^{(t)}|,$$
$$V_-^{(t)} \leq -3M \cdot |V_{n,i}^{(t)}|,$$
$$|V_{n,i}^{(t)}| = o(1),$$
$$\log\left(\exp(V_+^{(T_2)}) + \frac{\eta}{M}C_{17}(\|\boldsymbol{\mu}\|_2 - \tau)^2\|\boldsymbol{w}_O\|_2^2(t - T_2)\right) \leq V_+^{(t)} \leq 2\log\left(O\left(\frac{1}{\epsilon}\right)\right),$$
$$-2\log\left(O\left(\frac{1}{\epsilon}\right)\right) \leq V_-^{(t)} \leq -\log\left(\exp(-V_-^{(T_2)}) + \frac{\eta}{M}C_{17}(\|\boldsymbol{\mu}\|_2 - \tau)^2\|\boldsymbol{w}_O\|_2^2(t - T_2)\right)$$

for $i \in [M]\setminus\{1\}, n \in [N]$.

- $\mathcal{K}(t)$:

$$\|\boldsymbol{q}_\pm^{(t)}\|_2^2, \|\boldsymbol{k}_\pm^{(t)}\|_2^2 = \Theta(logC),$$
$$\|\boldsymbol{q}_{n,i}^{(t)}\|_2^2, \|\boldsymbol{k}_{n,i}^{(t)}\|_2^2 = \Theta\left(logC\right),$$
$$|\langle\boldsymbol{q}_+^{(t)}, \boldsymbol{q}_-^{(t)}\rangle|, |\langle\boldsymbol{q}_\pm^{(t)}, \boldsymbol{q}_{n,i}^{(t)}\rangle|, |\langle\boldsymbol{q}_{n,i}^{(t)}, \boldsymbol{q}_{n',j}^{(t)}\rangle| = o(1),$$
$$|\langle\boldsymbol{k}_+^{(t)}, \boldsymbol{k}_-^{(t)}\rangle|, |\langle\boldsymbol{k}_\pm^{(t)}, \boldsymbol{k}_{n,i}^{(t)}\rangle|, |\langle\boldsymbol{k}_{n,i}^{(t)}, \boldsymbol{k}_{n',j}^{(t)}\rangle| = o(1)$$

for $i, j \in [M]\setminus\{1\}, n, n' \in [N], i \neq j$ or $n \neq n', C = O(1)$.

- $\mathcal{L}(t)$:

$$|\langle\boldsymbol{q}_\pm^{(t)}, \boldsymbol{k}_\pm^{(t)}\rangle|, |\langle\boldsymbol{q}_\pm^{(t)}, \boldsymbol{k}_{n,j}^{(t)}\rangle|, |\langle\boldsymbol{q}_{n,i}^{(t)}, \boldsymbol{k}_\pm^{(t)}\rangle|, |\langle\boldsymbol{q}_{n,i}^{(t)}, \boldsymbol{k}_{n,j}^{(t)}\rangle| = \Theta(\log C),$$
$$|\langle\boldsymbol{q}_\pm^{(t)}, \boldsymbol{k}_\mp^{(t)}\rangle|, |\langle\boldsymbol{q}_{n,i}^{(t)}, \boldsymbol{k}_{\overline{n},j}^{(t)}\rangle| = o(1)$$

for $i, j \in [M]\setminus\{1\}, n, n' \in [N], n \neq \overline{n}, C = O(1)$.

By the results of Stage II, we know that $\mathcal{F}(T_1), \mathcal{G}(T_2), \mathcal{I}(T_2)$ are true. To prove that $\mathcal{F}(t), \mathcal{G}(t),$ $\mathcal{H}(t)$ and $\mathcal{I}(t)$ are true in stage III, we will prove the following claims holds for $t \in [T_2, T_3]$:

- Claim 9. $\mathcal{L}(T_2), \ldots, \mathcal{L}(t) \implies \mathcal{J}(t+1)$
- Claim 10. $\mathcal{J}(t), \mathcal{L}(t), \mathcal{K}(t) \implies \mathcal{K}(t+1)$
- Claim 11. $\mathcal{J}(t), \mathcal{K}(t), \mathcal{L}(t), \implies \mathcal{L}(t+1)$

### G.1 PROOF OF CLAIM 9

The proofs for $V_+^{(t)} \geq 3M \cdot |V_{n,i}^{(t)}|$ and $V_-^{(t)} \leq -3M \cdot |V_{n,i}^{(t)}|$ are the same as for F.2.1.

we provide the bounds for $-\widetilde{\ell}_n'^{(s)}$. Note that $\ell(z) = \log(1 + \exp(-z))$ and $-\ell'(z) = \exp(-z)/(1 + \exp(-z))$. Without loss of generality, assume $y_n = 1$. We have

$$-\widetilde{\ell}'(f(\widetilde{\boldsymbol{X}}_n, \theta(s))) = \frac{1}{1 + \exp\left(\frac{1}{M}\sum_{l=1}^M \varphi(\widetilde{\boldsymbol{x}}_{n,l}^\top \mathbf{W}_Q^{(s)}\mathbf{W}_K^{(s)\top}(\widetilde{\boldsymbol{X}}_n)^\top)\widetilde{\boldsymbol{X}}_n\mathbf{W}_V^{(s)}\boldsymbol{w}_O\right)}$$

$$= \frac{1}{M}\left(\text{softmax}(\langle\widetilde{\boldsymbol{q}}_\pm^{(s)}, \widetilde{\boldsymbol{k}}_\pm^{(s)}\rangle) + \sum_{l=2}^M \text{softmax}(\langle\widetilde{\boldsymbol{q}}_{n,l}^{(s)}, \widetilde{\boldsymbol{k}}_\pm^{(s)}\rangle)\right) \cdot \widetilde{\boldsymbol{\mu}}_+^\top \mathbf{W}_V^{(s)}\boldsymbol{w}_O$$

$$+ \sum_{j\in[M]\setminus\{1\}}\left(\text{softmax}(\langle\widetilde{\boldsymbol{q}}_{n,j}^{(s)}, \widetilde{\boldsymbol{k}}_{n,j}^{(s)}\rangle) + \sum_{l=2}^M \text{softmax}(\langle\widetilde{\boldsymbol{q}}_{n,l}^{(s)}, \widetilde{\boldsymbol{k}}_{n,j}^{(s)}\rangle)\right) \cdot \widetilde{\boldsymbol{\xi}}_{n,j}^\top \mathbf{W}_V^{(s)}\boldsymbol{w}_O$$

$$= \frac{1}{M}\left(M \cdot \frac{C}{C + M - 1} \cdot V_+^{(s)} + M \cdot \frac{1}{C + M - 1} \cdot \sum_{j\in[M]\setminus\{1\}} V_{n,i}^{(s)}\right)$$

$$\geq \frac{1}{2M}V_+^{(s)},$$

$$(101)$$

for $s \in [T_2, t]$, where the second equality is by $\mathcal{L}(t)$ that $\Lambda_{n,\pm,j}^{(s)} = \Theta(\log C)$ and $\Lambda_{n,i,\pm,j}^{(s)} = \Theta(\log C)$, and the inequality follows from $V_+^{(s)} \geq 3M \cdot |V_{n,i}^{(s)}|$.

Similarly, we have

$$\frac{1}{M} \sum_{l=1}^{M} \varphi(\widetilde{\boldsymbol{x}}_{n,l}^\top \mathbf{W}_Q^{(s)} \mathbf{W}_K^{(s)\top} (\widetilde{\boldsymbol{X}}_n)^\top) \widetilde{\boldsymbol{X}}_n \mathbf{W}_V^{(s)} \boldsymbol{w}_O \leq \max_{i \in [M] \setminus \{1\}} \{V_+^{(s)}, V_{n,i}^{(s)}\} \tag{102}$$

$$= V_+^{(s)}.$$

Then, we have

$$-\widetilde{\ell}'(f(\widetilde{\boldsymbol{X}}_n, \theta(s))) = \frac{1}{1 + \exp\left(\frac{1}{M} \sum_{l=1}^{M} \varphi(\widetilde{\boldsymbol{x}}_{n,l}^\top \mathbf{W}_Q^{(s)} \mathbf{W}_K^{(s)\top} (\widetilde{\boldsymbol{X}}_n)^\top) \widetilde{\boldsymbol{X}}_n \mathbf{W}_V^{(s)} \boldsymbol{w}_O\right)}$$

$$\geq \frac{1}{1 + \exp(V_+^{(s)})} \tag{103}$$

$$\geq \frac{C_{16}}{\exp(V_+^{(s)})}$$

For the last inequality, note that $V_+^{(T_2)} \geq 0$ and $V_+^{(s)}$ is monotonically increasing, so there exist a constant $C_{16}$ such that $\frac{1}{1+\exp(V_+^{(s)})} \geq \frac{C_{16}}{\exp(V_+^{(s)})}$. We also have the upper bound

$$-\ell'(f(X_n, \theta(s))) = \frac{1}{1 + \exp\left(\frac{1}{M} \sum_{l=1}^{M} \varphi(x_{n,l}^\top \mathbf{W}_Q^{(s)} \mathbf{W}_K^{(s)\top} (X_n)^\top) X_n \mathbf{W}_V^{(s)} \boldsymbol{w}_O\right)}$$

$$\leq \frac{1}{1 + \exp(V_+^{(s)}/2M)} \tag{104}$$

$$\leq \frac{1}{\exp(V_+^{(s)}/2M)}$$

By the update rule of $\gamma_{V,+}^{(t)}$ and in Lemma 8 and get

$$\gamma_{V,+}^{(s+1)} - \gamma_{V,+}^{(s)} = -\frac{\eta \langle \widetilde{\boldsymbol{\mu}}_+, \widetilde{\boldsymbol{\mu}}_+^{(t)} \rangle}{NM} \sum_{n \in S_+} \widetilde{\ell}_n'^{(t)} \left( \frac{\exp(\langle \widetilde{\mathbf{q}}_+^{(t)}, \widetilde{\mathbf{k}}_+^{(t)} \rangle)}{\exp(\langle \widetilde{\mathbf{q}}_+^{(t)}, \widetilde{\mathbf{k}}_+^{(t)} \rangle) + \sum_{k=2}^{M} \exp(\langle \widetilde{\mathbf{q}}_+^{(t)}, \widetilde{\mathbf{k}}_{n,k}^{(t)} \rangle)} \right.$$

$$\left. + \sum_{j=2}^{M} \frac{\exp(\langle \widetilde{\mathbf{q}}_{n,j}^{(t)}, \widetilde{\mathbf{k}}_+^{(t)} \rangle)}{\exp(\langle \widetilde{\mathbf{q}}_{n,j}^{(t)}, \widetilde{\mathbf{k}}_+^{(t)} \rangle) + \sum_{k=2}^{M} \exp(\langle \widetilde{\mathbf{q}}_{n,j}^{(t)}, \widetilde{\mathbf{k}}_{n,k}^{(t)} \rangle)} \right)$$

$$+ \sum_{n \in S_+} \widetilde{\ell}_n'^{(t)} \sum_{i=2}^{M} \frac{-\eta \langle \widetilde{\boldsymbol{\mu}}_+, \widetilde{\boldsymbol{\xi}}_{n,i}^{(t)} \rangle}{NM} \left( \frac{\exp(\langle \widetilde{\mathbf{q}}_+^{(t)}, \widetilde{\mathbf{k}}_{n,i}^{(t)} \rangle)}{\exp(\langle \widetilde{\mathbf{q}}_+^{(t)}, \widetilde{\mathbf{k}}_+^{(t)} \rangle) + \sum_{k=2}^{M} \exp(\langle \widetilde{\mathbf{q}}_+^{(t)}, \widetilde{\mathbf{k}}_{n,k}^{(t)} \rangle)} \right.$$

$$\left. + \sum_{j=2}^{M} \frac{\exp(\langle \widetilde{\mathbf{q}}_{n,j}^{(t)}, \widetilde{\mathbf{k}}_{n,i}^{(t)} \rangle)}{\exp(\langle \widetilde{\mathbf{q}}_{n,j}^{(t)}, \widetilde{\mathbf{k}}_+^{(t)} \rangle) + \sum_{k=2}^{M} \exp(\langle \widetilde{\mathbf{q}}_{n,j}^{(t)}, \widetilde{\mathbf{k}}_{n,k}^{(t)} \rangle)} \right)$$

$$\geq -\frac{\eta(\|\boldsymbol{\mu}\|_2 - \tau)^2}{NM} \sum_{n \in S_+} \ell_n'^{(s)} (M \cdot \frac{C}{C + M - 1}) + \frac{\eta \tau \|\boldsymbol{\mu}\|_2}{NM} \sum_{n \in S_+} \ell_n'^{(s)} (M \cdot \frac{1}{C + M - 1})$$

$$\geq \frac{\eta(\|\boldsymbol{\mu}\|_2 - \tau)^2}{NM} \cdot \frac{N}{4} \cdot \frac{C_{16}}{\exp(V_+^{(s)})} - \frac{\eta \tau \|\boldsymbol{\mu}\|_2}{NM} \cdot \frac{N}{4} \cdot \frac{C_{16}}{\exp(V_+^{(s)}/2M)}$$

$$\geq \frac{\eta C_{17}(\|\boldsymbol{\mu}\|_2 - \tau)^2}{M} \frac{1}{\exp(V_+^{(s)})}$$

where the second inequality is by (103)(104), the last inequality is by $N \cdot SNR^2 = \Omega(\frac{1}{\epsilon})$ Then by definition 4, we get

$$V_+^{(s+1)} - V_+^{(s)} = (\gamma_{V,+}^{(s+1)} - \gamma_{V,+}^{(s)}) \|\boldsymbol{w}_O\|_2^2 \geq \frac{\eta C_{17}(\|\boldsymbol{\mu}\|_2 - \tau)^2 \|\boldsymbol{w}_O\|_2^2}{M \exp(V_+^{(s)})} \tag{105}$$

Multiply both sides simultaneously by $\exp(V_+^{(s)})$ and get

$$\exp(V_+^{(s)})(V_+^{(s+1)} - V_+^{(s)}) \geq \frac{\eta}{M} C_{17}(\|\boldsymbol{\mu}\|_2 - \tau)^2 \|\boldsymbol{w}_O\|_2^2 \tag{106}$$

Taking a summation from $T_2$ to $t$ and get

$$\sum_{s=T_2}^{t} \exp(V_+^{(s)})(V_+^{(s+1)} - V_+^{(s)}) \geq \sum_{s=T_2}^{t} \frac{\eta}{M} C_{17}(\|\boldsymbol{\mu}\|_2 - \tau)^2 \|\boldsymbol{w}_O\|_2^2$$
$$\geq \frac{\eta}{M} C_{17}(\|\boldsymbol{\mu}\|_2 - \tau)^2 \|\boldsymbol{w}_O\|_2^2 (t - T_2 + 1) \tag{107}$$

By the property that $V_+^{(s)}$ is monotonically increasing, we have

$$\int_{V_+^{(T_2)}}^{V_+^{(t+1)}} \exp(x) dx \geq \sum_{s=T_2}^{t} \exp(V_+^{(s)})(V_+^{(s+1)} - V_+^{(s)})$$
$$\geq \frac{\eta}{M} C_{17}(\|\boldsymbol{\mu}\|_2 - \tau)^2 \|\boldsymbol{w}_O\|_2^2 (t - T_2 + 1) \tag{108}$$

By $\int_{V_+^{(T_2)}}^{V_+^{(t+1)}} \exp(x) dx = \exp(V_+^{(t+1)}) - \exp(V_+^{(T_2)})$ we get

$$V_+^{(t+1)} \geq \log\left(\exp(V_+^{(T_2)}) + \frac{\eta}{M} C_{17}(\|\boldsymbol{\mu}\|_2 - \tau)^2 \|\boldsymbol{w}_O\|_2^2 (t - T_2 + 1)\right) \tag{109}$$

Similarly, we have

$$V_-^{(t+1)} \leq -\log\left(\exp\left(V_-^{(T_2)}\right) + \eta C_{17}(\|\boldsymbol{\mu}\|_2 - \tau)^2 \|\boldsymbol{w}_O\|_2^2 (t - T_2 + 1)\right) \tag{110}$$

Similarly, we have the lower bound, the proofs are the same as for F.3.1

## G.2 PROOF OF CLAIM 10

We first consider the increment of $\langle \boldsymbol{q}, \boldsymbol{k} \rangle$ at the $t$-th update when using the original clean data. We then show that, at this step, the effect introduced by the perturbation under adversarial samples dominates the increment learned from the clean data; hence $\langle \boldsymbol{q}, \boldsymbol{k} \rangle$ remains stable and bounded.

By the update rule of $\langle \boldsymbol{q}, \boldsymbol{k} \rangle$ we have

$$\langle \boldsymbol{q}_+^{(t+1)}, \boldsymbol{k}_+^{(t+1)} \rangle - \langle \boldsymbol{q}_+^{(t)}, \boldsymbol{k}_+^{(t)} \rangle$$
$$= \alpha_{+,+}^{(t)} \|\boldsymbol{k}_+^{(t)}\|_2^2 + \beta_{+,+}^{(t)} \|\boldsymbol{q}_+^{(t)}\|_2^2 + \{\text{lower order term}\} \tag{111}$$

Subsequently, we establish upper bounds for $\alpha$ and $\beta$ for clean data.

$$
\begin{aligned}
\alpha_{+,+}^{(t)} &= \frac{\eta}{NM} \sum_{n \in S_+} -\ell_n^{'(t)} \|\boldsymbol{\mu}\|_2^2 \\
&\quad \cdot \left( V_+^{(t)} \frac{\exp(\langle \boldsymbol{q}_+^{(t)}, \boldsymbol{k}_+^{(t)} \rangle)}{\exp(\langle \boldsymbol{q}_+^{(t)}, \boldsymbol{k}_+^{(t)} \rangle) + \sum_{j=2}^{M} \exp(\langle \boldsymbol{q}_+^{(t)}, \boldsymbol{k}_{n,j}^{(t)} \rangle)} \right. \\
&\quad - \left( \frac{\exp(\langle \boldsymbol{q}_+^{(t)}, \boldsymbol{k}_+^{(t)} \rangle)}{\exp(\langle \boldsymbol{q}_+^{(t)}, \boldsymbol{k}_+^{(t)} \rangle) + \sum_{j=2}^{M} \exp(\langle \boldsymbol{q}_+^{(t)}, \boldsymbol{k}_{n,j}^{(t)} \rangle)} \right)^2 \\
&\quad - \sum_{i=2}^{M} (V_{n,i}^{(t)} \cdot \frac{\exp(\langle \boldsymbol{q}_+^{(t)}, \boldsymbol{k}_+^{(t)} \rangle)}{\exp(\langle \boldsymbol{q}_+^{(t)}, \boldsymbol{k}_+^{(t)} \rangle) + \sum_{j=2}^{M} \exp(\langle \boldsymbol{q}_+^{(t)}, \boldsymbol{k}_{n,j}^{(t)} \rangle)} \\
&\quad \left. \cdot \frac{\exp(\langle \boldsymbol{q}_+^{(t)}, \boldsymbol{k}_{n,i}^{(t)} \rangle)}{\exp(\langle \boldsymbol{q}_+^{(t)}, \boldsymbol{k}_+^{(t)} \rangle) + \sum_{j=2}^{M} \exp(\langle \boldsymbol{q}_+^{(t)}, \boldsymbol{k}_{n,j}^{(t)} \rangle)} \right) \\
&\leq \frac{\eta}{NM} \sum_{n \in S_+} \|\boldsymbol{\mu}\|_2^2 (V_+^{(t)}) \\
&\leq \frac{3\eta}{2M} \|\boldsymbol{\mu}\|_2^2 V_+^{(t)}
\end{aligned}
\tag{112}
$$

Then, we can then compute the incremental growth of $\langle \boldsymbol{q}_+^{(t)}, \boldsymbol{k}_+^{(t)} \rangle$ after one update step on clean data.

$$
\begin{aligned}
&\langle \boldsymbol{q}_+^{(t+1)}, \boldsymbol{k}_+^{(t+1)} \rangle - \langle \boldsymbol{q}_+^{(t)}, \boldsymbol{k}_+^{(t)} \rangle \\
&= \alpha_{+,+}^{(t)} \|\boldsymbol{k}_+^{(t)}\|_2^2 + \beta_{+,+}^{(t)} \|\boldsymbol{q}_+^{(t)}\|_2^2 + \{\text{lower order term}\} \\
&\leq \frac{3\eta}{2M} \|\boldsymbol{\mu}\|_2^2 V_+^{(t)} \Theta(\log C) + \frac{3\eta}{2M} \|\boldsymbol{\mu}\|_2^2 V_+^{(t)} \Theta(\log C)
\end{aligned}
\tag{113}
$$

Subsequently, we compute at the $t^{th}$ iteration the magnitude of the effect that the perturbation imposes on $\langle \boldsymbol{q}_+^{(t)}, \boldsymbol{k}_+^{(t)} \rangle$.

$$
\begin{aligned}
&max_{\widetilde{X}^{(t)} \in B(X^{(t)}, \tau)} \langle \widetilde{\boldsymbol{q}}_+^{(t)}, \widetilde{\boldsymbol{k}}_+^{(t)} \rangle - \langle \boldsymbol{q}_+^{(t)}, \boldsymbol{k}_+^{(t)} \rangle \\
&= ((1 + \frac{\tau}{\|\boldsymbol{\mu}\|_2})^2 - 1) \langle \boldsymbol{q}_+^{(t)}, \boldsymbol{k}_+^{(t)} \rangle \\
&= ((1 + \frac{\tau}{\|\boldsymbol{\mu}\|_2})^2 - 1) \Theta(logC)
\end{aligned}
\tag{114}
$$

As $\tau$ and $\|\boldsymbol{\mu}\|$ are same order, and $\frac{3\eta}{2M} \|\boldsymbol{\mu}\|_2^2 V_+^{(t)} = o(\frac{1}{N})$ by $V_+^{(t)} \leq 2\log\left(O(\frac{1}{\epsilon})\right)$, $d_h = \widetilde{\Omega}\left(\max\{\text{SNR}^4, \text{SNR}^{-4}\} N^2 \epsilon^{-2}\right)$ and $\eta \leq \widetilde{O}(\min\{\|\boldsymbol{\mu}\|_2^{-2}, (\sigma_p^2 d)^{-1}\} \cdot d_h^{-\frac{1}{2}})$. Thus, we have

$$
\langle \boldsymbol{q}_+^{(t+1)}, \boldsymbol{k}_+^{(t+1)} \rangle - \langle \boldsymbol{q}_+^{(t)}, \boldsymbol{k}_+^{(t)} \rangle \leq max_{\widetilde{\boldsymbol{X}}^{(t)} \in B(\boldsymbol{X}^{(t)}, \tau)} \langle \widetilde{\boldsymbol{q}}_+^{(t)}, \widetilde{\boldsymbol{k}}_+^{(t)} \rangle - \langle \boldsymbol{q}_+^{(t)}, \boldsymbol{k}_+^{(t)} \rangle
$$

which indicates that the perturbation's effect exceeds the one-step update on clean data. Therefore, $\langle \boldsymbol{q}_+^{(t)}, \boldsymbol{k}_-^{(t)} \rangle$ stay $\Theta(logC)$. By similar methods, we can bound the other $\langle \boldsymbol{q}^{(t)}, \boldsymbol{k}^{(t)} \rangle$, which complete the proof.

### G.3 PROOF OF CLAIM 11

We use similar methods in G.2. We first consider the increment of $\|\boldsymbol{q}\|_2^2$ and $\|\boldsymbol{k}\|_2^2$ at the $t$-th update when using the original clean data. We then show that, at this step, the effect introduced by the perturbation under adversarial samples dominates the increment learned from the clean data; hence $\|\boldsymbol{q}\|_2^2$ and $\|\boldsymbol{k}\|_2^2$ remains stable and bounded.

By the update rule of $\|\boldsymbol{q}\|_2^2$ and $\|\boldsymbol{k}\|_2^2$ we have

$$\|\boldsymbol{q}_+^{(t+1)}\|_2^2 - \|\boldsymbol{q}_+^{(t)}\|_2^2 = 2\langle \Delta\boldsymbol{q}_+^{(t)}, \boldsymbol{q}_+^{(t)}\rangle + \langle \Delta\boldsymbol{q}_+^{(t)}, \Delta\boldsymbol{q}_+^{(t)}\rangle$$

$$= 2\alpha_{+,+}^{(t)}\langle \boldsymbol{q}_+^{(t)}, \boldsymbol{k}_+^{(t)}\rangle + 2\sum_{n\in S_+}\sum_{i=2}^{M}\alpha_{n,+,i}^{(t)}\langle \boldsymbol{q}_+^{(t)}, \boldsymbol{k}_{n,i}^{(t)}\rangle$$

$$+ \left(\alpha_{+,+}^{(t)}\boldsymbol{k}_+^{(t)} + \sum_{n\in S_+}\sum_{i=2}^{M}\alpha_{n,+,i}^{(t)}\boldsymbol{k}_{n,i}^{(t)}\right)$$

$$\cdot \left(\alpha_{+,+}^{(t)}\boldsymbol{k}_+^{(t)\top} + \sum_{n\in S_+}\sum_{i=2}^{M}\alpha_{n,+,i}^{(t)}\boldsymbol{k}_{n,i}^{(t)\top}\right)$$

$$\leq 2|\alpha_{+,+}^{(t)}||\langle \boldsymbol{q}_+^{(t)}, \boldsymbol{k}_+^{(t)}\rangle| + 2\sum_{n\in S_+}\sum_{i=2}^{M}|\alpha_{n,+,i}^{(t)}||\langle \boldsymbol{q}_+^{(t)}, \boldsymbol{k}_{n,i}^{(t)}\rangle| + \{\text{lower order term}\}$$

$$\leq 2\frac{3\eta}{2M}\|\boldsymbol{\mu}\|_2^2\log\left(O(\frac{1}{\epsilon})\right)\Theta(\log C) + 2\sum_{n\in S_+}\sum_{i=2}^{M}\frac{3\eta}{2NM}\|\boldsymbol{\mu}\|_2^2\log\left(O(\frac{1}{\epsilon})\right)\Theta(\log C)$$

$$\leq 12\eta\|\boldsymbol{\mu}\|_2^2\log\left(O(\frac{1}{\epsilon})\right)\Theta(\log C)$$

(115)

where the second inequality comes form the upper bounds for $\alpha$ and $\beta$ on clean data similar to (112).

Subsequently, we compute at the $t^{th}$ iteration the magnitude of the effect that the perturbation imposes on $\|\boldsymbol{q}_+^{(t)}\|_2^2$.

$$max_{\widetilde{X}^{(t)}\in B(X^{(t)},\tau)}\|\widetilde{\boldsymbol{q}}_+^{(t)}\|_2^2 - \|\boldsymbol{q}_+^{(t)}\|_2^2$$

$$\leq ((1 + \frac{\tau}{\|\boldsymbol{\mu}\|_2})^2 - 1)\|\boldsymbol{q}_+^{(t)}\|_2^2$$

(116)

$$\leq ((1 + \frac{\tau}{\|\boldsymbol{\mu}\|_2})^2 - 1)\Theta(\log C)$$

As $\tau$ and $\|\boldsymbol{\mu}\|_2$ are same order, and $12\eta\|\boldsymbol{\mu}\|_2^2\log\left(O(\frac{1}{\epsilon})\right) = o(\frac{1}{NM})$ by $d_h = \widetilde{\Omega}\left(\max\{\text{SNR}^4, \text{SNR}^{-4}\}N^2\epsilon^{-2}\right)$ and $\eta \leq \widetilde{O}(\min\{\|\boldsymbol{\mu}\|_2^{-2}, (\sigma_p^2 d)^{-1}\}\cdot d_h^{-\frac{1}{2}})$. Thus, we have

$$\|\boldsymbol{q}_+^{(t+1)}\|_2^2 - \|\boldsymbol{q}_+^{(t)}\|_2^2 \leq max_{\widetilde{\boldsymbol{X}}^{(t)}\in B(\boldsymbol{X}^{(t)},\tau)}\|\widetilde{\boldsymbol{q}}_+^{(t)}\|_2^2 - \|\boldsymbol{q}_+^{(t)}\|_2^2$$

which indicates that the perturbation's effect exceeds the one-step update on clean data. Therefore, $\|\boldsymbol{q}_+^{(t+1)}\|_2^2$ stay $\Theta(logC)$. By similar methods, we can bound the other $\|\boldsymbol{q}\|_2^2$ and $\|\boldsymbol{k}\|_2^2$, which complete the proof.

The proof of convergence and test error is similar with Lemma 21 and Section F.4

# H  PROOF OF THEOREM 3

*Proof.* Fix an arbitrary $\theta$. Consider the two classes separately. For the positive class ($y = +$):

$$\delta_1^{(+)} = -\boldsymbol{\mu}_+, \quad \delta_2^{(+)} = 0$$

is a valid perturbation since $\|\delta_1^{(+)}\|_2 = \|\boldsymbol{\mu}_+\|_2 \leq \tau$, $\|\delta_2^{(+)}\|_2 = 0 \leq \tau$. Then the adversarially perturbed point

$$\widetilde{x}^{(+)} = [\boldsymbol{\mu}_+ - \boldsymbol{\mu}_+, \boldsymbol{\xi}_2, ..., \boldsymbol{\xi}_M] = [0, \boldsymbol{\xi}_2, ..., \boldsymbol{\xi}_M]$$

lies in $\mathcal{B}([\boldsymbol{\mu}, \boldsymbol{\xi}_2, ..., \boldsymbol{\xi}_M], \tau)$, so there exists a perturbation that can potentially flip the classifier's output for the positive class.

Similarly, for the negative class ($y = -$):

$$\delta_1^{(-)} = -\boldsymbol{\mu}_-, \quad \delta_2^{(-)} = 0$$

produces $\widetilde{x}^{(-)} = [0, \boldsymbol{\xi}_2, ..., \boldsymbol{\xi}_M] \in \mathcal{B}([\boldsymbol{\mu}, \boldsymbol{\xi}_2, ..., \boldsymbol{\xi}_M], \tau)$.

For each class independently, there exists a perturbation that can potentially flip its label.

Thus, at least one class can be adversarially fooled with probability at least $\min\{\Pr[y = +], \Pr[y = -]\}$. For uniform labels, this gives

$$L_D^{\text{rob}}(\theta) \geq \frac{1}{2} \cdot \frac{1}{2} = \frac{1}{4}.$$

This completes the proof. $\hfill\square$

# I COMPLETE CALCULATION PROCESS FOR BENIGN OVERFITTING

## I.1 CALCULATIONS FOR $\alpha$ AND $\beta$

In this subsection, we give the calculactions for $\alpha$ and $\beta$ defined in Definition 5.

$$
\begin{aligned}
\alpha_{+,+}^{(t)} = \frac{\eta}{NM} \sum_{n \in S_+} &-\widetilde{\ell}_n'(\theta)\langle\widetilde{\boldsymbol{\mu}}_+, \widetilde{\boldsymbol{\mu}}_+^{(t)}\rangle \\
&\cdot \left(V_+^{(t)}\left(\frac{\exp(\langle\widetilde{\boldsymbol{q}}_+^{(t)}, \widetilde{\boldsymbol{k}}_+^{(t)}\rangle)}{\exp(\langle\widetilde{\boldsymbol{q}}_+^{(t)}, \widetilde{\boldsymbol{k}}_+^{(t)}\rangle) + \sum_{j=2}^M \exp(\langle\widetilde{\boldsymbol{q}}_+^{(t)}, \widetilde{\boldsymbol{k}}_{n,j}^{(t)}\rangle)}\right.\right. \\
&\quad \left.- \left(\frac{\exp(\langle\widetilde{\boldsymbol{q}}_+^{(t)}, \widetilde{\boldsymbol{k}}_+^{(t)}\rangle)}{\exp(\langle\widetilde{\boldsymbol{q}}_+^{(t)}, \widetilde{\boldsymbol{k}}_+^{(t)}\rangle) + \sum_{j=2}^M \exp(\langle\widetilde{\boldsymbol{q}}_+^{(t)}, \widetilde{\boldsymbol{k}}_{n,j}^{(t)}\rangle)}\right)^2\right) \\
&\quad - \sum_{i=2}^M \left(V_{n,i}^{(t)} \cdot \frac{\exp(\langle\widetilde{\boldsymbol{q}}_+^{(t)}, \widetilde{\boldsymbol{k}}_+^{(t)}\rangle)}{\exp(\langle\widetilde{\boldsymbol{q}}_+^{(t)}, \widetilde{\boldsymbol{k}}_+^{(t)}\rangle) + \sum_{j=2}^M \exp(\langle\widetilde{\boldsymbol{q}}_+^{(t)}, \widetilde{\boldsymbol{k}}_{n,j}^{(t)}\rangle)}\right. \\
&\quad \left.\left.\cdot \frac{\exp(\langle\widetilde{\boldsymbol{q}}_+^{(t)}, \widetilde{\boldsymbol{k}}_{n,i}^{(t)}\rangle)}{\exp(\langle\widetilde{\boldsymbol{q}}_+^{(t)}, \widetilde{\boldsymbol{k}}_+^{(t)}\rangle) + \sum_{j=2}^M \exp(\langle\widetilde{\boldsymbol{q}}_+^{(t)}, \widetilde{\boldsymbol{k}}_{n,j}^{(t)}\rangle)}\right)\right) \\
&\quad + \sum_{i=2}^M \langle\widetilde{\boldsymbol{\mu}}_+, \widetilde{\boldsymbol{\xi}}_{n,i}^{(t)}\rangle \cdot \left(V_+^{(t)}\left(\frac{\exp(\langle\widetilde{\boldsymbol{q}}_{n,i}^{(t)}, \widetilde{\boldsymbol{k}}_+^{(t)}\rangle)}{\exp(\langle\widetilde{\boldsymbol{q}}_{n,i}^{(t)}, \widetilde{\boldsymbol{k}}_+^{(t)}\rangle) + \sum_{j=2}^M \exp(\langle\widetilde{\boldsymbol{q}}_{n,i}^{(t)}, \widetilde{\boldsymbol{k}}_{n,j}^{(t)}\rangle)}\right.\right. \\
&\quad \left.\left.- \left(\frac{\exp(\langle\widetilde{\boldsymbol{q}}_{n,i}^{(t)}, \widetilde{\boldsymbol{k}}_+^{(t)}\rangle)}{\exp(\langle\widetilde{\boldsymbol{q}}_{n,i}^{(t)}, \widetilde{\boldsymbol{k}}_+^{(t)}\rangle) + \sum_{j=2}^M \exp(\langle\widetilde{\boldsymbol{q}}_{n,i}^{(t)}, \widetilde{\boldsymbol{k}}_{n,j}^{(t)}\rangle)}\right)^2\right) \\
&\quad - \sum_{k=2}^M \left(V_{n,i}^{(t)} \cdot \frac{\exp(\langle\widetilde{\boldsymbol{q}}_{n,i}^{(t)}, \widetilde{\boldsymbol{k}}_+^{(t)}\rangle)}{\exp(\langle\widetilde{\boldsymbol{q}}_{n,i}^{(t)}, \widetilde{\boldsymbol{k}}_+^{(t)}\rangle) + \sum_{j=2}^M \exp(\langle\widetilde{\boldsymbol{q}}_{n,i}^{(t)}, \widetilde{\boldsymbol{k}}_{n,j}^{(t)}\rangle)}\right. \\
&\quad \left.\left.\cdot \frac{\exp(\langle\widetilde{\boldsymbol{q}}_{n,i}^{(t)}, \widetilde{\boldsymbol{k}}_{n,k}^{(t)}\rangle)}{\exp(\langle\widetilde{\boldsymbol{q}}_{n,i}^{(t)}, \widetilde{\boldsymbol{k}}_+^{(t)}\rangle) + \sum_{j=2}^M \exp(\langle\widetilde{\boldsymbol{q}}_{n,i}^{(t)}, \widetilde{\boldsymbol{k}}_{n,j}^{(t)}\rangle)}\right)\right)
\end{aligned}
$$

$$\alpha_{n,+,i}^{(t)} = -\frac{\eta}{NM}\ell_n'^{(t)}\langle\widetilde{\boldsymbol{\mu}}_+, \widetilde{\boldsymbol{\mu}}_+^{(t)}\rangle$$

$$\cdot\left(-V_+^{(t)}\cdot\frac{\exp(\langle\widetilde{\boldsymbol{q}}_+^{(t)},\widetilde{\boldsymbol{k}}_+^{(t)}\rangle)}{\exp(\langle\widetilde{\boldsymbol{q}}_+^{(t)},\widetilde{\boldsymbol{k}}_+^{(t)}\rangle) + \sum_{j=2}^{M}\exp(\langle\widetilde{\boldsymbol{q}}_+^{(t)},\widetilde{\boldsymbol{k}}_{n,j}^{(t)}\rangle)}\right.$$

$$+\frac{\exp(\langle\widetilde{\boldsymbol{q}}_+^{(t)},\widetilde{\boldsymbol{k}}_{n,i}^{(t)}\rangle)}{\exp(\langle\widetilde{\boldsymbol{q}}_+^{(t)},\widetilde{\boldsymbol{k}}_+^{(t)}\rangle) + \sum_{j=2}^{M}\exp(\langle\widetilde{\boldsymbol{q}}_+^{(t)},\widetilde{\boldsymbol{k}}_{n,j}^{(t)}\rangle)}$$

$$+ V_{n,i}^{(t)}\left(\frac{\exp(\langle\widetilde{\boldsymbol{q}}_+^{(t)},\widetilde{\boldsymbol{k}}_{n,i}^{(t)}\rangle)}{\exp(\langle\widetilde{\boldsymbol{q}}_+^{(t)},\widetilde{\boldsymbol{k}}_+^{(t)}\rangle) + \sum_{j=2}^{M}\exp(\langle\widetilde{\boldsymbol{q}}_+^{(t)},\widetilde{\boldsymbol{k}}_{n,j}^{(t)}\rangle)}\right.$$

$$\left.-\left(\frac{\exp(\langle\widetilde{\boldsymbol{q}}_+^{(t)},\widetilde{\boldsymbol{k}}_{n,i}^{(t)}\rangle)}{\exp(\langle\widetilde{\boldsymbol{q}}_+^{(t)},\widetilde{\boldsymbol{k}}_+^{(t)}\rangle) + \sum_{j=2}^{M}\exp(\langle\widetilde{\boldsymbol{q}}_+^{(t)},\widetilde{\boldsymbol{k}}_{n,j}^{(t)}\rangle)}\right)^2\right)$$

$$-\sum_{k\neq i}\left(V_{n,k}^{(t)}\cdot\frac{\exp(\langle\widetilde{\boldsymbol{q}}_+^{(t)},\widetilde{\boldsymbol{k}}_{n,i}^{(t)}\rangle)}{\exp(\langle\widetilde{\boldsymbol{q}}_+^{(t)},\widetilde{\boldsymbol{k}}_+^{(t)}\rangle) + \sum_{j=2}^{M}\exp(\langle\widetilde{\boldsymbol{q}}_+^{(t)},\widetilde{\boldsymbol{k}}_{n,j}^{(t)}\rangle)}\right.$$

$$\left.\left.\cdot\frac{\exp(\langle\widetilde{\boldsymbol{q}}_+^{(t)},\widetilde{\boldsymbol{k}}_{n,k}^{(t)}\rangle)}{\exp(\langle\widetilde{\boldsymbol{q}}_+^{(t)},\widetilde{\boldsymbol{k}}_+^{(t)}\rangle) + \sum_{j=2}^{M}\exp(\langle\widetilde{\boldsymbol{q}}_+^{(t)},\widetilde{\boldsymbol{k}}_{n,j}^{(t)}\rangle)}\right)\right)$$

$$+\sum_{k=2}^{M}\langle\widetilde{\boldsymbol{\mu}}_+, \widetilde{\boldsymbol{\xi}}_{n,k}^{(t)}\rangle\cdot\left(-V_+^{(t)}\cdot\frac{\exp(\langle\widetilde{\boldsymbol{q}}_{n,k}^{(t)},\widetilde{\boldsymbol{k}}_+^{(t)}\rangle)}{\exp(\langle\widetilde{\boldsymbol{q}}_{n,k}^{(t)},\widetilde{\boldsymbol{k}}_+^{(t)}\rangle) + \sum_{j=2}^{M}\exp(\langle\widetilde{\boldsymbol{q}}_{n,k}^{(t)},\widetilde{\boldsymbol{k}}_{n,j}^{(t)}\rangle)}\right.$$

$$\cdot\frac{\exp(\langle\widetilde{\boldsymbol{q}}_{n,k}^{(t)},\widetilde{\boldsymbol{k}}_{n,i}^{(t)}\rangle)}{\exp(\langle\widetilde{\boldsymbol{q}}_{n,k}^{(t)},\widetilde{\boldsymbol{k}}_+^{(t)}\rangle) + \sum_{j=2}^{M}\exp(\langle\widetilde{\boldsymbol{q}}_{n,k}^{(t)},\widetilde{\boldsymbol{k}}_{n,j}^{(t)}\rangle)}$$

$$+ V_{n,i}^{(t)}\left(\frac{\exp(\langle\widetilde{\boldsymbol{q}}_{n,k}^{(t)},\widetilde{\boldsymbol{k}}_{n,i}^{(t)}\rangle)}{\exp(\langle\widetilde{\boldsymbol{q}}_{n,k}^{(t)},\widetilde{\boldsymbol{k}}_+^{(t)}\rangle) + \sum_{j=2}^{M}\exp(\langle\widetilde{\boldsymbol{q}}_{n,k}^{(t)},\widetilde{\boldsymbol{k}}_{n,j}^{(t)}\rangle)}\right.$$

$$\left.-\left(\frac{\exp(\langle\widetilde{\boldsymbol{q}}_{n,k}^{(t)},\widetilde{\boldsymbol{k}}_{n,i}^{(t)}\rangle)}{\exp(\langle\widetilde{\boldsymbol{q}}_{n,k}^{(t)},\widetilde{\boldsymbol{k}}_+^{(t)}\rangle) + \sum_{j=2}^{M}\exp(\langle\widetilde{\boldsymbol{q}}_{n,k}^{(t)},\widetilde{\boldsymbol{k}}_{n,j}^{(t)}\rangle)}\right)^2\right)$$

$$-\sum_{l\neq i}\left(V_{n,l}^{(t)}\cdot\frac{\exp(\langle\widetilde{\boldsymbol{q}}_{n,k}^{(t)},\widetilde{\boldsymbol{k}}_{n,i}^{(t)}\rangle)}{\exp(\langle\widetilde{\boldsymbol{q}}_{n,k}^{(t)},\widetilde{\boldsymbol{k}}_+^{(t)}\rangle) + \sum_{j=2}^{M}\exp(\langle\widetilde{\boldsymbol{q}}_{n,k}^{(t)},\widetilde{\boldsymbol{k}}_{n,j}^{(t)}\rangle)}\right.$$

$$\left.\left.\cdot\frac{\exp(\langle\widetilde{\boldsymbol{q}}_{n,k}^{(t)},\widetilde{\boldsymbol{k}}_{n,l}^{(t)}\rangle)}{\exp(\langle\widetilde{\boldsymbol{q}}_{n,k}^{(t)},\widetilde{\boldsymbol{k}}_+^{(t)}\rangle) + \sum_{j=2}^{M}\exp(\langle\widetilde{\boldsymbol{q}}_{n,k}^{(t)},\widetilde{\boldsymbol{k}}_{n,j}^{(t)}\rangle)}\right)\right)$$

We can also derive the calculations for other $\alpha$ and $\beta$, since they follow the same procedure as in Section F.1 of Jiang et al. (2024).

### I.2 PROOF OF LEMMA 17

### I.3 UPDATE RULES FOR INNER PRODUCTS

In this subsection, we give the update rules for the inner products of $\boldsymbol{q}$ and $\boldsymbol{k}$.

$$\langle \boldsymbol{q}_+^{(t+1)}, \boldsymbol{k}_+^{(t+1)} \rangle - \langle \boldsymbol{q}_+^{(t)}, \boldsymbol{k}_+^{(t)} \rangle$$

$$= \alpha_{+,+}^{(t)} \|\boldsymbol{k}_+^{(t)}\|_2^2 + \sum_{n \in S_+} \sum_{i=2}^{M} \alpha_{n,+,i}^{(t)} \langle \boldsymbol{k}_+^{(t)}, \boldsymbol{k}_{n,i}^{(t)} \rangle$$

$$+ \beta_{+,+}^{(t)} \|\boldsymbol{q}_+^{(t)}\|_2^2 + \sum_{n \in S_+} \sum_{i=2}^{M} \beta_{n,+,i}^{(t)} \langle \boldsymbol{q}_+^{(t)}, \boldsymbol{q}_{n,i}^{(t)} \rangle$$

$$+ \left( \alpha_{+,+}^{(t)} \boldsymbol{k}_+^{(t)} + \sum_{n \in S_+} \sum_{i=2}^{M} \alpha_{n,+,i}^{(t)} \boldsymbol{k}_{n,i}^{(t)} \right)$$

$$\cdot \left( \beta_{+,+}^{(t)} \boldsymbol{q}_+^{(t)\top} + \sum_{n \in S_+} \sum_{i=2}^{M} \beta_{n,+,i}^{(t)} \boldsymbol{q}_{n,i}^{(t)\top} \right),$$

$$\langle \boldsymbol{q}_-^{(t+1)}, \boldsymbol{k}_-^{(t+1)} \rangle - \langle \boldsymbol{q}_-^{(t)}, \boldsymbol{k}_-^{(t)} \rangle$$

$$= \alpha_{-,-}^{(t)} \|\boldsymbol{k}_-^{(t)}\|_2^2 + \sum_{n \in S_-} \sum_{i=2}^{M} \alpha_{n,-,i}^{(t)} \langle \boldsymbol{k}_-^{(t)}, \boldsymbol{k}_{n,i}^{(t)} \rangle$$

$$+ \beta_{-,-}^{(t)} \|\boldsymbol{q}_-^{(t)}\|_2^2 + \sum_{n \in S_-} \sum_{i=2}^{M} \beta_{n,-,i}^{(t)} \langle \boldsymbol{q}_-^{(t)}, \boldsymbol{q}_{n,i}^{(t)} \rangle$$

$$+ \left( \alpha_{-,-}^{(t)} \boldsymbol{k}_-^{(t)} + \sum_{n \in S_-} \sum_{i=2}^{M} \alpha_{n,-,i}^{(t)} \boldsymbol{k}_{n,i}^{(t)} \right)$$

$$\cdot \left( \beta_{-,-}^{(t)} \boldsymbol{q}_-^{(t)\top} + \sum_{n \in S_-} \sum_{i=2}^{M} \beta_{n,-,i}^{(t)} \boldsymbol{q}_{n,i}^{(t)\top} \right),$$

We can also derive the update rules for other $\boldsymbol{q}$ and $\boldsymbol{k}$, since they follow the same procedure as in Section F.2 of Jiang et al. (2024).

### I.4 PROOF OF LEMMA 17

Let $T_0 = O\left(\frac{1}{\eta d_h^{\frac{1}{4}}(\|\boldsymbol{\mu}\|_2+\tau)^2\|\boldsymbol{w}_O\|_2^2}\right)$. By Lemma 16, we have $|V_+^{(t)}|, |V_-^{(t)}|, |V_{n,i}^{(t)}| = O(d_h^{-\frac{1}{4}})$ for $t \in [0, T_0]$ by Lemma 16. Plugging this into the expression for $\alpha$ and $\beta$ gives

$$
\begin{aligned}
|\alpha_{+,+}^{(t)}| = \Bigg| &\frac{\eta}{NM} \sum_{n \in S_+} -\widetilde{\ell}_n'(\theta)\langle\widetilde{\boldsymbol{\mu}}_+, \widetilde{\boldsymbol{\mu}}_+^{(t)}\rangle \\
&\cdot \Bigg( V_+^{(t)}\Bigg(\frac{\exp(\langle\boldsymbol{q}_+^{(t)}, \boldsymbol{k}_+^{(t)}\rangle)}{\exp(\langle\boldsymbol{q}_+^{(t)}, \boldsymbol{k}_+^{(t)}\rangle) + \sum_{j=2}^M \exp(\langle\boldsymbol{q}_+^{(t)}, \boldsymbol{k}_{n,j}^{(t)}\rangle)} \\
&- \Bigg(\frac{\exp(\langle\boldsymbol{q}_+^{(t)}, \boldsymbol{k}_+^{(t)}\rangle)}{\exp(\langle\boldsymbol{q}_+^{(t)}, \boldsymbol{k}_+^{(t)}\rangle) + \sum_{j=2}^M \exp(\langle\boldsymbol{q}_+^{(t)}, \boldsymbol{k}_{n,j}^{(t)}\rangle)}\Bigg)^2\Bigg) \\
&- \sum_{i=2}^M \Bigg( V_{n,i}^{(t)} \cdot \frac{\exp(\langle\boldsymbol{q}_+^{(t)}, \boldsymbol{k}_+^{(t)}\rangle)}{\exp(\langle\boldsymbol{q}_+^{(t)}, \boldsymbol{k}_+^{(t)}\rangle) + \sum_{j=2}^M \exp(\langle\boldsymbol{q}_+^{(t)}, \boldsymbol{k}_{n,j}^{(t)}\rangle)} \\
&\cdot \frac{\exp(\langle\boldsymbol{q}_+^{(t)}, \boldsymbol{k}_{n,i}^{(t)}\rangle)}{\exp(\langle\boldsymbol{q}_+^{(t)}, \boldsymbol{k}_+^{(t)}\rangle) + \sum_{j=2}^M \exp(\langle\boldsymbol{q}_+^{(t)}, \boldsymbol{k}_{n,j}^{(t)}\rangle)}\Bigg)\Bigg) \\
&+ \sum_{i=2}^M \langle\widetilde{\boldsymbol{\mu}}_+, \widetilde{\boldsymbol{\xi}}_{n,i}^{(t)}\rangle \cdot \Bigg( V_+^{(t)}\Bigg(\frac{\exp(\langle\boldsymbol{q}_{n,i}^{(t)}, \boldsymbol{k}_+^{(t)}\rangle)}{\exp(\langle\boldsymbol{q}_{n,i}^{(t)}, \boldsymbol{k}_+^{(t)}\rangle) + \sum_{j=2}^M \exp(\langle\boldsymbol{q}_{n,i}^{(t)}, \boldsymbol{k}_{n,j}^{(t)}\rangle)} \\
&- \Bigg(\frac{\exp(\langle\boldsymbol{q}_{n,i}^{(t)}, \boldsymbol{k}_+^{(t)}\rangle)}{\exp(\langle\boldsymbol{q}_{n,i}^{(t)}, \boldsymbol{k}_+^{(t)}\rangle) + \sum_{j=2}^M \exp(\langle\boldsymbol{q}_{n,i}^{(t)}, \boldsymbol{k}_{n,j}^{(t)}\rangle)}\Bigg)^2\Bigg) \\
&- \sum_{k=2}^M \Bigg( V_{n,i}^{(t)} \cdot \frac{\exp(\langle\boldsymbol{q}_{n,i}^{(t)}, \boldsymbol{k}_+^{(t)}\rangle)}{\exp(\langle\boldsymbol{q}_{n,i}^{(t)}, \boldsymbol{k}_+^{(t)}\rangle) + \sum_{j=2}^M \exp(\langle\boldsymbol{q}_{n,i}^{(t)}, \boldsymbol{k}_{n,j}^{(t)}\rangle)} \\
&\cdot \frac{\exp(\langle\boldsymbol{q}_{n,i}^{(t)}, \boldsymbol{k}_{n,k}^{(t)}\rangle)}{\exp(\langle\boldsymbol{q}_{n,i}^{(t)}, \boldsymbol{k}_+^{(t)}\rangle) + \sum_{j=2}^M \exp(\langle\boldsymbol{q}_{n,i}^{(t)}, \boldsymbol{k}_{n,j}^{(t)}\rangle)}\Bigg)\Bigg| \\
\le &\frac{\eta(\|\boldsymbol{\mu}\|_2+\tau)^2}{NM} \cdot \frac{3NM}{4} \cdot O(d_h^{-\frac{1}{4}}) + \frac{\eta(\|\boldsymbol{\mu}\|\tau + 2\sigma_p\tau\sqrt{2\log(4NM/\delta)}+\tau^2)}{NM} \cdot \frac{3NM(M-1)}{4} \cdot O(d_h^{-\frac{1}{4}}) \\
= &O\left(\frac{\eta(\|\boldsymbol{\mu}\|_2+\tau)^2}{d_h^{\frac{1}{4}}}\right)
\end{aligned}
$$

$$\tag{117}$$

where the inequality is by $-\widetilde{\ell}'^{(t)}_n \leq 1$ and the property that attention is smaller than 1 (e.g. $\frac{\exp(\langle \boldsymbol{q}^{(t)}_+, \boldsymbol{k}^{(t)}_+ \rangle)}{\exp(\langle \boldsymbol{q}^{(t)}_+, \boldsymbol{k}^{(t)}_+ \rangle) + \sum_{j=2}^{M} \exp(\langle \boldsymbol{q}^{(t)}_+, \boldsymbol{k}^{(t)}_{n,j} \rangle)} \leq 1$). We also have

$$
\begin{aligned}
|\alpha^{(t)}_{n,+,i}| = \Big| &-\frac{\eta}{NM} \ell'^{(t)}_n \langle \widetilde{\boldsymbol{\mu}}_+, \widetilde{\boldsymbol{\mu}}^{(t)}_+ \rangle \\
&\cdot \Bigg( -V^{(t)}_+ \cdot \frac{\exp(\langle \boldsymbol{q}^{(t)}_+, \boldsymbol{k}^{(t)}_+ \rangle)}{\exp(\langle \boldsymbol{q}^{(t)}_+, \boldsymbol{k}^{(t)}_+ \rangle) + \sum_{j=2}^{M} \exp(\langle \boldsymbol{q}^{(t)}_+, \boldsymbol{k}^{(t)}_{n,j} \rangle)} \\
&\quad + \frac{\exp(\langle \boldsymbol{q}^{(t)}_+, \boldsymbol{k}^{(t)}_{n,i} \rangle)}{\exp(\langle \boldsymbol{q}^{(t)}_+, \boldsymbol{k}^{(t)}_+ \rangle) + \sum_{j=2}^{M} \exp(\langle \boldsymbol{q}^{(t)}_+, \boldsymbol{k}^{(t)}_{n,j} \rangle)} \\
&\quad + V^{(t)}_{n,i} \Bigg( \frac{\exp(\langle \boldsymbol{q}^{(t)}_+, \boldsymbol{k}^{(t)}_{n,i} \rangle)}{\exp(\langle \boldsymbol{q}^{(t)}_+, \boldsymbol{k}^{(t)}_+ \rangle) + \sum_{j=2}^{M} \exp(\langle \boldsymbol{q}^{(t)}_+, \boldsymbol{k}^{(t)}_{n,j} \rangle)} \\
&\quad - \Bigg( \frac{\exp(\langle \boldsymbol{q}^{(t)}_+, \boldsymbol{k}^{(t)}_{n,i} \rangle)}{\exp(\langle \boldsymbol{q}^{(t)}_+, \boldsymbol{k}^{(t)}_+ \rangle) + \sum_{j=2}^{M} \exp(\langle \boldsymbol{q}^{(t)}_+, \boldsymbol{k}^{(t)}_{n,j} \rangle)} \Bigg)^2 \Bigg) \\
&\quad - \sum_{k \neq i} \Bigg( V^{(t)}_{n,k} \cdot \frac{\exp(\langle \boldsymbol{q}^{(t)}_+, \boldsymbol{k}^{(t)}_{n,i} \rangle)}{\exp(\langle \boldsymbol{q}^{(t)}_+, \boldsymbol{k}^{(t)}_+ \rangle) + \sum_{j=2}^{M} \exp(\langle \boldsymbol{q}^{(t)}_+, \boldsymbol{k}^{(t)}_{n,j} \rangle)} \\
&\qquad \cdot \frac{\exp(\langle \boldsymbol{q}^{(t)}_+, \boldsymbol{k}^{(t)}_{n,k} \rangle)}{\exp(\langle \boldsymbol{q}^{(t)}_+, \boldsymbol{k}^{(t)}_+ \rangle) + \sum_{j=2}^{M} \exp(\langle \boldsymbol{q}^{(t)}_+, \boldsymbol{k}^{(t)}_{n,j} \rangle)} \Bigg) \Bigg) \\
&+ \sum_{k=2}^{M} \langle \widetilde{\boldsymbol{\mu}}_+, \widetilde{\boldsymbol{\xi}}^{(t)}_{n,k} \rangle \cdot \Bigg( -V^{(t)}_+ \cdot \frac{\exp(\langle \boldsymbol{q}^{(t)}_{n,k}, \boldsymbol{k}^{(t)}_+ \rangle)}{\exp(\langle \boldsymbol{q}^{(t)}_{n,k}, \boldsymbol{k}^{(t)}_+ \rangle) + \sum_{j=2}^{M} \exp(\langle \boldsymbol{q}^{(t)}_{n,k}, \boldsymbol{k}^{(t)}_{n,j} \rangle)} \\
&\qquad \cdot \frac{\exp(\langle \boldsymbol{q}^{(t)}_{n,k}, \boldsymbol{k}^{(t)}_{n,i} \rangle)}{\exp(\langle \boldsymbol{q}^{(t)}_{n,k}, \boldsymbol{k}^{(t)}_+ \rangle) + \sum_{j=2}^{M} \exp(\langle \boldsymbol{q}^{(t)}_{n,k}, \boldsymbol{k}^{(t)}_{n,j} \rangle)} \\
&\quad + V^{(t)}_{n,i} \Bigg( \frac{\exp(\langle \boldsymbol{q}^{(t)}_{n,k}, \boldsymbol{k}^{(t)}_{n,i} \rangle)}{\exp(\langle \boldsymbol{q}^{(t)}_{n,k}, \boldsymbol{k}^{(t)}_+ \rangle) + \sum_{j=2}^{M} \exp(\langle \boldsymbol{q}^{(t)}_{n,k}, \boldsymbol{k}^{(t)}_{n,j} \rangle)} \\
&\quad - \Bigg( \frac{\exp(\langle \boldsymbol{q}^{(t)}_{n,k}, \boldsymbol{k}^{(t)}_{n,i} \rangle)}{\exp(\langle \boldsymbol{q}^{(t)}_{n,k}, \boldsymbol{k}^{(t)}_+ \rangle) + \sum_{j=2}^{M} \exp(\langle \boldsymbol{q}^{(t)}_{n,k}, \boldsymbol{k}^{(t)}_{n,j} \rangle)} \Bigg)^2 \Bigg) \\
&\quad - \sum_{l \neq i} \Bigg( V^{(t)}_{n,l} \cdot \frac{\exp(\langle \boldsymbol{q}^{(t)}_{n,k}, \boldsymbol{k}^{(t)}_{n,i} \rangle)}{\exp(\langle \boldsymbol{q}^{(t)}_{n,k}, \boldsymbol{k}^{(t)}_+ \rangle) + \sum_{j=2}^{M} \exp(\langle \boldsymbol{q}^{(t)}_{n,k}, \boldsymbol{k}^{(t)}_{n,j} \rangle)} \\
&\qquad \cdot \frac{\exp(\langle \boldsymbol{q}^{(t)}_{n,k}, \boldsymbol{k}^{(t)}_{n,l} \rangle)}{\exp(\langle \boldsymbol{q}^{(t)}_{n,k}, \boldsymbol{k}^{(t)}_+ \rangle) + \sum_{j=2}^{M} \exp(\langle \boldsymbol{q}^{(t)}_{n,k}, \boldsymbol{k}^{(t)}_{n,j} \rangle)} \Bigg) \Bigg| \\
&\leq \frac{\eta(\|\boldsymbol{\mu}\|_2 + \tau)^2}{NM} \cdot M \cdot O(d_h^{-\frac{1}{4}}) \\
&= O\left( \frac{\eta(\|\boldsymbol{\mu}\|_2 + \tau)^2}{d_h^{\frac{1}{4}} N} \right),
\end{aligned}
$$

where the inequality is by $-\widetilde{\ell}'^{(t)}_n \leq 1$, Lemma 13 and the property that attention is smaller than 1. Similarly, we have

$$
|\alpha^{(t)}_{-,-}|, |\beta^{(t)}_{+,+}|, |\beta^{(t)}_{-,-}| = O\left( \frac{\eta(\|\boldsymbol{\mu}\|_2 + \tau)^2}{d_h^{\frac{1}{4}}} \right),
$$

$$|\alpha_{n,-,l}^{(t)}|, |\beta_{n,+,l}^{(t)}|, |\beta_{n,-,l}^{(t)}| = O\left(\frac{\eta(\|\boldsymbol{\mu}\|_2 + \tau)^2}{d_h^{\frac{1}{4}} N}\right),$$

$$|\alpha_{n,l,-}^{(t)}|, |\beta_{n,l,+}^{(t)}|, |\beta_{n,l,-}^{(t)}|, |\alpha_{n,l,n',l'}^{(t)}|, |\beta_{n,l,n',l'}^{(t)}| = O\left(\frac{\eta(\|\boldsymbol{\mu}\|\tau + \sigma_p^2 d)}{d_h^{\frac{1}{4}} N}\right)$$

for $t \in [0, T_0]$. Next we use induction to show that the following proposition $\mathcal{A}(t)$ holds for $t \in [0, T_0]$. $\mathcal{A}(t)$:

$$|\langle q_\pm^{(t)}, k_\pm^{(t)}\rangle|, |\langle \boldsymbol{q}_{n,i}^{(t)}, k_\pm^{(t)}\rangle|, |\langle q_\pm^{(t)}, \boldsymbol{k}_{n,j}^{(t)}\rangle|, |\langle \boldsymbol{q}_{n,i}^{(t)}, \boldsymbol{k}_{n',j}^{(t)}\rangle|$$
$$= O\left(\max\{\|\boldsymbol{\mu}\|_2^2, \sigma_p^2 d\} \cdot \sigma_h^2 \cdot \sqrt{d_h \log(6N^2M^2/\delta)}\right),$$

$$|\langle q_\pm^{(t)}, q_\mp^{(t)}\rangle|, |\langle \boldsymbol{q}_{n,i}^{(t)}, q_\pm^{(t)}\rangle|, |\langle \boldsymbol{q}_{n,i}^{(t)}, \boldsymbol{q}_{n',j}^{(t)}\rangle|$$
$$= O\left(\max\{\|\boldsymbol{\mu}\|_2^2, \sigma_p^2 d\} \cdot \sigma_h^2 \cdot \sqrt{d_h \log(6N^2M^2/\delta)}\right),$$

$$|\langle k_\pm^{(t)}, k_\mp^{(t)}\rangle|, |\langle \boldsymbol{k}_{n,i}^{(t)}, k_\pm^{(t)}\rangle|, |\langle \boldsymbol{k}_{n,i}^{(t)}, \boldsymbol{k}_{n',j}^{(t)}\rangle|$$
$$= O\left(\max\{\|\boldsymbol{\mu}\|_2^2, \sigma_p^2 d\} \cdot \sigma_h^2 \cdot \sqrt{d_h \log(6N^2M^2/\delta)}\right),$$

$$\|q_\pm^{(t)}\|_2^2, \|k_\pm^{(t)}\|_2^2 = \Theta(\|\boldsymbol{\mu}\|_2^2 \sigma_h^2 d_h)$$

$$\|\boldsymbol{q}_{n,i}^{(t)}\|_2^2, \|\boldsymbol{k}_{n,i}^{(t)}\|_2^2 = \Theta(\sigma_p^2 \sigma_h^2 d d_h)$$

for $i, j \in [M]\backslash\{1\}, n, n' \in [N]$.

By Lemma 12 we know that $\mathcal{A}(0)$ is true. Now we assume $\mathcal{A}(0), \ldots, \mathcal{A}(T)$ is true, then we need to prove that $\mathcal{A}(T+1)$ is true. We first proof $|\langle \boldsymbol{q}_+^{(T+1)}, \boldsymbol{k}_+^{(T+1)}\rangle| = O\left(\max\{\|\boldsymbol{\mu}\|_2^2, \sigma_p^2 d\} \cdot \sigma_h^2 \cdot \sqrt{d_h \log(6N^2M^2/\delta)}\right)$, as an example.

$$|\langle \boldsymbol{q}_+^{(t+1)}, \boldsymbol{k}_+^{(t+1)}\rangle - \langle \boldsymbol{q}_+^{(t)}, \boldsymbol{k}_+^{(t)}\rangle| = \left| \alpha_{+,+}^{(t)} \|\boldsymbol{k}_+^{(t)}\|_2^2 + \sum_{n \in S_+} \sum_{i=2}^{M} \alpha_{n,+,i}^{(t)} \langle \boldsymbol{k}_+^{(t)}, \boldsymbol{k}_{n,i}^{(t)}\rangle \right.$$

$$+ \beta_{+,+}^{(t)} \|\boldsymbol{q}_+^{(t)}\|_2^2 + \sum_{n \in S_+} \sum_{i=2}^{M} \beta_{n,+,i}^{(t)} \langle \boldsymbol{q}_+^{(t)}, \boldsymbol{q}_{n,i}^{(t)}\rangle$$

$$+ \left( \alpha_{+,+}^{(t)} \boldsymbol{k}_+^{(t)} + \sum_{n \in S_+} \sum_{i=2}^{M} \alpha_{n,+,i}^{(t)} \boldsymbol{k}_{n,i}^{(t)} \right)$$

$$\left. \cdot \left( \beta_{+,+}^{(t)} \boldsymbol{q}_+^{(t)\top} + \sum_{n\top} \sum_{i=2}^{M} \beta_{n,+,i}^{(t)} \boldsymbol{q}_{n,i}^{(t)\top} \right) \right|$$

$$\leq O\left(\frac{\eta(\|\boldsymbol{\mu}\|_2 + \tau)^2}{d_h^{\frac{1}{4}}}\right) \cdot \Theta(\|\boldsymbol{\mu}\|_2^2 \sigma_h^2 d_h)$$

$$+ NM \cdot O\left(\frac{\eta(\|\boldsymbol{\mu}\|_2 + \tau)^2}{d_h^{\frac{1}{4}} N}\right) \cdot O\left(\max\{\|\boldsymbol{\mu}\|_2^2, \sigma_p^2 d\} \cdot \sigma_h^2 \cdot \sqrt{d_h \log(6N^2M^2/\delta)}\right)$$

$$+ \{\text{lower order term}\} = O\left(\eta\|\boldsymbol{\mu}\|_2^2(\|\boldsymbol{\mu}\|_2 + \tau)^2 \sigma_h^2 d_h^{\frac{3}{4}}\right)$$

Taking a summation, we obtain that

$$
\begin{aligned}
|\langle \boldsymbol{q}_+^{(T+1)}, \boldsymbol{k}_+^{(T+1)} \rangle| \leq & |\langle \boldsymbol{q}_+^{(0)}, \boldsymbol{k}_+^{(0)} \rangle| + \sum_{t=0}^{T} |\langle \boldsymbol{q}_+^{(t+1)}, \boldsymbol{k}_+^{(t+1)} \rangle - \langle \boldsymbol{q}_+^{(t)}, \boldsymbol{k}_+^{(t)} \rangle| \\
\leq & |\langle \boldsymbol{q}_+^{(0)}, \boldsymbol{k}_+^{(0)} \rangle| + \sum_{t=0}^{T_0-1} |\langle \boldsymbol{q}_+^{(t+1)}, \boldsymbol{k}_+^{(t+1)} \rangle - \langle \boldsymbol{q}_+^{(t)}, \boldsymbol{k}_+^{(t)} \rangle| \\
\leq & O\left( \max\{\|\boldsymbol{\mu}\|_2^2, \sigma_p^2 d\} \cdot \sigma_h^2 \cdot \sqrt{d_h \log(6N^2 M^2/\delta)} \right) \\
& + O\left( \frac{1}{\eta d_h^{\frac{1}{4}} (\|\boldsymbol{\mu}\|_2 + \tau)^2 \|\boldsymbol{w}_O\|_2^2} \right) \cdot O(\eta \|\boldsymbol{\mu}\|_2^2 (\|\boldsymbol{\mu}\|_2 + \tau)^2 \sigma_h^2 d_h^{\frac{3}{4}}) \\
= & O\left( \max\{\|\boldsymbol{\mu}\|_2^2, \sigma_p^2 d\} \cdot \sigma_h^2 \cdot \sqrt{d_h \log(6N^2 M^2/\delta)} \right) + O\left( \|\boldsymbol{\mu}\|_2^2 \sigma_h^2 d_h^{\frac{1}{2}} \right)
\end{aligned}
$$

Similarly to $\langle \boldsymbol{q}_+^{(t)}, \boldsymbol{k}_+^{(t)} \rangle$, it is easy to know that the inner product does not change by a magnitude more than the product of $\max\{\alpha, \beta\}$ and $\max\{\langle q, q \rangle, \langle k, k \rangle\}$ in a single iteration, which can be expressed as follows

$$
\begin{aligned}
& |\langle \boldsymbol{q}^{(t+1)}, \boldsymbol{k}^{(t+1)} \rangle - \langle \boldsymbol{q}^{(t)}, \boldsymbol{k}^{(t)} \rangle| \\
& = O\left( \max\left\{ \frac{\eta(\|\boldsymbol{\mu}\|_2 + \tau)^2}{d_h^{\frac{1}{4}}}, \frac{\eta(\sigma_p^2 d + \|\boldsymbol{\mu}\|\tau)}{d_h^{\frac{1}{4}} N} \right\} \right) \cdot \Theta(\max\{\|\boldsymbol{\mu}\|_2^2 \sigma_h^2 d_h, \sigma_p^2 \sigma_h^2 d d_h\}) \\
& = O\left( \frac{\eta(\|\boldsymbol{\mu}\|_2 + \tau)^2}{d_h^{\frac{1}{4}}} \right) \cdot \Theta(\max\{\|\boldsymbol{\mu}\|_2^2 \sigma_h^2 d_h, \sigma_p^2 \sigma_h^2 d d_h\}) \\
& = O\left( (\|\boldsymbol{\mu}\|_2 + \tau)^2 \sigma_h^2 d_h^{\frac{3}{4}} \cdot \max\{\|\boldsymbol{\mu}\|_2^2, \sigma_p^2 d\} \right)
\end{aligned}
$$

where the second equality is by the condition that $N \cdot \mathrm{SNR}^2 = \Omega(1)$.

Taking a summation, we obtain that

$$
\begin{aligned}
|\langle \boldsymbol{q}^{(T+1)}, \boldsymbol{k}^{(T+1)} \rangle - \langle \boldsymbol{q}^{(0)}, \boldsymbol{k}^{(0)} \rangle| \leq & \sum_{t=0}^{T-1} |\langle \boldsymbol{q}^{(t+1)}, \boldsymbol{k}^{(t+1)} \rangle - \langle \boldsymbol{q}^{(t)}, \boldsymbol{k}^{(t)} \rangle| \\
\leq & \sum_{t=0}^{T_0-1} O\left( \eta(\|\boldsymbol{\mu}\|_2 + \tau)^2 \sigma_h^2 d_h^{\frac{3}{4}} \cdot \max\{\|\boldsymbol{\mu}\|_2^2, \sigma_p^2 d\} \right) \\
= & O\left( \frac{1}{\eta d_h^{\frac{1}{4}} (\|\boldsymbol{\mu}\|_2 + \tau)^2 \|\boldsymbol{w}_O\|_2^2} \right) \cdot O\left( \eta(\|\boldsymbol{\mu}\|_2 + \tau)^2 \sigma_h^2 d_h^{\frac{3}{4}} \cdot \max\{\|\boldsymbol{\mu}\|_2^2, \sigma_p^2 d\} \right) \\
= & O\left( \max\{\|\boldsymbol{\mu}\|_2^2, \sigma_p^2 d\} \cdot \sigma_h^2 d_h^{\frac{1}{4}} \right).
\end{aligned}
$$

It is clear that the magnitude of $\langle \boldsymbol{q}^{(T+1)}, \boldsymbol{k}^{(T+1)} \rangle - \langle \boldsymbol{q}^{(0)}, \boldsymbol{k}^{(0)} \rangle$ is smaller than $\max\{\|\boldsymbol{\mu}\|_2^2, \sigma_p^2 d\} \cdot \sigma_h^2 \cdot \sqrt{d_h \log(6N^2 M^2/\delta)}$. Thus the magnitude of the bound for $\langle \boldsymbol{q}^{(T+1)}, \boldsymbol{k}^{(T+1)} \rangle$ is the same as that of $\langle \boldsymbol{q}^{(T)}, \boldsymbol{k}^{(T)} \rangle$. The proof for $\langle \boldsymbol{q}^{(T+1)}, \boldsymbol{q}^{(T+1)} \rangle$ and $\langle \boldsymbol{k}^{(T+1)}, \boldsymbol{k}^{(T+1)} \rangle$ is exactly the same, and we can conclude the proof by an induction.

### I.5   LOWER BOUNDS OF $\alpha$ AND $\beta$

In this subsection, we present some bounds for $\alpha$ and $\beta$ which can be used in F.2 and F.3. All the calculations in this subsection are based on the precise expression for $\alpha$ and $\beta$ in I.1 and assume that $\mathcal{B}(T_1), \ldots, \mathcal{B}(s), \mathcal{D}(T_1), \ldots, \mathcal{D}(s-1)$ hold ($s \in [T_1, t]$). Then the following propositions hold:

$$
V_+^{(s)} \geq 3M \cdot |V_{n,i}^{(s)}|,
$$

$$V_-^{(s)} \leq -3M \cdot |V_{n,i}^{(s)}|,$$

$$\text{softmax}(\langle \boldsymbol{q}_\pm^{(s)}, \boldsymbol{k}_\pm^{(s)} \rangle), \text{softmax}(\langle \boldsymbol{q}_{n,i}^{(s)}, \boldsymbol{k}_\pm^{(s)} \rangle) \geq \frac{1}{M} - o(1),$$

$$\text{softmax}(\langle \boldsymbol{q}_\pm^{(s)}, \boldsymbol{k}_{n,j}^{(s)} \rangle), \text{softmax}(\langle \boldsymbol{q}_{n,i}^{(s)}, \boldsymbol{k}_{n,j}^{(s)} \rangle) \leq \frac{1}{M} + o(1).$$

Now we give the bounds respectively for $\alpha_{+,+}^{(s)}, \alpha_{n,+,i}^{(s)}, \alpha_{-,-}^{(s)}, \alpha_{n,-,i}^{(s)}, \alpha_{n,i,+}^{(s)}, \alpha_{n,i,-}^{(s)}, \alpha_{n,i,n',i'}^{(s)},$ $\beta_{+,+}^{(s)}, \beta_{n,+,i}^{(s)}, \beta_{-,-}^{(s)}, \beta_{n,-,i}^{(s)}, \beta_{n,i,+}^{(s)}, \beta_{n,i,-}^{(s)}, \beta_{n,i,n',i'}^{(s)}.$

$$\alpha_{+,+}^{(s)} = \frac{\eta}{NM} \sum_{n \in S_+} -\widetilde{\ell}_n'(\theta) \langle \widetilde{\boldsymbol{\mu}}_+, \widetilde{\boldsymbol{\mu}}_+^{(s)} \rangle$$

$$\cdot \left( V_+^{(s)} \left( \frac{\exp(\langle \boldsymbol{q}_+^{(s)}, \boldsymbol{k}_+^{(s)} \rangle)}{\exp(\langle \boldsymbol{q}_+^{(s)}, \boldsymbol{k}_+^{(s)} \rangle) + \sum_{j=2}^M \exp(\langle \boldsymbol{q}_+^{(s)}, \boldsymbol{k}_{n,j}^{(s)} \rangle)} \right.\right.$$

$$\left.\left. - \left( \frac{\exp(\langle \boldsymbol{q}_+^{(s)}, \boldsymbol{k}_+^{(s)} \rangle)}{\exp(\langle \boldsymbol{q}_+^{(s)}, \boldsymbol{k}_+^{(s)} \rangle) + \sum_{j=2}^M \exp(\langle \boldsymbol{q}_+^{(s)}, \boldsymbol{k}_{n,j}^{(s)} \rangle)} \right)^2 \right) \right.$$

$$\left. - \sum_{i=2}^M \left( V_{n,i}^{(s)} \cdot \frac{\exp(\langle \boldsymbol{q}_+^{(s)}, \boldsymbol{k}_+^{(s)} \rangle)}{\exp(\langle \boldsymbol{q}_+^{(s)}, \boldsymbol{k}_+^{(s)} \rangle) + \sum_{j=2}^M \exp(\langle \boldsymbol{q}_+^{(s)}, \boldsymbol{k}_{n,j}^{(s)} \rangle)} \right.\right.$$

$$\left.\left. \cdot \frac{\exp(\langle \boldsymbol{q}_+^{(s)}, \boldsymbol{k}_{n,i}^{(s)} \rangle)}{\exp(\langle \boldsymbol{q}_+^{(s)}, \boldsymbol{k}_+^{(s)} \rangle) + \sum_{j=2}^M \exp(\langle \boldsymbol{q}_+^{(s)}, \boldsymbol{k}_{n,j}^{(s)} \rangle)} \right) \right)$$

$$+ \sum_{i=2}^M \langle \widetilde{\boldsymbol{\mu}}_+, \widetilde{\boldsymbol{\xi}}_{n,i}^{(s)} \rangle \cdot \left( V_+^{(s)} \left( \frac{\exp(\langle \boldsymbol{q}_{n,i}^{(s)}, \boldsymbol{k}_+^{(s)} \rangle)}{\exp(\langle \boldsymbol{q}_{n,i}^{(s)}, \boldsymbol{k}_+^{(s)} \rangle) + \sum_{j=2}^M \exp(\langle \boldsymbol{q}_{n,i}^{(s)}, \boldsymbol{k}_{n,j}^{(s)} \rangle)} \right.\right.$$

$$\left.\left. - \left( \frac{\exp(\langle \boldsymbol{q}_{n,i}^{(s)}, \boldsymbol{k}_+^{(s)} \rangle)}{\exp(\langle \boldsymbol{q}_{n,i}^{(s)}, \boldsymbol{k}_+^{(s)} \rangle) + \sum_{j=2}^M \exp(\langle \boldsymbol{q}_{n,i}^{(s)}, \boldsymbol{k}_{n,j}^{(s)} \rangle)} \right)^2 \right) \right.$$

$$\left. - \sum_{k=2}^M \left( V_{n,i}^{(s)} \cdot \frac{\exp(\langle \boldsymbol{q}_{n,i}^{(s)}, \boldsymbol{k}_+^{(s)} \rangle)}{\exp(\langle \boldsymbol{q}_{n,i}^{(s)}, \boldsymbol{k}_+^{(s)} \rangle) + \sum_{j=2}^M \exp(\langle \boldsymbol{q}_{n,i}^{(s)}, \boldsymbol{k}_{n,j}^{(s)} \rangle)} \right.\right.$$

$$\left.\left. \cdot \frac{\exp(\langle \boldsymbol{q}_{n,i}^{(s)}, \boldsymbol{k}_{n,k}^{(s)} \rangle)}{\exp(\langle \boldsymbol{q}_{n,i}^{(s)}, \boldsymbol{k}_+^{(s)} \rangle) + \sum_{j=2}^M \exp(\langle \boldsymbol{q}_{n,i}^{(s)}, \boldsymbol{k}_{n,j}^{(s)} \rangle)} \right) \right)$$

$$\geq \frac{\eta}{NM} \sum_{n \in S_+} -\widetilde{\ell}_n'^{(s)} \langle \widetilde{\boldsymbol{\mu}}_+, \widetilde{\boldsymbol{\mu}}_+^{(s)} \rangle \cdot \frac{\exp(\langle \boldsymbol{q}_+^{(s)}, \boldsymbol{k}_+^{(s)} \rangle)}{\exp(\langle \boldsymbol{q}_+^{(s)}, \boldsymbol{k}_+^{(s)} \rangle) + \sum_{j=2}^M \exp(\langle \boldsymbol{q}_+^{(s)}, \boldsymbol{k}_{n,j}^{(s)} \rangle)}$$

$$\cdot \left( V_+^{(s)} \left( 1 - \frac{\exp(\langle \boldsymbol{q}_+^{(s)}, \boldsymbol{k}_+^{(s)} \rangle)}{\exp(\langle \boldsymbol{q}_+^{(s)}, \boldsymbol{k}_+^{(s)} \rangle) + \sum_{j=2}^M \exp(\langle \boldsymbol{q}_+^{(s)}, \boldsymbol{k}_{n,j}^{(s)} \rangle)} \right) \right.$$

$$\left. - \frac{1}{2} \cdot V_+^{(s)} \sum_{i=2}^M \frac{\exp(\langle \boldsymbol{q}_+^{(s)}, \boldsymbol{k}_{n,i}^{(s)} \rangle)}{\exp(\langle \boldsymbol{q}_+^{(s)}, \boldsymbol{k}_+^{(s)} \rangle) + \sum_{j=2}^M \exp(\langle \boldsymbol{q}_+^{(s)}, \boldsymbol{k}_{n,j}^{(s)} \rangle)} \right)$$

$$+ \frac{\eta}{NM} \sum_{n \in S_+} -\widetilde{\ell}_n'^{(s)} \langle \widetilde{\boldsymbol{\mu}}_+, \widetilde{\boldsymbol{\xi}}_{n,i}^{(s)} \rangle \cdot \frac{\exp(\langle \boldsymbol{q}_{n,i}^{(s)}, \boldsymbol{k}_+^{(s)} \rangle)}{\exp(\langle \boldsymbol{q}_{n,i}^{(s)}, \boldsymbol{k}_+^{(s)} \rangle) + \sum_{j=2}^M \exp(\langle \boldsymbol{q}_{n,i}^{(s)}, \boldsymbol{k}_{n,j}^{(s)} \rangle)}$$

$$\cdot \left( V_+^{(s)} \left( 1 - \frac{\exp(\langle \boldsymbol{q}_{n,i}^{(s)}, \boldsymbol{k}_+^{(s)} \rangle)}{\exp(\langle \boldsymbol{q}_{n,i}^{(s)}, \boldsymbol{k}_+^{(s)} \rangle) + \sum_{j=2}^M \exp(\langle \boldsymbol{q}_{n,i}^{(s)}, \boldsymbol{k}_{n,j}^{(s)} \rangle)} \right) \right)$$

$$-\frac{1}{2} \cdot V_+^{(s)} \sum_{k=2}^{M} \frac{\exp(\langle \boldsymbol{q}_{n,i}^{(s)}, \boldsymbol{k}_{n,k}^{(s)} \rangle)}{\exp(\langle \boldsymbol{q}_{n,i}^{(s)}, \boldsymbol{k}_+^{(s)} \rangle) + \sum_{j=2}^{M} \exp(\langle \boldsymbol{q}_{n,i}^{(s)}, \boldsymbol{k}_{n,j}^{(s)} \rangle)} \Bigg)$$

$$\geq \frac{\eta}{2NM} \sum_{n \in S_+} -\ell_n'^{(s)}(\|\boldsymbol{\mu}\|_2 - \tau)^2 V_+^{(s)} \cdot \frac{\exp(\langle \boldsymbol{q}_+^{(s)}, \boldsymbol{k}_+^{(s)} \rangle)}{\exp(\langle \boldsymbol{q}_+^{(s)}, \boldsymbol{k}_+^{(s)} \rangle) + \sum_{j=2}^{M} \exp(\langle \boldsymbol{q}_+^{(s)}, \boldsymbol{k}_{n,j}^{(s)} \rangle)}$$

$$\cdot \left( 1 - \frac{\exp(\langle \boldsymbol{q}_+^{(s)}, \boldsymbol{k}_+^{(s)} \rangle)}{\exp(\langle \boldsymbol{q}_+^{(s)}, \boldsymbol{k}_+^{(s)} \rangle) + \sum_{j=2}^{M} \exp(\langle \boldsymbol{q}_+^{(s)}, \boldsymbol{k}_{n,j}^{(s)} \rangle)} \right)$$

$$+ \sum_{i=2}^{M} (\|\boldsymbol{\mu}\|\tau + \sigma_p \tau \sqrt{2\log(4NM/\delta)} + \tau^2) \cdot \Bigg( V_+^{(s)} \left( \frac{\exp(\langle \boldsymbol{q}_{n,i}^{(s)}, \boldsymbol{k}_+^{(s)} \rangle)}{\exp(\langle \boldsymbol{q}_{n,i}^{(s)}, \boldsymbol{k}_+^{(s)} \rangle) + \sum_{j=2}^{M} \exp(\langle \boldsymbol{q}_{n,i}^{(s)}, \boldsymbol{k}_{n,j}^{(s)} \rangle)} \right.$$

$$\left. - \left( \frac{\exp(\langle \boldsymbol{q}_{n,i}^{(s)}, \boldsymbol{k}_+^{(s)} \rangle)}{\exp(\langle \boldsymbol{q}_{n,i}^{(s)}, \boldsymbol{k}_+^{(s)} \rangle) + \sum_{j=2}^{M} \exp(\langle \boldsymbol{q}_{n,i}^{(s)}, \boldsymbol{k}_{n,j}^{(s)} \rangle)} \right)^2 \right) \Bigg)$$

where the first inequality is by $V_+^{(s)} \geq 3M \cdot |V_{n,i}^{(s)}|$, the second inequality is by the fact that the sum of attention equal to 1 and Lemma 13. Similarly, we have

$$\beta_{+,+}^{(s)} \geq \frac{\eta}{2NM} \sum_{n \in S_+} -\widetilde{\ell}_n'^{(s)}(\|\boldsymbol{\mu}\|_2 - \tau)^2 V_+^{(s)} \cdot \frac{\exp(\langle \boldsymbol{q}_+^{(s)}, \boldsymbol{k}_+^{(s)} \rangle)}{\exp(\langle \boldsymbol{q}_+^{(s)}, \boldsymbol{k}_+^{(s)} \rangle) + (M-1)\exp(\max_j \langle \boldsymbol{q}_+^{(s)}, \boldsymbol{k}_{n,j}^{(s)} \rangle)}$$

$$\cdot \frac{\exp(\max_j \langle \boldsymbol{q}_+^{(s)}, \boldsymbol{k}_{n,j}^{(s)} \rangle)}{\exp(\langle \boldsymbol{q}_+^{(s)}, \boldsymbol{k}_+^{(s)} \rangle) + (M-1)\exp(\max_j \langle \boldsymbol{q}_+^{(s)}, \boldsymbol{k}_{n,j}^{(s)} \rangle)} + \{\text{lower order term}\} \tag{118}$$

$$\alpha_{-,-}^{(s)} \geq \frac{\eta}{2NM} \sum_{n \in S_-} \widetilde{\ell}_n'^{(s)}(\|\boldsymbol{\mu}\|_2 - \tau)^2 V_-^{(s)} \cdot \frac{\exp(\langle \boldsymbol{q}_-^{(s)}, \boldsymbol{k}_-^{(s)} \rangle)}{\exp(\langle \boldsymbol{q}_-^{(s)}, \boldsymbol{k}_-^{(s)} \rangle) + (M-1)\exp(\max_j \langle \boldsymbol{q}_-^{(s)}, \boldsymbol{k}_{n,j}^{(s)} \rangle)}$$

$$\cdot \frac{\exp(\max_j \langle \boldsymbol{q}_-^{(s)}, \boldsymbol{k}_{n,j}^{(s)} \rangle)}{\exp(\langle \boldsymbol{q}_-^{(s)}, \boldsymbol{k}_-^{(s)} \rangle) + (M-1)\exp(\max_j \langle \boldsymbol{q}_-^{(s)}, \boldsymbol{k}_{n,j}^{(s)} \rangle)} + \{\text{lower order term}\} \tag{119}$$

Similarly, by applying the update rules in Section I.3, we can derive the following bounds on $\alpha$ and $\beta$.

$$\alpha_{+,+}^{(s)}, \alpha_{-,-}^{(s)}, \beta_{+,+}^{(s)}, \beta_{-,-}^{(s)}, \alpha_{n,i,+}^{(s)}, \alpha_{n,i,-}^{(s)}, \beta_{n,+,i}^{(s)}, \beta_{n,-,i}^{(s)} \geq 0,$$

$$\alpha_{n,+,i}^{(s)}, \alpha_{n,-,i}^{(s)}, \alpha_{n,i,n,j}^{(s)}, \beta_{n,i,+}^{(s)}, \beta_{n,i,-}^{(s)}, \beta_{n,j,n,i}^{(s)} \leq 0.$$

## I.6 Lower Bounds of $\langle \mathbf{q}, \mathbf{k} \rangle$

In order to give the lower bounds for $\langle q, k \rangle$, we need to rewrite the bounds of $\alpha$ and $\beta$ in a more concise form. We first expand the equations in I.5 under the assumption that $\mathcal{B}(s)$ and $\mathcal{E}(s)$ holds for $s \in [T_1, t]$.

$$\alpha_{+,+}^{(s)} \geq \frac{\eta}{2NM} \sum_{n \in S_+} -\widetilde{\ell}_n'^{(s)}(\|\boldsymbol{\mu}\|_2 - \tau)^2 V_+^{(s)} \cdot \frac{\exp(\langle \boldsymbol{q}_+^{(s)}, \boldsymbol{k}_+^{(s)} \rangle)}{\exp(\langle \boldsymbol{q}_+^{(s)}, \boldsymbol{k}_+^{(s)} \rangle) + \sum_{j'=2}^{M} \exp(\langle \boldsymbol{q}_+^{(s)}, \boldsymbol{k}_{n,j'}^{(s)} \rangle)}$$

$$\cdot \left( 1 - \frac{\exp(\langle \boldsymbol{q}_+^{(s)}, \boldsymbol{k}_+^{(s)} \rangle)}{\exp(\langle \boldsymbol{q}_+^{(s)}, \boldsymbol{k}_+^{(s)} \rangle) + \sum_{j'=2}^{M} \exp(\langle \boldsymbol{q}_+^{(s)}, \boldsymbol{k}_{n,j'}^{(s)} \rangle)} \right)$$

$$+ \sum_{i=2}^{M} (\|\boldsymbol{\mu}\|_2\tau + \sigma_p \tau \sqrt{2\log(4NM/\delta)} + \tau^2) \cdot \Bigg( V_+^{(s)} \left( \frac{\exp(\langle \boldsymbol{q}_{n,i}^{(s)}, \boldsymbol{k}_+^{(s)} \rangle)}{\exp(\langle \boldsymbol{q}_{n,i}^{(s)}, \boldsymbol{k}_+^{(s)} \rangle) + \sum_{j=2}^{M} \exp(\langle \boldsymbol{q}_{n,i}^{(s)}, \boldsymbol{k}_{n,j}^{(s)} \rangle)} \right.$$

$$-\left(\frac{\exp(\langle \boldsymbol{q}_{n,i}^{(s)}, \boldsymbol{k}_{+}^{(s)}\rangle)}{\exp(\langle \boldsymbol{q}_{n,i}^{(s)}, \boldsymbol{k}_{+}^{(s)}\rangle) + \sum_{j=2}^{M}\exp(\langle \boldsymbol{q}_{n,i}^{(s)}, \boldsymbol{k}_{n,j}^{(s)}\rangle)}\right)^2\Bigg)$$

$$= \frac{\eta}{2NM}\sum_{n\in S_+} -\widetilde{\ell}_n'^{(s)}(\|\boldsymbol{\mu}\|_2 - \tau)^2 V_+^{(s)}\cdot \frac{\exp(\langle \boldsymbol{q}_{+}^{(s)}, \boldsymbol{k}_{+}^{(s)}\rangle)}{\exp(\langle \boldsymbol{q}_{+}^{(s)}, \boldsymbol{k}_{+}^{(s)}\rangle) + \sum_{j'=2}^{M}\exp(\langle \boldsymbol{q}_{+}^{(s)}, \boldsymbol{k}_{n,j'}^{(s)}\rangle)}$$

$$\cdot \frac{\sum_{j=2}^{M}\exp(\langle \boldsymbol{q}_{+}^{(s)}, \boldsymbol{k}_{n,j}^{(s)}\rangle)}{\exp(\langle \boldsymbol{q}_{+}^{(s)}, \boldsymbol{k}_{+}^{(s)}\rangle) + \sum_{j'=2}^{M}\exp(\langle \boldsymbol{q}_{+}^{(s)}, \boldsymbol{k}_{n,j'}^{(s)}\rangle)} + \{\text{lower term}\}$$

$$\geq \frac{\eta}{2NM} -\widetilde{\ell}_n'^{(s)}(\|\boldsymbol{\mu}\|_2 - \tau)^2 V_+^{(s)}\cdot \frac{\exp(\langle \boldsymbol{q}_{+}^{(s)}, \boldsymbol{k}_{+}^{(s)}\rangle)}{\exp(\langle \boldsymbol{q}_{+}^{(s)}, \boldsymbol{k}_{+}^{(s)}\rangle) + \sum_{j'=2}^{M}\exp(\langle \boldsymbol{q}_{+}^{(s)}, \boldsymbol{k}_{n,j'}^{(s)}\rangle)}$$

$$\cdot \frac{\exp(\langle \boldsymbol{q}_{+}^{(s)}, \boldsymbol{k}_{n,j}^{(s)}\rangle)}{\exp(\langle \boldsymbol{q}_{+}^{(s)}, \boldsymbol{k}_{+}^{(s)}\rangle) + \sum_{j'=2}^{M}\exp(\langle \boldsymbol{q}_{+}^{(s)}, \boldsymbol{k}_{n,j'}^{(s)}\rangle)} + \{\text{lower term}\}$$

$$\geq \frac{\eta}{2NM} -\widetilde{\ell}_n'^{(s)}(\|\boldsymbol{\mu}\|_2 - \tau)^2 V_+^{(s)}\cdot \left(\frac{1}{M} - o(1)\right)\cdot \frac{\exp(\langle \boldsymbol{q}_{+}^{(s)}, \boldsymbol{k}_{n,j}^{(s)}\rangle)}{\exp(\langle \boldsymbol{q}_{+}^{(s)}, \boldsymbol{k}_{+}^{(s)}\rangle) + \sum_{j'=2}^{M}\exp(\langle \boldsymbol{q}_{+}^{(s)}, \boldsymbol{k}_{n,j'}^{(s)}\rangle)}$$

$$\geq \frac{\eta}{2NM} -\widetilde{\ell}_n'^{(s)}(\|\boldsymbol{\mu}\|_2 - \tau)^2 V_+^{(s)}\cdot \left(\frac{1}{M} - o(1)\right)\cdot \frac{\exp(\langle \boldsymbol{q}_{+}^{(s)}, \boldsymbol{k}_{n,j}^{(s)}\rangle)}{C\exp(\langle \boldsymbol{q}_{+}^{(s)}, \boldsymbol{k}_{+}^{(s)}\rangle)} + \{\text{lower term}\}$$

$$\geq \frac{\eta^2 C_5(\|\boldsymbol{\mu}\|_2 - \tau)^4\|\boldsymbol{w}_O\|_2^2(s - T_1)}{N}\cdot \frac{1}{\exp(\Lambda_{n,+,j}^{(s)})},$$

where the first inequality is by Lemma 20(as $\mathcal{E}(s)$ holds) and $\text{softmax}(\langle \boldsymbol{q}_{+}^{(s)}, \boldsymbol{k}_{+}^{(s)}\rangle) \geq \left(\frac{1}{M} - o(1)\right)$. In the fourth inequality, by $\langle \boldsymbol{q}^{(T_1)}, \boldsymbol{k}^{(T_1)}\rangle = o(1)$ and the monotonicity of $\langle \boldsymbol{q}_{+}^{(s)}, \boldsymbol{k}_{+}^{(s)}\rangle$ ($\langle \boldsymbol{q}_{+}^{(s)}, \boldsymbol{k}_{+}^{(s)}\rangle$ is increasing and $\langle \boldsymbol{q}_{+}^{(s)}, \boldsymbol{k}_{n,j}^{(s)}\rangle$ is decreasing), there exist a constant $C$ such that $C\exp(\langle \boldsymbol{q}_{+}^{(s)}, \boldsymbol{k}_{+}^{(s)}\rangle) \geq \exp(\langle \boldsymbol{q}_{+}^{(s)}, \boldsymbol{k}_{+}^{(s)}\rangle) + \sum_{j'=2}^{M}\exp(\langle \boldsymbol{q}_{+}^{(s)}, \boldsymbol{k}_{n,j'}^{(s)}\rangle)$. In the last inequality, we plugging the lower bounds of $V_+^{(s)}$ and $-\widetilde{\ell}_n'^{(s)}$ and then absorb all the constant factors. Similarly, we have

$$\beta_{+,+}^{(s)} \geq \frac{\eta^2 C_5(\|\boldsymbol{\mu}\|_2 - \tau)^4\|\boldsymbol{w}_O\|_2^2(s - T_1)}{N}\cdot \frac{1}{\exp(\Lambda_{n,+,j}^{(s)})}, \tag{120}$$

$$\alpha_{-,-}^{(s)} \geq \frac{\eta^2 C_5(\|\boldsymbol{\mu}\|_2 - \tau)^4\|\boldsymbol{w}_O\|_2^2(s - T_1)}{N}\cdot \frac{1}{\exp(\Lambda_{n,-,j}^{(s)})}, \tag{121}$$

$$\beta_{-,-}^{(s)} \geq \frac{\eta^2 C_5(\|\boldsymbol{\mu}\|_2 - \tau)^4\|\boldsymbol{w}_O\|_2^2(s - T_1)}{N}\cdot \frac{1}{\exp(\Lambda_{n,-,j}^{(s)})}. \tag{122}$$

With the concise lower bounds for $\alpha$ and $\beta$ above and proposition $\mathcal{C}(s)$, we will give the lower bounds for the dynamics of $\langle \boldsymbol{q}, \boldsymbol{k}\rangle$.

$$\langle \boldsymbol{q}_{+}^{(s+1)}, \boldsymbol{k}_{+}^{(s+1)}\rangle - \langle \boldsymbol{q}_{+}^{(s)}, \boldsymbol{k}_{+}^{(s)}\rangle = \alpha_{+,+}^{(s)}\|\boldsymbol{k}_{+}^{(s)}\|_2^2 + \sum_{n\in S_+}\sum_{i=2}^{M}\alpha_{n,+,i}^{(s)}\langle \boldsymbol{k}_{+}^{(s)}, \boldsymbol{k}_{n,i}^{(s)}\rangle$$

$$+ \beta_{+,+}^{(s)}\|\boldsymbol{q}_{+}^{(s)}\|_2^2 + \sum_{n\in S_+}\sum_{i=2}^{M}\beta_{n,+,i}^{(s)}\langle \boldsymbol{q}_{+}^{(s)}, \boldsymbol{q}_{n,i}^{(s)}\rangle$$

$$+ \left(\alpha_{+,+}^{(s)}\boldsymbol{k}_{+}^{(s)} + \sum_{n\in S_+}\sum_{i=2}^{M}\alpha_{n,+,i}^{(s)}\boldsymbol{k}_{n,i}^{(s)}\right)\cdot\left(\beta_{+,+}^{(s)}\boldsymbol{q}_{+}^{(s)} + \sum_{n\in S_+}\sum_{i=2}^{M}\beta_{n,+,i}^{(s)}\boldsymbol{q}_{n,i}^{(s)}\right)$$

$$= \alpha_{+,+}^{(s)} \|\boldsymbol{k}_+^{(s)}\|_2^2 + \beta_{+,+}^{(s)} \|\boldsymbol{q}_+^{(s)}\|_2^2 + \{\text{lower order term}\}$$

$$\geq \frac{2\eta^2 C_5 (\|\boldsymbol{\mu}\|_2 - \tau)^4 \|\boldsymbol{w}_O\|_2^2 (s - T_1)}{N} \cdot \frac{1}{\exp(\Lambda_{n,+,j}^{(s)})} \cdot \Theta(\|\boldsymbol{\mu}\|_2^2 \sigma_h^2 d_h)$$

$$+ \{\text{lower order term}\}$$

$$\geq \frac{\eta^2 C_6 (\|\boldsymbol{\mu}\|_2 - \tau)_2^4 \|\boldsymbol{\mu}\|_2^2 \|\boldsymbol{w}_O\|_2^2 \sigma_h^2 d_h (s - T_1)}{N} \cdot \frac{1}{\exp(\Lambda_{n,+,j}^{(s)})},$$

$$(123)$$

Similarly, we have the lower bounds for the dynamics of other $\langle \boldsymbol{q}, \boldsymbol{k} \rangle$.

## I.7 Upper Bounds of $\langle q, k \rangle$

In order to give the upper bounds of $\langle q, k \rangle$ in stage II, we need to give the upper bounds of $\alpha$ and $\beta$ based on the equations in Section I.1 under the assumption that $\mathcal{D}(T_1), \ldots, \mathcal{D}(s-1)$ hold for $s \in [T_1, t]$.

$$\alpha_{+,+}^{(s)} \leq \frac{\eta}{NM} \sum_{n \in S_+} -\widetilde{\ell}_n'(\theta)\langle \widetilde{\boldsymbol{\mu}}_+, \widetilde{\boldsymbol{\mu}}_+^{(t)}\rangle$$

$$\cdot \left( V_+^{(t)} \left( \frac{\exp(\langle \boldsymbol{q}_+^{(t)}, \boldsymbol{k}_+^{(t)}\rangle)}{\exp(\langle \boldsymbol{q}_+^{(t)}, \boldsymbol{k}_+^{(t)}\rangle) + \sum_{j=2}^M \exp(\langle \boldsymbol{q}_+^{(t)}, \boldsymbol{k}_{n,j}^{(t)}\rangle)} \right. \right.$$

$$\left. - \left( \frac{\exp(\langle \boldsymbol{q}_+^{(t)}, \boldsymbol{k}_+^{(t)}\rangle)}{\exp(\langle \boldsymbol{q}_+^{(t)}, \boldsymbol{k}_+^{(t)}\rangle) + \sum_{j=2}^M \exp(\langle \boldsymbol{q}_+^{(t)}, \boldsymbol{k}_{n,j}^{(t)}\rangle)} \right)^2 \right)$$

$$- \sum_{i=2}^M \left( V_{n,i}^{(t)} \cdot \frac{\exp(\langle \boldsymbol{q}_+^{(t)}, \boldsymbol{k}_+^{(t)}\rangle)}{\exp(\langle \boldsymbol{q}_+^{(t)}, \boldsymbol{k}_+^{(t)}\rangle) + \sum_{j=2}^M \exp(\langle \boldsymbol{q}_+^{(t)}, \boldsymbol{k}_{n,j}^{(t)}\rangle)} \right.$$

$$\left. \left. \cdot \frac{\exp(\langle \boldsymbol{q}_+^{(t)}, \boldsymbol{k}_{n,i}^{(t)}\rangle)}{\exp(\langle \boldsymbol{q}_+^{(t)}, \boldsymbol{k}_+^{(t)}\rangle) + \sum_{j=2}^M \exp(\langle \boldsymbol{q}_+^{(t)}, \boldsymbol{k}_{n,j}^{(t)}\rangle)} \right) \right)$$

$$+ \sum_{i=2}^M \langle \widetilde{\boldsymbol{\mu}}_+, \widetilde{\boldsymbol{\xi}}_{n,i}^{(t)}\rangle \cdot \left( V_+^{(t)} \left( \frac{\exp(\langle \boldsymbol{q}_{n,i}^{(t)}, \boldsymbol{k}_+^{(t)}\rangle)}{\exp(\langle \boldsymbol{q}_{n,i}^{(t)}, \boldsymbol{k}_+^{(t)}\rangle) + \sum_{j=2}^M \exp(\langle \boldsymbol{q}_{n,i}^{(t)}, \boldsymbol{k}_{n,j}^{(t)}\rangle)} \right. \right.$$

$$\left. \left. - \left( \frac{\exp(\langle \boldsymbol{q}_{n,i}^{(t)}, \boldsymbol{k}_+^{(t)}\rangle)}{\exp(\langle \boldsymbol{q}_{n,i}^{(t)}, \boldsymbol{k}_+^{(t)}\rangle) + \sum_{j=2}^M \exp(\langle \boldsymbol{q}_{n,i}^{(t)}, \boldsymbol{k}_{n,j}^{(t)}\rangle)} \right)^2 \right) \right)$$

$$- \sum_{k=2}^M \left( V_{n,i}^{(t)} \cdot \frac{\exp(\langle \boldsymbol{q}_{n,i}^{(t)}, \boldsymbol{k}_+^{(t)}\rangle)}{\exp(\langle \boldsymbol{q}_{n,i}^{(t)}, \boldsymbol{k}_+^{(t)}\rangle) + \sum_{j=2}^M \exp(\langle \boldsymbol{q}_{n,i}^{(t)}, \boldsymbol{k}_{n,j}^{(t)}\rangle)} \right.$$

$$\left. \cdot \frac{\exp(\langle \boldsymbol{q}_{n,i}^{(t)}, \boldsymbol{k}_{n,k}^{(t)}\rangle)}{\exp(\langle \boldsymbol{q}_{n,i}^{(t)}, \boldsymbol{k}_+^{(t)}\rangle) + \sum_{j=2}^M \exp(\langle \boldsymbol{q}_{n,i}^{(t)}, \boldsymbol{k}_{n,j}^{(t)}\rangle)} \right)$$

$$\leq \frac{\eta}{NM} \sum_{n \in S_+} (\|\boldsymbol{\mu}\|_2 + \tau)^2 (V_+^{(s)} \cdot \frac{\exp(\langle \boldsymbol{q}_+^{(s)}, \boldsymbol{k}_+^{(s)}\rangle)}{\exp(\langle \boldsymbol{q}_+^{(s)}, \boldsymbol{k}_+^{(s)}\rangle) + \sum_{j=2}^M \exp(\langle \boldsymbol{q}_+^{(s)}, \boldsymbol{k}_{n,j}^{(s)}\rangle)}$$

$$+ \max_{i=2} |V_{n,i}^{(s)}| \cdot \frac{\sum_{j=2}^M \exp(\langle \boldsymbol{q}_+^{(s)}, \boldsymbol{k}_{n,j}^{(s)}\rangle)}{\exp(\langle \boldsymbol{q}_+^{(s)}, \boldsymbol{k}_+^{(s)}\rangle) + \sum_{j=2}^M \exp(\langle \boldsymbol{q}_+^{(s)}, \boldsymbol{k}_{n,j}^{(s)}\rangle)})$$

$$+ \sum_{i=2}^M (\|\boldsymbol{\mu}\|\tau + \sigma_p \tau \sqrt{2\log(4NM/\delta)} + \tau^2)(V_+^{(s)} \cdot \frac{\exp(\langle \boldsymbol{q}_{n,i}^{(s)}, \boldsymbol{k}_+^{(s)}\rangle)}{\exp(\langle \boldsymbol{q}_{n,i}^{(s)}, \boldsymbol{k}_+^{(s)}\rangle) + \sum_{j=2}^M \exp(\langle \boldsymbol{q}_{n,i}^{(s)}, \boldsymbol{k}_{n,j}^{(s)}\rangle)}$$

$$+ \max_{i=2} |V_{n,i}^{(s)}| \cdot \frac{\sum_{j=2}^M \exp(\langle \boldsymbol{q}_{n,i}^{(s)}, \boldsymbol{k}_{n,j}^{(s)}\rangle)}{\exp(\langle \boldsymbol{q}_{n,i}^{(s)}, \boldsymbol{k}_+^{(s)}\rangle) + \sum_{j=2}^M \exp(\langle \boldsymbol{q}_{n,i}^{(s)}, \boldsymbol{k}_{n,j}^{(s)}\rangle)})$$

$$\leq \frac{\eta}{NM} \cdot \frac{3N}{4} \cdot (\|\boldsymbol{\mu}\|_2 + \tau)^2 \cdot \left( V_+^{(s)} \cdot \frac{C}{\exp(\langle \boldsymbol{q}_+^{(s)}, \boldsymbol{k}_+^{(s)}\rangle)} + \max_i |V_{n,i}^{(s)}| \cdot \frac{C}{\exp(\langle \boldsymbol{q}_+^{(s)}, \boldsymbol{k}_+^{(s)}\rangle)} \right)$$

$$+ \sum_{i=2}^M (\|\boldsymbol{\mu}\|\tau + \sigma_p \tau \sqrt{2\log(4NM/\delta)} + \tau^2) \cdot \left( V_+^{(s)} \cdot \frac{C}{\exp(\langle \boldsymbol{q}_{n,i}^{(s)}, \boldsymbol{k}_+^{(s)}\rangle)} + \max_i |V_{n,i}^{(s)}| \cdot \frac{C}{\exp(\langle \boldsymbol{q}_{n,i}^{(s)}, \boldsymbol{k}_+^{(s)}\rangle)} \right)$$

$$\leq \frac{\eta^2 C_9 (\|\boldsymbol{\mu}\|_2 + \tau)^2}{\exp(\langle \boldsymbol{q}_+^{(s)}, \boldsymbol{k}_+^{(s)}\rangle)} + \sum_{i=2}^M \frac{\eta^2 C_9 (\|\boldsymbol{\mu}\|_2 \tau + \sigma_p \tau \sqrt{2\log(4NM/\delta)} + \tau^2)}{\exp(\langle \boldsymbol{q}_{n,i}^{(s)}, \boldsymbol{k}_+^{(s)}\rangle)}$$

$$(124)$$

where the first inequality is by $-\widetilde{\ell}_n'^{(s)} \leq 1$ and $\mathrm{softmax}(\langle \boldsymbol{q}_+^{(s)}, \boldsymbol{k}_+^{(s)}\rangle) \leq 1$. For the second inequality, we first consider $\frac{\sum_{j=2}^M \exp(\langle \boldsymbol{q}_+^{(s)}, \boldsymbol{k}_{n,j}^{(s)}\rangle)}{\exp(\langle \boldsymbol{q}_+^{(s)}, \boldsymbol{k}_+^{(s)}\rangle) + \sum_{j=2}^M \exp(\langle \boldsymbol{q}_+^{(s)}, \boldsymbol{k}_{n,j}^{(s)}\rangle)} \leq \frac{\sum_{j=2}^M \exp(\langle \boldsymbol{q}_+^{(s)}, \boldsymbol{k}_{n,j}^{(s)}\rangle)}{\exp(\langle \boldsymbol{q}_+^{(s)}, \boldsymbol{k}_+^{(s)}\rangle)}$.

Then by the monotonicity of $\langle \boldsymbol{q}_+^{(s)}, \boldsymbol{k}_{n,j}^{(s)} \rangle$ and $\langle \boldsymbol{q}_+^{(T_1)}, \boldsymbol{k}_{n,j}^{(T_1)} \rangle = o(1)$ we have $\sum_{j=2}^{M} \exp(\langle \boldsymbol{q}_+^{(s)}, \boldsymbol{k}_{n,j}^{(s)} \rangle) \leq C$ for $s \in [T_1, t]$. The last inequality is by $V_+^{(s)}, V_{n,i}^{(s)} = o(1)$ for $s \in [T_1, t]$ and absorbing the constant factors. Similarly, we have

Similar to Section I.6, we apply the bounds of $\alpha$ and $\beta$ above to give the upper bounds for the dynamics $\langle \boldsymbol{q}, \boldsymbol{k} \rangle$.

$$\langle \boldsymbol{q}_+^{(s+1)}, \boldsymbol{k}_+^{(s+1)} \rangle - \langle \boldsymbol{q}_+^{(s)}, \boldsymbol{k}_+^{(s)} \rangle = \alpha_{+,+}^{(s)} \|\boldsymbol{k}_+^{(s)}\|_2^2 + \sum_{n \in S_+} \sum_{i=2}^{M} \alpha_{n,+,i}^{(s)} \langle \boldsymbol{k}_+^{(s)}, \boldsymbol{k}_{n,i}^{(s)} \rangle$$

$$+ \beta_{+,+}^{(s)} \|\boldsymbol{q}_+^{(s)}\|_2^2 + \sum_{n \in S_+} \sum_{i=2}^{M} \beta_{n,+,i}^{(s)} \langle \boldsymbol{q}_+^{(s)}, \boldsymbol{q}_{n,i}^{(s)} \rangle$$

$$+ \left( \alpha_{+,+}^{(s)} \boldsymbol{k}_+^{(s)} + \sum_{n \in S_+} \sum_{i=2}^{M} \alpha_{n,+,i}^{(s)} \boldsymbol{k}_{n,i}^{(s)} \right)^{\top}$$

$$\cdot \left( \beta_{+,+}^{(s)} \boldsymbol{q}_+^{(s)} + \sum_{n \in S_+} \sum_{i=2}^{M} \beta_{n,+,i}^{(s)} \boldsymbol{q}_{n,i}^{(s)} \right)$$

$$+ \sum_{n \in S_+} \sum_{i=2}^{M} \alpha_{n,+,i}^{(s)} \|\boldsymbol{k}_{n,i}^{(s)}\|_2^2$$

$$= \alpha_{+,+}^{(s)} \|\boldsymbol{k}_+^{(s)}\|_2 + \beta_{+,+}^{(s)} \|\boldsymbol{q}_+^{(s)}\|_2 + \{\text{lower order terms}\}$$

$$\leq \left( \frac{\eta^2 C_9 (\|\boldsymbol{\mu}\|_2 + \tau)^2}{\exp(\langle \boldsymbol{q}_+^{(s)}, \boldsymbol{k}_+^{(s)} \rangle)} + \sum_{i=2}^{M} \frac{\eta^2 C_9 (\|\boldsymbol{\mu}\|_2 \tau + \sigma_p \tau \sqrt{2 \log(4NM/\delta)} + \tau^2)}{\exp(\langle \boldsymbol{q}_{n,i}^{(s)}, \boldsymbol{k}_+^{(s)} \rangle)} \right) \cdot \Theta(\|\boldsymbol{\mu}\|_2^2 \sigma_h^2 d_h) + \{\text{lower order terms}\}$$

$$\leq \frac{\eta C_{10} \|\boldsymbol{\mu}\|_2^2 (\|\boldsymbol{\mu}\|_2 + \tau)^2 \sigma_h^2 d_h}{\exp(\langle \boldsymbol{q}_+^{(s)}, \boldsymbol{k}_+^{(s)} \rangle)} + \sum_{i=2}^{M} \frac{\eta C_{10} \|\boldsymbol{\mu}\|_2^2 (\|\boldsymbol{\mu}\|_2 \tau + \sigma_p \tau \sqrt{2 \log(4NM/\delta)} + \tau^2) \sigma_h^2 d_h}{\exp(\langle \boldsymbol{q}_{n,i}^{(s)}, \boldsymbol{k}_+^{(s)} \rangle)}.$$

$$(125)$$

Similarly, we have the upper bounds for the dynamics of other $\langle \boldsymbol{q}, \boldsymbol{k} \rangle$.

### I.8 BOUNDS FOR THE SUM OF $\alpha$ AND $\beta$

The gradients of the inner products of $q$ and $k$ contain a lot of coefficients $\alpha$ and $\beta$, and in order to conveniently give the upper bounds of some lower order inner products, we will give upper bounds for the summation of $\alpha$ and $\beta$ (e.g. $\sum_{s=T_1}^{t} |\alpha_{+,+}^{(s)}|$).

Note that in the Jacobi matrix of the Softmax function, the elements on the diagonal are softmax$(a_i) \cdot (1 - \text{softmax}(a_i))$ and the elements on the off-diagonal are softmax$(a_i) \cdot \text{softmax}(a_j)$. In Stage II, the attentions on signals $\boldsymbol{\mu}_\pm$ increase and the attentions on noises $\boldsymbol{\xi}$ decrease, then we can consider the following cases:

- if $a_i = \langle \boldsymbol{q}_+, \boldsymbol{k}_+ \rangle$ or $a_i = \langle \boldsymbol{q}_i, \boldsymbol{k}_+ \rangle$, softmax$(a_i)$ has a constant upper bound 1, $(1 - \text{softmax}(a_i))$ decreases as softmax$(a_i)$ increases. So the upper bound of softmax$(a_i) \cdot (1 - \text{softmax}(a_i))$ decreases as softmax$(a_i)$ increases.
- if $a_i = \langle \boldsymbol{q}_+, \boldsymbol{k}_j \rangle$ or $a_i = \langle \boldsymbol{q}_i, \boldsymbol{k}_j \rangle$, $(1 - \text{softmax}(a_i))$ has a constant upper bound 1. So the upper bound of softmax$(a_i) \cdot (1 - \text{softmax}(a_i))$ decreases as softmax$(a_i)$ decreases.
- if $a_j = \langle \boldsymbol{q}_+, \boldsymbol{k}_j \rangle$ or $a_j = \langle \boldsymbol{q}_i, \boldsymbol{k}_j \rangle$, softmax$(a_i)$ has a constant upper bound 1. So the upper bound of softmax$(a_i) \cdot \text{softmax}(a_j)$ decreases as softmax$(a_j)$ decreases.

Based on the above cases, we first study the bounds of the following terms

- $1 - \text{softmax}(\langle \boldsymbol{q}_+^{(s)}, \boldsymbol{k}_+^{(s)} \rangle)$

- $1 - \mathrm{softmax}(\langle \boldsymbol{q}_{n,i}^{(s)}, \boldsymbol{k}_+^{(s)} \rangle)$

- $\mathrm{softmax}(\langle \boldsymbol{q}_+^{(s)}, \boldsymbol{k}_{n,j}^{(s)} \rangle)$

- $\mathrm{softmax}(\langle \boldsymbol{q}_{n,i}^{(s)}, \boldsymbol{k}_{n,j}^{(s)} \rangle)$

Note that $1 - \mathrm{softmax}(\langle \boldsymbol{q}_+^{(s)}, \boldsymbol{k}_+^{(s)} \rangle) = \sum_j \mathrm{softmax}(\langle \boldsymbol{q}_+^{(s)}, \boldsymbol{k}_{n,j}^{(s)} \rangle)$ and $1 - \mathrm{softmax}(\langle \boldsymbol{q}_{n,i}^{(s)}, \boldsymbol{k}_+^{(s)} \rangle) = \sum_j \mathrm{softmax}(\langle \boldsymbol{q}_{n,i}^{(s)}, \boldsymbol{k}_{n,j}^{(s)} \rangle)$, we only need to give the upper bounds for $\mathrm{softmax}(\langle \boldsymbol{q}_+^{(s)}, \boldsymbol{k}_{n,j}^{(s)} \rangle)$ and $\mathrm{softmax}(\langle \boldsymbol{q}_{n,i}^{(s)}, \boldsymbol{k}_{n,j}^{(s)} \rangle)$.

Assume that the propositions $\mathcal{B}(T_1), \ldots, \mathcal{B}(s), \mathcal{D}(T_1), \ldots, \mathcal{D}(s-1)$ hold ($s \in [T_1, t]$), we have

$$|V_\pm^{(s)}|, |V_{n,i}^{(s)}| \leq O(d_h^{-\frac{1}{4}}) + \eta C_4 (\|\boldsymbol{\mu}\|_2 + \tau)^2 \|\boldsymbol{w}_O\|_2^2 (s - T_1), \tag{126}$$

$$\Lambda_{n,\pm,j}^{(s)} \geq \log\left( \exp(\Lambda_{n,\pm,j}^{(T_1)}) + \frac{\eta^2 C_8 (\|\boldsymbol{\mu}\|_2 - \tau)^2 \|\boldsymbol{\mu}\|_2^2 \|\boldsymbol{w}_O\|_2^2 d_h^{\frac{1}{2}}}{N \left(\log(6N^2 M^2/\delta)\right)^2} \cdot (s - T_1)(s - T_1 - 1) \right), \tag{127}$$

$$\Lambda_{n,i,\pm,j}^{(s)} \geq \log\left( \exp(\Lambda_{n,i,\pm,j}^{(T_1)}) + \frac{\eta^2 C_8 (\sigma_P^2 d + \sigma_P \tau \sqrt{2\log(4NM/\delta)} + \tau^2) \|\boldsymbol{\mu}\|_2^2 \|\boldsymbol{w}_O\|_2^2 d_h^{\frac{1}{2}}}{N \left(\log(6N^2 M^2/\delta)\right)^2} \cdot (s - T_1)(s - T_1 - 1) \right), \tag{128}$$

for $i, j \in [M] \setminus \{1\}, n \in [N], s \in [T_1, t]$.

Then we have

$$\frac{\exp(\langle \boldsymbol{q}_\pm^{(s)}, \boldsymbol{k}_{n,j}^{(s)} \rangle)}{\exp(\langle \boldsymbol{q}_\pm^{(s)}, \boldsymbol{k}_\pm^{(s)} \rangle) + \sum_{j'=2}^{M} \exp(\langle \boldsymbol{q}_\pm^{(s)}, \boldsymbol{k}_{n,j'}^{(s)} \rangle)}$$

$$\leq \frac{\exp(\langle \boldsymbol{q}_\pm^{(s)}, \boldsymbol{k}_{n,j}^{(s)} \rangle)}{C \exp(\langle \boldsymbol{q}_\pm^{(s)}, \boldsymbol{k}_\pm^{(s)} \rangle)}$$

$$= \frac{1}{C \exp(\Lambda_{n,\pm,j}^{(s)})} \tag{129}$$

$$\leq \frac{1}{C \exp(\Lambda_{n,\pm,j}^{(T_1)}) + \frac{\eta^2 C_8 C (\|\boldsymbol{\mu}\|_2 - \tau)^2 \|\boldsymbol{\mu}\|_2^2 \|\boldsymbol{w}_O\|_2^2 d_h^{\frac{1}{2}}}{N \left(\log(6N^2 M^2/\delta)\right)^2} \cdot (s - T_1)(s - T_1 - 1)}$$

$$\leq \frac{1}{C_{13} + \frac{\eta^2 C_{13} (\|\boldsymbol{\mu}\|_2 - \tau)^2 \|\boldsymbol{\mu}\|_2^2 \|\boldsymbol{w}_O\|_2^2 d_h^{\frac{1}{2}}}{N \left(\log(6N^2 M^2/\delta)\right)^2} \cdot (s - T_1)(s - T_1 - 1)}.$$

For the first inequality, by $\langle \boldsymbol{q}^{(T_1)}, \boldsymbol{k}^{(T_1)} \rangle = o(1)$ and the monotonicity of $\langle \boldsymbol{q}^{(s)}, \boldsymbol{k}^{(s)} \rangle$ ($\langle \boldsymbol{q}^{(s)}, \boldsymbol{k}^{(s)} \rangle$ is increasing and $\langle \boldsymbol{q}_\pm^{(s)}, \boldsymbol{k}_{n,j}^{(s)} \rangle$ is decreasing), there exists a constant $C$ such that $C \exp(\langle \boldsymbol{q}_\pm^{(s)}, \boldsymbol{k}_\pm^{(s)} \rangle) \geq \exp(\langle \boldsymbol{q}_\pm^{(s)}, \boldsymbol{k}_\pm^{(s)} \rangle) + \sum_{j'=2}^{M} \exp(\langle \boldsymbol{q}_\pm^{(s)}, \boldsymbol{k}_{n,j'}^{(s)} \rangle)$.. The second inequality is by plugging (127). For the last inequality, by $\Lambda_{n,\pm,j}^{(T_1)} = o(1)$, there exist a constant $C_{13}$ such that $C_{13} \leq C \exp(\Lambda_{n,\pm,j}^{(T_1)})$ and $C_{13} \leq C_8 C$. Similarly, we have

$$\frac{\exp(\langle \boldsymbol{q}_{n,i}^{(s)}, \boldsymbol{k}_{n,j}^{(s)} \rangle)}{\exp(\langle \boldsymbol{q}_{n,i}^{(s)}, \boldsymbol{k}_+^{(s)} \rangle) + \sum_{j'=2}^{M} \exp(\langle \boldsymbol{q}_{n,i}^{(s)}, \boldsymbol{k}_{n,j'}^{(s)} \rangle)}$$

$$\leq \frac{1}{C \exp(\Lambda_{n,i,+,j}^{(s)})} \tag{130}$$

$$\leq \frac{1}{C_{13} + \frac{\eta^2 C_{13} (\sigma_P^2 d + \sigma_P \tau \sqrt{2\log(4NM/\delta)} + \tau^2) \|\boldsymbol{\mu}\|_2^2 \|\boldsymbol{w}_O\|_2^2 d_h^{1/2}}{N (\log(6N^2 M^2/\delta))^2} \cdot (s - T_1)(s - T_1 - 1)}.$$

Plugging above equations into the expressions of $\alpha, \beta$ we have

$$
\begin{aligned}
|\alpha_{+,+}^{(s)}| \leq & \left| \frac{\eta}{NM} \sum_{n \in S_+} -\widetilde{\ell}_n'(\theta) \langle \widetilde{\boldsymbol{\mu}}_+, \widetilde{\boldsymbol{\mu}}_+^{(s)} \rangle \right. \\
& \cdot \left( V_+^{(s)} \left( \frac{\exp(\langle \boldsymbol{q}_+^{(s)}, \boldsymbol{k}_+^{(s)} \rangle)}{\exp(\langle \boldsymbol{q}_+^{(s)}, \boldsymbol{k}_+^{(s)} \rangle) + \sum_{j=2}^M \exp(\langle \boldsymbol{q}_+^{(s)}, \boldsymbol{k}_{n,j}^{(s)} \rangle)} \right. \right. \\
& \left. - \left( \frac{\exp(\langle \boldsymbol{q}_+^{(s)}, \boldsymbol{k}_+^{(s)} \rangle)}{\exp(\langle \boldsymbol{q}_+^{(s)}, \boldsymbol{k}_+^{(s)} \rangle) + \sum_{j=2}^M \exp(\langle \boldsymbol{q}_+^{(s)}, \boldsymbol{k}_{n,j}^{(s)} \rangle)} \right)^2 \right) \\
& - \sum_{i=2}^M \left( V_{n,i}^{(s)} \cdot \frac{\exp(\langle \boldsymbol{q}_+^{(s)}, \boldsymbol{k}_+^{(s)} \rangle)}{\exp(\langle \boldsymbol{q}_+^{(s)}, \boldsymbol{k}_+^{(s)} \rangle) + \sum_{j=2}^M \exp(\langle \boldsymbol{q}_+^{(s)}, \boldsymbol{k}_{n,j}^{(s)} \rangle)} \right. \\
& \left. \left. \cdot \frac{\exp(\langle \boldsymbol{q}_+^{(s)}, \boldsymbol{k}_{n,i}^{(s)} \rangle)}{\exp(\langle \boldsymbol{q}_+^{(s)}, \boldsymbol{k}_+^{(s)} \rangle) + \sum_{j=2}^M \exp(\langle \boldsymbol{q}_+^{(s)}, \boldsymbol{k}_{n,j}^{(s)} \rangle)} \right) \right) \\
& + \sum_{i=2}^M \langle \widetilde{\boldsymbol{\mu}}_+, \widetilde{\boldsymbol{\xi}}_{n,i}^{(s)} \rangle \cdot \left( V_+^{(s)} \left( \frac{\exp(\langle \boldsymbol{q}_{n,i}^{(s)}, \boldsymbol{k}_+^{(s)} \rangle)}{\exp(\langle \boldsymbol{q}_{n,i}^{(s)}, \boldsymbol{k}_+^{(s)} \rangle) + \sum_{j=2}^M \exp(\langle \boldsymbol{q}_{n,i}^{(s)}, \boldsymbol{k}_{n,j}^{(s)} \rangle)} \right. \right. \\
& \left. - \left( \frac{\exp(\langle \boldsymbol{q}_{n,i}^{(s)}, \boldsymbol{k}_+^{(s)} \rangle)}{\exp(\langle \boldsymbol{q}_{n,i}^{(s)}, \boldsymbol{k}_+^{(s)} \rangle) + \sum_{j=2}^M \exp(\langle \boldsymbol{q}_{n,i}^{(s)}, \boldsymbol{k}_{n,j}^{(s)} \rangle)} \right)^2 \right) \\
& - \sum_{k=2}^M \left( V_{n,i}^{(s)} \cdot \frac{\exp(\langle \boldsymbol{q}_{n,i}^{(s)}, \boldsymbol{k}_+^{(s)} \rangle)}{\exp(\langle \boldsymbol{q}_{n,i}^{(s)}, \boldsymbol{k}_+^{(s)} \rangle) + \sum_{j=2}^M \exp(\langle \boldsymbol{q}_{n,i}^{(s)}, \boldsymbol{k}_{n,j}^{(s)} \rangle)} \right. \\
& \left. \left. \left. \cdot \frac{\exp(\langle \boldsymbol{q}_{n,i}^{(s)}, \boldsymbol{k}_{n,k}^{(s)} \rangle)}{\exp(\langle \boldsymbol{q}_{n,i}^{(s)}, \boldsymbol{k}_+^{(s)} \rangle) + \sum_{j=2}^M \exp(\langle \boldsymbol{q}_{n,i}^{(s)}, \boldsymbol{k}_{n,j}^{(s)} \rangle)} \right) \right) \right| \\
\leq & \frac{\eta(\|\boldsymbol{\mu}\|_2 + \tau)^2 \cdot 3N}{NM} \cdot \left( O(d_h^{-\frac{1}{4}}) + \eta C_4 (\|\boldsymbol{\mu}\|_2 + \tau)^2 \|\boldsymbol{w}_O\|_2^2 (s - T_1) \right) \\
& \cdot O\left( \frac{1}{C_{13} + \frac{\eta^2 C_{13}((\|\boldsymbol{\mu}\|_2 + \tau)^2 - \tau)^2 \|\boldsymbol{\mu}\|_2^2 \|\boldsymbol{w}_O\|_2^2 d_h^{\frac{1}{2}}}{N(\log(6N^2 M^2/\delta))^2} \cdot (s - T_1)(s - T_1 - 1)} \right) \\
= & O\left( \frac{\eta \|\boldsymbol{\mu}\|_2^2 d_h^{-\frac{1}{4}}}{C_{13} + \frac{\eta^2 C_{13}(\|\boldsymbol{\mu}\|_2 - \tau)^2 \|\boldsymbol{\mu}\|_2^2 \|\boldsymbol{w}_O\|_2^2 d_h^{\frac{1}{2}}}{N(\log(6N^2 M^2/\delta))^2} \cdot (s - T_1)(s - T_1 - 1)} \right) \\
& + O\left( \frac{\eta^2 (\|\boldsymbol{\mu}\|_2 + \tau)^4 \|\boldsymbol{w}_O\|_2^2 (s - T_1)}{C_{13} + \frac{\eta^2 C_{13}(\|\boldsymbol{\mu}\|_2 - \tau)^2 \|\boldsymbol{\mu}\|_2^2 \|\boldsymbol{w}_O\|_2^2 d_h^{\frac{1}{2}}}{N(\log(6N^2 M^2/\delta))^2} \cdot (s - T_1)(s - T_1 - 1)} \right) \\
= & O\left( \eta(\|\boldsymbol{\mu}\|_2 + \tau)^2 d_h^{-\frac{1}{4}} \right) + O\left( \frac{\eta^2 (\|\boldsymbol{\mu}\|_2 + \tau)^4 \|\boldsymbol{w}_O\|_2^2 (s - T_1)}{C_{13} + \frac{\eta^2 C_{13}(\|\boldsymbol{\mu}\|_2 - \tau)^2 \|\boldsymbol{\mu}\|_2^2 \|\boldsymbol{w}_O\|_2^2 d_h^{\frac{1}{2}}}{N(\log(6N^2 M^2/\delta))^2} \cdot (s - T_1)(s - T_1 - 1)} \right) .
\end{aligned}
$$

$$(131)$$

where the third equality is by $\frac{\eta^2 C_{13}(\|\boldsymbol{\mu}\|_2-\tau)^2\|\boldsymbol{\mu}\|_2^2\|\boldsymbol{w}_O\|_2^2 d_h^{\frac{1}{2}}}{N(\log(6N^2M^2/\delta))^2}\cdot(s-T_1)(s-T_1-1)\geq 0$ for $s\in[T_1,t]$.

Next, we give an upper bound for $\dfrac{\eta^2(\|\boldsymbol{\mu}\|_2+\tau)^4\|\boldsymbol{w}_O\|_2^2(s-T_1)}{C_{13}+\frac{\eta^2 C_{13}(\|\boldsymbol{\mu}\|_2-\tau)^2\|\boldsymbol{\mu}\|_2^2\|\boldsymbol{w}_O\|_2^2 d_h^{\frac{1}{2}}}{N(\log(6N^2M^2/\delta))^2}\cdot(s-T_1)(s-T_1-1)}$ as follows:

$$
\frac{\eta^2(\|\boldsymbol{\mu}\|_2+\tau)^4\|\boldsymbol{w}_O\|_2^2(s-T_1)}{C_{13}+\frac{\eta^2 C_{13}(\|\boldsymbol{\mu}\|_2-\tau)^2\|\boldsymbol{\mu}\|_2^2\|\boldsymbol{w}_O\|_2^2 d_h^{\frac{1}{2}}}{N(\log(6N^2M^2/\delta))^2}\cdot(s-T_1)(s-T_1-1)}
$$

$$
=\frac{\eta^2(\|\boldsymbol{\mu}\|_2+\tau)^4\|\boldsymbol{w}_O\|_2^2}{\frac{C_{13}}{(s-T_1)}+\frac{\eta^2 C_{13}(\|\boldsymbol{\mu}\|_2-\tau)^2\|\boldsymbol{\mu}\|_2^2\|\boldsymbol{w}_O\|_2^2 d_h^{\frac{1}{2}}}{N(\log(6N^2M^2/\delta))^2}\cdot(s-T_1)-\frac{\eta^2 C_{13}(\|\boldsymbol{\mu}\|_2-\tau)^2\|\boldsymbol{\mu}\|_2^2\|\boldsymbol{w}_O\|_2^2 d_h^{\frac{1}{2}}}{N(\log(6N^2M^2/\delta))^2}}
$$

$$
\leq\frac{\eta^2(\|\boldsymbol{\mu}\|_2+\tau)^4\|\boldsymbol{w}_O\|_2^2}{2\sqrt{\frac{\eta^2 C_{13}^2(\|\boldsymbol{\mu}\|_2-\tau)^2\|\boldsymbol{\mu}\|_2^2\|\boldsymbol{w}_O\|_2^2 d_h^{\frac{1}{2}}}{N(\log(6N^2M^2/\delta))^2}}-\frac{\eta^2 C_{13}(\|\boldsymbol{\mu}\|_2-\tau)^2\|\boldsymbol{\mu}\|_2^2\|\boldsymbol{w}_O\|_2^2 d_h^{\frac{1}{2}}}{N(\log(6N^2M^2/\delta))^2}}
$$

$$
=\frac{\eta^2(\|\boldsymbol{\mu}\|_2+\tau)^4\|\boldsymbol{w}_O\|_2^2}{\frac{2\eta C_{13}(\|\boldsymbol{\mu}\|_2-\tau)\|\boldsymbol{\mu}\|_2\|\boldsymbol{w}_O\|_2^2 d_h^{\frac{1}{4}}}{N^{\frac{1}{2}}(\log(6N^2M^2/\delta))}-\frac{\eta^2 C_{13}(\|\boldsymbol{\mu}\|_2-\tau)^2\|\boldsymbol{\mu}\|_2^2\|\boldsymbol{w}_O\|_2^2 d_h^{\frac{1}{2}}}{N(\log(6N^2M^2/\delta))^2}}
$$
(132)
$$
=\frac{\eta^2(\|\boldsymbol{\mu}\|_2+\tau)^4\|\boldsymbol{w}_O\|_2^2}{\Theta\left(\frac{\eta(\|\boldsymbol{\mu}\|_2-\tau)\|\boldsymbol{\mu}\|_2\|\boldsymbol{w}_O\|_2 d_h^{\frac{1}{4}}}{N^{\frac{1}{2}}\log(6N^2M^2/\delta)}\right)}
$$

$$
=O\left(\eta(\|\boldsymbol{\mu}\|_2+\tau)^2 N^{\frac{1}{2}}d_h^{-\frac{1}{4}}\log(6N^2M^2/\delta)\right),
$$

where the inequality is by $ax+\frac{b}{x}\geq 2\sqrt{ab}$ for $x>0$, the third equality is by absorbing the lower order term $\frac{\eta^2 C_{13}(\|\boldsymbol{\mu}\|_2+\tau)^4\|\boldsymbol{w}_O\|_2^2 d_h^{\frac{1}{2}}}{N(\log(6N^2M^2/\delta))^2}$, the last equality is by $\|\boldsymbol{w}_O\|_2=\Theta(1)$. Plugging this into (131) and get

$$
|\alpha_{+,+}^{(s)}|=O\left(\eta(\|\boldsymbol{\mu}\|_2+\tau)^2 d_h^{-\frac{1}{4}}\right)+O\left(\eta(\|\boldsymbol{\mu}\|_2+\tau)^2 N^{\frac{1}{2}}d_h^{-\frac{1}{4}}\log(6N^2M^2/\delta)\right)
$$
(133)
$$
=O\left(\eta(\|\boldsymbol{\mu}\|_2+\tau)^2 N^{\frac{1}{2}}d_h^{-\frac{1}{4}}\log(6N^2M^2/\delta)\right).
$$

Similarly, we have

$$
|\alpha_{-,-}^{(s)}|,|\beta_{+,+}^{(s)}|,|\beta_{-,-}^{(s)}|=O\left(\eta(\|\boldsymbol{\mu}\|_2+\tau)^2 N^{\frac{1}{2}}d_h^{-\frac{1}{4}}\log(6N^2M^2/\delta)\right),
$$

$$
|\alpha_{n,+,i}^{(s)}|,|\alpha_{n,-,i}^{(s)}|=O\left(\eta(\|\boldsymbol{\mu}\|_2+\tau)^2 N^{-\frac{1}{2}}d_h^{-\frac{1}{4}}\log(6N^2M^2/\delta)\right),
$$

$$
|\beta_{n,+,i}^{(s)}|,|\beta_{n,-,i}^{(s)}|=O\left(\frac{\eta(\|\boldsymbol{\mu}\|_2+\tau)^2\|\boldsymbol{\mu}\|_2\log(6N^2M^2/\delta)}{\sigma_p d^{\frac{1}{2}}N^{\frac{1}{2}}d_h^{\frac{1}{4}}}\right)=
$$
(134)
$$
O\left(\eta(\|\boldsymbol{\mu}\|_2+\tau)^2\cdot\mathrm{SNR}\cdot N^{-\frac{1}{2}}d_h^{-\frac{1}{4}}\log(6N^2M^2/\delta)\right),
$$

for $i\in[M]\setminus\{1\},n\in S_+$.

$$
|\alpha_{n,i,+}^{(s)}|,|\alpha_{n,i,-}^{(s)}|=O\left(\frac{\eta\|\boldsymbol{\mu}\|_2\sigma_p d^{\frac{1}{2}}\log(6N^2M^2/\delta)}{N^{\frac{1}{2}}d_h^{\frac{1}{4}}}\right)=O\left(\eta(\|\boldsymbol{\mu}\|_2+\tau)^2 d_h^{-\frac{1}{4}}\log(6N^2M^2/\delta)\right),
$$
(135)

for $i\in[M]\setminus\{1\},n\in S_+$, the last equality is by $N\cdot\mathrm{SNR}^2\geq\Omega(1)$.

$$
|\beta_{n,i,+}^{(s)}|,|\beta_{n,i,-}^{(s)}|=O\left(\frac{\eta\sigma_p^2 d\log(6N^2M^2/\delta)}{N^{\frac{1}{2}}d_h^{\frac{1}{4}}}\right)=O\left(\eta(\|\boldsymbol{\mu}\|_2+\tau)^2 N^{\frac{1}{2}}d_h^{-\frac{1}{4}}\log(6N^2M^2/\delta)\right),
$$
(136)

for $i \in [M] \setminus \{1\}, n \in S_+$, the last equality is by $N \cdot \mathrm{SNR}^2 \geq \Omega(1)$.

$$|\alpha_{n,i,n,j}^{(s)}|, |\beta_{n,j,n,i}^{(s)}| = O\left(\frac{\eta(\|\boldsymbol{\mu}\|_2 + \tau)\sigma_p d^{\frac{1}{2}}\log(6N^2M^2/\delta)}{N^{\frac{1}{2}}d_h^{\frac{1}{4}}}\right)$$
$$= O\left(\eta(\|\boldsymbol{\mu}\|_2 + \tau)^2 d_h^{-\frac{1}{4}}\log(6N^2M^2/\delta)\right), \tag{137}$$

for $i, j \in [M] \setminus \{1\}, n \in [N]$, the last equality is by $N \cdot \mathrm{SNR}^2 \geq \Omega(1)$.

$$|\alpha_{n,i,n',j}^{(s)}|, |\beta_{n,j,n',i}^{(s)}| = O\left(\frac{\eta(\|\boldsymbol{\mu}\|_2 + \tau)\sigma_p \log(6N^2M^2/\delta)\log(6N^2M^2/\delta)}{N^{\frac{1}{2}}d_h^{\frac{1}{4}}}\right)$$
$$= O\left(\eta(\|\boldsymbol{\mu}\|_2 + \tau)^2 d^{-\frac{1}{2}}d_h^{-\frac{1}{4}}(\log(6N^2M^2/\delta))^2\right), \tag{138}$$

for $i, j \in [M] \setminus \{1\}, n, n' \in [N], n \neq n'$, the last equality is by $N \cdot \mathrm{SNR}^2 \geq \Omega(1)$. Taking a summation we obtain that

$$\sum_{s=T_1}^{t} |\alpha_{+,+}^{(s)}| = O\left(\eta(\|\boldsymbol{\mu}\|_2 + \tau)^2 \|\boldsymbol{w}_O\|_2^2 \log(6N^2M^2/\delta)\right) \cdot O\left(\eta(\|\boldsymbol{\mu}\|_2 + \tau)^2 N^{\frac{1}{2}}d_h^{-\frac{1}{4}}\log(6N^2M^2/\delta)\right)$$
$$= O\left(N^{\frac{1}{2}}d_h^{-\frac{1}{4}}\right), \tag{139}$$

where the last equality is by $\|\boldsymbol{w}_O\| = \Theta(1)$. Similarly, we have

$$\sum_{s=T_1}^{t} |\alpha_{-,-}^{(s)}|, \sum_{s=T_1}^{t} |\beta_{+,+}^{(s)}|, \sum_{s=T_1}^{t} |\beta_{-,-}^{(s)}|, \sum_{s=T_1}^{t} |\beta_{n,i,+}^{(s)}|, \sum_{s=T_1}^{t} |\beta_{n,i,-}^{(s)}| = O\left(N^{\frac{1}{2}}d_h^{-\frac{1}{4}}\right), \tag{140}$$

for $i \in [M] \setminus \{1\}, n \in S_+$.

$$\sum_{s=T_1}^{t} |\alpha_{n,+,i}^{(s)}|, \sum_{s=T_1}^{t} |\alpha_{n,-,i}^{(s)}| = O\left(N^{-\frac{1}{2}}d_h^{-\frac{1}{4}}\right), \tag{141}$$

for $i \in [M] \setminus \{1\}, n \in S_+$.

$$\sum_{s=T_1}^{t} |\beta_{n,+,i}^{(s)}|, \sum_{s=T_1}^{t} |\beta_{n,-,i}^{(s)}| = O\left(\mathrm{SNR} \cdot N^{-\frac{1}{2}}d_h^{-\frac{1}{4}}\right) \tag{142}$$

for $i \in [M] \setminus \{1\}, n \in S_+$.

$$\sum_{s=T_1}^{t} |\alpha_{n,i,+}^{(s)}|, \sum_{s=T_1}^{t} |\alpha_{n,i,-}^{(s)}|, \sum_{s=T_1}^{t} |\alpha_{n,i,n,j}^{(s)}|, \sum_{s=T_1}^{t} |\beta_{n,j,n,i}^{(s)}| = O\left(d_h^{-\frac{1}{4}}\right) \tag{143}$$

for $i, j \in [M] \setminus \{1\}, n \in S_+$.

$$\sum_{s=T_1}^{t} |\alpha_{n,i,n',j}^{(s)}|, \sum_{s=T_1}^{t} |\beta_{n,j,n',i}^{(s)}| = O\left(d^{-\frac{1}{2}}d_h^{-\frac{1}{4}}\log(6N^2M^2/\delta)\right) \tag{144}$$

for $i, j \in [M] \setminus \{1\}, n, n' \in [N], n \neq n'$.

With these sums of $\alpha$ and $\beta$ above, we can easily prove Claim 3 and Claim 4.

## I.9   PROOF OF CLAIM 3

In this subsection, we assume that $\mathcal{E}(T_1), \ldots, \mathcal{E}(t)$ hold, and then proof that $\mathcal{C}(t+1)$ is true with the result of I.8.

$$\left|\|\boldsymbol{q}_+^{(t+1)}\|_2^2 - \|\boldsymbol{q}_+^{(T_1)}\|_2^2\right| \leq \sum_{s=T_1}^{t} \left|\|\boldsymbol{q}_+^{(s+1)}\|_2^2 - \|\boldsymbol{q}_+^{(s)}\|_2^2\right| \tag{145}$$

$$\leq \sum_{s=T_1}^{t} \left| 2\alpha_{+,+}^{(s)} \langle \boldsymbol{q}_+^{(s)}, \boldsymbol{k}_+^{(s)} \rangle + 2 \sum_{n \in S_+} \sum_{i=2}^{M} \alpha_{n,+,i}^{(s)} \langle \boldsymbol{q}_+^{(s)}, \boldsymbol{k}_{n,i}^{(s)} \rangle \right.$$

$$\left. + (\alpha_{+,+}^{(s)} \boldsymbol{k}_+^{(s)} + \sum_{n \in S_+} \sum_{i=2}^{M} \alpha_{n,+,i}^{(s)} \boldsymbol{k}_{n,i}^{(s)}) \cdot (\alpha_{+,+}^{(s)} \boldsymbol{k}_+^{(s)\top} + \sum_{n \in S_+} \sum_{i=2}^{M} \alpha_{n,+,i}^{(s)} \boldsymbol{k}_{n,i}^{(s)\top}) \right|$$

$$\leq 2 \sum_{s=T_1}^{t} |\alpha_{+,+}^{(s)}| |\langle \boldsymbol{q}_+^{(s)}, \boldsymbol{k}_+^{(s)} \rangle| + 2 \sum_{n \in S_+} \sum_{i=2}^{M} \sum_{s=T_1}^{t} |\alpha_{n,+,i}^{(s)}| |\langle \boldsymbol{q}_+^{(s)}, \boldsymbol{k}_{n,i}^{(s)} \rangle| \tag{146}$$

$$+ \{\text{lower order term}\}$$

$$= O\left( N^{\frac{1}{2}} d_h^{-\frac{1}{4}} \right) \cdot O(\log(d_h^{\frac{1}{2}})) + N \cdot M \cdot O\left( N^{-\frac{1}{2}} d_h^{-\frac{1}{4}} \right) \cdot O(\log(d_h^{\frac{1}{2}}))$$

$$= O\left( N^{\frac{1}{2}} d_h^{-\frac{1}{4}} \log(d_h^{\frac{1}{2}}) \right)$$

where the first inequality is by triangle inequality. Since $\sigma_h^2 \geq (\max\{\sigma_p^2 d, \|\boldsymbol{\mu}\|_2^2\})^{-1} \cdot d_h^{-\frac{1}{2}} (\log(6N^2 M^2/\delta))^{-2}$ and $d_h = \widetilde{\Omega}(\max\{\text{SNR}^4, \text{SNR}^{-4}\} N^2 \epsilon^{-2})$, we have $N^{\frac{1}{2}} d_h^{-\frac{1}{4}} \log(d_h^{\frac{1}{2}}) = o(\|\boldsymbol{\mu}\|_2^2 \sigma_h^2 d_h)$, so $\|\boldsymbol{q}_+^{(t+1)}\|_2^2 = \|\boldsymbol{q}_+^{(T_1)}\|_2^2 + o(\|\boldsymbol{\mu}\|_2^2 \sigma_h^2 d_h) = \Theta(\|\boldsymbol{\mu}\|_2^2 \sigma_h^2 d_h)$. Similarly, we have

$$\left| \|\boldsymbol{q}_-^{(t+1)}\|_2^2 - \|\boldsymbol{q}_-^{(T_1)}\|_2^2 \right| = O\left( N^{\frac{1}{2}} d_h^{-\frac{1}{4}} \log(d_h^{\frac{1}{2}}) \right) = o(\|\boldsymbol{\mu}\|_2^2 \sigma_h^2 d_h),$$

$$\left| \|\boldsymbol{k}_+^{(t+1)}\|_2^2 - \|\boldsymbol{k}_+^{(T_1)}\|_2^2 \right| = O\left( (1+\text{SNR}) N^{\frac{1}{2}} d_h^{-\frac{1}{4}} \log(d_h^{\frac{1}{2}}) \right) = o(\|\boldsymbol{\mu}\|_2^2 \sigma_h^2 d_h),$$

$$\left| \|\boldsymbol{q}_{n,i}^{(t+1)}\|_2^2 - \|\boldsymbol{q}_{n,i}^{(T_1)}\|_2^2 \right| = O\left( d_h^{-\frac{1}{4}} \log(d_h^{\frac{1}{2}}) \right) = o(\sigma_p^2 \sigma_h^2 d_h d_n), \tag{147}$$

$$\left| \|\boldsymbol{k}_{n,i}^{(t+1)}\|_2^2 - \|\boldsymbol{k}_{n,i}^{(T_1)}\|_2^2 \right| = O\left( N^{\frac{1}{2}} d_h^{-\frac{1}{4}} \log(d_h^{\frac{1}{2}}) \right) = o(\sigma_p^2 \sigma_h^2 d_h d_n),$$

so we have

$$\|q_\pm^{(t+1)}\|_2^2, \|k_\pm^{(t+1)}\|_2^2 = \Theta(\|\boldsymbol{\mu}\|_2^2 \sigma_h^2 d_h),$$
$$\|\boldsymbol{q}_{n,i}^{(t+1)}\|_2^2, \|\boldsymbol{k}_{n,i}^{(t+1)}\|_2^2 = \Theta(\sigma_p^2 \sigma_h^2 d d_h) \tag{148}$$

for $i \in [M] \setminus \{1\}, n \in [N]$.

$$|\langle \boldsymbol{q}_+^{(t+1)}, \boldsymbol{q}_-^{(t+1)} \rangle| \leq |\langle \boldsymbol{q}_+^{(T_1)}, \boldsymbol{q}_-^{(T_1)} \rangle| + \sum_{s=T_1}^{t} |\langle \boldsymbol{q}_+^{(s+1)}, \boldsymbol{q}_-^{(s+1)} \rangle - \langle \boldsymbol{q}_+^{(s)}, \boldsymbol{q}_-^{(s)} \rangle|$$

$$\leq |\langle \boldsymbol{q}_+^{(T_1)}, \boldsymbol{q}_-^{(T_1)} \rangle|$$

$$+ \sum_{s=T_1}^{t} \left| \alpha_{+,+}^{(s)} \langle \boldsymbol{q}_+^{(s)}, \boldsymbol{k}_+^{(s)} \rangle + \sum_{n \in S_+} \sum_{i=2}^{M} \alpha_{n,+,i}^{(s)} \langle \boldsymbol{q}_+^{(s)}, \boldsymbol{k}_{n,i}^{(s)} \rangle \right.$$

$$+ \alpha_{-,-}^{(s)} \langle \boldsymbol{q}_-^{(s)}, \boldsymbol{k}_-^{(s)} \rangle + \sum_{n \in S_-} \sum_{i=2}^{M} \alpha_{n,-,i}^{(s)} \langle \boldsymbol{q}_+^{(s)}, \boldsymbol{k}_{n,i}^{(s)} \rangle \tag{149}$$

$$+ \left( \alpha_{+,+}^{(s)} \boldsymbol{k}_+^{(s)} + \sum_{n \in S_+} \sum_{i=2}^{M} \alpha_{n,+,i}^{(s)} \boldsymbol{k}_{n,i}^{(s)} \right)$$

$$\left. \cdot \left( \alpha_{-,-}^{(s)} \boldsymbol{k}_-^{(s)} + \sum_{n \in S_-} \sum_{i=2}^{M} \alpha_{n,-,i}^{(s)} \boldsymbol{k}_{n,i}^{(s)} \right)^{\top} \right|$$

$$
\begin{aligned}
&\leq |\langle \boldsymbol{q}_+^{(T_1)}, \boldsymbol{q}_-^{(T_1)}\rangle| \\
&\quad + \sum_{s=T_1}^{t} |\alpha_{+,+}^{(s)}||\langle \boldsymbol{q}_+^{(s)}, \boldsymbol{k}_+^{(s)}\rangle| + \sum_{n\in S_+} \sum_{i=2}^{M} \sum_{s=T_1}^{t} |\alpha_{n,+,i}^{(s)}||\langle \boldsymbol{q}_+^{(s)}, \boldsymbol{k}_{n,i}^{(s)}\rangle| \\
&\quad + \sum_{s=T_1}^{t} |\alpha_{-,-}^{(s)}||\langle \boldsymbol{q}_-^{(s)}, \boldsymbol{k}_-^{(s)}\rangle| + \sum_{n\in S_-} \sum_{i=2}^{M} \sum_{s=T_1}^{t} |\alpha_{n,-,i}^{(s)}||\langle \boldsymbol{q}_+^{(s)}, \boldsymbol{k}_{n,i}^{(s)}\rangle| \\
&\quad + \{\text{lower order term}\} \\
&\leq |\langle \boldsymbol{q}_+^{(T_1)}, \boldsymbol{q}_-^{(T_1)}\rangle| \\
&\quad + O\left( N^{\frac{1}{2}} d_h^{-\frac{1}{4}} \cdot o(1) + N\cdot M \cdot O\left(N^{-\frac{1}{2}} d_h^{-\frac{1}{4}}\right) \cdot \log(d_h^{\frac{2}{2}})\right) \\
&= |\langle \boldsymbol{q}_+^{(T_1)}, \boldsymbol{q}_-^{(T_1)}\rangle| + O\left(N^{\frac{1}{2}} d_h^{-\frac{1}{4}} \log(d_h^{\frac{2}{2}})\right) \\
&= o(1),
\end{aligned}
\tag{150}
$$

where the first inequality is triangle inequality, the last equality is by $d_h = \widetilde{\Omega}(\max\{\text{SNR}^4, \text{SNR}^{-4}\} N^2 \epsilon^{-2})$. Similarly, we can prove:

$$
\begin{aligned}
\|\boldsymbol{q}_\pm^{(t)}\|_2^2, \|\boldsymbol{k}_\pm^{(t)}\|_2^2 &= \Theta\left(\|\mu\|_2^2 \sigma_h^2 d_h\right), \\
\|\boldsymbol{q}_{n,i}^{(t)}\|_2^2, \|\boldsymbol{k}_{n,i}^{(t)}\|_2^2 &= \Theta\left(\sigma_p^2 \sigma_h^2 d d_h\right), \\
|\langle \boldsymbol{q}_+^{(t)}, \boldsymbol{q}_-^{(t)}\rangle|, |\langle \boldsymbol{q}_\pm^{(t)}, \boldsymbol{q}_{n,i}^{(t)}\rangle|, |\langle \boldsymbol{q}_{n,i}^{(t)}, \boldsymbol{q}_{n',j}^{(t)}\rangle| &= o(1), \\
|\langle \boldsymbol{k}_+^{(t)}, \boldsymbol{k}_-^{(t)}\rangle|, |\langle \boldsymbol{k}_\pm^{(t)}, \boldsymbol{k}_{n,i}^{(t)}\rangle|, |\langle \boldsymbol{k}_{n,i}^{(t)}, \boldsymbol{k}_{n',j}^{(t)}\rangle| &= o(1),
\end{aligned}
$$

for $i, j \in [M]\setminus\{1\}, n, n' \in [N], i \neq j$ or $n \neq n'$.

## I.10 UPPER BOUNDS OF $\langle q, k\rangle$

In order to give the upper bounds for $\langle \boldsymbol{q}, \boldsymbol{k}\rangle$ in stage III, we need to give the upper bounds of $\alpha$ and $\beta$ based on the equations in I.1. The main difference between this subsection and I.7 is that the bounds of $|V_\pm|, |V_{n,i}|$ is $\log\left(O\left(\frac{1}{\epsilon}\right)\right)$ in this subsection, while the bounds of $|V_\pm|, |V_{n,i}|$ is $\log\left(O\left(\frac{1}{\epsilon}\right)\right)$ in

I.7, resulting in different bounds for $\alpha$ and $\beta$. Now we take $\alpha_{+,+}^{(s)}$ as an example

$$
\alpha_{+,+}^{(s)} \leq \frac{\eta}{NM} \sum_{n \in S_+} -\widetilde{\ell}_n'(\theta) \langle \widetilde{\boldsymbol{\mu}}_+, \widetilde{\boldsymbol{\mu}}_+^{(s)} \rangle
$$

$$
\cdot \left( V_+^{(s)} \left( \frac{\exp(\langle \boldsymbol{q}_+^{(s)}, \boldsymbol{k}_+^{(s)} \rangle)}{\exp(\langle \boldsymbol{q}_+^{(s)}, \boldsymbol{k}_+^{(s)} \rangle) + \sum_{j=2}^M \exp(\langle \boldsymbol{q}_+^{(s)}, \boldsymbol{k}_{n,j}^{(s)} \rangle)} \right. \right.
$$

$$
\left. - \left( \frac{\exp(\langle \boldsymbol{q}_+^{(s)}, \boldsymbol{k}_+^{(s)} \rangle)}{\exp(\langle \boldsymbol{q}_+^{(s)}, \boldsymbol{k}_+^{(s)} \rangle) + \sum_{j=2}^M \exp(\langle \boldsymbol{q}_+^{(s)}, \boldsymbol{k}_{n,j}^{(s)} \rangle)} \right)^2 \right)
$$

$$
- \sum_{i=2}^M \left( V_{n,i}^{(s)} \cdot \frac{\exp(\langle \boldsymbol{q}_+^{(s)}, \boldsymbol{k}_+^{(s)} \rangle)}{\exp(\langle \boldsymbol{q}_+^{(s)}, \boldsymbol{k}_+^{(s)} \rangle) + \sum_{j=2}^M \exp(\langle \boldsymbol{q}_+^{(s)}, \boldsymbol{k}_{n,j}^{(s)} \rangle)} \right.
$$

$$
\left. \left. \cdot \frac{\exp(\langle \boldsymbol{q}_+^{(s)}, \boldsymbol{k}_{n,i}^{(s)} \rangle)}{\exp(\langle \boldsymbol{q}_+^{(s)}, \boldsymbol{k}_+^{(s)} \rangle) + \sum_{j=2}^M \exp(\langle \boldsymbol{q}_+^{(s)}, \boldsymbol{k}_{n,j}^{(s)} \rangle)} \right) \right)
$$

$$
+ \sum_{i=2}^M \langle \widetilde{\boldsymbol{\mu}}_+, \widetilde{\boldsymbol{\xi}}_{n,i}^{(s)} \rangle \cdot \left( V_+^{(s)} \left( \frac{\exp(\langle \boldsymbol{q}_{n,i}^{(s)}, \boldsymbol{k}_+^{(s)} \rangle)}{\exp(\langle \boldsymbol{q}_{n,i}^{(s)}, \boldsymbol{k}_+^{(s)} \rangle) + \sum_{j=2}^M \exp(\langle \boldsymbol{q}_{n,i}^{(s)}, \boldsymbol{k}_{n,j}^{(s)} \rangle)} \right. \right.
$$

$$
\left. - \left( \frac{\exp(\langle \boldsymbol{q}_{n,i}^{(s)}, \boldsymbol{k}_+^{(s)} \rangle)}{\exp(\langle \boldsymbol{q}_{n,i}^{(s)}, \boldsymbol{k}_+^{(s)} \rangle) + \sum_{j=2}^M \exp(\langle \boldsymbol{q}_{n,i}^{(s)}, \boldsymbol{k}_{n,j}^{(s)} \rangle)} \right)^2 \right)
$$

$$
- \sum_{k=2}^M \left( V_{n,i}^{(s)} \cdot \frac{\exp(\langle \boldsymbol{q}_{n,i}^{(s)}, \boldsymbol{k}_+^{(s)} \rangle)}{\exp(\langle \boldsymbol{q}_{n,i}^{(s)}, \boldsymbol{k}_+^{(s)} \rangle) + \sum_{j=2}^M \exp(\langle \boldsymbol{q}_{n,i}^{(s)}, \boldsymbol{k}_{n,j}^{(s)} \rangle)} \right.
$$

$$
\left. \cdot \frac{\exp(\langle \boldsymbol{q}_{n,i}^{(s)}, \boldsymbol{k}_{n,k}^{(s)} \rangle)}{\exp(\langle \boldsymbol{q}_{n,i}^{(s)}, \boldsymbol{k}_+^{(s)} \rangle) + \sum_{j=2}^M \exp(\langle \boldsymbol{q}_{n,i}^{(s)}, \boldsymbol{k}_{n,j}^{(s)} \rangle)} \right)
$$

$$
\leq \frac{\eta}{NM} \sum_{n \in S_+} (\|\boldsymbol{\mu}\|_2 + \tau)^2 (V_+^{(s)} \cdot \frac{\exp(\langle \boldsymbol{q}_+^{(s)}, \boldsymbol{k}_+^{(s)} \rangle)}{\exp(\langle \boldsymbol{q}_+^{(s)}, \boldsymbol{k}_+^{(s)} \rangle) + \sum_{j=2}^M \exp(\langle \boldsymbol{q}_+^{(s)}, \boldsymbol{k}_{n,j}^{(s)} \rangle)}
$$

$$
+ \max_{i=2} |V_{n,i}^{(s)}| \cdot \frac{\sum_{j=2}^M \exp(\langle \boldsymbol{q}_+^{(s)}, \boldsymbol{k}_{n,j}^{(s)} \rangle)}{\exp(\langle \boldsymbol{q}_+^{(s)}, \boldsymbol{k}_+^{(s)} \rangle) + \sum_{j=2}^M \exp(\langle \boldsymbol{q}_+^{(s)}, \boldsymbol{k}_{n,j}^{(s)} \rangle)})
$$

$$
+ \sum_{i=2}^M (\|\boldsymbol{\mu}\|\tau + \sigma_p \tau \sqrt{2\log(4NM/\delta)} + \tau^2)(V_+^{(s)} \cdot \frac{\exp(\langle \boldsymbol{q}_{n,i}^{(s)}, \boldsymbol{k}_+^{(s)} \rangle)}{\exp(\langle \boldsymbol{q}_{n,i}^{(s)}, \boldsymbol{k}_+^{(s)} \rangle) + \sum_{j=2}^M \exp(\langle \boldsymbol{q}_{n,i}^{(s)}, \boldsymbol{k}_{n,j}^{(s)} \rangle)}
$$

$$
+ \max_{i=2} |V_{n,i}^{(s)}| \cdot \frac{\sum_{j=2}^M \exp(\langle \boldsymbol{q}_{n,i}^{(s)}, \boldsymbol{k}_{n,j}^{(s)} \rangle)}{\exp(\langle \boldsymbol{q}_{n,i}^{(s)}, \boldsymbol{k}_+^{(s)} \rangle) + \sum_{j=2}^M \exp(\langle \boldsymbol{q}_{n,i}^{(s)}, \boldsymbol{k}_{n,j}^{(s)} \rangle)})
$$

$$
\leq \frac{\eta}{NM} \cdot \frac{3N}{4} \cdot (\|\boldsymbol{\mu}\|_2 + \tau)^2 \cdot \left( V_+^{(s)} \cdot \frac{C}{\exp(\langle \boldsymbol{q}_+^{(s)}, \boldsymbol{k}_+^{(s)} \rangle)} + \max_i |V_{n,i}^{(s)}| \cdot \frac{C}{\exp(\langle \boldsymbol{q}_+^{(s)}, \boldsymbol{k}_+^{(s)} \rangle)} \right)
$$

$$
+ \sum_{i=2}^M (\|\boldsymbol{\mu}\|\tau + \sigma_p \tau \sqrt{2\log(4NM/\delta)} + \tau^2) \cdot \left( V_+^{(s)} \cdot \frac{C}{\exp(\langle \boldsymbol{q}_{n,i}^{(s)}, \boldsymbol{k}_+^{(s)} \rangle)} + \max_i |V_{n,i}^{(s)}| \cdot \frac{C}{\exp(\langle \boldsymbol{q}_{n,i}^{(s)}, \boldsymbol{k}_+^{(s)} \rangle)} \right)
$$

$$
\leq \frac{\eta C_9 (\|\boldsymbol{\mu}\|_2 + \tau)^2 \log\left(O\left(\frac{1}{\epsilon}\right)\right)}{\exp(\langle \boldsymbol{q}_+^{(s)}, \boldsymbol{k}_+^{(s)} \rangle)},
$$

$$
(151)
$$

where the first inequality is by $-\widetilde{\ell}_n'^{(s)} \leq 1$ and $\text{softmax}(\langle \boldsymbol{q}_+^{(s)}, \boldsymbol{k}_+^{(s)} \rangle) \leq 1$. For the second inequality, we first consider $\frac{\sum_{j=2}^M \exp(\langle \boldsymbol{q}_+^{(s)}, \boldsymbol{k}_{n,j}^{(s)} \rangle)}{\exp(\langle \boldsymbol{q}_+^{(s)}, \boldsymbol{k}_+^{(s)} \rangle) + \sum_{j=2}^M \exp(\langle \boldsymbol{q}_+^{(s)}, \boldsymbol{k}_{n,j}^{(s)} \rangle)} \leq \frac{\sum_{j=2}^M \exp(\langle \boldsymbol{q}_+^{(s)}, \boldsymbol{k}_{n,j}^{(s)} \rangle)}{\exp(\langle \boldsymbol{q}_+^{(s)}, \boldsymbol{k}_+^{(s)} \rangle)}$, then by the monotonicity of $\langle \boldsymbol{q}_+^{(s)}, \boldsymbol{k}_{n,j}^{(s)} \rangle$ and $\langle \boldsymbol{q}_+^{(T_1)}, \boldsymbol{k}_{n,j}^{(T_1)} \rangle = o(1)$ we have $\sum_{j=2}^M \exp(\langle \boldsymbol{q}_+^{(s)}, \boldsymbol{k}_{n,j}^{(s)} \rangle) \leq C$ for

$t \in [T_1, T_3]$. The last inequality is by $V_+^{(s)}, |V_{n,i}^{(s)}| \leq 2\log\left(O\left(\frac{1}{\epsilon}\right)\right)$ for $t \in [T_2, T_3]$ and absorbing the constant factors. Similar to I.7, we can give the bounds for the other $\alpha$ and $\beta$ as follows:

$$
\begin{aligned}
\alpha_{-,-}^{(s)} &\leq \frac{\eta C_9(\|\boldsymbol{\mu}\|_2 + \tau)^2 \log\left(O\left(\frac{1}{\epsilon}\right)\right)}{\exp(\langle \boldsymbol{q}_-^{(s)}, \boldsymbol{k}_-^{(s)}\rangle)}, \\
\beta_{+,+}^{(s)} &\leq \frac{\eta C_9(\|\boldsymbol{\mu}\|_2 - \tau)^2 \log\left(O\left(\frac{1}{\epsilon}\right)\right)}{\exp(\langle \boldsymbol{q}_+^{(s)}, \boldsymbol{k}_+^{(s)}\rangle)},
\end{aligned}
\tag{152}
$$

Similar to I.6, we apply the bounds of $\alpha$ and $\beta$ above to give the upper bounds for the dynamics $\langle q, k\rangle$.

$$
\begin{aligned}
&\langle \boldsymbol{q}_+^{(s+1)}, \boldsymbol{k}_+^{(s+1)}\rangle - \langle \boldsymbol{q}_+^{(s)}, \boldsymbol{k}_+^{(s)}\rangle \\
&= \alpha_{+,+}^{(s)}\|\boldsymbol{k}_+^{(s)}\|_2^2 + \sum_{n \in S_+}\sum_{i=2}^{M}\alpha_{n,+,i}^{(s)}\langle \boldsymbol{k}_+^{(s)}, \boldsymbol{k}_{n,i}^{(s)}\rangle \\
&\quad + \beta_{+,+}^{(s)}\|\boldsymbol{q}_+^{(s)}\|_2^2 + \sum_{n \in S_+}\sum_{i=2}^{M}\beta_{n,+,i}^{(s)}\langle \boldsymbol{q}_+^{(s)}, \boldsymbol{q}_{n,i}^{(s)}\rangle \\
&\quad + \left(\alpha_{+,+}^{(s)}\boldsymbol{k}_+^{(s)} + \sum_{n \in S_+}\sum_{i=2}^{M}\alpha_{n,+,i}^{(s)}\boldsymbol{k}_{n,i}^{(s)}\right) \\
&\quad \cdot \left(\beta_{+,+}^{(s)}\boldsymbol{q}_+^{(s)\top} + \sum_{n \in S_+}\sum_{i=2}^{M}\beta_{n,+,i}^{(s)}\boldsymbol{q}_{n,i}^{(s)\top}\right) \\
&= \alpha_{+,+}^{(s)}\|\boldsymbol{k}_+^{(s)}\|_2^2 + \beta_{+,+}^{(s)}\|\boldsymbol{q}_+^{(s)}\|_2^2 + \{\text{lower order term}\} \\
&\leq \frac{2\eta C_9(\|\boldsymbol{\mu}\|_2 + \tau)^2 \log\left(O\left(\frac{1}{\epsilon}\right)\right)}{\exp(\langle \boldsymbol{q}_+^{(s)}, \boldsymbol{k}_+^{(s)}\rangle)} \cdot \Theta(\|\boldsymbol{\mu}\|_2^2 \sigma_h^2 d_h) + \{\text{lower order term}\} \\
&\leq \eta C_{10}\|\boldsymbol{\mu}\|_2^2(\|\boldsymbol{\mu}\|_2 + \tau)\sigma_h^2 d_h \log\left(O\left(\frac{1}{\epsilon}\right)\right)\frac{1}{\exp(\langle \boldsymbol{q}_+^{(s)}, \boldsymbol{k}_+^{(s)}\rangle)},
\end{aligned}
\tag{153}
$$

Similarly, we have the upper bounds for the dynamics of other $\langle \boldsymbol{q}, \boldsymbol{k}\rangle$.

### I.11 BOUNDS FOR THE SUM OF $\alpha$ AND $\beta$

Assume that the propositions $\mathcal{F}(T_2), \ldots, \mathcal{F}(s), \mathcal{H}(T_2), \ldots, \mathcal{H}(s-1)$ hold $(s \in [T_1, t])$, we have

$$
|V_\pm^{(s)}| \leq 2\log\left(O\left(\frac{1}{\epsilon}\right)\right),
\tag{154}
$$

$$
|V_{n,i}^{(s)}| = O(1),
\tag{279}
$$

$$
\Lambda_{n,\pm,j}^{(s)} \geq \Lambda_{n,\pm,j}^{(T_2)} \geq \log\left(\exp(\Lambda_{n,\pm,j}^{(T_1)}) + \Theta\left(\frac{d_h^{\frac{1}{2}}}{N(\log(6N^2M^2/\delta))^3}\right)\right),
\tag{155}
$$

$$
\Lambda_{n,i,\pm,j}^{(s)} \geq \Lambda_{n,i,\pm,j}^{(T_2)} \geq \log\left(\exp(\Lambda_{n,i,\pm,j}^{(T_1)}) + \Theta\left(\frac{\sigma_p^2 ddd_h^{\frac{1}{2}}}{N\|\boldsymbol{\mu}\|_2^2(\log(6N^2M^2/\delta))^3}\right)\right)
\tag{156}
$$

for $i, j \in [M]\backslash\{1\}, n \in [N], s \in [T_2, t]$. Similar to (57) and (58), we have

$$
\frac{\exp(\langle q_\pm^{(s)}, k_{n,j}^{(s)}\rangle)}{\exp(\langle q_\pm^{(s)}, k_\pm^{(s)}\rangle) + \sum_{j'=2}^{M}\exp(\langle q_\pm^{(s)}, k_{n,j'}^{(s)}\rangle)} = O\left(\frac{N(\log(6N^2M^2/\delta))^3}{d_h^{\frac{1}{2}}}\right)
\tag{157}
$$

$$\frac{\exp(\langle \boldsymbol{q}_{n,i}^{(s)}, \boldsymbol{k}_{n,j}^{(s)}\rangle)}{\exp(\langle \boldsymbol{q}_{n,i}^{(s)}, \boldsymbol{k}_{+}^{(s)}\rangle) + \sum_{j'=2}^{M}\exp(\langle \boldsymbol{q}_{n,i}^{(s)}, \boldsymbol{k}_{n,j'}^{(s)}\rangle)} = O\left(\frac{N\|\boldsymbol{\mu}\|_2^2(\log(6N^2M^2/\delta))^3}{\sigma_p^2 dd_h^{\frac{1}{2}}}\right) \tag{158}$$

Plugging above equations into the expressions of $\alpha, \beta$ and letting $O\left(\log\left(O\left(\frac{1}{\epsilon}\right)\right)\right)$ be the upper bound for $|V_{\pm}^{(s)}|, |V_{n,i}^{(s)}|$ we have

$$\begin{aligned}
\alpha_{+,+}^{(s)} &\le \frac{\eta}{NM}\sum_{n\in S_+} -\widetilde{\ell}'_n(\theta)\langle \widetilde{\boldsymbol{\mu}}_+, \widetilde{\boldsymbol{\mu}}_+^{(s)}\rangle \\
&\quad \cdot \left(V_+^{(s)}\left(\frac{\exp(\langle \boldsymbol{q}_+^{(s)}, \boldsymbol{k}_+^{(s)}\rangle)}{\exp(\langle \boldsymbol{q}_+^{(s)}, \boldsymbol{k}_+^{(s)}\rangle) + \sum_{j=2}^{M}\exp(\langle \boldsymbol{q}_+^{(s)}, \boldsymbol{k}_{n,j}^{(s)}\rangle)}\right.\right. \\
&\quad\quad \left.\left. - \left(\frac{\exp(\langle \boldsymbol{q}_+^{(s)}, \boldsymbol{k}_+^{(s)}\rangle)}{\exp(\langle \boldsymbol{q}_+^{(s)}, \boldsymbol{k}_+^{(s)}\rangle) + \sum_{j=2}^{M}\exp(\langle \boldsymbol{q}_+^{(s)}, \boldsymbol{k}_{n,j}^{(s)}\rangle)}\right)^2\right)\right. \\
&\quad\quad \left. - \sum_{i=2}^{M}\left(V_{n,i}^{(s)}\cdot \frac{\exp(\langle \boldsymbol{q}_+^{(s)}, \boldsymbol{k}_+^{(s)}\rangle)}{\exp(\langle \boldsymbol{q}_+^{(s)}, \boldsymbol{k}_+^{(s)}\rangle) + \sum_{j=2}^{M}\exp(\langle \boldsymbol{q}_+^{(s)}, \boldsymbol{k}_{n,j}^{(s)}\rangle)}\right.\right. \\
&\quad\quad\quad \left.\left. \cdot \frac{\exp(\langle \boldsymbol{q}_+^{(s)}, \boldsymbol{k}_{n,i}^{(s)}\rangle)}{\exp(\langle \boldsymbol{q}_+^{(s)}, \boldsymbol{k}_+^{(s)}\rangle) + \sum_{j=2}^{M}\exp(\langle \boldsymbol{q}_+^{(s)}, \boldsymbol{k}_{n,j}^{(s)}\rangle)}\right)\right) \\
&\quad + \sum_{i=2}^{M}\langle \widetilde{\boldsymbol{\mu}}_+, \widetilde{\boldsymbol{\xi}}_{n,i}^{(s)}\rangle \cdot \left(V_+^{(s)}\left(\frac{\exp(\langle \boldsymbol{q}_{n,i}^{(s)}, \boldsymbol{k}_+^{(s)}\rangle)}{\exp(\langle \boldsymbol{q}_{n,i}^{(s)}, \boldsymbol{k}_+^{(s)}\rangle) + \sum_{j=2}^{M}\exp(\langle \boldsymbol{q}_{n,i}^{(s)}, \boldsymbol{k}_{n,j}^{(s)}\rangle)}\right.\right. \\
&\quad\quad \left.\left. - \left(\frac{\exp(\langle \boldsymbol{q}_{n,i}^{(s)}, \boldsymbol{k}_+^{(s)}\rangle)}{\exp(\langle \boldsymbol{q}_{n,i}^{(s)}, \boldsymbol{k}_+^{(s)}\rangle) + \sum_{j=2}^{M}\exp(\langle \boldsymbol{q}_{n,i}^{(s)}, \boldsymbol{k}_{n,j}^{(s)}\rangle)}\right)^2\right)\right. \\
&\quad\quad \left. - \sum_{k=2}^{M}\left(V_{n,i}^{(s)}\cdot \frac{\exp(\langle \boldsymbol{q}_{n,i}^{(s)}, \boldsymbol{k}_+^{(s)}\rangle)}{\exp(\langle \boldsymbol{q}_{n,i}^{(s)}, \boldsymbol{k}_+^{(s)}\rangle) + \sum_{j=2}^{M}\exp(\langle \boldsymbol{q}_{n,i}^{(s)}, \boldsymbol{k}_{n,j}^{(s)}\rangle)}\right.\right. \\
&\quad\quad\quad \left.\left. \cdot \frac{\exp(\langle \boldsymbol{q}_{n,i}^{(s)}, \boldsymbol{k}_{n,k}^{(s)}\rangle)}{\exp(\langle \boldsymbol{q}_{n,i}^{(s)}, \boldsymbol{k}_+^{(s)}\rangle) + \sum_{j=2}^{M}\exp(\langle \boldsymbol{q}_{n,i}^{(s)}, \boldsymbol{k}_{n,j}^{(s)}\rangle)}\right)\right) \\
&\le \frac{\eta(\|\boldsymbol{\mu}\|_2 + \tau)^2}{NM}\cdot \frac{3N}{4}\cdot O\left(\log\left(O\left(\frac{1}{\epsilon}\right)\right)\right)\cdot O\left(\frac{N\left(\log(6N^2M^2/\delta)\right)^3}{d_h^{\frac{3}{2}}}\right) \\
&= O\left(\frac{\eta N(\|\boldsymbol{\mu}\|_2 + \tau)^2\left(\log(6N^2M^2/\delta)\right)^3\log\left(O\left(\frac{1}{\epsilon}\right)\right)}{d_h^{\frac{3}{2}}}\right)
\end{aligned} \tag{159}$$

Taking a summation we obtain that

$$\begin{aligned}
\sum_{s=T_2}^{t}|\alpha_{+,+}^{(s)}| &= O\left(\frac{1}{\eta\epsilon\|\boldsymbol{\mu}\|_2^2\|\boldsymbol{w}_O\|_2^2}\right)\cdot O\left(\eta\|\boldsymbol{\mu}\|_2^2 N(\log(6N^2M^2/\delta))^3\log\left(O\left(\frac{1}{\epsilon}\right)\right)\frac{\log\left(O\left(\frac{1}{\epsilon}\right)\right)}{d_h^{\frac{3}{2}}}\right) \\
&= O\left(\frac{N(\log(6N^2M^2/\delta))^3\log\left(O\left(\frac{1}{\epsilon}\right)\right)}{\epsilon d_h^{\frac{3}{2}}}\right),
\end{aligned} \tag{160}$$

where the last equality is by $\|\boldsymbol{w}_O\| = \Theta(1)$. Similarly, we have the bounds for other sum of $\alpha$ and $\beta$. Thus, we can easily prove Claim 7 and Claim 8.

### I.12 PROOF OF CLAIM 7

In this subsection, we assume that $\mathcal{I}(T_2), \ldots, \mathcal{I}(t)$ hold, and then proof that $\mathcal{G}(t+1)$ is true with the result of I.11.

$$
\begin{aligned}
\left\| \boldsymbol{q}_+^{(t+1)} \right\|_2^2 - \left\| \boldsymbol{q}_+^{(t)} \right\|_2^2 &\le \sum_{s=T_2}^{t} \left| \left\| \boldsymbol{q}_+^{(s+1)} \right\|_2^2 - \left\| \boldsymbol{q}_+^{(s)} \right\|_2^2 \right| \\
&\le \sum_{s=T_2}^{t} \left| 2\alpha_{+,+}^{(s)} \langle \boldsymbol{q}_+^{(s)}, \boldsymbol{k}_+^{(s)} \rangle + 2 \sum_{n \in S_+} \sum_{i=2}^{M} \alpha_{n,+,i}^{(s)} \langle \boldsymbol{q}_+^{(s)}, \boldsymbol{k}_{n,i}^{(s)} \rangle \right. \\
&\qquad + \left. \langle \alpha_{+,+}^{(s)} \boldsymbol{k}_+^{(s)} + \sum_{n \in S_+} \sum_{i=2}^{M} \alpha_{n,+,i}^{(s)} \boldsymbol{k}_{n,i}^{(s)} \cdot \left( \alpha_{+,+}^{(s)} \boldsymbol{k}_+^{(s)} + \sum_{n \in S_+} \sum_{i=2}^{M} \alpha_{n,+,i}^{(s)} \boldsymbol{k}_{n,i}^{(s)} \right)^{\top} \right| \\
&\le 2 \sum_{s=T_2}^{t} \left| \alpha_{+,+}^{(s)} \langle \boldsymbol{q}_+^{(s)}, \boldsymbol{k}_+^{(s)} \rangle \right| + 2 \sum_{n \in S_+} \sum_{i=2}^{M} \sum_{s=T_2}^{t} \left| \alpha_{n,+,i}^{(s)} \langle \boldsymbol{q}_+^{(s)}, \boldsymbol{k}_{n,i}^{(s)} \rangle \right| \\
&\qquad + \{\text{lower order term}\} \\
&= O\left( \frac{N(\log(6N^2M^2/\delta))^3 \log\left(O\left(\frac{1}{\epsilon}\right)\right)}{\epsilon d_h^{\frac{3}{2}}} \right) \cdot \log\left( \epsilon^{-1} d_h^{\frac{1}{2}} \right) \\
&\qquad + N \cdot M \cdot O\left( \frac{(\log(6N^2M^2/\delta))^3 \log\left(O\left(\frac{1}{\epsilon}\right)\right)}{\epsilon d_h^{\frac{3}{2}}} \right) \cdot \log\left( \epsilon^{-1} d_h^{\frac{1}{2}} \right) \\
&= O\left( \frac{N(\log(6N^2M^2/\delta))^3 \log\left(O\left(\frac{1}{\epsilon}\right)\right) \log\left( \epsilon^{-1} d_h^{\frac{1}{2}} \right)}{\epsilon d_h^{\frac{3}{2}}} \right)
\end{aligned}
$$
(161)

where the first inequality is by triangle inequality, the second inequality is by the update rules in D.2, the third inequality is by $t \le T_3$. Since $\sigma_h^2 \ge \left( \max\{\sigma_p^2 d, \|\boldsymbol{\mu}\|_2^2\} \right)^{-1} \cdot d_h^{-\frac{1}{2}} (\log(6N^2M^2/\delta))^{-2}$ and $d_h = \widetilde{\Omega}\left( \max\{\text{SNR}^4, \text{SNR}^{-4}\} N^2 \epsilon^{-2} \right)$, we have

$$
\frac{N(\log(6N^2M^2/\delta))^3 \log\left(O\left(\frac{1}{\epsilon}\right)\right) \log(\epsilon^{-1} d_h^{\frac{1}{2}})}{\epsilon d_h^{\frac{1}{2}}} = o(\|\boldsymbol{\mu}\|_2^2 \sigma_h^2 d_h),
$$
(162)

so $\|\boldsymbol{q}_+^{(t+1)}\|_2^2 = \|\boldsymbol{q}_+^{(T_2)}\|_2^2 + o(\|\boldsymbol{\mu}\|_2^2 \sigma_h^2 d_h) = \Theta(\|\boldsymbol{\mu}\|_2^2 \sigma_h^2 d_h)$. Similarly, we have

$$
\|\boldsymbol{q}_\pm^{(t)}\|_2^2, \|\boldsymbol{k}_\pm^{(t)}\|_2^2 = \Theta(\|\boldsymbol{\mu}\|_2^2 \sigma_h^2 d_h),
$$
$$
\|\boldsymbol{q}_{n,i}^{(t)}\|_2^2, \|\boldsymbol{k}_{n,i}^{(t)}\|_2^2 = \Theta\left( \sigma_p^2 \sigma_h^2 d d_h \right),
$$
$$
|\langle \boldsymbol{q}_+^{(t)}, \boldsymbol{q}_-^{(t)} \rangle|, |\langle \boldsymbol{q}_\pm^{(t)}, \boldsymbol{q}_{n,i}^{(t)} \rangle|, |\langle \boldsymbol{q}_{n,i}^{(t)}, \boldsymbol{q}_{n',j}^{(t)} \rangle| = o(1),
$$
$$
|\langle \boldsymbol{k}_+^{(t)}, \boldsymbol{k}_-^{(t)} \rangle|, |\langle \boldsymbol{k}_\pm^{(t)}, \boldsymbol{k}_{n,i}^{(t)} \rangle|, |\langle \boldsymbol{k}_{n,i}^{(t)}, \boldsymbol{k}_{n',j}^{(t)} \rangle| = o(1)
$$

for $i, j \in [M] \setminus \{1\}, n, n' \in [N], i \ne j$ or $n \ne n'$.

