# OpenReview forum: "Benign Overfitting in Adversarial Training for Vision Transformers"
_ICLR.cc/2026/Conference — Submitted to ICLR 2026_

### Official Review · Reviewer_Uksd · 2025-10-27

**Soundness:** 2
**Presentation:** 3
**Contribution:** 1
**Rating:** 2
**Confidence:** 2

**Summary:**

A theoretical analysis of a two layer vision transformer with adversarial training is done. The adversarial threat model for this work is with respect to the l-infinity norm. Empirically the method is validated on a heavily modified version of the MNIST dataset and a synthetic dataset.

**Strengths:**

The paper is well written and easy to follow.

**Weaknesses:**

I am not an expert in theoretical machine learning and much of the paper’s strength lies in the fact that they have done a lot of mathematical derivations. Other reviewers will have to comment on whether that alone is enough to merit acceptance.

However, from an experimental perspective I find the results entirely unconvincing for the following reasons:

1. The choice of datasets are too small and not impactful. The authors only test on a synthetic dataset and MNIST. Actually, even the MNIST dataset they use is not the true MNIST dataset. The dataset the authors test on is a binarized version of the MNIST dataset. These days CIFAR-10/100 experiments are the minimum for experimentally demonstrating a viable adversarial robustness technique (with Tiny-ImageNet and ImageNet also becoming norms).

2. The attacks used in the paper are not SOTA. APGD would be a much better choice (instead of PGD) and an L2 version of the APGD code has been available for multiple years:
https://github.com/fra31/auto-attack

I also question WHY the authors only use the l2 metric. Why not also show what happens for l-inf, l-0 or l-1 attacks? Here are the related attack links:

L0: https://github.com/fra31/sparse-imperceivable-attacks
L1: https://arxiv.org/abs/2103.01208
L2: https://arxiv.org/pdf/2003.01690

 There has been much work that shows it may not be enough to just prevent one norm attack, so considering multi-norm attacks (even if the results are poor) is a much more interesting scope: https://proceedings.mlr.press/v119/maini20a/maini20a.pdf

As a reviewer I cannot mandate that you do any more experiments. However, I would say that it is nearly impossible for me to be an advocate for your paper because the scope of the current work is not justifiable in my opinion. If you could extend your framework and your experiments to the multi-norm case (even if the results then become less robust), that would be a much much stronger work.

**Questions:**

Please address the issues I mention in the weakness section of my review. Specifically:

A. Why aren't SOTA attacks used in the experimental results?
B. Why only focus on L2 norm? Can you give any better justification for the scope of your current work?
C. Why aren't more complex datasets used?

---

> ### Author Response · Authors · 2025-11-22
>
> - **Q1: Question on SOTA attack**
>     >Why aren't SOTA attacks used in the experimental results?
>
> ### Response to Q1:
> Thanks for the comment. Note that our theoretical framework (Algorithm 1, Step 4) explicitly guarantees optimization toward the strongest adversarial perturbation under the chosen norm.
>
> And we have added **additional experiments on MINST and CIFAR-10 with APGD**, as detailed in Appendix B.2. We show that our theoretical insights continue to hold for larger and more complex datasets, as well as under stronger adversarial attacks. We still observe a clear phase transition phenomenon. Moreover, as both the sample size $N$ and the $SNR$ increase, the clean and robust test error consistently decreases.
>
> ---
>
>  - **Q2: Question about attack norm**
>     >Why only focus on L2 norm? Can you give any better justification for the scope of your current work?
>
> ### Response to Q2:
>
> Thanks for the comment. In fact, our analysis actually covers perturbations under all norm types you mentioned. We focus on the $\ell_{2}$ threat model because it is the most commonly adopted norm in both theoretical and empirical robustness studies.
>
> 1. For $\ell_{\infty}$ or $\ell_{1}$-norms, they can be mapped to $\ell_{2}$ through standard norm-equivalence. For any $x \in \mathbb{R}^d$ and $1 \le p \le q \le \infty$, the following inequality holds:
>
> $\lVert x \rVert_q \le d^{(\frac{1}{q} - \frac{1}{p})} \lVert x \rVert_p.$
>
> In particular, the $\ell_{\infty}$ or $\ell_{1}$ norms satisfy:
>
> $\lVert x \rVert_\infty \le \lVert x \rVert_2 \le \sqrt{d}\lVert x \rVert_\infty, \lVert x \rVert_2 \le \lVert x \rVert_1 \le \sqrt{d}\lVert x \rVert_2.$
>
> Thus, an $\ell_{\infty}$ or $\ell_{1}$ perturbation budget $\tau$ corresponds to an $\ell_{2}$ budget scaled by at most $\sqrt{d}$. That means, all the proofs are still hold with the only difference is the perturbation radius $\tau$.
>
> 2.  For the $\ell_{0}$-norm perturbation model, our theoretical results of lower bounds has implicitly showed that it cannot provide benign overfitting guarantees. According to Theorem 3, once the $\ell_{2}$-norm perturbation radius becomes sufficiently large (i.e., $\tau \ge \lVert \mu \rVert_2$), the model incurs a large robust test error. This implies that even an $\ell_{0}$-norm radius of 1 can still lead to substantial robust test error in the worst case.
>
> 3. We include **additional experiments on MNIST and CIFAR-10 under multi-norm ($\ell_{1}$, $\ell_{2}$, $\ell_{\infty}$) attacks** in Appendix B.3. Following the training setup of [1], we select at each iteration the most challenging adversarial example among the three norms. The results show that although the model’s robust test accuracy decreases under multi-norm attacks, our theoretical insights continue to hold.
>
>
> [1]Maini et al. Adversarial Robustness Against the Union of Multiple Perturbation Models
>
> ---
>
>  - **Q3：Question on dataset**
>    >Why aren't more complex datasets used?
>
> ### Response to Q3:
>
> Thanks for the comment. We have added **additional experiments on MINST and CIFAR-10 with APGD**, as detailed in Appendix B.2. We show that our theoretical insights continue to hold for larger and more complex datasets, as well as under stronger adversarial attacks. We still observe a clear phase transition phenomenon. Moreover, as both the sample size $N$ and the $SNR$ increase, the clean and robust test error consistently decreases.

---

> ### Author Response · Authors · 2025-11-24
>
> Dear Reviewer,
>
> Thank you again for your valuable feedback. We wanted to confirm that we have fully addressed all of your comments. If there are no remaining concerns, we would greatly appreciate your consideration in updating the score to reflect the improvements.
>
> Best regards,
>
> Authors

---

> > ### Comment · Reviewer_Uksd · 2025-11-24
> >
> > " it is the most commonly adopted norm"
> >
> > =Do you have any citation to back up such a bold claim? I don't think that is correct, I would consider the l-infinity norm to be the most widely studied norm in the adversarial machine learning image domain but I can only say this from experience. Can you give better reasoning?
> >
> >
> > Why are the l1, and l-inf results in the appendix instead of in the main body of the paper?
> >
> > I am still uncomfortable with the fact that the biggest dataset is CIFAR-10 but I realize you may not be able to do more experiments in the time given.

---

> ### Author Response · Authors · 2025-11-25
>
> Thank you for the remark. We agree that the original statement *“it is the most commonly adopted norm”* was too strong. However, we note that the $ \ell_{2}$-norm attack is indeed very common in theoretical analyses of adversarial robustness[1,2].
>
> Moreover, **our results extend directly to the$ \ell_{\infty}$-norm setting**. This extension does not affect our benign overfitting conclusions or the dependence on N and the SNR conditions. That is, under
> $\tau_{\infty} \le \frac{\lVert \mu \rVert_2}{\sqrt{d}\log d_h}$
> and
> $N \cdot \mathrm{SNR}^2 = \Omega(1)$,
> we still obtain that the **robust training loss converges to $\epsilon$**, and both the **clean and robust test errors approach nearly 0**.
>
> [1]Wang et al. Benign Overfitting in Adversarial Training of Neural Networks
>
> [2]Hao et al. The Surprising Harmfulness of Benign Overfitting for Adversarial Robustness
>
> ---
>
> Thanks for the comment. The paper’s core conceptual contribution and main technical challenges are already fully captured in the $ \ell_{2}$ norm setting. The extensions to the $ \ell_{1}$ and $ \ell_{\infty}$ norms can be established almost identically using norm-equivalence arguments and therefore do not introduce additional conceptual insights.
>
> We are happy to move a shortened summary of these results into the main text if you believe it would improve accessibility.
>
> ---
>
> Thanks for the comment. First, we would like to emphasize that **our main contribution lies in analyzing the training dynamics and generalization behavior of ViTs under adversarial training**. The experiments we provide demonstrate that our theory holds in more general settings.
>
> To the best of our knowledge, MNIST and CIFAR-10 are usually used real-world benchmark datasets in the feature-learning literature[1,2]. We have already included CIFAR-10 to show that our theoretical findings extend to a more complex real-world dataset beyond MNIST. We expect that similar conclusions would hold on CIFAR-100 as well.
>
> [1]Wang et al. Benign Overfitting in Adversarial Training of Neural Networks
>
> [2]Chen et al. Benign Overfitting in Adversarially Robust Linear Classification

---

> > ### Author Response · Authors · 2025-12-01
> >
> > We have **added experiments on Tiny-ImageNet** in Appendix B.2 and B.4. The results demonstrate that the phase transition phenomenon remains clearly present, and that increasing either the sample size $N$ or the SNR consistently reduces both the clean and robust test error, fully in line with our theoretical predictions.
> >
> > We validate our theoretical findings across multiple complex real-world datasets, including MNIST, CIFAR-10, and Tiny-ImageNet.

---

### Official Review · Reviewer_U4KV · 2025-10-31

**Soundness:** 2
**Presentation:** 3
**Contribution:** 2
**Rating:** 4
**Confidence:** 3

**Summary:**

In this paper, with a solid theoretical analysis, the authors demonstrate that the phenomenon of benign overfitting also exists for ViTs. Experiments on a real-world dataset (MNIST) highlight the correctness of the finding.

**Strengths:**

1 The theoretical proof is solid.

2 This paper is well motivated.

3 The writing is good.

**Weaknesses:**

1 Acknowledging the theoretical contributions of this paper, the findings do not bring new insights to the community. For example, as listed in the second point in the contribution of this paper, the authors claim that:

1) Small perturbations yield trajectories close to clean training. (According to the definition of Adversarial training, if the perturbation is small enough, AT will collapse to natural training).

2) Moderate perturbations cause the attention mechanism to fail, such that the ViT collapses to a linear model; (Due to the misleading effect of the moderate adversarial samples, it will disrupt the attention mechanism.)

3) Large perturbations lead to significant generalization error beyond benign overfitting.  (In this circumstance, the robust overfitting will happen and resisting attacks crafted with a large attack budget is also challenging, increasing the generalization error.)

2 Theories with practical implications tend to be more appreciated. Unfortunately, this paper does not give the take-away tips on how to better perform AT on ViT Transformers.

3 The verified dataset is the MNIST dataset, which is the simplest dataset in image classification. Without the experiments on more complex datasets such as CIFAR-10 and ImageNet, the correctness of the theory can not be verified in the application of ViTs in real scenarios.

4 In Line 418, the signal and noise vectors is concatenated to form a new vector which is quite different from the application of AT in real-world datasets.

5 The theory makes an analysis on a simple Transformer architecture, ignoring the role of the linear projection layer and the MLP head.

**Questions:**

1 Why, in the verified experiment, only the samples of "0" and "1" labels are chosen to perform experiments? Can the theory be generalized to datasets with more classes?

2 Can the experiments be generalized to explain the appearance of the robust overfitting in Vision Transformer?

---

> ### Author Response · Authors · 2025-11-22
>
> - **W1：Question about insight**
>   >The findings do not bring new insights to the community.
>
> ### Response to W1:
>
> We kindly do not agree. Before our paper, all the three findings you mentioned are intuitive and there is no any regiorous understanding of "why". Also, some findings are inaccurate (comment 3). Below we will response the findings one by one.
>
>   >1.Small perturbations yield trajectories close to clean training. (According to the definition of Adversarial training, if the perturbation is small enough, AT will collapse to natural training).
>
>
> First, "close to clean training" does not imply that benign overfitting occurs. We have rigorously proven the specific requirements of $N$, SNR, and $\tau$ for benign overfitting under adversarial training.
>
> Second, intuitively, small perturbation may close to clean training. However,
> mathmatically, the exact perturbation radius required to trigger each phase remains unexplored, and the relationship between ViT’s benign overfitting under adversarial training and other factors, such as $N$ and SNR, has not been investigated.
>
>   >2.Moderate perturbations cause the attention mechanism to fail, such that the ViT collapses to a linear model; (Due to the misleading effect of the moderate adversarial samples, it will disrupt the attention mechanism.)
>
> To the best of our knowledge, no prior work has demonstrated or discussed the effect of moderate perturbations on the attention mechanism, nor shown that it can cause the ViT to collapse to a linear model. We will be appreciated if the reviewer could provide some references.
>
>   >3.Large perturbations lead to significant generalization error beyond benign overfitting. (In this circumstance, the robust overfitting will happen and resisting attacks crafted with a large attack budget is also challenging, increasing the generalization error.)
>
> Note that, unlike your comments, in our theoretical setting, robust overfitting does not occur. According to Step 4 of Algorithm 1, for each epoch we select the perturbation that maximizes the loss, which is different from the fixed-pattern attacks commonly used in empirical settings. So such phenomenon may not be caused by robust overfitting.
> As a result, the model cannot exploit a single attack pattern or learn its structure, and therefore robust overfitting does not arise in our analysis.
>
>
>
> ---
>
> - **W2：Question about practical implications**
>   >This paper does not give the take-away tips on how to better perform AT on ViT Transformers.
>
> ### Response to W2:
>
> Thanks for the comment. From a feature learning perspective, our primary focus is on understanding rather than modifying the training process. The fact that we do not change the training procedure does not imply a lack of contribution. **Providing understanding** does not mean no contribution.
> In fact, the entire field of feature learning is largely centered around understanding.
>
> Nevertheless, we are happy to discuss several practical tips on how to better perform adversarial training on ViT-based Transformers.
>
> 1. **We provide guidance on the choice of the perturbation budget in adversarial training.** Our theoretical analysis (Theorem 2) suggests that to maintain effective adversarial learning, one should choose the perturbation budget $\tau \leq \frac{\lVert{\mu}\rVert_2}{\log d_h}$.
>
> 2. **We provide guidance on selecting $N$ and the signal-to-noise ratio $SNR$ in adversarial training.** Another key condition for preventing benign overfitting during adversarial training is ensuring that both $N$ and the SNR are sufficiently large. In practice, researchers sometimes employ data augmentation by injecting controlled noise into the dataset. From the perspective of our theoretical results, this approach decreases the $SNR$ while increasing $N$, since noise injection yields “new" data points. Because reducing the SNR may be harmful to generalization, it is important to ensure that a sufficiently large number of data points is used to train the model.
>
> ---
>
>
>  - **W3：Question on dataset**
>    >Without the experiments on more complex datasets such as CIFAR-10 and ImageNet, the correctness of the theory can not be verified in the application of ViTs in real scenarios.
>
> ### Response to W3:
>
> Thanks for the comment. **We have added additional experiments on MINST and CIFAR-10 with APGD, as detailed in Appendix B.2.** We show that our theoretical insights continue to hold for larger and more complex datasets, as well as under stronger adversarial attacks. We still observe a clear phase transition phenomenon. Moreover, as both the sample size $N$ and the $SNR$ increase, the clean and robust test error consistently decreases.

---

> > ### Author Response · Authors · 2025-11-22
> >
> > - **W4：Question about application**
> >    >In Line 418, the signal and noise vectors is concatenated to form a new vector which is quite different from the application of AT in real-world datasets.
> >
> > ### Response to W4:
> >
> > **This is inaccurate.** Please note that this is not the perturbation setting used in our adversarial training, but rather a way to simulate noise patches in the original image. Since the SNR is not accessible in real-world datasets, we treat the entire image as the signal patch and add an additional noise patch as the noise. In fact, the images with added noise become more challenging to learn than the original images.
> >
> > ---
> >
> >  - **W5: Question on ViT architecture**
> >     >The theory makes an analysis on a simple Transformer architecture, ignoring the role of the linear projection layer and the MLP head.
> >
> > ### Response to W5:
> >
> > Thanks for the comment. **Our theory directly extends to architectures with a linear projection and an MLP head.** We do not include it because our paper is already 80 pages. Adding these results will make our paper more complicated and hard to read.
> >
> >
> > A linear projection $P$ does not change the fundamental geometry of adversarial training. It simply reparameterizes the input space: the model operates on $Px$ rather than $x$. All geometric quantities (inner products, margins, adversarial directions) are computed in this projected space, where $\langle Px_i, Px_j \rangle = x_i^\top (P^\top P) x_j.$
> > Thus, the attention learning dynamics maintain exactly the same form after substituting $x$ with $Px$. The projection layer changes the coordinate system but not the core mechanisms analyzed in our theory.
> >
> > As for the MLP components, their effect can be absorbed into the value matrix $W_V$。
> >
> > Besides, our theoretical analysis adopts a 2-layer Transformer model, which is a more realistic and more complex architecture than those considered in most prior theoretical works. Previous analyses often assumed linear attention [1] or merged $W_Q$ and $W_K$ into a shared matrix [2,3], whereas we employ a more faithful attention structure and carefully analyze the training dynamics of $W_Q$ and $W_K$, which introduces significant theoretical challenges.
> >
> >
> >
> > [1]Frei et al. Trained Transformer Classifiers Generalize and Exhibit Benign Overfitting In-Context
> >
> > [2]Magen et al. Benign Overfitting in Single-Head Attention
> >
> > [3]Sakamoto et al. Benign Overfitting in Token Selection of Attention Mechanism
> >
> >
> > ---
> >
> >  - **Q1: Question on multi-classes**
> >     >Why, in the verified experiment, only the samples of "0" and "1" labels are chosen to perform experiments? Can the theory be generalized to datasets with more classes?
> >
> > ### Response to Q1:
> >
> > Thanks for the comment. Our theoretical analysis focuses on a binary classification setting. Therefore, in the MNIST experiments, we simply selected two classes for validation, and choosing different labels does not affect the results.
> >
> > First, the primary objective of this work is to study the dynamics, convergence, and generalization of the attention mechanism in ViTs under adversarial training, rather than tackling multi-class classification. Existing work has already analyzed benign overfitting in multi-class settings for two-layer neural networks [1], and we believe that their analysis can be extended to our Transformer architecture. However, this direction is not the main focus of the present paper.
> >
> > \[1]Xu et al. Rethinking Benign Overfitting in Two-Layer Neural Networks
> >
> > ---
> >
> >
> >  - **Q2: Question on robust overfitting**
> >     >Can the experiments be generalized to explain the appearance of the robust overfitting in Vision Transformer?
> >
> > ### Response to Q2:
> >
> > Thanks for the comment. In our theoretical setting, robust overfitting does not occur. According to Step 4 of Algorithm 1, for each epoch we select the perturbation that maximizes the loss, which is different from the fixed-pattern attacks commonly used in empirical settings. As a result, the model cannot exploit a single attack pattern or learn its structure, and therefore robust overfitting does not arise in our analysis.

---

> ### Author Response · Authors · 2025-11-24
>
> Dear Reviewer,
>
> Thank you again for your valuable feedback. We wanted to confirm that we have fully addressed all of your comments. If there are no remaining concerns, we would greatly appreciate your consideration in updating the score to reflect the improvements.
>
> Best regards,
>
> Authors

---

> ### Comment · Reviewer_U4KV · 2025-11-27
>
> Dear authors,
>
> Thank you so much for your rebuttal. Most of my concerns are well addressed. However, noticing that there are **highly related** papers that are not discussed in this paper. I suggest adding them to the revised version.
>
> 1 When Adversarial Training Meets Vision Transformers: Recipes from Training to Architecture
>
> 2 Are Transformers More Robust than CNNs?
>
> 3 Comparative Study of Adversarial Defenses: Adversarial Training and Regularization in Vision Transformers and CNNs
>
> I will increase my score if the above concern are solved.
>
> Best regards,
>
> Reviewer U4KV

---

> > ### Author Response · Authors · 2025-11-27
> >
> > Thank you for your response and for acknowledging that our rebuttal has addressed your concerns. We greatly appreciate your constructive suggestions.
> >
> > We have added discussions of the three highly relevant papers you mentioned into the revised version of the Related Work section, as shown below:
> >
> >  >**Differences in Adversarial Robustness Between Transformer and CNN.** Several studies have compared Transformers and CNNs under adversarial attacks. [1] observe that under unified training settings, Transformers are not inherently more robust, with their OOD generalization mainly attributed to self-attention. [2] show that under standard training, Transformers do not necessarily outperform CNNs under adversarial attack, and propose training strategies to improve ViT robustness. [3] propose a regularization method that enables ViTs to exhibit stronger adversarial robustness than CNNs. These studies are largely empirical, focusing on performance differences, and do not analyze the learning dynamics or the role of attention in adversarial training.
> >
> > We hope that we have fully addressed all of your concerns.
> >
> > [1] Bai et al. Are Transformers More Robust Than CNNs?
> >
> > [2] Mo et al. When Adversarial Training Meets Vision Transformers: Recipes from Training to Architecture
> >
> > [3] Dingeto et al. Comparative Study of Adversarial Defenses: Adversarial Training and Regularization in Vision Transformers and CNNs

---

> > > ### Comment · Reviewer_U4KV · 2025-11-27
> > >
> > > Thank you for the further revision, I have increased my score to 8.

---

### Official Review · Reviewer_zyXo · 2025-11-01

**Soundness:** 3
**Presentation:** 3
**Contribution:** 2
**Rating:** 6
**Confidence:** 2

**Summary:**

This paper provides the first theoretical analysis of benign overfitting under adversarial training for Vision Transformers (ViTs). The authors construct a simplified two-layer ViT model, derive convergence and generalization guarantees, and identify distinct regimes of adversarial perturbation magnitudes that influence learning dynamics. Empirical validations on synthetic and MNIST data confirm the theoretical predictions.

**Strengths:**

The paper provides a comprehensive and rigorous theoretical analysis of benign overfitting in the context of adversarial training for Vision Transformers (ViTs). Specifically, it extends the study of benign overfitting to the transformer architecture, offering new insights into how the interplay between attention mechanisms, signal-to-noise ratio, and perturbation magnitude determines both robust generalization and overfitting behavior.

**Weaknesses:**

The main limitation of the paper lies in its experimental evaluation. The experiments are conducted only on a subset of MNIST using shallow Vision Transformers (ViTs), which is too simplistic to convincingly verify the proposed theoretical results. MNIST lacks the complexity required to test the robustness and generalization behaviors predicted by the theory. At minimum, the authors should include experiments with a standard multi-layer ViT on a more challenging dataset such as CIFAR-10, to better demonstrate the practical relevance and validity of their theoretical findings.

**Questions:**

## 1. Generality beyond Vision Transformers:
I am wondering whether the current theoretical investigation could be extended beyond the simplified two-layer ViT model to encompass more general Transformer architectures—for example, models with multiple self-attention layers, residual connections, layer normalization, or feed-forward blocks. It would be valuable to understand whether the derived benign overfitting behavior and robustness–generalization relationships continue to hold under these more realistic architectural settings, and whether the theoretical scaling laws remain consistent when evaluated on larger and more complex datasets beyond MNIST.

## 2. Applicability to ViT variants:
I am also curious about how the proposed theorems and analysis apply to different variants of Vision Transformers, such as Swin Transformer, DeiT, or hierarchical ViTs that modify the attention mechanism or token structure. Do the key theoretical conditions, particularly those involving the signal-to-noise ratio and perturbation magnitude, still characterize the transition between benign and harmful overfitting in these variants? Some clarification or discussion on the generality of the theoretical framework across ViT architectures would strengthen the paper’s impact and scope.

## 3. Empirical validation of theoretical regimes:
The paper identifies three distinct regimes of adversarial perturbation (clean-like, linear-collapse, and failure). Could the authors provide more detailed empirical evidence or visualizations to confirm these transitions—perhaps by monitoring changes in attention distributions, feature alignment, or representation collapse across varying perturbation strengths? Such results would make the theoretical phase transition more tangible and strengthen the connection between theory and practice.

---

> ### Author Response · Authors · 2025-11-22
>
> - **W and Q1：Question on dataset**
>    >At minimum, the authors should include experiments with a standard multi-layer ViT on a more challenging dataset such as CIFAR-10, to better demonstrate the practical relevance and validity of their theoretical findings. Whether the theoretical scaling laws remain consistent when evaluated on larger and more complex datasets beyond MNIST.
>
> ### Response to W and Q1:
>
> Thanks for the comment. We have added additional experiments on CIFAR-10, as detailed in Appendix B.2. We show that our theoretical insights continue to hold for larger and more complex datasets, as well as under stronger adversarial attacks. We still observe a clear phase transition phenomenon. Moreover, as both the sample size $N$ and the $SNR$ increase, the clean and robust test error consistently decreases.
>
> Our main contribution lies in the theoretical analysis rather than the empirical results. We provide a detailed theoretical characterization of the training dynamics and convergence–generalization behavior of a 2-layer Transformer under different adversarial perturbation magnitudes. The experiments are conducted solely to validate our theoretical predictions.
>
> Moreover, the models used in our MNIST and CIFAR-10 experiments already go beyond the theoretical setting analyzed in this paper: we employ two attention layers, each with four self-attention heads, followed by an MLP layer with ReLU activation, which corresponds to a standard Transformer architecture.
>
> ---
> - **Q2：Applicability to ViT variants**
>   >I am also curious about how the proposed theorems and analysis apply to different variants of Vision Transformers, such as Swin Transformer, DeiT, or hierarchical ViTs that modify the attention mechanism or token structure.
>
>
> ### Response to Q2:
>
> Thanks for the comment. First, existing theoretical analyses of feature learning are all developed under the vanilla Transformer setting [1,2,3], and our work intentionally follows this standard setup.
> In addition, prior adversarial-training theory has almost exclusively focused on linear models or simple 2-layer neural networks [4,5]. To the best of our knowledge, this is the first work that develops a generalization theory for adversarial training on models equipped with an attention mechanism, namely the Transformer architectures. This represents the core contribution of our work.
>
> [1]Jiang et al. Unveil Benign Overfitting for Transformer in Vision: Training Dynamics, Convergence, and Generalization
>
> [2]Magen et al. Benign Overfitting in Single-Head Attention
>
> [3]Sakamoto et al. Benign Overfitting in Token Selection of Attention Mechanism
>
> [4]Wang et al. Benign Overfitting in Adversarial Training of Neural Networks
>
> [5]Chen et al. Benign Overfitting in Adversarially Robust Linear Classification
>
> ---
>
> - **Q3：Empirical validation of theoretical regimes**
>   >Could the authors provide more detailed empirical evidence or visualizations to confirm these transitions—perhaps by monitoring changes in attention distributions, feature alignment, or representation collapse across varying perturbation strengths.
>
>
> ### Response to Q3:
>
> Thanks for the comment. We have added experiments on training dynamics with different perturbation $\tau$ radius, as detailed in Appendix B.1.
>
> 1. In Figure 3a, when $\frac{\tau}{\lVert \mu \rVert_2} = 0.02$ or $0.1$, the adversarial training loss converges to zero, indicating that the model successfully interpolates all noise-corrupted training samples, consistent with the benign overfitting behavior predicted in Theorem 2. In contrast, when $\frac{\tau}{\lVert \mu \rVert_2} = 0.5$, the adversarial training loss fails to decrease, aligning with the non-convergence regime characterized in Theorem 3. Moreover, the case $\frac{\tau}{\lVert \mu \rVert_2} = 0.02$ exhibits a faster convergence rate than the $\frac{\tau}{\lVert \mu \rVert_2} = 0.1$ setting, highlighting the role of the attention mechanism in accelerating convergence, which is in agreement with our theoretical predictions in Theorem 2.
>
>
> 2. In Figure 3b, when $\frac{\tau}{\lVert \mu \rVert_2} = 0.02$, the attention entropy decreases to nearly zero, indicating that the attention mechanism correctly concentrates on the signal patch. This behavior is consistent with Case 1 of Theorem 2, where the perturbation level is sufficiently small for the model to recover the underlying signal structure.
> In contrast, when $\frac{\tau}{\lVert \mu \rVert_2} = 0.1$, the attention entropy fails to decrease and instead remains high, demonstrating that moderate perturbations hinder the learning of attention weights. As a result, the attention distribution remains nearly uniform rather than focusing on the signal patch. This phenomenon aligns with Case 2 of Theorem 2, where the perturbation magnitude prevents the attention mechanism from identifying the true signal.

---

> > ### Comment · Reviewer_zyXo · 2025-11-26
> > **Any additional experiments beyond 2-layer ViT?**
> >
> > I would say that you have partially addressed my concerns. In Appendix B.2, you extend your experiments on MNIST and CIFAR-10. **I am wondering whether you are still using the simplified two-layer ViT for these datasets?** From my perspective, extending the analysis to more realistic models would significantly increase the impact of the work.
> >
> > I also notice that you included some dynamics to illustrate the transitions, which is helpful. However, a more detailed explanation of the figure would significantly improve clarity.

---

> > > ### Author Response · Authors · 2025-12-01
> > >
> > > In Appendix B.4, we additionally **present experiments using a realistic model (ViT-Base)** on MNIST, CIFAR-10, and Tiny-ImageNet. These results further validate our theory on both more scaled models and more complex datasets.
> > >
> > > Besides, in the revised version, we have added a more detailed explanation of the dynamics and clarified the transitions accordingly.

---

> ### Author Response · Authors · 2025-11-22
>
> 3. In Figure 3c, when $\frac{\tau}{\lVert \mu \rVert_2} = 0.1$, the $\lVert W_V \rVert_2$ norm exhibits the largest growth. This behavior is consistent with Case 2 of Theorem 2, where the ViT effectively collapses into a linear model and the value projection $W_V$ becomes the dominant component driving the learning dynamics.
> In contrast, for $\frac{\tau}{\lVert \mu \rVert_2} = 0.02$, the attention mechanism remains effective, so only mild updates to $W_V$ are required for the model to fit the noisy training data and achieve benign overfitting. When $\frac{\tau}{\lVert \mu \rVert_2} = 0.5$, the perturbation is too large for the model to learn meaningful structure, resulting in $W_V$ failing to make progress during training.
>
> ---
>
> - **Q1：Generality beyond Vision Transformers**
>   >I am wondering whether the current theoretical investigation could be extended beyond the simplified two-layer ViT model to encompass more general Transformer architectures—for example, models with multiple self-attention layers, residual connections, layer normalization, or feed-forward blocks.
>
> ### Response to Q1:
>
> Thanks for the comment. First, our theoretical analysis adopts a 2-layer Transformer model, which is a more realistic and more complex architecture than those considered in most prior theoretical works. Previous analyses often assumed linear attention [1] or merged $W_Q$ and $W_K$ into a shared matrix [2,3], whereas we employ a more faithful attention structure and carefully analyze the training dynamics of $W_Q$ and $W_K$, which introduces significant theoretical challenges.
>
> In addition, the models used in our MNIST and CIFAR experiments already go beyond the theoretical model analyzed in this paper: we use two attention layers, each with four self-attention heads, followed by an MLP layer with ReLU activation, which corresponds to a standard Transformer architecture. The empirical results are consistent with our theoretical analysis, indicating that our theoretical conclusions also apply to more general settings.
>
> 1. For multi-head attention (MHA), we are fortunate to find that our theoretical framework extends to multiple heads relatively straightforwardly. We put the formal anlaysis in Appendix C. We let the parameters be
>
> $\theta := \{(W_{Q,h}, W_{K,h}, W_{V,h})\}_{h=1}^H$
>
> , where $W_{Q,h}, W_{K,h} \in \mathbb{R}^{d \times d_h}$ and $W_{V,h} \in \mathbb{R}^{d \times d_v}$ for each $h \in [H]$. Here $H$ denotes the number of attention heads, which we treat as a fixed constant. Under this parameterization, the network can be written as:$f({X}, \theta)=\sum_{h=1}^Hf_h({X}, \theta)$
> where,
>
> $$
> f_h(X, \theta)
> = \frac{1}{M} \sum_{l=1}^{M}
> \varphi(x_l^{\top} W_{Q,h} W_{K,h}^{\top} X^{\top})
> \, X W_{V,h} w_O.
> $$
>
> The gradients in the multi-head attention module,
> $\frac{\partial f_{h}}{\partial W_{K,h}}, \frac{\partial f_{h}}{\partial W_{Q,h}}, \frac{\partial f_{h}}{\partial W_{V,h}}$,
>
> remain unchanged. However, the gradient of the loss with respect to the output of each single head, i.e., $\frac{\partial \ell}{\partial f_h}$, does change.
> Intuitively, the model output increases by approximately an $H$-fold factor, which causes the scale of the loss $\ell'$ to decrease accordingly.
> More concretely, following our analysis of the signal attention head, $\ell'^{(t)}=\frac{1}{M}\pm o(1)$ stay when $t \leq T_2 = \Theta\left(\frac{1}{\eta(\lVert \mu \rVert_2 + \tau)^2 \lVert w_O \rVert_2^2}\right)$. Thus, this implies $f_h^{(T_2)}(\mathbf{X}, \theta)=o(1)$. The
> $H$-fold increase in the multi-head model outputs does not alter this result, so the effect of the changes in $\frac{\partial \ell}{\partial f_h}$ can be ignored.
> Therefore, under the MHA setting, the training dynamics of the model still follow those of the single-head attention case, and our conclusions remain unchanged. Moreover, in our MNIST and CIFAR-10 experiments, we adopted a 4-head attention mechanism, and the results still exhibit the same phase transition behavior, providing empirical evidence that our theoretical analysis applies to MHA.
>
> 2. As for the FFN, its effect can be absorbed into the value matrix $W_V$.
>
> 3. Residual connections mainly serve as mechanisms for stabilizing gradient flow and improving training stability, rather than altering the fundamental principles governing how the model learns features from adversarial examples. Therefore, we simplify them in our theoretical setting.
>
>
>
> [1]Frei et al. Trained Transformer Classifiers Generalize and Exhibit Benign Overfitting In-Context
>
> [2]Magen et al. Benign Overfitting in Single-Head Attention
>
> [3]Sakamoto et al. Benign Overfitting in Token Selection of Attention Mechanism

---

> ### Author Response · Authors · 2025-11-24
>
> Dear Reviewer,
>
> Thank you again for your valuable feedback. We wanted to confirm that we have fully addressed all of your comments. If there are no remaining concerns, we would greatly appreciate your consideration in updating the score to reflect the improvements.
>
> Best regards,
>
> Authors

---

### Official Review · Reviewer_kd1t · 2025-11-03

**Soundness:** 3
**Presentation:** 3
**Contribution:** 3
**Rating:** 6
**Confidence:** 2

**Summary:**

The paper gives a theoretical account of robust benign overfitting in a simplified two‑layer ViT trained with ℓ₂ adversarial training. It proves that, under specific relationships between dataset size and signal‑to‑noise ratio (SNR), and for a moderate perturbation radius τ, the model can interpolate the training data (vanishing robust training loss) while maintaining small robust test error. Experiments on synthetic data and MNIST visualize phase transitions via heatmaps that align with the theory (e.g., boundary roughly 𝑁⋅SNR^2=Ω(1)).

**Strengths:**

++ The paper pinpoints when adversarially trained ViTs interpolate yet generalize robustly (e.g., N·SNR² = Ω(1) with τ ≲ ‖μ‖²₂ / log(dh)), and how robust error scales with d, SNR, τ, yielding explicit bounds and a practical “safe” τ range.

++ The analysis explains why moderate τ can “flatten” attention into near‑uniform weights—collapsing a ViT to a linear model—and contrasts convergence/SNR requirements with that degenerate baseline. This isolates an attention‑driven mechanism behind phase transitions.

++ The paper shows that no classifier can achieve nontrivial robust accuracy when τ ≥ ‖μ‖²₂ gives a sharp ceiling on what adversarial training can achieve, cleanly bracketing the benign region.

++ Heatmaps on synthetic and MNIST reproduce the predicted boundary N·SNR² = Ω(1) and show that robust gains appear only once both SNR and N clear the theoretical thresholds. The figures concretize the phase transition narrative.

**Weaknesses:**

-- Validation uses synthetic data and MNIST; no CIFAR/ImageNet‑scale tests or modern ViT training recipes, so the practical reach of the theory is not stress‑tested under real‑world pipelines, augmentations, or stronger attacks.

-- The two‑layer ViT and assumptions (e.g., multi‑patch distribution, specific τ/SNR scalings) help analysis but may not capture architectural and optimization nuances (depth, MHA heads, layernorm, schedules) that affect robustness in practice.

-- Results emphasize l2 training/attacks and do not discuss other threat models (l∞, l1, corruptions) or multi-step PGD details that affect robust outcomes; generalization across norms remains open.

**Questions:**

1. Do the phase boundaries or impossibility result change meaningfully for ℓ∞ or autoattack‑style suites? Any conjecture or preliminary evidence?

2. How do the conditions scale with number of heads M, head dimension, and depth? Can you extend the analysis (even heuristically) to stacked blocks or to pre‑norm residual forms common in ViTs?

3. Could you reproduce the heatmap phase boundary on CIFAR‑10/100 with small ViTs and ℓ₂ PGD to show qualitative agreement (even if not strictly in‑distribution with the theory)?

---

> ### Author Response · Authors · 2025-11-22
>
> - **W3 and Q1: Question about attack norm**
>    > Results emphasize $\ell_{2}$ training/attacks and do not discuss other threat models ($\ell_{\infty}$, $\ell_{1}$, corruptions) or multi-step PGD details that affect robust outcomes.
>
> ### Response to W3 and Q1:
>
>  Thanks for the comment. We focus on the $\ell_{2}$ threat model because it is the most commonly adopted norm in both theoretical and empirical robustness studies. And it does not limit generality, because perturbations under other norms can be mapped to $\ell_{2}$ through standard norm-equivalence. For any $x \in \mathbb{R}^d$ and $1 \le p \le q \le \infty$, the following inequality holds:$\lVert x \rVert_q \le d^{\left(\frac{1}{q} - \frac{1}{p}\right)} \lVert x \rVert_p.$
>
> In particular, the $\ell_{\infty}$ or $\ell_{1}$ norms satisfy:
>
> $\lVert x \rVert_{\infty} \le \lVert x \rVert_2 \le \sqrt{d}\lVert x \rVert_{\infty}$
>
> $\lVert x \rVert_2 \le \lVert x \rVert_1 \le \sqrt{d} \lVert x \rVert_2.$
>
> Thus, an $\ell_{\infty}$ or $\ell_{1}$ perturbation budget $\tau$ corresponds to an $\ell_{2}$ budget scaled by at most $\sqrt{d}$. That means, all the proofs are still hold with the only difference is the perturbation radius $\tau$.
>
> We include **additional experiments on MNIST and CIFAR-10 under multi-norm ($\ell_{1}$, $\ell_{2}$, $\ell_{\infty}$) attacks** in Appendix B.3. Following the training setup of [1], we select at each iteration the most challenging adversarial example among the three norms. The results show that although the model’s robust test accuracy decreases under multi-norm attacks, our theoretical insights continue to hold.
>
>
> Regarding the reviewer’s concern about attack strength, our theoretical framework (Algorithm 1, Step 4) explicitly guarantees optimization toward the strongest adversarial perturbation under the chosen norm. Thus, our adverarial examples are the worst case, stronger than the multi-step PGD you mentioned.
>
> In practice, we approximate the inner maximization via a 20-step PGD attack, which is a widely used and sufficiently strong procedure to capture near-optimal adversarial examples. Thus, the experiments faithfully reflect the threat model assumed in our theory.
>
>  [1]Maini et al. Adversarial Robustness Against the Union of Multiple Perturbation Models
>
>  ---
>
>  - **W1 and Q3: Question on dataset**
>     >Could you reproduce the heatmap phase boundary on CIFAR‑10/100 with small ViTs and $\ell_{2}$ PGD to show qualitative agreement.
>
> ### Response to W1 and Q3:
>
> Thanks for the comment. We have added **additional experiments on CIFAR-10 with APGD**, as detailed in Appendix B.2. We show that our theoretical insights continue to hold for larger and more complex datasets, as well as under stronger adversarial attacks. We still observe a clear phase transition phenomenon. Moreover, as both the sample size $N$ and the $SNR$ increase, the clean and robust test error consistently decreases.
>
>
> ---

---

> ### Author Response · Authors · 2025-11-22
>
> - **W2 and Q2: Question on ViT architecture**
>     >The two‑layer ViT may not capture architectural and optimization nuances (depth, MHA heads, layernorm, schedules) that affect robustness in practice. How do the conditions scale with number of heads M, head dimension, and depth?
>
> ### Response to W2 and Q2:
>
> Thanks for the comment. First, our theoretical analysis adopts a 2-layer Transformer model, which is a more realistic and more complex architecture than those considered in most prior theoretical works. Previous analyses often assumed linear attention [1] or merged $W_Q$ and $W_K$ into a shared matrix [2,3], whereas we employ a more faithful attention structure and carefully analyze the training dynamics of $W_Q$ and $W_K$, which introduces significant theoretical challenges.
>
> 1. For multi-head attention (MHA), we are fortunate to find that our theoretical framework extends to multiple heads relatively straightforwardly. We put the formal anlaysis in Appendix C. We let the parameters be
>
> $\theta := \{(W_{Q,h}, W_{K,h}, W_{V,h})\}_{h=1}^H$
>
> , where $W_{Q,h}, W_{K,h} \in \mathbb{R}^{d \times d_h}$ and $W_{V,h} \in \mathbb{R}^{d \times d_v}$ for each $h \in [H]$. Here $H$ denotes the number of attention heads, which we treat as a fixed constant. Under this parameterization, the network can be written as:$f({X}, \theta)=\sum_{h=1}^Hf_h({X}, \theta)$
> where,
>
> $$
> f_h(X, \theta)
> = \frac{1}{M} \sum_{l=1}^{M}
> \varphi(x_l^{\top} W_{Q,h} W_{K,h}^{\top} X^{\top})
> \, X W_{V,h} w_O.
> $$
>
> The gradients in the multi-head attention module,
> $\frac{\partial f_{h}}{\partial W_{K,h}}, \frac{\partial f_{h}}{\partial W_{Q,h}}, \frac{\partial f_{h}}{\partial W_{V,h}}$,
>
> remain unchanged. However, the gradient of the loss with respect to the output of each single head, i.e., $\frac{\partial \ell}{\partial f_h}$, does change.
> Intuitively, the model output increases by approximately an $H$-fold factor, which causes the scale of the loss $\ell'$ to decrease accordingly.
> More concretely, following our analysis of the signal attention head, $\ell'^{(t)}=\frac{1}{M}\pm o(1)$ stay when $t \leq T_2 = \Theta\left(\frac{1}{\eta(\lVert \mu \rVert_2 + \tau)^2 \lVert w_O \rVert_2^2}\right)$. Thus, this implies $f_h^{(T_2)}(\mathbf{X}, \theta)=o(1)$. The
> $H$-fold increase in the multi-head model outputs does not alter this result, so the effect of the changes in $\frac{\partial \ell}{\partial f_h}$ can be ignored.
> Therefore, under the MHA setting, the training dynamics of the model still follow those of the single-head attention case, and our conclusions remain unchanged. Moreover, in our MNIST and CIFAR-10 experiments, we adopted a 4-head attention mechanism, and the results still exhibit the same phase transition behavior, providing empirical evidence that our theoretical analysis applies to MHA.
>
> 2. Regarding the head dimension, we set it to $d_h$ in our formulation(1),$W_{Q}, W_{K} \in \mathbb{R}^{d \times d_h}$, and we do not simplify or reduce it.
>
> 3. As for model depth, unlike expermental studies, the developments of theory for transformers are quite slow, extending the analysis to deeper architectures is theoretically challenging.
>  Even for simpler models such as MLPs, extending convergence or learning-dynamics analyses to deeper networks often requires non-standard or impractical training strategies, such as layer-wise training [4,5]. We believe that extending our theoretical framework to multi-layer Transformers is an important and meaningful direction for future work, and we are optimistic that these challenges can eventually be addressed.
>
> 4. LayerNorm mainly affects training stability and scaling rather than fundamental expressive power. In practical models, LayerNorm improves optimization convergence and numerical stability by regulating activation distributions and gradient flow; however, its influence on the structural ability of attention to implement algorithmic mappings is typically limited. Hence, we consider it orthogonal to the core theoretical questions of this paper.
>
> 5. Training schedules affect the optimization trajectory but do not change the model’s expressive capabilities at the functional level. Different learning-rate schedules, warmup strategies, or weight decay may alter training speed or the selection of local minima, but they do not fundamentally create or remove representational capacity.
>
> [1]Frei et al. Trained Transformer Classifiers Generalize and Exhibit Benign Overfitting In-Context
>
> [2]Magen et al. Benign Overfitting in Single-Head Attention
>
> [3]Sakamoto et al. Benign Overfitting in Token Selection of Attention Mechanism
>
> [4]Nichani et al. Provable Guarantees for Nonlinear Feature Learning in Three-Layer Neural Networks
>
> [5]Wang et al. Learning Hierarchical Polynomials with Three-Layer Neural Networks

---

> ### Author Response · Authors · 2025-11-24
>
> Dear Reviewer,
>
> Thank you again for your valuable feedback. We wanted to confirm that we have fully addressed all of your comments. If there are no remaining concerns, we would greatly appreciate your consideration in updating the score to reflect the improvements.
>
> Best regards,
>
> Authors

---

### Meta-Review · Area_Chair_hCR7 · 2026-01-07

**Summary:**

The paper received mixed evaluations, with two positive scores and two negative scores. A common concern among the reviewers is the weakness of the experimental evaluation: the experiments are conducted on a small dataset using a simplified ViT model, and comparisons against stronger attack methods as well as multi-norm attack are missing.

In addition, Reviewer U4KV notes that the paper does not offer sufficiently novel insights to the community, despite acknowledging its theoretical contributions.

**Reviewer Concerns:**

After reviewing the rebuttal and the reviews, the AC appreciates the authors’ efforts and believes that many of the raised concerns have been addressed. However, the AC finds that the issue of weak evaluation is still outstanding despite the fact that a lot more experiments are included in the response. The experiments requested by reviewers (e.g., scale up to larger datasets and realistic vit models) are necessary for readers to fully understand and assess the theoretical contributions of the work. A more systematic experimental design is required, and the experimental section would need substantial revision to meet the acceptance bar this time. However, the AC believes this paper has a good chance of getting accepted by the next top conference if revised carefully.

**Reviewer Scores:**

Reviewer kd1t and Reviewer zyXo keep their scores unchanged.

Reviewer U4KV increases the score to 8, as communicated in the response.

Reviewer Uksd maintains the negative score of 2.

---

### Decision · Program_Chairs · 2026-01-26

Reject